# NoisyVideo: A Comprehensive Benchmark for Robust Video Understanding under Visual Noise

## Abstract

The recent progress of Video Multimodal Large Language Models (Video MLLMs) has greatly advanced multimodal understanding and reasoning in video analysis. However, the robustness of these models under diverse noise conditions that commonly occur in real-world scenarios remains largely unexplored, and existing research lacks systematic evaluations to comprehensively assess Video MLLM performance on question answering tasks under various noise conditions, both of which limit their reliability in practical deployments. To bridge this gap, we propose a thorough robustness benchmark encompassing 36 noise types in 8 categories, spanning diverse video categories and question types, resulting in 21,924 noise-corrupted test videos for comprehensive evaluation of Video MLLMs' robustness. We evaluate 10 state-of-the-art MLLMs on this novel benchmark for the initial systematic evaluation from multiple perspectives. Our multi-faceted analysis uncovers existing bottlenecks and performance degradation under certain noise scenarios, particularly for tasks requiring fine-grained understanding and reasoning. Additionally, we also examine the effectiveness of current image restoration techniques in mitigating noise effects and discuss their limitations. By constructing this extensive benchmark, our work lays a foundation for the systematic evaluations of Video MLLMs, offering insightful findings for future research on robust video-language understanding.

## 1 Introduction

Video understanding Cho et al. (2025); Shaker et al. (2025); Tang et al. (2023) is a core capability for modern intelligent systems, underpinning a wide range of downstream tasks such as action recognition Abdelkawy et al. (2025), optical character recognition Nagaonkar et al. (2025), and scene graph generation Kim et al. (2025). Recent advances in large language models (LLMs) Touvron et al. (2023); Achiam et al. (2023) have fueled a paradigm shift in video understanding, where researchers have achieved remarkable progress by introducing visual mapping layers Liu et al. (2023) that project visual signals into the LLM processing space. This approach has led to the rapid development of Video Multimodal Large Language Models (Video MLLMs) Maaz et al. (2023); Lin et al. (2023); Li et al. (2024c), resulting in a diversity of available models for many tasks. To evaluate the performance of these models on video understanding, some benchmarks have been introduced, focusing more on some aspects, such as longer video contexts Tan et al. (2025); Zhou et al. (2024); Fang et al. (2024) and specific application domains Xu et al. (2024b); Gao et al. (2025); Xie et al. (2025).

Despite the significant progress brought by these studies, a pervasive challenge in practical video analysis (i.e., the presence of noise corruption) may be overlooked and remain largely underexplored. Noise is known to impact the performance of LLMs remarkably Chen et al. (2023). Thus, the above motivations lead to our study: *Do Video MLLMs remain robust when faced with various types of noise corruption that commonly occur in practical deployment scenarios?*

In practice, visual noise can be occasionally introduced by imperfect filming techniques, hardware limitations, or transmission errors Boncelet (2009), and may be unnoticed in real-world situations Wang & Zhu (1998). Consequently, visual noise remains hidden in videos, potentially undermining the performance of Video MLLMs Schiappa et al. (2022). Previous works also show that MLLMs are

Figure 1: Overview of the eight major categories of noise included in our benchmark, with detailed noise types under each category. When applying Video MLLMs for video understanding, we expect to obtain reliable answers. However, in practice, due to the impact of visual noise, Video MLLMs may produce incorrect answers.

Table 1: Comparisons between our *NoisyVideo* benchmark and peer benchmarks about visual noise. Icons represent different noise types: Quality, Temporal, Blur, Lighting/Color, Scene Interference, Digital, Occlusion, and Compression.

| Model | Quality | Temporal | Blur | Lighting | Scene | Digital | Occlusion | Compression | Total | Year | Type | Task/Model |
|---|---|---|---|---|---|---|---|---|---|---|---|---|
| Hendrycks et al.Hendrycks & Dietterich (2019) | 3 | – | 4 | 2 | 3 | 2 | – | 1 | 15 | 2019 | Image | Image Classification |
| Schiappa et al.Schiappa et al. (2022) | 4 | 5 | 3 | – | – | 3 | – | 3 | 18 | 2023 | Video | Video-Language Model |
| Sun et al.Sun et al. (2023) | 3 | – | 4 | 2 | 3 | 2 | – | 1 | 15 | 2023 | Image | Text-Image retrieval |
| Chen et al.Chen et al. (2023) | 4 | 1 | 5 | 2 | 3 | 4 | – | 1 | 20 | 2023 | Image | Vision-Language Model |
| Shirnin et al.Shirnin et al. (2024) | – | – | 1 | – | 2 | 2 | – | – | 5 | 2024 | Image | Vision-Language Model |
| Zeng et al.Zeng et al. (2024) | – | 2 | 1 | 1 | – | – | 1 | – | 5 | 2024 | Video | Temporal Action Detection |
| Xie et al.Xie et al. (2025) | – | 2 | 2 | 3 | 5 | 1 | – | 2 | 15 | 2025 | Video | Autonomous Driving |
| **NoisyVideo (Ours)** | **4** | **4** | **5** | **6** | **6** | **6** | **2** | **3** | **36** | – | Video | Video Understanding |

more sensitive to visual perturbations than textual perturbations Qiu et al. (2022). This implies that analyzing the robustness of Video MLLMs on visual noise is a crucial and impactful research direction. However, current benchmarks related to visual noise either do not support comprehensive video understanding tasks Hendrycks & Dietterich (2019); Sun et al. (2023), or cover a narrow set of visual noise types Schiappa et al. (2022); Zeng et al. (2024), as also shown in Table 1. We aim to explore further in this direction.

To fill this gap, we propose a comprehensive benchmark that systematically evaluates the robustness of Video MLLMs under a wide range of visually noisy conditions for video understanding tasks. Our benchmark offers several distinctive features. First, we construct an extensive and fine-grained noise set with 36 corruption types across 8 major categories (see Figure 1), covering degradations such as quality, blur, temporal disruptions, lighting and color variation, scene interference (e.g., rain or fog), digital noise, occlusion, and compression. This diversity enables rigorous and realistic testing. Second, we employ multiple evaluation metrics, including subjective GPT-based scores, objective SBERT-based semantic similarity for all question types, and conventional accuracy on True/False questions, ensuring a holistic assessment of model robustness. Third, we conduct fine-grained analysis across different model architectures, video genres, and question types, revealing nuanced patterns of vulnerability and strength, such as the sensitivity of perception-based versus reasoning-based questions, the unique challenges of specific video categories, and the influence of model structure on robustness to visual noise. Furthermore, we examine whether recent image restoration techniques can effectively recover the Video MLLMs' understanding ability with visual noise.

Our analysis shows that pixel-level structural corruption substantially impairs the robustness of most models. Additionally, we observe that certain models are highly sensitive to specific video instances, noise types, or question categories, resulting in significant performance drops. While

denoising methods can partially restore performance on simple question answering tasks, they remain inadequate for recovering the fine-grained understanding required for complex reasoning. These findings highlight the significant challenges in achieving robust video understanding. Our work provides a practical and systematic benchmark for evaluating Video MLLMs and offers valuable insights to guide the development of more robust models.

Our contributions are summarized as follows:

- **A comprehensive visual noise robustness benchmark for Video MLLMs:** We propose *the first* systematic benchmark to evaluate the robustness of Video MLLMs, encompassing 8 broad categories and 36 noise types that represent various forms of video corruption relevant to practical deployment scenarios.

- **Multi-dimensional evaluation of robustness:** We consolidate holistic measurements that integrate subjective, objective, and accuracy-based metrics, showing a thorough quantitative evaluation of Video MLLMs' performance under various noise conditions.

- **Fine-grained analysis across multiple perspectives:** We conduct detailed analysis with model architecture, video genre, and question type, revealing nuanced patterns among different Video MLLMs and providing insightful findings for future research.

- **Fully open-source benchmark and tools:** We make our benchmark and its tools for construction fully open-source to boost the research in video understanding. `https://anonymous.4open.science/r/NoisyVideo-1AD5`

## 2 NOISYVIDEO

In this section, we provide the detailed construction of our benchmark. Specifically, we elaborate on the criteria of the visual noise types and evaluation metrics in our benchmark. To show its effectiveness, we conduct the evaluation of representative Video MLLMs on this benchmark.

### 2.1 VIDEO DEGRADATION METHODOLOGY

We introduce a wide range of noise types that are representative of common video degradation patterns to evaluate the robustness of Video MLLMs. While evaluating on naturally occurring noisy videos would be ideal, most real-world noise conditions present significant challenges for systematic analysis: they typically contain complex mixtures of multiple degradation sources that cannot be easily disentangled, and lack corresponding clean reference videos for controlled comparison. Therefore, following established practices in robustness evaluation Schiappa et al. (2022); Xie et al. (2025), we adopt a hybrid approach that combines authentic reproduction methods with well-validated synthetic generation techniques. For degradations that can be faithfully reproduced using standard processing pipelines (such as compression artifacts), we employ real-world encoding workflows to generate authentic corruptions. For other noise types, we implement established synthetic generation methods documented in the literature to ensure controlled experimental conditions that enable systematic analysis.

To ensure comprehensive coverage of potential degradation scenarios, we harvest noise types from different sources in both image and video domains. From the literature Perry (2018); Ramesh et al. (2022); Pyatykh et al. (2012); Lim (2006); Boncelet (2009); Foi et al. (2008); Zlokolica et al. (2006); Amer & Dubois (2005); Amrani et al. (2021); Wang et al. (2019a); Zhang et al. (2021); Guo et al. (2024); Hendrycks & Dietterich (2019), we identify *36 types of different noise* due to capturing, processing, and saving. Each noise reflects a realistic degradation that may influence the performance of Video MLLMs in practical application scenarios.

For degradations that can be faithfully reproduced, we employ industry-grade encoders to generate the distorted videos, ensuring these corruptions closely match their real-world counterparts. For instance, to simulate H.265 compression artifacts, we utilize the same FFmpeg workflows used in industry practice, producing artifacts that are representative of real-world compression noise.

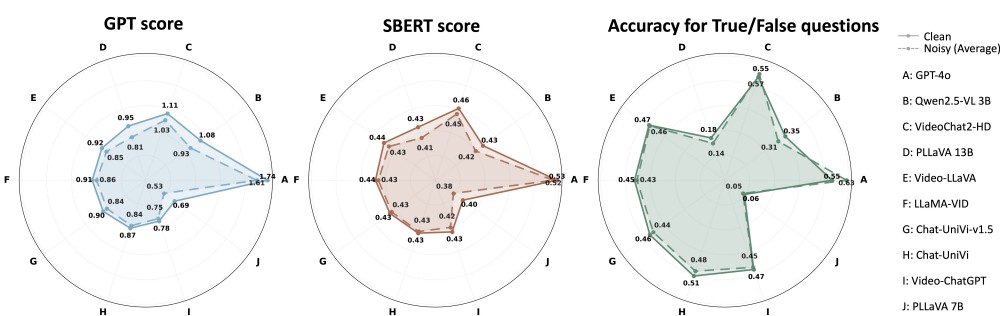

Figure 2: A preliminary study on different metrics in a controlled experiment.

## 2.2 NOISE TYPE

To better analyze the impacts of different noises, we further categorize them into *8 groups* by their characteristics. These categories are well recognized in the existing studies Hendrycks & Dietterich (2019); Zeng et al. (2024); Wang et al. (2019b). Namely, they are distinct noises related to quality, temporality, blurring, lighting/color, scene interference, digitality, occlusion, and compression. Figure 1 depicts the relationships in this taxonomy.

We notice that some noise types have corruptions in the pixel continuity. More specifically, Quality noise (e.g., Gaussian or impulse noise) emulates sensor imperfections or external interference, which often occurs in low-light environments or with low-quality sensors. Blur noise like motion blur and defocus blur replicates scenarios where camera movement or improper focus impact video clarity, such as in handheld or dynamic recordings. Temporal noise, including frame drops, frame repeats, and jitter, represents irregularities in video playback or transmission. These are common in videos transmitted over unstable networks or in hardware-constrained systems. Alternatively, compression (e.g., H.265 compression and JPEG artifacts) simulates distortions caused by lossy encoding processes during video upload, download, or storage, which are frequently encountered on streaming platforms or in bandwidth-limited scenarios. Occlusion is another important type of noise, which simulates situations where the camera lens is partially blocked out by stains, or when the shooting angle causes the target object to be partly covered. Digital distortions in videos cover a wide range of artifacts. Some are related to geometric deformations, such as rolling shutter and elastic distortions, which alter the shape and structure of objects within the frame. Others involve changes in resolution, such as resolution degradation, which reduces visual clarity and detail. Additionally, certain distortions affect color properties, like color quantization. Although the abovementioned noises originate from different sources, it should be noted that from the perspective of their visual manifestations, most of them ultimately result in the destruction of continuity information within the frames.

The other types of noise do not damage the pixel integrity of objects in the frames. Lighting/color noise, such as flicker, overexposure, or underexposure, primarily simulates challenging illumination conditions rather than altering the underlying structure of objects. These lighting-related noises are commonly used in data augmentation to improve the robustness of models against varying lighting environments. Scene interference, such as rain, fog, and shadows, is designed to mimic environmental conditions that are frequently encountered during outdoor video capture, for instance, in autonomous driving or surveillance scenarios. These effects introduce additional visual complexity but do not necessarily disrupt the object's structure.

The gathered noise types are carefully constructed with our best efforts to cover diverse and realistic scenarios, ensuring a comprehensive evaluation of Video MLLMs' robustness. By thorough testing under these noise conditions, we aim to examine the strengths and limitations of Video MLLMs. Detailed descriptions and parameter settings for each noise type are provided in Appendix A.2.

## 2.3 EVALUATION METRICS

We find that existing studies generally rely on an LLM judge to evaluate the performance of Video MLLMs Fang et al. (2024); Maaz et al. (2023); Xu et al. (2024b). We have conducted a preliminary study with three measurements on video understanding with all visual noise as shown in Figure 2.

It is worth noticing that different models exhibit varying changes in performance across different metrics (i.e., columns A, B, and C), implying that relying on limited metrics could be biased. Therefore, we tend to extend the benchmark with more measurements to improve the fairness and comprehensiveness.

We follow the existing study Fang et al. (2024) to include a traditional measurement in our benchmark. Specifically, the metric is scored by conventional LLMs, also known as *GPT score*. However, since this scoring is subjective to single models, we aim to include alternative measurements. From the literature, we identify the *SBERT score* Reimers & Gurevych (2019), which is objective and commonly used in many LLM-related works Supriya et al. (2023); Zhong et al. (2020); Moon et al. (2020). We include this metric in our benchmark.

To further enhance the comprehensiveness of the measurements, we consider the statistical measurements, besides the predictive ones (i.e., GPT score) and the semantic ones (i.e., SBERT score). More specifically, the *accuracy* is measured on the selection-based questions (i.e., True or False questions in our benchmark) with concrete correct answers. It provides a new angle to assess the robustness of Video MLLMs under noise conditions.

In summary, our evaluation framework incorporates:

- ✓ **GPT Score**: Subjective assessment following established practices in Video MLLM evaluation, where GPT-4o OpenAI (2024) evaluates response quality based on relevance, correctness, and completeness.

- ✓ **SBERT Score**: Objective semantic similarity measurement that computes cosine similarity between model responses and ground truth answers using sentence embeddings, providing reproducible and bias-free evaluation.

- ✓ **Accuracy**: Concrete performance measurement on selection-based questions (e.g., True/False, multiple-choice) with definitive correct answers, offering clear and interpretable results for fundamental understanding assessment.

The NoisyVideo benchmark measures the abovementioned three metrics individually on both settings with and without noise presence. The difference between the two measurements reveals the intrinsic robustness of Video MLLMs, eliminating the other factors in a controlled experiment manner.

## 2.4 EVALUATION SUBJECTS

To demonstrate the effectiveness of our benchmark and answer the research question in our study, we conduct a comprehensive evaluation on various subjects.

We adopted 609 videos exhaustively from MMBench-video Fang et al. (2024), which consists of 19 video types and 9 question types to cover diverse practical application scenarios. All 36 types of noise are applied to these videos individually, producing a exhaustive set of 21,924 videos as our evaluation subjects.

For the Video MLLM subjects, since a full evaluation of all constructed video tasks would require substantial computational resources, we focus our assessment on representative models. We select models from two perspectives: *general MLLMs* and *specialized Video MLLMs*.

General MLLMs, such as GPT-4o OpenAI (2024) and Qwen2.5-VL 3B Yang et al. (2024), are chosen because they are widely adopted as base models for a variety of tasks, including video understanding Li et al. (2024a); Chen et al. (2025). Although these MLLMs are not specifically designed for video, their broad applicability makes them valuable references. For specialized Video MLLMs, we specifically include two spatio-temporal enhanced models (i.e., Chat-UniVi Jin et al. (2024) and Video-ChatGPT Maaz et al. (2023)), both of which focus on effective integration of temporal dynamics and spatial features. Four more state-of-the-art Video MLLMs (i.e., PLLaVA Xu et al. (2024a), VideoChat2 Li et al. (2023), Video-LLaVA Lin et al. (2023), and LLaMA-VID Li et al. (2024c)) are selected due to their popularity and recognition in related works. We also consider the impact of model size on the video understanding capabilities. The aforementioned specialized Video MLLMs are 7B-parameter models, which are widely adopted as the baselines in the MLLMs Li et al. (2024a); Liu et al. (2023). Moreover, we include the 13B variant of PLLaVA and the enhanced data

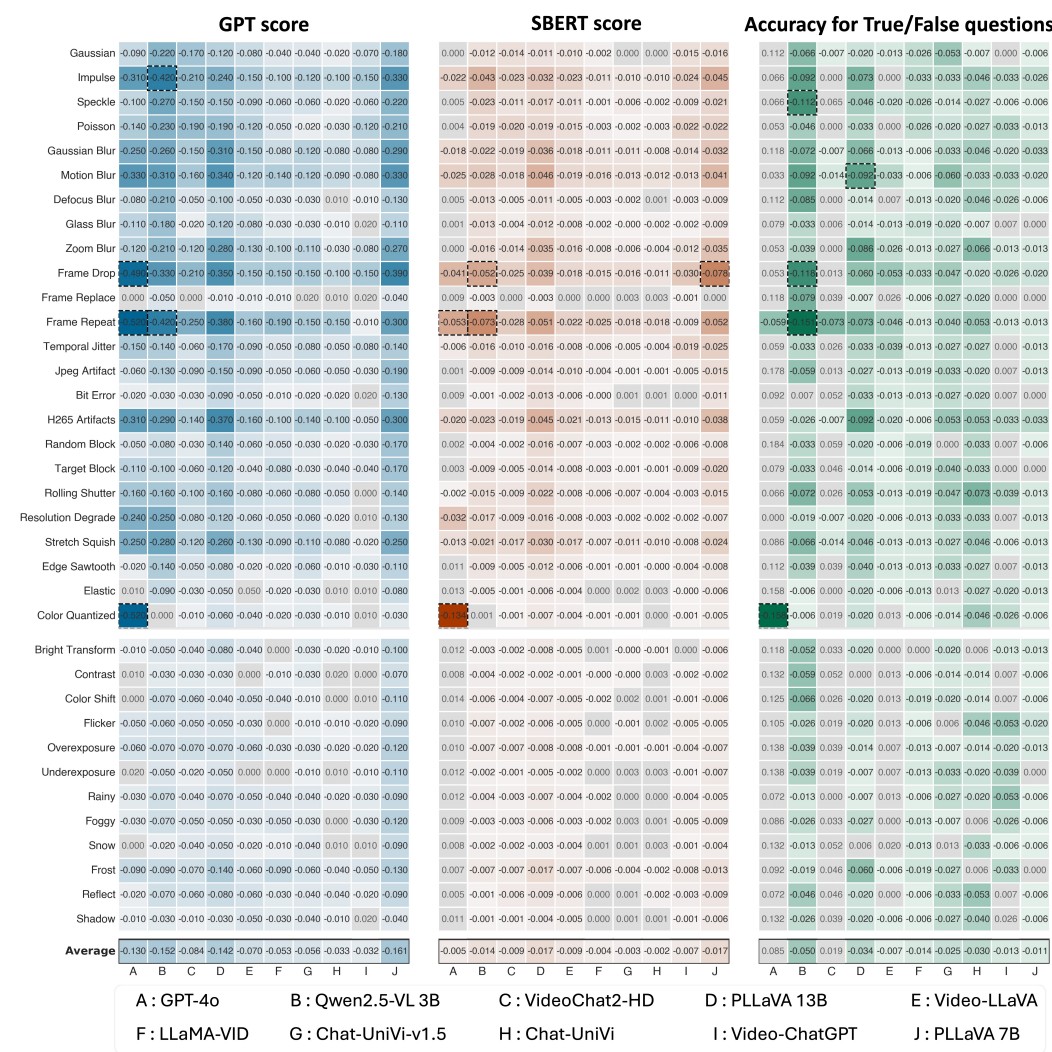

Figure 3: Heatmap of performance drops for general MLLMs and the specialized Video MLLMs under different noise conditions.

version of Chat-UniVi-v1.5 to enrich the evaluation, thanks to their public availability. The public links can be seen in Appendix A.3.

This setup enables us to analyze the variance across video scenarios, model architectures, model size, and training data, contributing to the video understanding robustness.

## 3 EXPERIMENTS

This section presents a comprehensive evaluation of Video MLLMs' robustness under various noise conditions using our NoisyVideo benchmark. We systematically analyze the performance degradation patterns across different dimensions to understand the vulnerability and resilience characteristics of current Video MLLMs. Additionally, we investigate whether existing video denoising methods can restore noisy videos to achieve comparable video understanding performance as clean videos, providing valuable insights for practical deployment scenarios. Due to space constraints, more experimental results are presented in the Appendix, with all visual results (figures and charts) relocated there to maintain focus on the core findings and analysis in the main text.

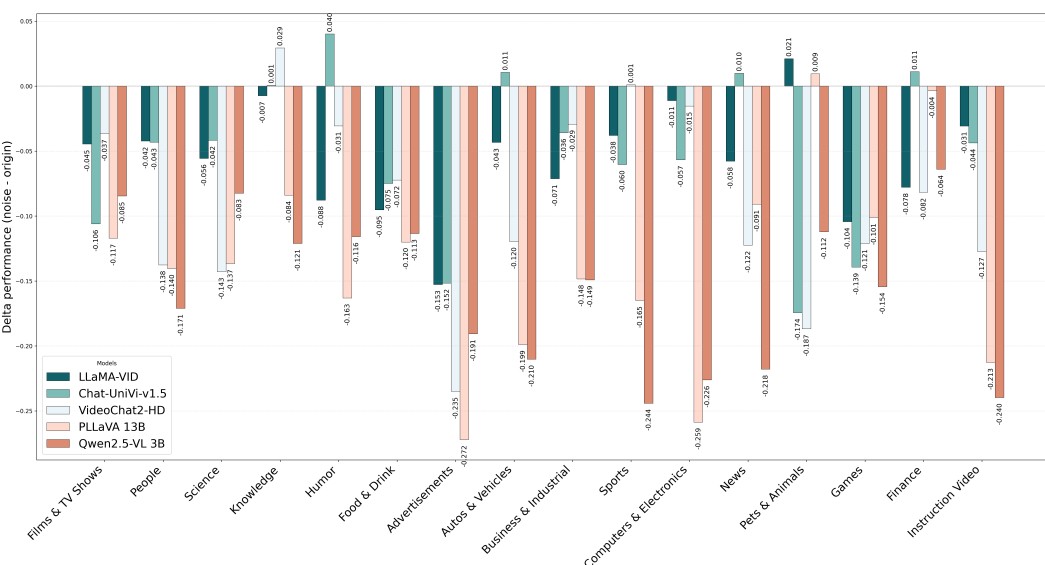

Figure 4: Delta performance (GPT Score) of general and Video MLLMs across different video categories. More detailed SBERT and accuracy results are provided in the Appendix A.7.

## 3.1 EVALUATION OF VIDEO MLLMS UNDER NOISE CONDITIONS

We evaluate the impacts of different visual noise types on video understanding tasks and assess the robustness of Video MLLMs under these noise conditions. Specifically, given the original performance of subject Video MLLMs on quality-assured videos (i.e., without noise in the videos), we examine the performance of the same subject models on the noisy version of the same videos. Then, we perform the comparison study on the two acquired performances to evaluate the impacts of noise. To ease our presentation, we refer to the quality-assured videos as *clean* videos and the noise-amended videos as *noisy* videos. We also perform a comparative analysis on different groups to investigate the potential factors related to robustness. We summarize the key results in Figure 3.

### 3.1.1 IMPACTS OF NOISE TYPES ON ROBUSTNESS OF VIDEO MLLMS

Figure 3 presents the delta performance of 10 models across 36 noise types. The columns of models are sorted by their original performance on clean videos in descending order (The general MLLMs and the specialized Video MLLMs are ranked separately). The rows of noise types are grouped by their influence on pixel continuity, as we show in Section 2.2. The last row summarizes the average impacts of different noises on the performance.

As illustrated in Figure 3, it is evident from all three metrics that noise generally exerts a detrimental effect on the performance of Video MLLMs. Out of 1,080 cases, 163, 44, and 873 cases show increases, retention, and drops in performance, respectively. The performance drops occupy 80.83% of cases in the experiments. Going deeper, general MLLMs (i.e., GPT-4o and Qwen2.5-VL 3B) suffer heavier performance drops than specialized Video MLLMs. These two general MLLMs receive better performance on clean videos but perform relatively worse on noisy videos on GPT score, which implies that *high video understanding performance does not ensure robustness*. Users may choose specialized Video MLLMs in the application scenarios with noise.

Another finding from this large-scale experiment is that *noises corrupting the pixel continuity generally have a greater negative impact on robustness*. For the two groups, the performance drops in the corruption group range from $-0.20$ to $-0.04$ for GPT Score ($-0.12 \pm 0.07$), and from $-2.25e{-}2$ to $-4.10e{-}3$ for SBERT Score ($-1.23e{-}2 \pm 6.70e{-}3$). In contrast, in the other group, GPT Score drops range from $-0.10$ to $-0.01$ ($-0.04 \pm 0.03$), and SBERT Score drops from $-6.80e{-}3$ to $9.60e{-}3$ ($-1.80e{-}3 \pm 4.80e{-}3$), which is significantly slighter than the former. Furthermore, as shown in the accuracy statistics, the mean delta performance of accuracy for the models under corruption group is $-0.01 \pm 0.04$. However, for the other group, the performance increases to $2.9e{-}3 \pm 0.04$. This indicates that, for these simple question-answering tasks, the impact of noise

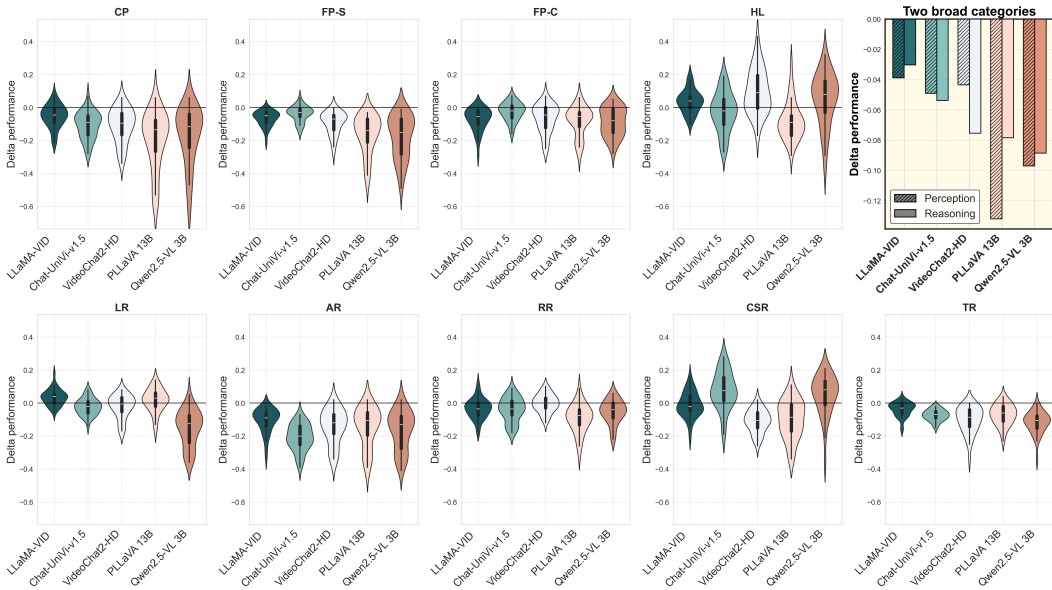

Figure 5: Delta performance (GPT Score) of general and Video MLLMs across different question categories. More detailed SBERT and accuracy results are provided in the Appendix A.7.

types related to color is minimal, and model performance remains largely unaffected. Overall, these results highlight the critical role of pixel continuity in the video understanding capabilities of Video MLLMs. Developers should therefore pay particular attention to noise that disrupts pixel continuity when designing and deploying such models in practical application scenarios.

### 3.1.2 ANALYSIS OF ROBUSTNESS OF VIDEO MLLMS ACROSS VIDEO TYPES

Figure 4 presents the delta performance of representative Video MLLMs on average in groups of different video types. Echoing the video categories mentioned in Section 2.4, we investigate the impact of noise on robustness in different video contents. 16 video categories are identified from the 609 videos[1]. According to the performance in Figure 3, we focus on five representative models: two highly robust models (i.e., LLaMA-VID and Chat-UniVi-v1.5), one moderate model (i.e., VideoChat2-HD), and two less robust ones (i.e., PLLaVA 13B and Qwen2.5-VL 3B). Results show that robustness varies substantially by video categories and models. Overall, most models perform poorly on advertisements, suggesting that there may be limited effective training data in popular training corpora. Qwen2.5-VL 3B and PLLaVA 13B are especially non-robust in noisy sports and electronics videos, while Chat-UniVi-v1.5 remains robust on news content. This result shows that each model has its strengths and weaknesses across different video categories. Developers may evaluate and select the most suitable Video MLLM for their specific application, considering the video type factor.

### 3.1.3 ANALYSIS OF ROBUSTNESS OF VIDEO MLLMS ACROSS QUESTION TYPES

Figure 5 presents the delta performance of representative Video MLLMs in distributions and groups of different question types. The questions are divided into nine types within two broad categories: a) four perception questions, coarse perception (CP), fine-grained perception-single instance (FP-S), fine-grained perception-cross instance (FP-C), and hallucination (HL), and b) five reasoning questions, logic (LR), attribute (AR), relation (RR), common sense (CSR), and temporal reasoning (TR). The results show that model robustness varies notably by question type. For instance, PLLaVA's GPT score declines by over 13% on perception questions when noise is present, but only drops about 8% on reasoning questions. This difference may be due to the fact that its descriptive answers are more resilient to noise in reasoning tasks. All models struggle significantly with Attribute

---

[1]Three categories are filtered out due to insufficient videos for analysis, i.e., less than 10 videos.

Reasoning questions, exhibiting low robustness and high variance. This suggests that detailed attribute recognition is particularly sensitive to visual noise. Since these tasks require the identification of fine-grained properties such as color, shape, or emotion, noise can severely distort inputs and lead to a considerable decrease in performance. Overall, the effect of visual noise varies by question type, and fine-grained attribute reasoning continues to pose a major challenge for current Video MLLMs in noisy conditions.

## 3.2 DISCUSSION

Recent advances in image restoration have led to the development of effective methods for reducing visual noise in images. A key question is whether current Video MLLMs can achieve the same level of understanding when analyzing restored (denoised) videos compared to clean videos. To explore this, we employ two different approaches for video enhancement: the classic NAFNet model Chen et al. (2022) for CNN-based denoising and the SeedVR2-7B Wang et al. (2025) which is one diffusion model. We focus on three types of noise for our analysis: impulse noise, motion blur and H.265 which had a significant impact in our previous experiments.

As shown in Figure 6, we compare the results for each noise type across four settings: clean, noisy, and two types of restored videos. Our results reveal that NAFNet denoising maintains similar GPT and SBERT scores to noisy videos while improving accuracy, whereas SeedVR2-7B shows inconsistent performance across different noise types.

These findings suggest that CNN-based restoration enables the model to recover sufficient information for simple True/False questions, but fails to provide the finer details required for precise descriptions or complex reasoning. The

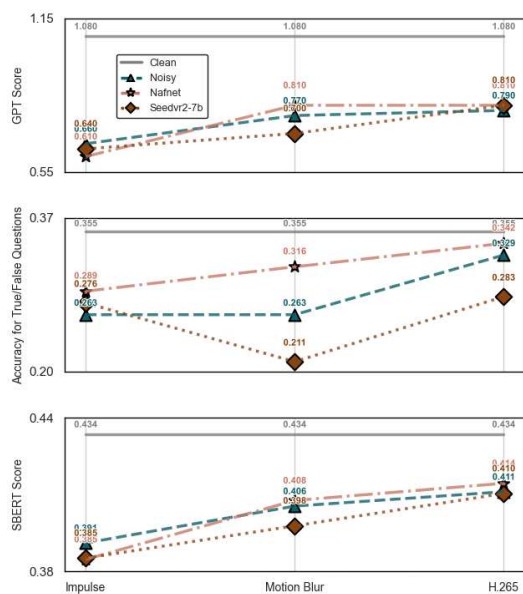

Figure 6: Impact of noise and denoising on Qwen2.5-VL 3B model performance across multiple metrics.

diffusion-based approach demonstrates limitations in handling specific noise types, particularly motion blur. Both restoration methods indicate that video understanding tasks demand more detailed and nuanced visual information than simple question answering, and such fine-grained details remain challenging to fully restore with current techniques. This underscores a fundamental challenge: bridging the gap between perceptual visual quality improvement and robust, accurate video-language understanding.

## 4 CONCLUSION

We present a comprehensive evaluation of state-of-the-art Video MLLMs under various visual noise conditions. By examining both general-purpose and specialized models across diverse video categories and question types, we reveal significant robustness differences and specific challenges. Our results show that robustness largely depends on video domain and task type, with fine-grained reasoning especially sensitive to noise. While image restoration methods can partially recover performance on simple factual questions, they remain insufficient for restoring details needed for accurate descriptions and complex reasoning. These findings highlight the need for more robust Video MLLMs and improved restoration techniques that better preserve semantic and fine-grained information. Our benchmark and insights provide valuable guidance for evaluating and selecting Video MLLMs in practical application scenarios and support future progress in robust video understanding.

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

# A APPENDIX

This appendix provides supplementary information to our main paper. We first shows the related work, second we present the noise parameter settings used in NoisyVideo to facilitate reproducibility. Thirdly, we describe model implementation details, followed by a discussion of the limitations of our work. Then we declare the use of LLM in our benchmark. We also present the additional experimental results and visualizations not included in the main manuscript. Finally, we present the License of the our work.

## A.1 RELATED WORK

### A.1.1 VIDEO MLLMS FOR VIDEO UNDERSTANDING

In recent years, LLMs have rapidly evolved towards multimodal capabilities, exemplified by GPT OpenAI (2023), Gemini Gemini Team (2023), and Qwen Yang et al. (2024), which aim to provide general understanding ability across multiple modalities. To better address video-related tasks, a new line of research has emerged focusing on specialized LLMs for video understanding. Early works such as Video-ChatGPT Maaz et al. (2023) aligned frame-level visual features with language embeddings through linear projectors, while VideoChat Li et al. (2023) added a learnable Q-former structure. Video-LLaMA Zhang et al. (2023) integrated audio features, and Video-LLaVA Lin et al. (2023) learned from mixed image and video datasets. In this work, we construct a benchmark to evaluate the video understanding capabilities of both general MLLMs and video-specialized models under various visual noise conditions.

### A.1.2 VIDEO UNDERSTANDING

Currently, video question answering (VideoQA) is the predominant kind of task for evaluating a model's video understanding capabilities Xu et al. (2022); Xiao et al. (2025). Early benchmarks typically focused on a single type of video content Tapaswi et al. (2016); Garcia et al. (2020); Mangalam et al. (2023). In recent years, VideoQA tasks have expanded to cover a much broader range of video genres and question types Li et al. (2024b); Ning et al. (2023); Fu et al. (2024); Fang et al. (2024). Video-MME Fu et al. (2024) is the comprehensive evaluation benchmark for Video MLLM. MLVU Zhou et al. (2024) focuses on long videos. MMBench-Video Fang et al. (2024) covers 16 types of videos and 26 categories of questions. Due to the complexity and diversity of question types in current VideoQA datasets, including the challenging type of open-ended questions, LLMs such as GPT are usually employed as evaluation methods Maaz et al. (2023). Unlike the existing benchmarks, we focus on visual noise in videos and conduct a comprehensive and multi-faceted evaluation for video understanding under noise conditions.

### A.1.3 VISUAL MODEL ROBUSTNESS

Recently, robustness analysis in multimodal tasks has emerged, which considers the real-world situations and takes a further step towards a more reliable system. Yet, this field is still in its infancy. Li et al. Li et al. (2020) take the first step to analyze the robustness in a multimodal task, Visual Question Answering (VQA), against 4 generic corruptions, including linguistic variation and visual content manipulation. Schiappa et al. Schiappa et al. (2022) introduce naturally corrupted visual and textual benchmarks on text-to-video retrieval. Chen et al. Chen et al. (2023) assess the robustness of 11 widely-used adaptation methods across 4 vision-language datasets under mixed corruptions. Xu et al. Xu et al. (2024b) propose a benchmark to evaluate the model robustness based on edited videos. Zeng et al. Zeng et al. (2024) analyze the robustness of seven leading temporal action detection (TAD) methods. However, few of these benchmarks have systematically analyzed the robustness of Video MLLMs to visual noise. Our benchmark fills this gap by exploring a comprehensive evaluation in this area.

## A.2 NOISE IMPLEMENTATION DETAILS

Our benchmark categorizes noise into 8 major types, and we employ PSNR and SSIM as control metrics to ensure fair comparison across different categories. We implement a two-level control strategy: first, we normalize perturbations within each noise category to achieve similar PSNR

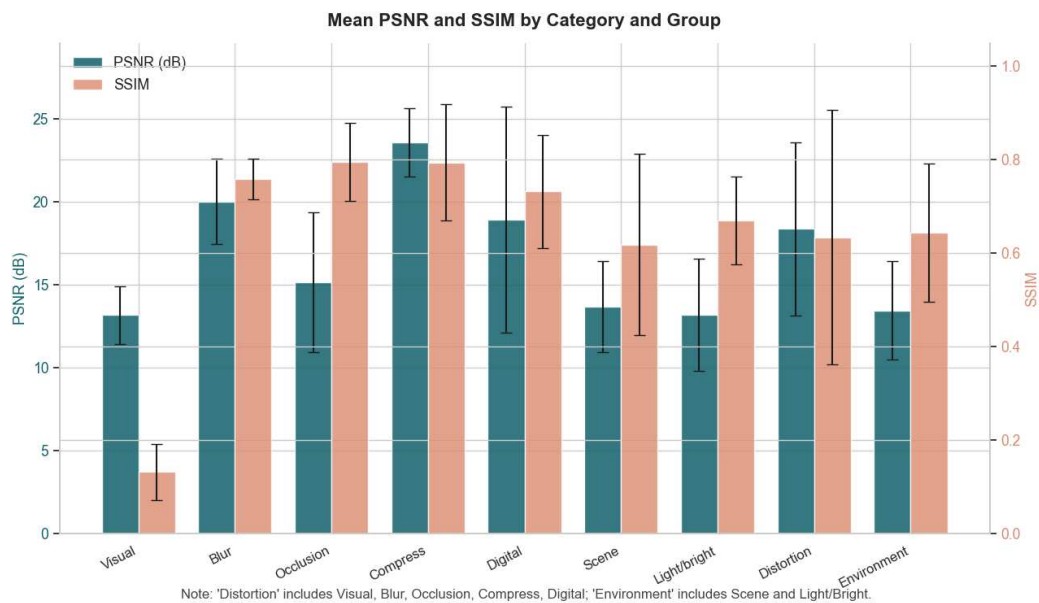

Figure 7: PSNR and SSIM for one video demo

and SSIM values; second, we further classify these 8 noise categories into two higher-level groups based on whether they introduce corruptions in pixel continuity. Specifically, we categorize noises that disrupt pixel continuity as "distortions", including Quality noise, Blur noise, Temporal noise, Compression, Occlusion, and Digital distortions. The remaining categories—Lighting/color noise and Scene interference—are classified as "environment noise" as they preserve pixel continuity. Since our classification criterion focuses on pixel continuity, we particularly emphasize controlling SSIM values to be nearly identical between the distortion and environment noise groups, while also maintaining PSNR differences within reasonable ranges. This controlled approach ensures that we mitigate the effects of different noise strengths, allowing for fair comparison across categories.

Note that some references use 3 Zeng et al. (2024) or 5 levels Hendrycks & Dietterich (2019) for each perturbation and report averaged results across these levels, while we directly control perturbation strength through PSNR and SSIM metrics to achieve more precise normalization. Both methodologies serve the same fundamental purpose: controlling variables to enable fair comparison across different perturbations. We use one video to show the demo PSNR/SSIM value, as shown in Figure 7. We detailed noise setting is shown as below:

**Gaussian noise**: Additive noise sampled from a zero-mean normal distribution with standard deviation 100 is added to each frame, and results are clamped to the valid pixel range [0, 255].

**Impulse noise**: A uniform mask is sampled for each frame. With default probability 0.7, pixels with mask values below 0.35 are set to black (0), those between 0.35 and 0.7 are set to white (255), and the rest remain unchanged.

**Speckle noise**: Multiplicative noise with intensity 0.7 is applied. The noise map is generated at full resolution for single-pixel grains or upsampled from a lower resolution patch for larger grains. Noisy pixels are computed as $I_{\text{noisy}} = I \times (1 + \text{noise})$, with results clamped to [0, 255].

**Poisson noise**: Each frame is scaled by gain (default 0.01), used as the rate parameter for Poisson sampling, then divided by gain to restore scale, and the results are clamped to [0, 255].

**Gaussian blur**: A 2D Gaussian kernel of size 101 with $\sigma = 20$ is generated and applied via depth-wise convolution across all channels for each affected frame, smoothing image details according to the specified kernel.

**Motion blur**: A 2D motion kernel of size 101 oriented at $45°$ is generated and applied via depth-wise convolution across all channels for each frame, averaging pixel values along the specified direction to simulate directional motion blur.

**Defocus blur**: A disk-shaped convolution kernel is generated by thresholding a circular mask of radius $r_s$ (with $r_s \in \{3, 4, 6, 8, 10\}$ for severity levels $s = 1, \ldots, 5$) and applying Gaussian anti-aliasing with standard deviation $\sigma_s \in \{0.1, 0.5, 0.5, 0.5, 0.5\}$. This smoothed disk kernel is then applied per channel via 2D filtering to simulate an out-of-focus effect, with pixel values clamped to the valid range $[0, 255]$.

**Glass blur**: A two-stage Gaussian smoothing (with $\sigma \in \{0.7, 0.9, 1.0, 1.1, 1.5\}$ for severity levels 1–5) is applied before and after a local pixel shuffle (max displacement $\Delta \in \{1, 2, 2, 3, 4\}$ pixels over $N \in \{2, 1, 3, 2, 2\}$ iterations), creating a refractive "frosted glass" effect.

**Zoom blur**: For severity $s \in \{1, \ldots, 5\}$, define the zoom factors $c_s = \{1.0, 1.0 + \Delta_s, \ldots, 1.0 + 0.1s\}$ with $\Delta_s \in \{0.01, 0.01, 0.02, 0.02, 0.03\}$ for $s = 1, \ldots, 5$; for each $f \in c_s$, crop the central $\frac{H}{f} \times \frac{W}{f}$ region, bilinearly resize it to $H \times W$, sum these zoomed frames with the original, divide by $|c_s| + 1$, and clamp to $[0, 255]$ to produce the radial zoom-blur effect.

**Frame drop**: A random subset of frames is replaced with blank frames by setting every pixel channel to 0, simulating the complete loss of those frames.

**Frame replace**: A subset of frames is selected according to the noise ratio; those frames are randomly permuted among themselves, and the shuffled frames are reinserted in place to simulate out-of-order frame delivery.

**Frame repeat**: A subset of frames is selected as reference frames, and each remaining frame is replaced by its temporally nearest reference frame, producing repeated-frame artifacts throughout the video.

**Temporal jitter**: A random split of the noise ratio $\rho$ is computed by sampling $\alpha = \rho \cdot U(0, 1)$ and setting $\beta = 1 - \alpha$; then frame replacement (with fraction $\alpha$) and frame drop (with fraction $\beta$) are applied in sequence according to the specified execution order (default `replace_first`), yielding a mixture of shuffled and omitted frames.

**JPEG artifact**: A subset of frames selected by the noise ratio is encoded to JPEG at quality level $q \in [1, 100]$ (default $q = 1$) and then decoded, introducing quantization and blocking distortions characteristic of low-quality JPEG compression.

**Bit error**: A subset of frames (controlled by the noise ratio) is selected, and in each a random rectangular region covering between $\frac{1}{4}$ and $\frac{1}{2}$ of the frame is chosen; within that region vertical stripes of width $w \in \{15, 10, 5\}$ for severity levels $\{low, medium, high\}$ are created by copying pixels from randomly sampled columns, yielding a streaked stripe-artifact effect.

**H.265 artifacts**: A two-pass H.265 encoder is applied with the quantization parameter CRF $\in \{32, 38, 45, 51\}$ and bitrate $\in \{\text{400k}, \text{200k}, \text{100k}, \text{50k}\}$ for low-extreme severity levels, where frames are first downscaled to $\frac{1}{4}$ resolution before reencoding at full size, producing pronounced blocking, ringing, and other compression artifacts.

**Random block**: For each selected frame, a rectangular region of height $h \in [0.3H, 0.7H]$ and width $w \in [0.3W, 0.7W]$ is sampled at a random position and all its pixel values are set to zero, simulating partial occlusion.

**Target block**: For each selected frame, a pre-trained detection model predicts the bounding box of the primary object $(x_1, y_1, x_2, y_2)$, and all pixels within that region are set to zero to occlude the main subject.

**Rolling shutter**: A subset of frames (ratio $\rho$) is chosen and for each selected frame at index $t$, a random orientation (horizontal or vertical) and direction are picked; then for each column $x$ (or row $y$), a temporal offset

$$\delta = \min\left(\left\lfloor \tfrac{x}{W-1} \, d \, \min(t, B) \right\rfloor, \; B\right)$$

(with width $W$ or height $H$, delay factor $d$, and buffer size $B$) is computed, and the pixel strip at $x$ (or $y$) is sampled from frame $t - \delta$, yielding a characteristic rolling-shutter smear.

**Resolution degrade**: Each selected frame is downsampled via bicubic interpolation to $\left(\lfloor sH \rfloor, \lfloor sW \rfloor\right)$ with scale factor $s$ (default $s = 0.1$) and then upsampled back to $(H, W)$ via bicubic interpolation, with pixel values clamped to the valid range $[0, 255]$.

**Stretch/squish**: A random orientation (horizontal or vertical) is selected, then each chosen frame is downsampled via bicubic interpolation along that axis to size $\left(H, \lfloor sW \rfloor\right)$ or $\left(\lfloor sH \rfloor, W\right)$ with scale factor $s = \frac{1}{30}$, and finally upsampled back to $(H, W)$ via bicubic interpolation, with pixel values clamped to $[0, 255]$.

**Edge sawtooth**: Edges are detected via Canny with thresholds $(T_{\text{low}} = 50, T_{\text{high}} = 150)$, dilated by a $2 \times 2$ kernel, then a surround mask of neighboring pixels is sampled with probability $\rho = 0.3$, and all masked pixels are replaced with uniformly random RGB values to produce serrated, noisy edges.

**Color quantization**: Each selected frame's pixel values are uniformly quantized to $2^b$ levels by computing

$$\tilde{I} = \text{round}\left(\frac{I}{\Delta}\right) \times \Delta, \quad \Delta = \frac{255}{2^b - 1} \quad (b = 3 \text{ by default}),$$

and then clamping $\tilde{I}$ to the valid range $[0, 255]$.

**Elastic transform**: For severity $s \in \{1, \ldots, 5\}$, generate random displacement fields $\Delta_x, \Delta_y$ by sampling uniformly in $[-\delta_{\max}, \delta_{\max}]$ with $\delta_{\max} = 0.005 \min(H, W)$, smoothing each via a Gaussian filter with $\sigma = 0.01 \min(H, W)$, and scaling by $\alpha_s \in \{12.5, 16.25, 21.25, 25, 30\}$; the warped image is then obtained by remapping pixels from $(x, y)$ to $\left\lfloor x + \Delta_x(x, y), \, y + \Delta_y(x, y) \right\rfloor$ using bilinear interpolation, and clamped to $[0, 255]$.

**Brightness transform**: For severity levels $s \in \{1, \ldots, 5\}$, let $c_s = 0.1s$; each selected frame is converted to HSV (for color), its V channel is increased by $c_s$ (clipped to $[0, 1]$) before converting back to RGB, or for grayscale frames the intensity is simply offset by $c_s$ (clipped to $[0, 1]$), then rescaled to $[0,255]$.

**Contrast transform**: For each selected frame, compute its mean intensity $\mu$, sample a factor $\alpha \sim \mathcal{U}(m, M)$ with $(m, M) = (-5, 5)$, replace $\alpha$ by $-1/\alpha$ if $\alpha < 0$, and apply

$$I_{\text{out}} = \text{clip}\left(\alpha (I - \mu) + \mu, \, 0, \, 255\right),$$

thereby symmetrically scaling pixel deviations around the original mean.

**Color shift**: For each selected frame, sample multiplicative gains $\alpha_k \sim \mathcal{U}(1 - s, \, 1 + s)$ for $k = 1, 2, 3$ (or a single $\alpha$ if `per_channel=false`), multiply each channel by its $\alpha_k$, and clamp the result to $[0, 255]$, where $s$ is the shift magnitude (default $s = 1$).

**Flicker**: For each frame, sample $\alpha \sim \mathcal{N}(1, \rho^2)$ with $\rho$ equal to the flicker ratio (default 0.2), multiply all pixel values by $\alpha$, and clamp to $[0, 255]$, producing random frame-level brightness fluctuations.

**Overexposure**: For each selected frame, sample a brightness offset $b \sim \mathcal{U}(0.1, 0.3)$ and a gamma exponent $\gamma \sim \mathcal{U}(1.1, 1.4)$; first add $b$ to normalized intensity $(I/255)$, then apply inverse gamma correction $(I')^{1/\gamma}$, and rescale to $[0,255]$, yielding locally blown-out highlights.

**Underexposure**: For each selected frame, sample a brightness offset $b \sim \mathcal{U}(-0.3, -0.1)$ and a gamma exponent $\gamma \sim \mathcal{U}(0.6, 0.9)$; first add $b$ to normalized intensity $(I/255)$, then apply inverse gamma correction $(I')^{1/\gamma}$, and rescale to $[0,255]$, producing dimmed, murky shadows.

**Rain**. For severity level $s \in \{1, \ldots, 5\}$, define $(m_s, L_s, \rho_s)$ as the $s$th row of

$$\begin{pmatrix} 0.05 & 8 & 1 \\ 0.07 & 10 & 2 \\ 0.09 & 15 & 3 \\ 0.11 & 18 & 4 \\ 0.13 & 22 & 5 \end{pmatrix},$$

where the three columns correspond to $m_s$, $L_s$ and $\rho_s$, respectively. Blur this density map with a Gaussian filter ($\rho = 0.5$), then blend the original image of each frame $I$ and the rain layer $R$ by

$$I_{\text{rain}} = \text{clip}\left(I (1 - 0.3 R) + 200 R b_s, \, 0, \, 255\right),$$

using the brightness–contrast pairs $(b_s, c_s)$ from

$$\{(0.20, 1.03), (0.25, 1.04), (0.30, 1.05), (0.35, 1.06), (0.40, 1.07)\},$$

and finally add the fog weighted by $f_s \in \{0.05, 0.07, 0.10, 0.13, 0.15\}$.

**Fog**: For severity levels $s \in \{1, \ldots, 5\}$, generate a plasma fractal map $F$ of size $2^{\lceil \log_2(\max(H,W)) \rceil}$ via the diamond-square algorithm with decay parameter $d_s \in \{2, 2, 1.7, 1.5, 1.4\}$ for each $s$, normalize $F \in [0, 1]$ and crop to $(H, W)$, then for each selected frame normalized to $I \in [0, 1]$, compute

$$I_{\text{fog}} = \text{clip}\Big(\frac{I + c_s F}{1 + c_s}, 0, 1\Big) \times 255,$$

where $c_s \in \{1.5, 2.0, 2.5, 2.5, 3.0\}$, producing a depth-varying haze effect.

**Snow**. For severity level $s \in \{1, \ldots, 5\}$, the seven-tuple $(\mu_s, \sigma_s, r_s, t_s, R_s, \Sigma_s, \beta_s)$ is defined by the $s$th row of

$$\begin{pmatrix} 0.10 & 0.30 & 3.0 & 0.50 & 10 & 4 & 0.80 \\ 0.20 & 0.30 & 2.0 & 0.50 & 12 & 4 & 0.70 \\ 0.55 & 0.30 & 4.0 & 0.90 & 12 & 8 & 0.70 \\ 0.55 & 0.30 & 4.5 & 0.85 & 12 & 8 & 0.65 \\ 0.55 & 0.30 & 2.5 & 0.85 & 12 & 12 & 0.55 \end{pmatrix},$$

where columns correspond to $\mu_s, \sigma_s, r_s, t_s, R_s, \Sigma_s, \beta_s$. Generate a Gaussian snow layer $S \sim \mathcal{N}(\mu_s, \sigma_s^2)$, clip values below $t_s$, zoom by factor $r_s$, and apply a 1D motion blur with $K = 2\lceil \sigma_s \rceil + 1$ and $\rho = R_s$ at a random angle. Finally blend via

$$I_{\text{snow}} = \text{clip}\Big(\beta_s\, I + (1 - \beta_s) \max\big(I,\, 1.5\, \text{gray}(I) + 0.5\big) + S + \text{rot90}(S, 2),\, 0,\, 1\Big) \times 255.$$

**Frost**: For severity levels $s \in \{1, \ldots, 5\}$, let

$$(\lambda_s, \mu_s) \in \{(1, 0.4), (0.8, 0.6), (0.7, 0.7), (0.65, 0.7), (0.6, 0.75)\}$$

respectively; a frost texture $F$ is randomly selected, resized, and tiled to cover the frame, and the output is

$$I_{\text{frost}} = \text{clip}(\lambda_s I + \mu_s F, 0, 255),$$

blending original content with ice-crystal patterns.

**Specular reflection**: For each selected frame, compute a highlight mask by thresholding the mean intensity per pixel above 100, apply Gaussian blur in depth (kernel size 15, $\sigma = 5$) to the frame, blend original and blurred via $1.5 \times \text{orig} + 0.5 \times \text{blur}$ in highlight regions, then combine with a pregenerated Perlin noise layer via $0.9 \times \text{frame} + 0.1 \times \text{noise}$ and clamp to $[0, 255]$ to simulate mirror-like reflections.

**Shadow**: For each selected frame of size $H \times W$, sample

$$x_0 \sim \mathcal{U}\Big(\frac{W}{4}, \frac{3W}{4}\Big), \quad y_0 \sim \mathcal{U}\Big(\frac{H}{4}, \frac{3H}{4}\Big), \quad a \sim \mathcal{U}\Big(\frac{W}{4}, \frac{W}{2}\Big), \quad b \sim \mathcal{U}\Big(\frac{H}{4}, \frac{H}{2}\Big), \quad \theta \sim \mathcal{U}(0, 180°).$$

sample an elliptical mask $M$ defined by center $(x_0, y_0)$, axes $(a, b)$, and rotation $\theta$, blur $M$ with a Gaussian kernel of size $51 \times 51$ to obtain $\widetilde{M}$, and compute

$$I_{\text{shadow}} = \text{clip}\big(I\,(1 - 0.7\,\widetilde{M}),\, 0,\, 255\big),$$

producing soft, occluding shadows.

Since video models process multiple frames as input, we conducted a preliminary experiment to investigate how the proportion of noisy frames affects model performance. Specifically, we varied the percentage of frames corrupted by noise and observed that model accuracy decreased most significantly when noise was applied to 90% of the frames (see Figure 8). Based on this observation, and to better amplify the impact of noise, we set the severe noise level in our benchmark to correspond to corrupting 90% of the input frames.

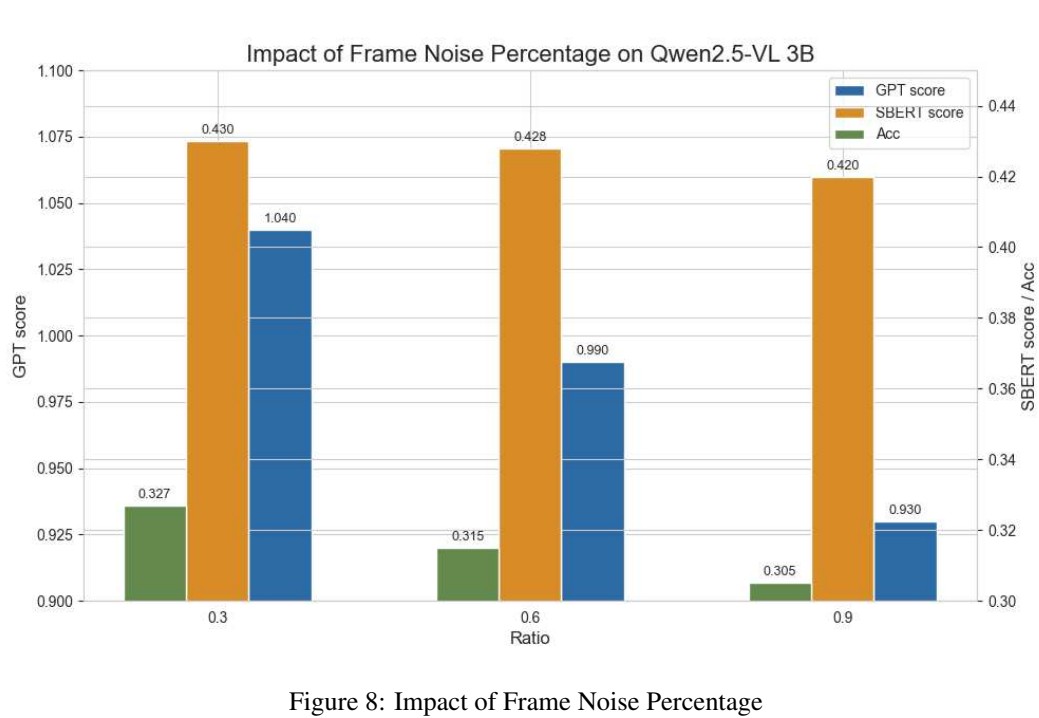

Figure 8: Impact of Frame Noise Percentage

## A.3 MODEL IMPLEMENTATION DETAILS

We evaluated a diverse set of state-of-the-art video-language models, each with distinct architectural designs and input image resolutions:

- **Video-LLaVA-7B**: Extends the LLaVA architecture to videos by processing multiple frames using a CLIP ViT-L/14 vision encoder and aligning visual features with a Vicuna language model, enabling strong zero-shot performance on video understanding tasks. [Code]

- **VideoChat2-HD**: Following the BLIP2 framework, this model utilizes a QFormer to bridge vision and language, supporting high-resolution video inputs for enhanced video-language understanding and dialogue. [Code]

- **Chat-UniVi-7B / 7B-v1.5**: Employs a unified visual representation with dynamic visual tokens, using a shared visual encoder and Vicuna LLM for both image and video tasks, thus achieving versatility across different visual modalities. [Code]

- **LLaMA-VID-7B**: Utilizes a token compression mechanism to efficiently process long video sequences, equipped with an EVA-G visual encoder and a Vicuna-7B language backbone for improved scalability and efficiency in video analysis. [Code]

- **Video-ChatGPT**: Incorporates both spatial and temporal pooling after CLIP-based visual encoding, followed by a Vicuna v1.1 language backbone, enabling robust video-language interaction and reasoning. [Code]

- **PLLaVA-7B / 13B**: A parameter-free extension of LLaVA that introduces temporal pooling on top of a CLIP vision encoder, designed to enhance the robustness and efficiency of video captioning without increasing model parameters. [Code]

- **Qwen2.5-VL-3B-Instruct**: Qwen's multimodal instruction-following model, equipped with a vision transformer encoder and a 3B-parameter language model, designed for strong performance on vision-language tasks. [Code]

- **GPT-4o**: The latest flagship multimodal model from OpenAI, capable of high-performance visual and language understanding across both images and videos, featuring a proprietary architecture and dynamic input resolution. (Closed-source)

All models were evaluated on 8 NVIDIA A100 GPUs with Intel(R) Xeon(R) Platinum 8358 CPUs. For fair comparison, we set the number of input frames to 8 for all models. Our benchmark also allows flexible frame number selection, enabling users to specify the frame count as needed.

## A.4 LIMITATIONS

We develop a benchmark to evaluate the robustness of video-language models (video-LLMs) against visual noise. The main limitations of our benchmark are as follows: it is challenging to obtain real-world noisy videos, so we simulate visual noise to approximate real-world conditions. Since we use GPT-4o as the judge for our benchmark, there are associated costs. Due to budget constraints, we selected only a few representative video-LLMs for comparison. Our benchmark includes a large number of noise types. Running the full benchmark is equivalent to running 36 visual clean benchmarks, which limits the duration and quantity of clean videos in our selected dataset. However, our codebase allows users to freely choose any clean dataset according to their needs and budget.

## A.5 DECLARATION OF LLM USAGE

Our benchmark follows previous video understanding benchmarks by using GPT-4o OpenAI (2024) as a judge to evaluate the performance of Video MLLMs when faced with visual noise. Through our experiments, we found that relying solely on GPT as a judge can introduce bias. Therefore, we designed our evaluation metrics to assess model performance from both subjective and objective perspectives. The details of metrics are listed in Section 2.3. Additionally, our also describe the Video MLLMs in Section 2.4 and Appendix A.3.

## A.6 MODEL FACTORS ON ROBUSTNESS OF VIDEO MLLMs

We observe that *the model architecture has strong connections with the robustness in video understanding*. We intensively include the spatio-temporal enhanced models (i.e., Chat-UniVi models and Video-ChatGPT). Generally, they achieve the smallest absolute delta GPT scores on average. This likely benefits from their architectural designs that explicitly decouple temporal and spatial information. In contrast, PLLaVA models perform the worst in specialized Video MLLMs. This may be due to PLLaVA employing adaptive pooling to synchronize the reduction of both temporal and spatial dimensions.

Based on Figure 3, we further compare the PLLaVA 13B (column D) with its 7B version (column J) and find out that *increasing the model size does not necessarily improve the robustness*. For many cases in these two columns, the performance drops in column D are larger than those in column J, suggesting that model size is not the key bottleneck towards robustness.

Comparing the data-enhanced version of Chat-UniVi-v1.5 (column G) with its vanilla version (i.e., column H of Chat-UniVi), we find that *training with more data does not contribute to higher robustness*. We conduct the Wilcoxon signed-rank test Wilcoxon (1992) (paired test) on the two sets of values and cannot conclude that there is a significant difference at the confidence level of 0.05.

## A.7 OTHER EXPERIMENT RESULTS

In this section, we provide the full experimental results corresponding to the analyses presented in Section 4 of the main text, specifically the "Analysis of Robustness of Video MLLMs across Video Types" and the "Analysis of Robustness of Video MLLMs across Question Types." Our evaluation covers 10 representative models; users interested in other models are encouraged to conduct additional tests as needed.

Tables 2–11 report robustness results across different video types, while Tables 12–21 summarize the results for different question types. Differing from the main text, which focuses on delta performance, the tables here display the raw output scores for each model. For clarity, the lowest value in each column—corresponding to the greatest performance drop under a particular noise type—is highlighted in gray.

We select five models and analyze each type of noise as shown in Figure 9–Figure 11. Across all three metrics (**Acc**, **SBERT Score**, and **GPT Score**), we observe that the impact of noise types on model

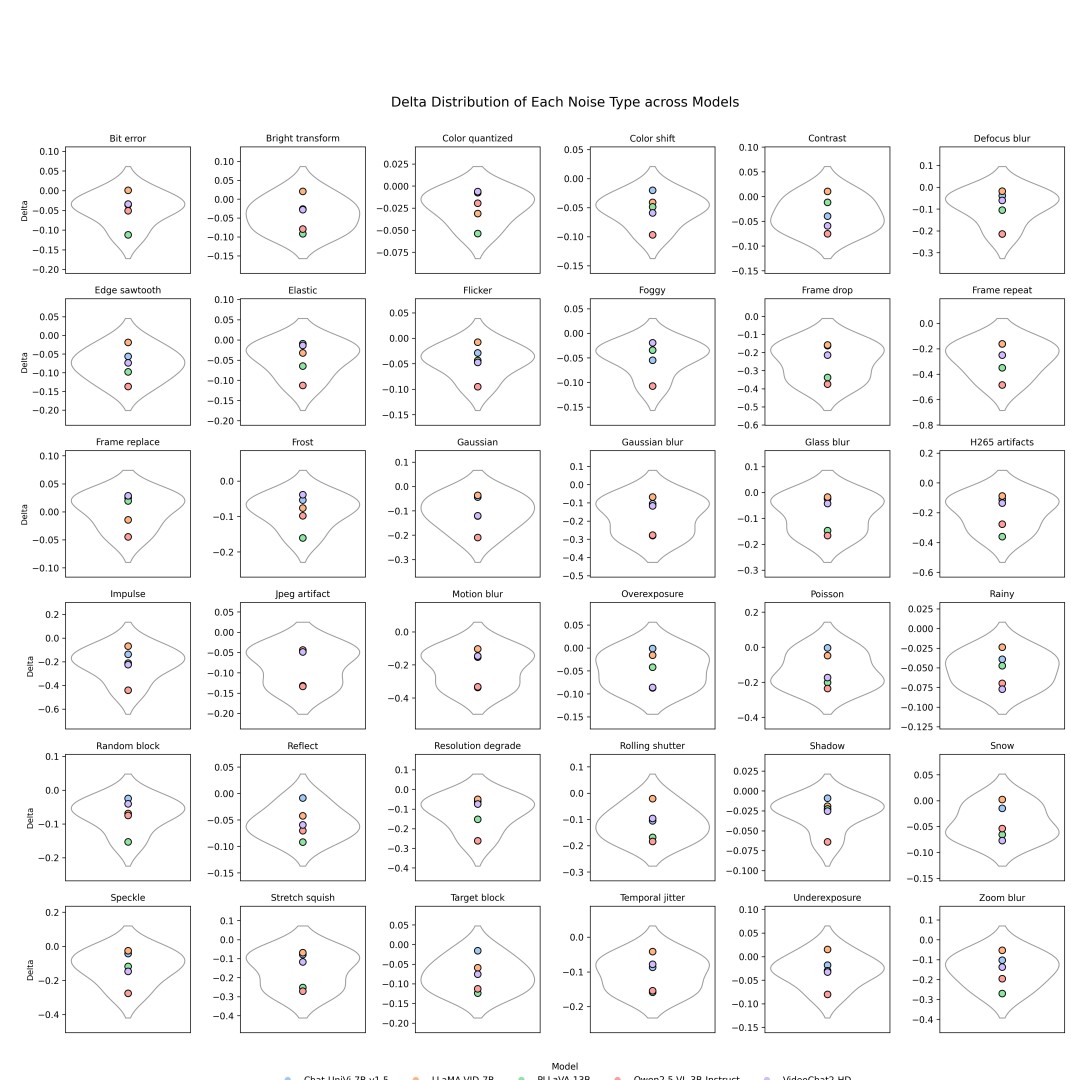

Figure 9: Delta Distribution of Each Noise Type across Models (GPT Score).

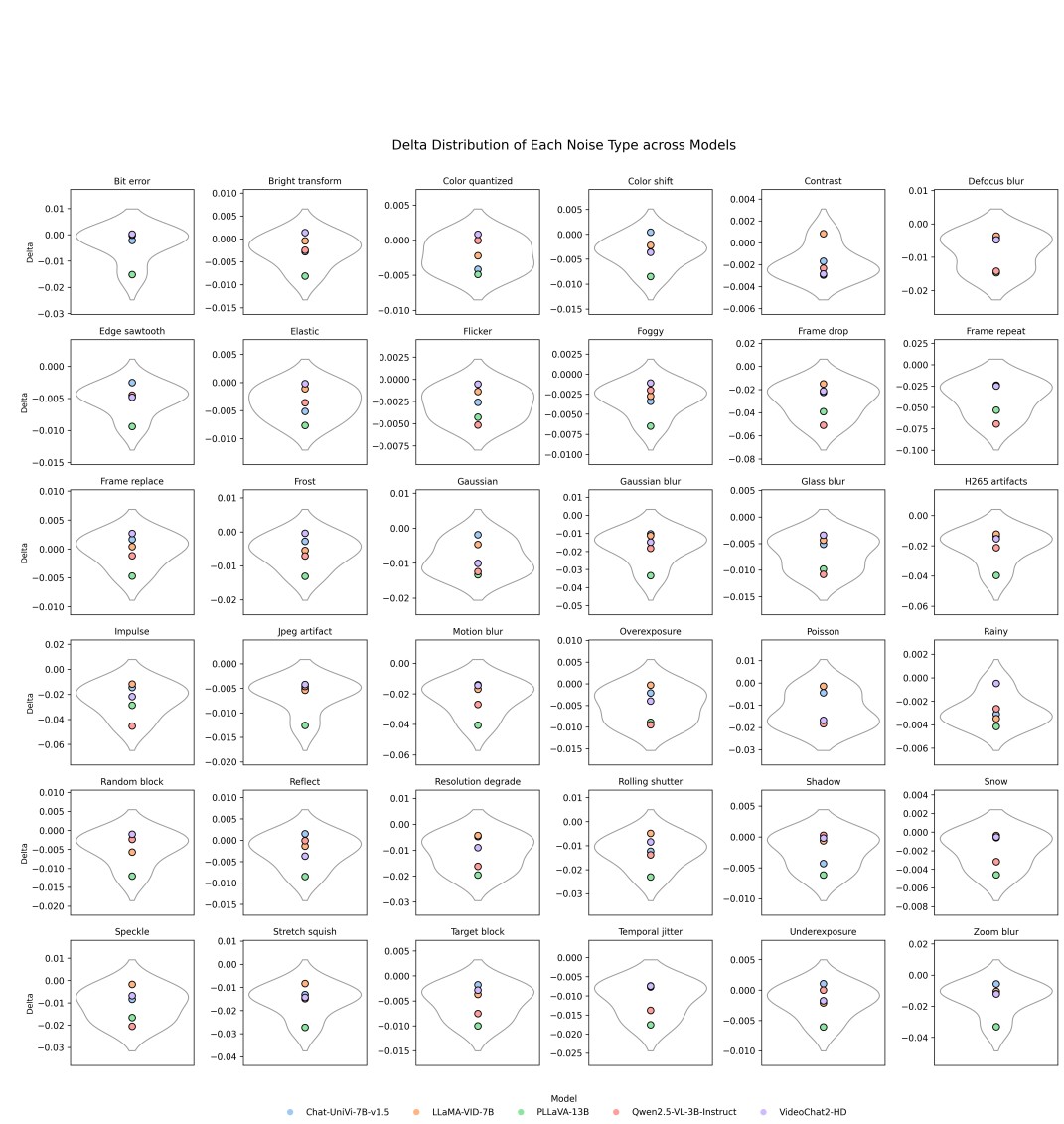

Figure 10: Delta Distribution of Each Noise Type across Models (SBERT Score).

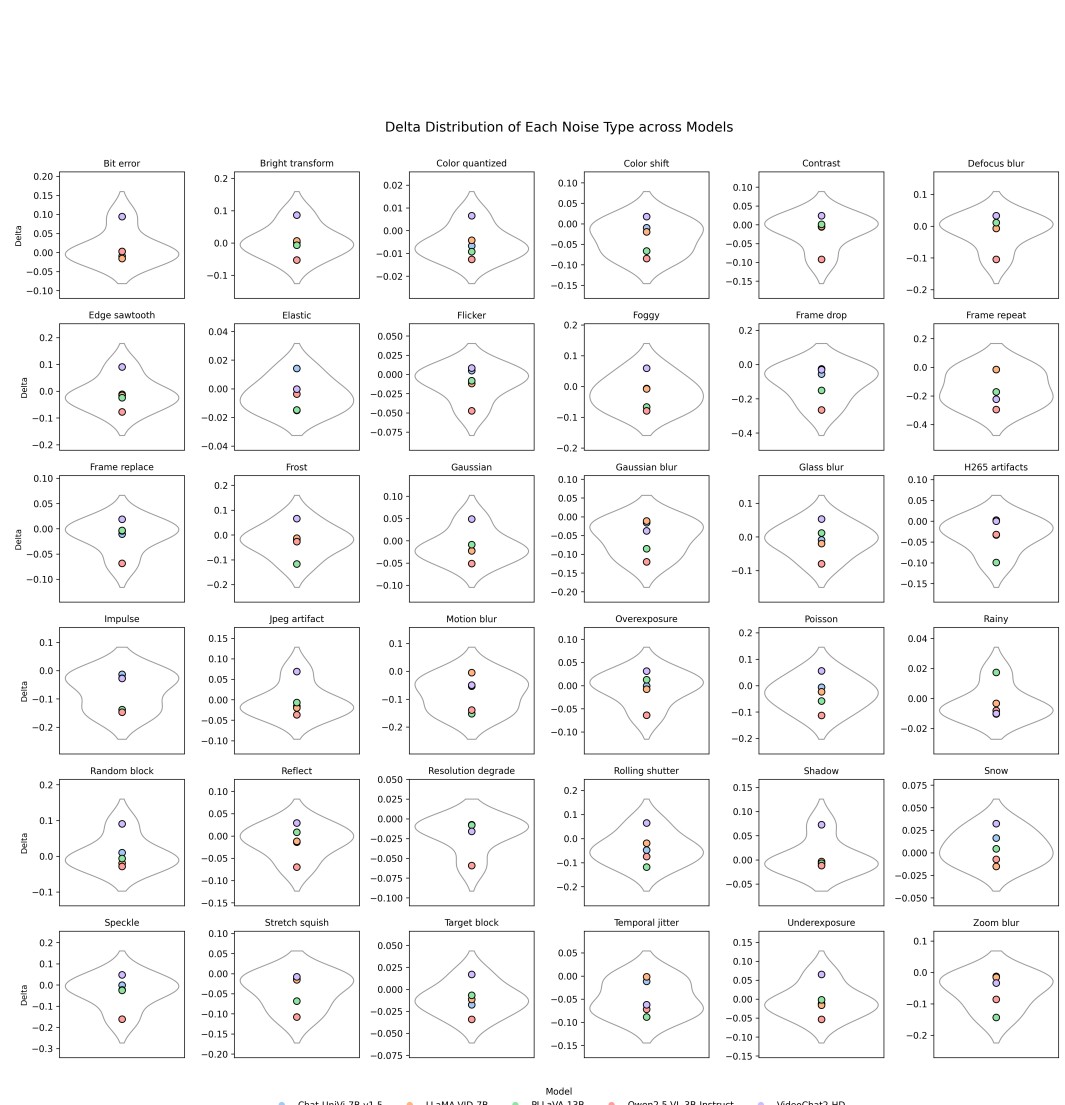

Figure 11: Delta Distribution of Each Noise Type across Models (Acc Score).

robustness is both metric- and model-dependent, revealing a number of nuanced behaviors. For blur-related corruptions such as *Gaussian blur* and *Motion blur*, all models suffer noticeable performance degradation, but the effect is most pronounced on the **GPT Score** metric, where PLLaVA-13B's delta can reach as low as $-0.35$, surpassing the drops observed in **Acc** (around $-0.15$). In contrast, *VideoChat2-HD* consistently demonstrates the strongest resilience, with deltas often 30% less negative than the next best model. This pattern suggests that while all models are sensitive to spatial blur, the severity is amplified under semantic or generative evaluation (**GPT Score**), and that *VideoChat2-HD* is inherently more robust to such distortions.

Temporal disruptions, such as *Frame drop* and *Frame repeat*, expose differences in temporal modeling capability. PLLaVA-13B is particularly vulnerable to these noises, as evidenced by its sharply lower delta values in both **Acc** and **GPT Score** metrics ($-0.20$ and $-0.30$, respectively), while other models, though affected, can better preserve their performance. This suggests that certain architectures may lack effective mechanisms for handling discontinuities or redundancies in the video stream.

For compression artifacts like *H265 artifacts*, the gap between models is striking. On the **GPT Score** metric, LLaMA-VID shows a dramatic delta drop, approaching $-0.65$, whereas *VideoChat2-HD* and *Qwen2.5-VL 3B* experience much milder degradation. This finding highlights a specific weakness of LLaMA-VID when dealing with heavy compression, possibly due to its handling of blocky or low-frequency corrupted frames.

Geometric distortions present another interesting case: under *Stretch squish* noise, PLLaVA 13B's performance sharply deteriorates, especially on the **GPT Score** metric (delta near $-0.40$), while other models remain relatively stable. This outlier behavior points to a lack of geometric invariance in PLLaVA 13B, which could be addressed by targeted data augmentation or architectural changes.

Some noise types, such as *Impulse* and *Jpeg artifact*, yield much smaller and more uniform delta values across all models and metrics, indicating that these models are well-adapted to cope with minor pixel-level corruptions. Meanwhile, for extreme exposure conditions (*Overexposure* and *Underexposure*), all models exhibit similar levels of performance decrease, suggesting that this is a common weakness among current architectures, regardless of their design or training.

### A.8 THE VISUALIZATION OF THE QA OVER DIFFERENT VISUAL NOISE

In this section, we present eight demonstration cases to illustrate the impact of different visual noises on model performance, as shown in Figure 12. For each noise category, we randomly selected one type of noise and applied it to a randomly chosen video. Both the original clean video and its noisy counterpart are shown for comparison. To simulate the typical process of frame extraction in video models, we uniformly sampled 8 frames from each video and applied noise to 90% of these frames. For consistency, all visualizations are generated using the Qwen2.5-VL 3B model.

These demonstrations clearly show that visual noise can severely degrade the model's ability to discern fine details. On clean frames, the model is able to accurately recognize objects and count the number of instances within the video. However, when noise is present, such as in the color shift and motion blur examples, the model frequently fails to distinguish between similar objects or to count them correctly. Notably, in cases where noise severely degrades information (e.g., with resolution degradation), the model may produce hallucinated or irrelevant answers due to the lack of useful visual cues.

### A.9 LICENSE OF THE USED ASSETS

MMbench-video is under CC BY-NC 4.0 license. Imagecorruption is under Apache-2.0 license.

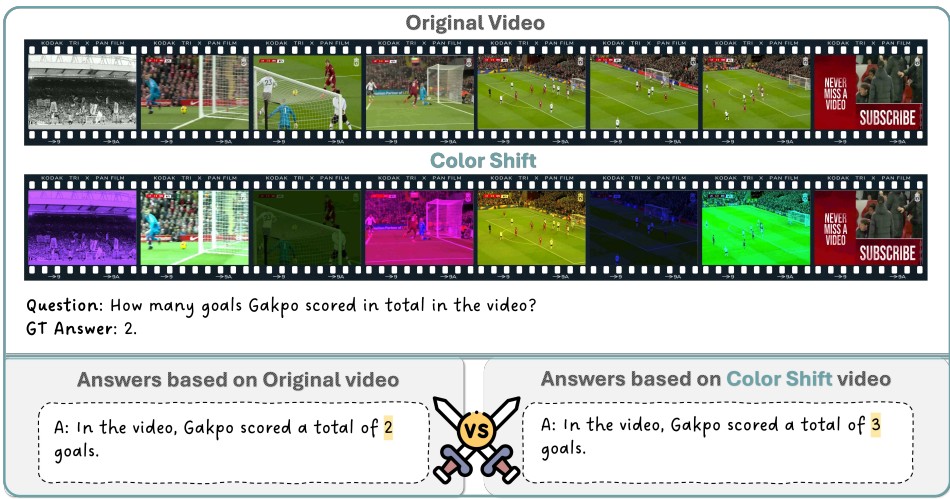

(a) The visualization of color shift noise

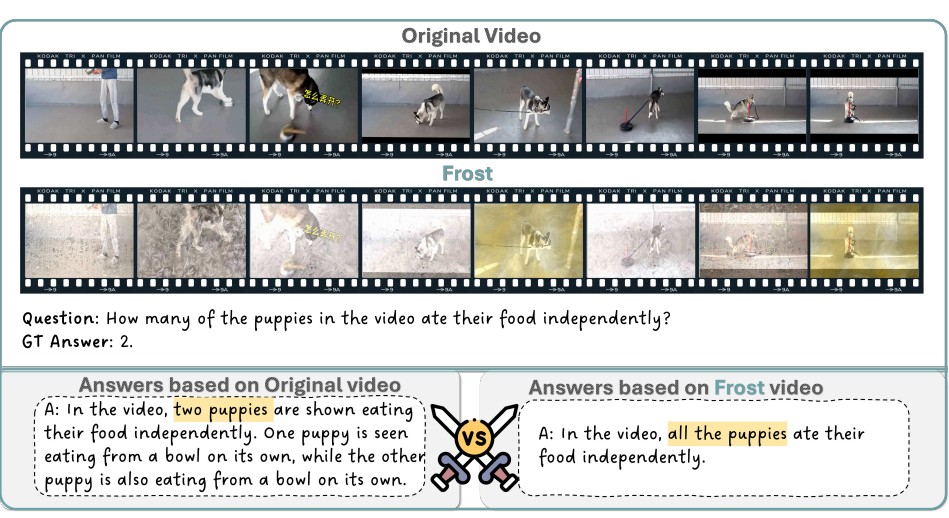

(b) The visualization of frost noise

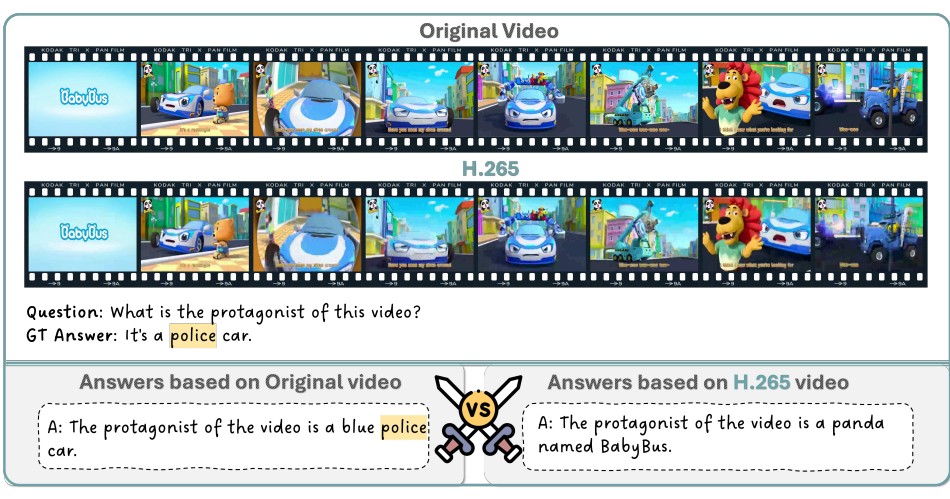

(c) The visualization of H.265 noise

Figure 12: Visualization of different types of noise (a)–(c)

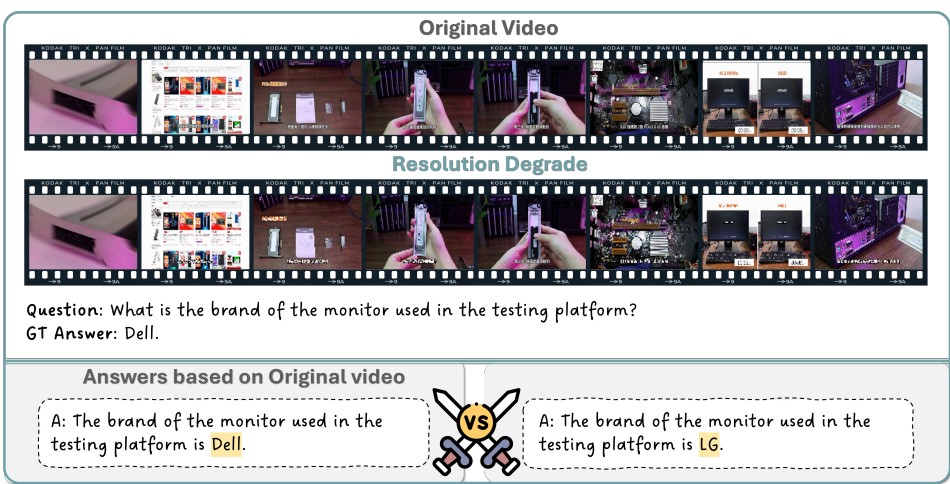

(d) The visualization of resolution degrade noise

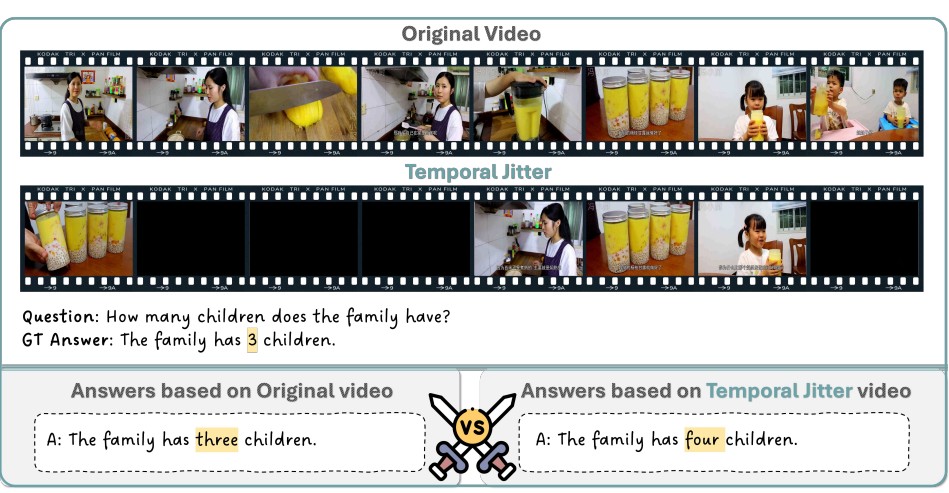

(e) The visualization of temporal jitter noise

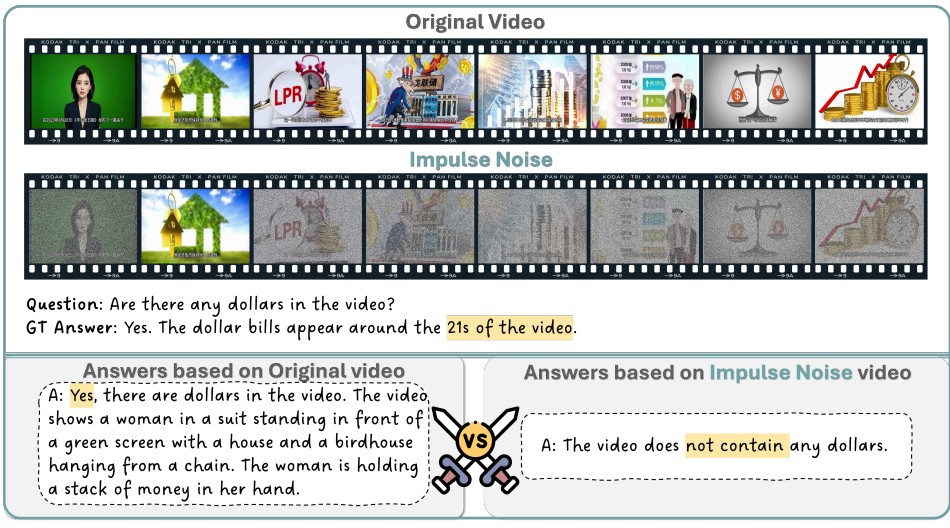

(f) The visualization of impluse noise

Figure 12: Visualization of different types of noise (d)–(f)

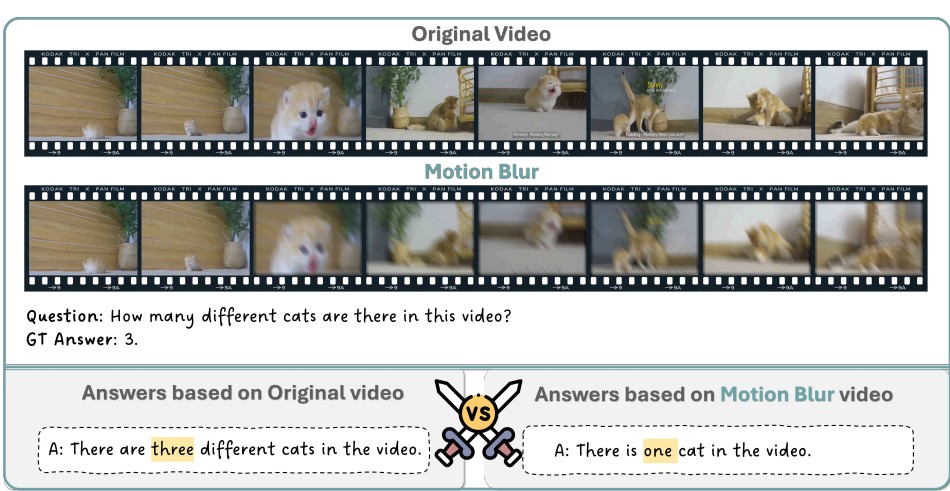

(g) The visualization of motion blur

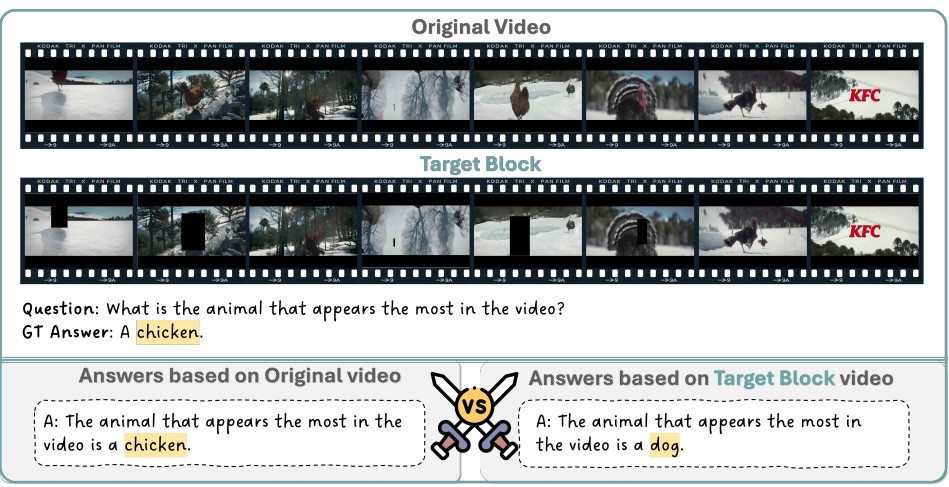

(h) The visualization of target block

Figure 12: Visualization of different types of noise (g)–(h)

Table 2: Chat-UniVi-v1.5 performance across different video types

| Noise Type | A GPT | A SBERT | A Acc | B GPT | B SBERT | B Acc | C GPT | C SBERT | C Acc | D GPT | D SBERT | D Acc | E GPT | E SBERT | E Acc | F GPT | F SBERT | F Acc | G GPT | G SBERT | G Acc | H GPT | H SBERT | H Acc | I GPT | I SBERT | I Acc | J GPT | J SBERT | J Acc |
|---|---|---|---|---|---|---|---|---|---|---|---|---|---|---|---|---|---|---|---|---|---|---|---|---|---|---|---|---|---|---|
| Clean | 0.976 | 0.416 | 0.167 | 1.000 | 0.357 | 1.000 | 1.143 | 0.423 | 0.846 | 0.827 | 0.346 | 0.278 | 0.827 | 0.344 | 0.500 | 1.250 | 0.450 | 0.500 | 0.847 | 0.484 | 0.344 | 0.938 | 0.462 | 0.344 | 1.147 | 0.524 | 0.812 | 0.811 | 0.340 | 1.000 |
| Gaussian | 0.959 | 0.413 | 0.167 | 0.875 | 0.355 | 1.000 | 1.170 | 0.434 | 0.846 | 0.827 | 0.345 | 0.222 | 0.712 | 0.338 | 0.500 | 1.000 | 0.421 | 0.500 | 0.634 | 0.479 | 0.250 | 1.038 | 0.471 | 0.281 | 1.039 | 0.523 | 0.688 | 0.700 | 0.324 | 1.000 |
| Impulse | 0.732 | 0.393 | 0.167 | 0.250 | 0.298 | 1.000 | 1.182 | 0.432 | 0.846 | 0.811 | 0.337 | 0.222 | 0.712 | 0.339 | 0.500 | 0.780 | 0.399 | 0.500 | 0.664 | 0.464 | 0.250 | 0.863 | 0.455 | 0.281 | 1.070 | 0.526 | 0.750 | 0.678 | 0.316 | 1.000 |
| Speckle | 0.764 | 0.393 | 0.167 | 0.750 | 0.306 | 1.000 | 1.170 | 0.422 | 0.923 | 0.755 | 0.343 | 0.278 | 0.654 | 0.339 | 0.500 | 0.882 | 0.415 | 0.500 | 0.677 | 0.474 | 0.250 | 1.000 | 0.457 | 0.312 | 1.039 | 0.518 | 0.750 | 0.711 | 0.326 | 1.000 |
| Poisson | 0.902 | 0.408 | 0.167 | 0.500 | 0.342 | 1.000 | 1.277 | 0.432 | 0.923 | 0.809 | 0.351 | 0.278 | 0.817 | 0.345 | 0.500 | 0.726 | 0.428 | 0.500 | 0.630 | 0.465 | 0.312 | 0.912 | 0.463 | 0.312 | 1.062 | 0.521 | 0.750 | 0.722 | 0.312 | 1.000 |
| Gaussian Blur | 0.683 | 0.383 | 0.167 | 0.625 | 0.348 | 1.000 | 1.125 | 0.428 | 0.846 | 0.691 | 0.353 | 0.278 | 0.865 | 0.345 | 0.500 | 0.769 | 0.451 | 1.000 | 0.604 | 0.458 | 0.312 | 0.925 | 0.469 | 0.312 | 1.047 | 0.520 | 0.688 | 0.767 | 0.306 | 1.000 |
| Motion Blur | 0.634 | 0.369 | 0.167 | 0.375 | 0.332 | 1.000 | 1.196 | 0.428 | 0.846 | 0.845 | 0.343 | 0.278 | 0.846 | 0.340 | 0.500 | 0.731 | 0.401 | 1.000 | 0.644 | 0.474 | 0.281 | 1.025 | 0.464 | 0.344 | 1.233 | 0.532 | 0.750 | 0.767 | 0.325 | 1.000 |
| Defocus Blur | 0.919 | 0.400 | 0.167 | 0.500 | 0.320 | 1.000 | 1.170 | 0.430 | 0.846 | 0.745 | 0.363 | 0.222 | 0.837 | 0.329 | 0.500 | 0.710 | 0.385 | 1.000 | 0.723 | 0.480 | 0.344 | 1.062 | 0.460 | 0.281 | 1.178 | 0.519 | 0.688 | 0.711 | 0.306 | 1.000 |
| Glass Blur | 0.943 | 0.418 | 0.167 | 0.375 | 0.341 | 1.000 | 1.134 | 0.427 | 0.923 | 0.873 | 0.354 | 0.278 | 0.760 | 0.344 | 0.000 | 0.726 | 0.343 | 0.500 | 0.668 | 0.470 | 0.281 | 0.800 | 0.464 | 0.312 | 1.016 | 0.529 | 0.750 | 0.622 | 0.328 | 1.000 |
| Zoom Blur | 0.724 | 0.391 | 0.167 | 0.500 | 0.391 | 1.000 | 1.196 | 0.430 | 0.846 | 0.718 | 0.363 | 0.333 | 0.731 | 0.338 | 0.000 | 0.923 | 0.385 | 0.500 | 0.668 | 0.465 | 0.344 | 0.825 | 0.460 | 0.312 | 0.977 | 0.531 | 0.750 | 0.578 | 0.301 | 1.000 |
| Frame Drop | 0.512 | 0.353 | 0.167 | 0.875 | 0.372 | 1.000 | 1.196 | 0.438 | 0.846 | 0.773 | 0.344 | 0.333 | 0.923 | 0.349 | 0.500 | 1.250 | 0.430 | 0.500 | 0.634 | 0.470 | 0.281 | 0.838 | 0.455 | 0.312 | 0.938 | 0.532 | 0.812 | 0.800 | 0.330 | 1.000 |
| Frame Replace | 0.919 | 0.419 | 0.167 | 0.333 | 0.276 | 1.000 | 1.054 | 0.414 | 0.846 | 0.791 | 0.341 | 0.222 | 0.798 | 0.338 | 0.500 | 1.000 | 0.343 | 0.500 | 0.906 | 0.467 | 0.281 | 0.863 | 0.460 | 0.312 | 1.023 | 0.532 | 0.625 | 0.544 | 0.312 | 1.000 |
| Frame Repeat | 0.618 | 0.355 | 0.167 | 0.625 | 0.325 | 1.000 | 1.161 | 0.430 | 0.846 | 0.727 | 0.342 | 0.278 | 0.644 | 0.334 | 0.500 | 0.250 | 0.357 | 0.500 | 0.757 | 0.475 | 0.312 | 0.938 | 0.461 | 0.312 | 1.062 | 0.523 | 0.750 | 0.678 | 0.329 | 1.000 |
| Temporal Jitter | 0.675 | 0.390 | 0.167 | 1.000 | 0.353 | 1.000 | 1.214 | 0.428 | 0.846 | 0.864 | 0.365 | 0.278 | 0.702 | 0.352 | 0.500 | 1.250 | 0.398 | 0.500 | 0.715 | 0.480 | 0.312 | 1.075 | 0.473 | 0.312 | 1.155 | 0.535 | 0.750 | 0.733 | 0.320 | 1.000 |
| Jpeg Artifact | 0.829 | 0.397 | 0.167 | 0.750 | 0.324 | 1.000 | 1.170 | 0.431 | 0.846 | 0.900 | 0.351 | 0.222 | 0.837 | 0.355 | 0.500 | 0.750 | 0.401 | 1.000 | 0.804 | 0.477 | 0.281 | 0.938 | 0.481 | 0.312 | 0.922 | 0.522 | 0.688 | 0.678 | 0.320 | 1.000 |
| Bit Error | 0.870 | 0.412 | 0.167 | 0.750 | 0.397 | 1.000 | 1.205 | 0.425 | 0.846 | 0.827 | 0.355 | 0.389 | 0.731 | 0.354 | 0.500 | 1.250 | 0.469 | 0.500 | 0.600 | 0.462 | 0.281 | 0.675 | 0.432 | 0.333 | 0.922 | 0.512 | 0.688 | 0.678 | 0.320 | 1.000 |
| H265 Artifacts | 0.772 | 0.397 | 0.167 | 0.250 | 0.308 | 1.000 | 1.161 | 0.417 | 0.846 | 0.827 | 0.352 | 0.222 | 0.865 | 0.330 | 0.500 | 1.000 | 0.422 | 0.500 | 0.783 | 0.481 | 0.281 | 1.012 | 0.468 | 0.281 | 1.016 | 0.528 | 0.750 | 0.744 | 0.325 | 1.000 |
| Random Block | 0.829 | 0.405 | 0.167 | 0.750 | 0.319 | 1.000 | 1.277 | 0.424 | 0.846 | 0.791 | 0.343 | 0.222 | 0.798 | 0.336 | 0.500 | 1.250 | 0.425 | 0.500 | 0.706 | 0.479 | 0.281 | 0.988 | 0.477 | 0.281 | 1.225 | 0.528 | 0.750 | 0.567 | 0.310 | 1.000 |
| Target Block | 0.789 | 0.401 | 0.167 | 1.000 | 0.342 | 1.000 | 1.205 | 0.440 | 0.846 | 0.773 | 0.356 | 0.333 | 0.817 | 0.343 | 0.000 | 1.000 | 0.432 | 0.500 | 0.732 | 0.483 | 0.312 | 0.875 | 0.458 | 0.312 | 1.085 | 0.528 | 0.750 | 0.733 | 0.330 | 1.000 |
| Rolling Shutter | 0.707 | 0.389 | 0.167 | 0.500 | 0.314 | 1.000 | 1.170 | 0.430 | 0.846 | 0.709 | 0.356 | 0.222 | 0.817 | 0.343 | 0.500 | 1.000 | 0.293 | 0.500 | 0.830 | 0.486 | 0.344 | 0.912 | 0.464 | 0.344 | 1.132 | 0.518 | 0.750 | 0.567 | 0.320 | 1.000 |
| Resolution Degrade | 0.821 | 0.405 | 0.167 | 0.125 | 0.357 | 1.000 | 1.205 | 0.427 | 0.846 | 0.745 | 0.352 | 0.333 | 0.731 | 0.351 | 0.500 | 1.000 | 0.410 | 0.500 | 0.855 | 0.480 | 0.344 | 0.963 | 0.464 | 0.250 | 0.984 | 0.518 | 0.750 | 0.700 | 0.320 | 1.000 |
| Stretch Squish | 0.756 | 0.385 | 0.167 | 0.250 | 0.327 | 1.000 | 1.241 | 0.430 | 0.846 | 0.809 | 0.351 | 0.222 | 0.760 | 0.347 | 0.500 | 1.000 | 0.457 | 0.500 | 0.800 | 0.481 | 0.250 | 1.025 | 0.457 | 0.312 | 1.116 | 0.525 | 0.750 | 0.678 | 0.312 | 1.000 |
| Edge Sawtooth | 0.886 | 0.418 | 0.167 | 0.875 | 0.332 | 1.000 | 1.134 | 0.430 | 0.846 | 0.827 | 0.359 | 0.333 | 0.788 | 0.347 | 0.500 | 1.000 | 0.413 | 0.500 | 0.757 | 0.481 | 0.312 | 1.147 | 0.461 | 0.312 | 1.209 | 0.522 | 0.750 | 0.656 | 0.321 | 1.000 |
| Elastic | 0.927 | 0.412 | 0.333 | 0.625 | 0.340 | 1.000 | 1.125 | 0.430 | 0.846 | 0.864 | 0.362 | 0.333 | 0.846 | 0.357 | 0.500 | 1.000 | 0.424 | 0.500 | 0.817 | 0.474 | 0.312 | 1.109 | 0.470 | 0.312 | 1.109 | 0.539 | 0.750 | 0.767 | 0.326 | 1.000 |
| Color Quantized | 0.951 | 0.413 | 0.333 | 0.500 | 0.335 | 1.000 | 1.080 | 0.433 | 0.846 | 0.809 | 0.359 | 0.222 | 0.875 | 0.357 | 0.500 | 1.250 | 0.438 | 0.500 | 0.830 | 0.470 | 0.250 | 1.038 | 0.468 | 0.281 | 1.047 | 0.524 | 0.812 | 0.656 | 0.329 | 1.000 |
| Bright Transform | 0.878 | 0.404 | 0.333 | 1.000 | 0.364 | 1.000 | 1.214 | 0.429 | 0.846 | 0.845 | 0.359 | 0.333 | 0.712 | 0.342 | 0.500 | 1.000 | 0.410 | 0.500 | 0.787 | 0.482 | 0.281 | 0.988 | 0.454 | 0.312 | 0.922 | 0.523 | 0.750 | 0.656 | 0.322 | 1.000 |
| Contrast | 0.829 | 0.417 | 0.167 | 0.750 | 0.335 | 1.000 | 1.045 | 0.433 | 0.846 | 0.782 | 0.359 | 0.333 | 0.798 | 0.352 | 0.500 | 0.750 | 0.363 | 0.500 | 0.826 | 0.489 | 0.312 | 0.975 | 0.483 | 0.312 | 1.054 | 0.533 | 0.812 | 0.633 | 0.317 | 1.000 |
| Color Shift | 0.902 | 0.415 | 0.167 | 0.625 | 0.331 | 1.000 | 0.911 | 0.429 | 0.846 | 0.791 | 0.355 | 0.222 | 0.692 | 0.352 | 0.000 | 1.000 | 0.436 | 0.500 | 0.860 | 0.489 | 0.375 | 0.938 | 0.456 | 0.312 | 1.085 | 0.537 | 0.750 | 0.633 | 0.313 | 1.000 |
| Flicker | 0.870 | 0.403 | 0.167 | 1.000 | 0.344 | 1.000 | 1.098 | 0.427 | 0.846 | 0.727 | 0.348 | 0.278 | 0.750 | 0.346 | 0.500 | 0.750 | 0.467 | 0.500 | 0.715 | 0.475 | 0.312 | 1.050 | 0.471 | 0.281 | 1.109 | 0.538 | 0.750 | 0.778 | 0.319 | 1.000 |
| Overexposure | 0.919 | 0.403 | 0.167 | 0.750 | 0.355 | 1.000 | 1.134 | 0.424 | 0.846 | 0.736 | 0.344 | 0.222 | 0.817 | 0.345 | 0.500 | 1.750 | 0.387 | 0.500 | 0.804 | 0.485 | 0.281 | 0.875 | 0.461 | 0.312 | 1.016 | 0.529 | 0.750 | 0.611 | 0.314 | 1.000 |
| Underexposure | 0.894 | 0.410 | 0.167 | 0.250 | 0.314 | 0.167 | 1.064 | 0.430 | 0.846 | 0.773 | 0.353 | 0.222 | 0.817 | 0.347 | 0.000 | 1.000 | 0.410 | 0.500 | 0.834 | 0.481 | 0.312 | 0.963 | 0.457 | 0.312 | 1.147 | 0.522 | 0.750 | 0.656 | 0.326 | 1.000 |
| Rainy | 0.919 | 0.419 | 0.167 | — | — | — | 1.064 | 0.430 | 0.846 | 0.809 | 0.362 | 0.333 | 0.846 | 0.357 | 0.500 | 1.000 | 0.413 | 0.500 | 0.817 | 0.490 | 0.312 | 1.025 | 0.470 | 0.312 | 1.209 | 0.524 | 0.812 | 0.767 | 0.327 | 1.000 |
| Foggy | 0.911 | 0.414 | 0.167 | — | — | — | 1.080 | 0.433 | 0.846 | 0.845 | 0.359 | 0.333 | 0.875 | 0.357 | 0.500 | 1.000 | 0.424 | 0.500 | 0.740 | 0.474 | 0.312 | 1.038 | 0.468 | 0.250 | 1.140 | 0.532 | 0.812 | 0.733 | 0.322 | 1.000 |
| Snow | 0.870 | 0.417 | 0.333 | — | — | — | 1.214 | 0.433 | 0.846 | 0.782 | 0.359 | 0.222 | 0.712 | 0.342 | 0.500 | 0.750 | 0.410 | 0.500 | 0.787 | 0.482 | 0.281 | 0.988 | 0.454 | 0.312 | 0.922 | 0.523 | 0.750 | 0.656 | 0.317 | 1.000 |
| Frost | 0.797 | 0.398 | 0.167 | — | — | — | 1.098 | 0.429 | 0.846 | 0.791 | 0.348 | 0.278 | 0.798 | 0.352 | 0.500 | 0.750 | 0.436 | 0.500 | 0.826 | 0.489 | 0.375 | 0.975 | 0.456 | 0.312 | 1.054 | 0.533 | 0.812 | 0.633 | 0.317 | 1.000 |
| Reflect | 0.927 | 0.410 | 0.167 | — | — | — | 1.134 | 0.424 | 0.846 | 0.727 | 0.348 | 0.278 | 0.750 | 0.346 | 0.000 | 1.250 | 0.467 | 0.500 | 0.715 | 0.475 | 0.312 | 1.050 | 0.471 | 0.281 | 1.109 | 0.538 | 0.750 | 0.778 | 0.319 | 1.000 |
| Shadow | 0.837 | 0.419 | 0.167 | — | — | — | 1.064 | 0.442 | 0.846 | 0.773 | 0.353 | 0.222 | 0.817 | 0.347 | 0.500 | 1.000 | 0.387 | 0.500 | 0.804 | 0.485 | 0.281 | 0.875 | 0.461 | 0.312 | 1.062 | 0.530 | 0.812 | 0.611 | 0.314 | 1.000 |

| Noise Type | K GPT | K SBERT | K Acc | L GPT | L SBERT | L Acc | M GPT | M SBERT | M Acc | N GPT | N SBERT | N Acc | O GPT | O SBERT | O Acc | Q GPT | Q SBERT | Q Acc | R GPT | R SBERT | R Acc | S GPT | S SBERT | S Acc | T GPT | T SBERT | T Acc |
|---|---|---|---|---|---|---|---|---|---|---|---|---|---|---|---|---|---|---|---|---|---|---|---|---|---|---|---|
| Clean | 0.453 | 0.333 | 0.667 | 1.364 | 0.395 | 1.000 | 0.721 | 0.506 | 1.000 | 0.758 | 0.321 | 0.600 | 0.538 | 0.342 | 0.000 | 1.220 | 0.412 | 0.263 | 0.741 | 0.544 | 0.461 | 1.311 | 0.475 | 0.571 | 0.897 | 0.430 | 0.461 |
| Gaussian | 0.511 | 0.346 | 0.667 | 1.273 | 0.399 | 1.000 | 0.782 | 0.511 | 1.000 | 0.811 | 0.321 | 0.600 | 0.615 | 0.340 | 0.158 | 1.132 | 0.405 | 0.158 | 0.741 | 0.549 | 0.400 | 1.349 | 0.459 | 0.571 | 0.860 | 0.430 | 0.408 |
| Impulse | 0.431 | 0.330 | 0.667 | 1.182 | 0.398 | 1.000 | 0.680 | 0.500 | 1.000 | 0.663 | 0.305 | 0.600 | 0.538 | 0.315 | 0.000 | 0.956 | 0.402 | 0.263 | 0.636 | 0.540 | 0.400 | 1.179 | 0.461 | 0.571 | 0.782 | 0.420 | 0.428 |
| Speckle | 0.438 | 0.337 | 0.667 | 1.227 | 0.399 | 1.000 | 0.728 | 0.512 | 1.000 | 0.663 | 0.311 | 0.600 | 1.231 | 0.363 | 0.158 | 1.066 | 0.406 | 0.395 | 0.728 | 0.540 | 0.400 | 1.425 | 0.468 | 0.571 | 0.841 | 0.423 | 0.447 |
| Poisson | 0.504 | 0.344 | 0.667 | 0.242 | 0.411 | 1.000 | 0.776 | 0.502 | 1.000 | 0.853 | 0.322 | 0.600 | 0.692 | 0.326 | 0.158 | 0.967 | 0.406 | 0.263 | 0.667 | 0.536 | 0.400 | 1.198 | 0.468 | 0.571 | 0.875 | 0.419 | 0.428 |
| Gaussian Blur | 0.336 | 0.320 | 0.667 | 0.227 | 0.396 | 1.000 | 0.619 | 0.497 | 1.000 | 0.642 | 0.314 | 0.600 | 0.769 | 0.309 | 0.105 | 0.934 | 0.399 | 0.105 | 0.660 | 0.536 | 0.400 | 0.453 | 0.468 | 0.571 | 0.783 | 0.416 | 0.428 |
| Motion Blur | 0.606 | 0.351 | 0.667 | 1.076 | 0.384 | 1.000 | 0.605 | 0.505 | 1.000 | 0.716 | 0.314 | 0.600 | 0.462 | 0.334 | 0.211 | 1.066 | 0.410 | 0.158 | 0.784 | 0.542 | 0.400 | 1.245 | 0.455 | 0.571 | 0.776 | 0.427 | 0.441 |
| Defocus Blur | 0.569 | 0.343 | 0.667 | 1.258 | 0.409 | 1.000 | 0.741 | 0.505 | 1.000 | 0.842 | 0.328 | 0.600 | 0.692 | 0.334 | 0.158 | 1.000 | 0.407 | 0.211 | 0.660 | 0.542 | 0.400 | 1.189 | 0.465 | 0.571 | 0.873 | 0.415 | 0.401 |
| Glass Blur | 0.496 | 0.341 | 0.667 | 0.470 | 0.411 | 1.000 | 0.741 | 0.500 | 1.000 | 0.653 | 0.321 | 0.600 | 0.923 | 0.335 | 0.158 | 0.879 | 0.390 | 0.158 | 0.698 | 0.545 | 0.400 | 1.170 | 0.469 | 0.571 | 0.871 | 0.424 | 0.441 |
| Zoom Blur | 0.350 | 0.321 | 0.667 | 0.197 | 0.397 | 1.000 | 0.667 | 0.505 | 1.000 | 0.747 | 0.306 | 0.600 | 0.923 | 0.357 | 0.158 | 0.868 | 0.390 | 0.263 | 0.784 | 0.539 | 0.400 | 1.321 | 0.447 | 0.571 | 0.791 | 0.424 | 0.434 |
| Frame Drop | 0.584 | 0.350 | 0.667 | 1.545 | 0.381 | 1.000 | 0.707 | 0.508 | 1.000 | 0.779 | 0.329 | 0.600 | 0.769 | 0.325 | 0.263 | 1.209 | 0.409 | 0.158 | 0.784 | 0.552 | 0.400 | 0.934 | 0.469 | 0.571 | 0.921 | 0.432 | 0.434 |
| Frame Replace | 0.401 | 0.321 | 0.667 | 1.091 | 0.370 | 1.000 | 0.844 | 0.508 | 1.000 | 0.611 | 0.296 | 0.600 | 0.846 | 0.367 | 0.158 | 1.055 | 0.388 | 0.158 | 0.630 | 0.519 | 0.429 | 0.243 | 0.463 | 0.571 | 0.753 | 0.411 | 0.421 |
| Frame Repeat | 0.482 | 0.347 | 0.667 | 1.485 | 0.396 | 1.000 | 0.707 | 0.503 | 1.000 | 0.737 | 0.314 | 0.600 | 0.615 | 0.323 | 0.158 | 1.088 | 0.414 | 0.211 | 0.735 | 0.549 | 0.429 | 0.255 | 0.472 | 0.571 | 0.824 | 0.425 | 0.434 |
| Temporal Jitter | 0.496 | 0.347 | 0.667 | 1.273 | 0.394 | 1.000 | 0.714 | 0.509 | 1.000 | 0.695 | 0.319 | 0.600 | 0.538 | 0.347 | 0.211 | 1.044 | 0.403 | 0.158 | 0.667 | 0.541 | 0.400 | 1.330 | 0.466 | 0.571 | 0.839 | 0.428 | 0.428 |
| Jpeg Artifact | 0.547 | 0.348 | 0.667 | 0.379 | 0.384 | 1.000 | 0.680 | 0.509 | 1.000 | 0.768 | 0.321 | 0.600 | 0.692 | 0.347 | 0.105 | 1.110 | 0.389 | 0.158 | 0.716 | 0.540 | 0.400 | 1.123 | 0.456 | 0.571 | 0.881 | 0.431 | 0.434 |
| Bit Error | 0.489 | 0.345 | 0.667 | 0.152 | 0.393 | 1.000 | 0.653 | 0.502 | 1.000 | 0.800 | 0.323 | 0.600 | 0.846 | 0.366 | 0.158 | 0.989 | 0.391 | 0.158 | 0.722 | 0.535 | 0.400 | 1.283 | 0.455 | 0.571 | 0.763 | 0.428 | 0.408 |
| H265 Artifacts | 0.511 | 0.336 | 0.667 | 1.364 | 0.406 | 1.000 | 0.707 | 0.502 | 1.000 | 0.758 | 0.324 | 0.600 | 0.692 | 0.348 | 0.158 | 1.055 | 0.410 | 0.211 | 0.698 | 0.543 | 0.400 | 1.179 | 0.465 | 0.571 | 0.865 | 0.431 | 0.461 |
| Random Block | 0.489 | 0.350 | 0.667 | 0.379 | 0.418 | 1.000 | 0.816 | 0.510 | 1.000 | 0.832 | 0.322 | 0.600 | 0.923 | 0.366 | 0.158 | 1.121 | 0.391 | 0.211 | 0.654 | 0.542 | 0.400 | 1.208 | 0.452 | 0.571 | 0.868 | 0.429 | 0.441 |
| Target Block | 0.467 | 0.338 | 0.667 | 1.409 | 0.403 | 1.000 | 0.680 | 0.510 | 1.000 | 0.821 | 0.317 | 0.600 | 0.615 | 0.345 | 0.211 | 1.121 | 0.402 | 0.211 | 0.735 | 0.550 | 0.400 | 1.274 | 0.465 | 0.571 | 0.865 | 0.429 | 0.447 |
| Rolling Shutter | 0.511 | 0.338 | 0.667 | 1.424 | 0.411 | 1.000 | 0.701 | 0.499 | 1.000 | 0.905 | 0.325 | 0.600 | 0.462 | 0.354 | 0.211 | 1.077 | 0.409 | 0.158 | 0.728 | 0.545 | 0.400 | 1.292 | 0.461 | 0.571 | 0.888 | 0.432 | 0.441 |
| Resolution Degrade | 0.460 | 0.332 | 0.667 | 1.273 | 0.394 | 1.000 | 0.741 | 0.508 | 1.000 | 0.768 | 0.330 | 0.600 | 0.769 | 0.340 | 0.158 | 1.154 | 0.410 | 0.158 | 0.802 | 0.543 | 0.400 | 1.283 | 0.470 | 0.571 | 0.893 | 0.432 | 0.467 |
| Stretch Squish | 0.511 | 0.354 | 0.667 | 0.530 | 0.427 | 1.000 | 0.707 | 0.507 | 1.000 | 0.905 | 0.323 | 0.600 | 0.923 | 0.331 | 0.158 | 1.176 | 0.408 | 0.158 | 0.685 | 0.543 | 0.400 | 1.217 | 0.470 | 0.571 | 0.860 | 0.430 | 0.428 |
| Edge Sawtooth | 0.672 | 0.354 | 0.667 | 0.318 | 0.402 | 1.000 | 0.714 | 0.518 | 1.000 | 0.863 | 0.323 | 0.600 | 0.846 | 0.350 | 0.158 | 1.011 | 0.397 | 0.158 | 0.673 | 0.540 | 0.400 | 0.387 | 0.473 | 0.571 | 0.861 | 0.433 | 0.454 |
| Elastic | 0.569 | 0.344 | 0.667 | 0.303 | 0.402 | 1.000 | 0.721 | 0.514 | 1.000 | 0.811 | 0.214 | 0.600 | 0.846 | 0.350 | 0.211 | 1.088 | 0.392 | 0.158 | 0.722 | 0.547 | 0.400 | 1.217 | 0.454 | 0.571 | 0.873 | 0.433 | 0.454 |
| Color Quantized | 0.489 | 0.336 | 0.667 | 0.439 | 0.406 | 1.000 | 0.823 | 0.508 | 1.000 | 0.821 | 0.324 | 0.600 | 0.846 | 0.364 | 0.158 | 0.923 | 0.410 | 0.211 | 0.648 | 0.543 | 0.400 | 1.321 | 0.431 | 0.571 | 0.837 | 0.425 | 0.474 |
| Bright Transform | 0.453 | 0.339 | 0.667 | 0.227 | 0.382 | 1.000 | 0.755 | 0.510 | 1.000 | 0.800 | 0.324 | 0.600 | 0.615 | 0.348 | 0.158 | 1.121 | 0.402 | 0.211 | 0.636 | 0.550 | 0.400 | 1.330 | 0.462 | 0.571 | 0.862 | 0.423 | 0.408 |
| Contrast | 0.526 | 0.347 | 0.667 | 1.333 | 0.400 | 1.000 | 0.741 | 0.512 | 1.000 | 0.874 | 0.326 | 0.600 | 1.154 | 0.352 | 0.158 | 0.967 | 0.403 | 0.158 | 0.741 | 0.543 | 0.571 | 1.255 | 0.464 | 0.571 | 0.860 | 0.431 | 0.434 |

Note: A: Advertisements, B: Algorithm & Models, C: Autos & Vehicles, D: Business & Industrial, E: Computers & Electronics, F: Fairy Tale, G: Films & TV Shows, H: Finance, I: Food & Drink, J: Games, K: Humor, L: Instruction Video (how to), M: Knowledge, N: News, O: Others, P: People, Q: Pets & Animals, R: Science, S: Sports, T: Overall

Table 3: Chat-UniVi performance across different video types

| Noise Type | A GPT | A SBERT | A Acc | B GPT | B SBERT | B Acc | C GPT | C SBERT | C Acc | D GPT | D SBERT | D Acc | E GPT | E SBERT | E Acc |
|---|---|---|---|---|---|---|---|---|---|---|---|---|---|---|---|
| Clean | 0.894 | 0.403 | 0.500 | 0.625 | 0.295 | – | 0.964 | 0.429 | 0.769 | 0.800 | 0.348 | 0.389 | 0.827 | 0.342 | 0.000 |
| Gaussian | 0.780 | 0.399 | 0.333 | 0.625 | 0.308 | – | 0.866 | 0.427 | 0.769 | 0.918 | 0.349 | 0.556 | 0.788 | 0.349 | 0.500 |
| Impulse | 0.707 | 0.386 | 0.167 | 0.375 | 0.303 | – | 0.875 | 0.425 | 0.769 | 0.727 | 0.333 | 0.389 | 0.837 | 0.347 | 0.500 |
| Speckle | 0.846 | 0.397 | 0.167 | 0.500 | 0.278 | – | 1.018 | 0.438 | 0.769 | 0.936 | 0.358 | 0.556 | 0.817 | 0.344 | 0.500 |
| Poisson | 0.780 | 0.394 | 0.167 | 0.750 | 0.278 | – | 0.955 | 0.437 | 0.769 | 0.745 | 0.344 | 0.611 | 0.808 | 0.350 | 0.500 |
| Gaussian Blur | 0.650 | 0.381 | 0.333 | 0.625 | 0.287 | – | 0.964 | 0.431 | 0.769 | 0.700 | 0.338 | 0.389 | 0.760 | 0.345 | 0.500 |
| Motion Blur | 0.724 | 0.380 | 0.333 | 0.500 | 0.278 | – | 0.973 | 0.428 | 0.846 | 0.755 | 0.338 | 0.444 | 0.875 | 0.351 | 0.500 |
| Defocus Blur | 0.894 | 0.401 | 0.167 | 0.375 | 0.285 | – | 1.062 | 0.438 | 0.769 | 0.745 | 0.342 | 0.389 | 0.769 | 0.337 | 0.500 |
| Glass Blur | 0.829 | 0.405 | 0.333 | 0.750 | 0.282 | – | 1.000 | 0.439 | 0.769 | 0.873 | 0.355 | 0.611 | 0.731 | 0.346 | 0.500 |
| Zoom Blur | 0.862 | 0.400 | 0.333 | 0.750 | 0.303 | – | 1.062 | 0.439 | 0.769 | 0.727 | 0.343 | 0.333 | 0.808 | 0.346 | 0.000 |
| Frame Drop | 0.618 | 0.381 | 0.167 | 0.375 | 0.326 | – | 1.009 | 0.432 | 0.769 | 0.800 | 0.342 | 0.556 | 0.740 | 0.343 | 0.000 |
| Frame Replace | 0.813 | 0.403 | 0.333 | 0.750 | 0.321 | – | 1.089 | 0.443 | 0.769 | 0.945 | 0.359 | 0.556 | 0.760 | 0.347 | 0.000 |
| Frame Repeat | 0.488 | 0.349 | 0.333 | 0.375 | 0.278 | – | 0.812 | 0.416 | 0.692 | 0.627 | 0.329 | 0.500 | 0.760 | 0.335 | 0.000 |
| Temporal Jitter | 0.748 | 0.391 | 0.500 | 0.375 | 0.316 | – | 0.893 | 0.441 | 0.769 | 0.727 | 0.339 | 0.333 | 0.808 | 0.351 | 0.500 |
| Jpeg Artifact | 0.724 | 0.399 | 0.167 | 0.625 | 0.307 | – | 0.982 | 0.433 | 0.769 | 0.782 | 0.343 | 0.556 | 0.750 | 0.345 | 0.500 |
| Bit Error | 0.829 | 0.398 | 0.167 | 0.875 | 0.298 | – | 1.018 | 0.440 | 0.769 | 0.827 | 0.345 | 0.444 | 0.635 | 0.341 | 0.000 |
| H265 Artifacts | 0.699 | 0.378 | 0.167 | 0.750 | 0.291 | – | 0.857 | 0.426 | 0.769 | 0.827 | 0.350 | 0.444 | 0.702 | 0.341 | 0.500 |
| Random Block | 0.724 | 0.388 | 0.167 | 0.500 | 0.296 | – | 0.911 | 0.434 | 0.769 | 0.682 | 0.330 | 0.333 | 0.779 | 0.349 | 0.500 |
| Target Block | 0.642 | 0.388 | 0.167 | 1.000 | 0.339 | – | 0.955 | 0.439 | 0.769 | 0.791 | 0.343 | 0.500 | 0.875 | 0.351 | 0.000 |
| Rolling Shutter | 0.683 | 0.370 | 0.500 | 0.375 | 0.269 | – | 0.902 | 0.432 | 0.769 | 0.791 | 0.346 | 0.500 | 0.760 | 0.336 | 0.000 |
| Resolution Degrade | 0.829 | 0.395 | 0.333 | 0.250 | 0.314 | – | 1.009 | 0.438 | 0.769 | 0.827 | 0.336 | 0.611 | 0.750 | 0.341 | 0.500 |
| Stretch Squish | 0.772 | 0.379 | 0.333 | 0.250 | 0.299 | – | 0.938 | 0.430 | 0.769 | 0.809 | 0.342 | 0.500 | 0.817 | 0.345 | 0.000 |
| Edge Sawtooth | 0.797 | 0.397 | 0.167 | 0.875 | 0.275 | – | 1.018 | 0.430 | 0.769 | 0.836 | 0.335 | 0.556 | 0.692 | 0.347 | 0.000 |
| Elastic | 0.837 | 0.404 | 0.167 | 0.875 | 0.304 | – | 0.920 | 0.439 | 0.692 | 0.809 | 0.342 | 0.611 | 0.788 | 0.346 | 0.000 |
| Color Quantized | 0.805 | 0.398 | 0.167 | 0.750 | 0.318 | – | 0.884 | 0.426 | 0.769 | 0.800 | 0.335 | 0.500 | 0.808 | 0.347 | 0.000 |
| Bright Transform | 0.764 | 0.397 | 0.333 | 0.375 | 0.290 | – | 0.982 | 0.433 | 0.769 | 0.764 | 0.335 | 0.444 | 0.779 | 0.343 | 0.500 |
| Contrast | 0.894 | 0.406 | 0.167 | 0.625 | 0.293 | – | 0.955 | 0.438 | 0.769 | 0.809 | 0.336 | 0.611 | 0.779 | 0.351 | 0.500 |
| Color Shift | 0.862 | 0.407 | 0.167 | 0.750 | 0.294 | – | 1.009 | 0.440 | 0.769 | 0.700 | 0.333 | 0.556 | 0.865 | 0.346 | 0.500 |
| Flicker | 0.878 | 0.411 | 0.333 | 0.375 | 0.286 | – | 0.964 | 0.436 | 0.769 | 0.773 | 0.345 | 0.389 | 0.846 | 0.353 | 0.500 |
| Overexposure | 0.732 | 0.391 | 0.167 | 0.375 | 0.299 | – | 0.884 | 0.432 | 0.769 | 0.655 | 0.333 | 0.556 | 0.740 | 0.356 | 0.000 |
| Underexposure | 0.854 | 0.406 | 0.333 | 0.875 | 0.281 | – | 1.062 | 0.440 | 0.769 | 0.791 | 0.343 | 0.500 | 0.894 | 0.356 | 0.500 |
| Rainy | 0.813 | 0.397 | 0.167 | 0.500 | 0.296 | – | 0.929 | 0.439 | 0.769 | 0.782 | 0.336 | 0.500 | 0.740 | 0.350 | 0.500 |
| Foggy | 0.797 | 0.407 | 0.333 | 0.625 | 0.301 | – | 1.089 | 0.443 | 0.769 | 0.773 | 0.341 | 0.556 | 0.808 | 0.345 | 0.500 |
| Snow | 0.829 | 0.409 | 0.333 | 0.500 | 0.296 | – | 1.009 | 0.442 | 0.769 | 0.745 | 0.332 | 0.389 | 0.779 | 0.348 | 0.000 |
| Frost | 0.780 | 0.399 | 0.167 | 0.625 | 0.283 | – | 0.786 | 0.424 | 0.769 | 0.773 | 0.342 | 0.556 | 0.817 | 0.347 | 0.500 |
| Reflect | 0.748 | 0.385 | 0.167 | 0.375 | 0.300 | – | 1.009 | 0.433 | 0.769 | 0.782 | 0.335 | 0.500 | 0.817 | 0.356 | 0.500 |
| Shadow | 0.837 | 0.402 | 0.167 | 1.125 | 0.304 | – | 0.946 | 0.441 | 0.692 | 0.827 | 0.346 | 0.500 | 0.798 | 0.357 | 0.000 |

| Noise Type | F GPT | F SBERT | F Acc | G GPT | G SBERT | G Acc | H GPT | H SBERT | H Acc | I GPT | I SBERT | I Acc | J GPT | J SBERT | J Acc |
|---|---|---|---|---|---|---|---|---|---|---|---|---|---|---|---|
| Clean | 1.250 | 0.360 | – | 0.855 | 0.483 | 0.469 | 0.825 | 0.477 | 0.333 | 1.054 | 0.532 | 0.625 | 0.489 | 0.302 | 1.000 |
| Gaussian | 1.250 | 0.382 | – | 0.821 | 0.481 | 0.375 | 0.963 | 0.482 | 0.500 | 1.070 | 0.532 | 0.688 | 0.578 | 0.313 | 1.000 |
| Impulse | 1.000 | 0.339 | – | 0.740 | 0.469 | 0.406 | 0.762 | 0.459 | 0.333 | 0.984 | 0.515 | 0.750 | 0.411 | 0.296 | 1.000 |
| Speckle | 1.750 | 0.377 | – | 0.783 | 0.481 | 0.312 | 0.938 | 0.472 | 0.500 | 0.961 | 0.516 | 0.688 | 0.633 | 0.315 | 1.000 |
| Poisson | 1.250 | 0.361 | – | 0.834 | 0.478 | 0.344 | 0.887 | 0.469 | 0.333 | 1.116 | 0.520 | 0.750 | 0.478 | 0.301 | 1.000 |
| Gaussian Blur | 1.000 | 0.363 | – | 0.719 | 0.467 | 0.312 | 0.812 | 0.472 | 0.333 | 1.078 | 0.518 | 0.750 | 0.522 | 0.302 | 1.000 |
| Motion Blur | 1.000 | 0.372 | – | 0.643 | 0.447 | 0.375 | 0.825 | 0.471 | 0.333 | 0.915 | 0.508 | 0.625 | 0.478 | 0.303 | 1.000 |
| Defocus Blur | 1.500 | 0.321 | – | 0.855 | 0.477 | 0.406 | 0.825 | 0.474 | 0.333 | 1.062 | 0.529 | 0.688 | 0.544 | 0.316 | 1.000 |
| Glass Blur | 0.250 | 0.323 | – | 0.838 | 0.476 | 0.344 | 0.863 | 0.460 | 0.500 | 1.023 | 0.522 | 0.750 | 0.500 | 0.311 | 1.000 |
| Zoom Blur | 1.750 | 0.392 | – | 0.740 | 0.464 | 0.375 | 1.000 | 0.471 | 0.333 | 0.984 | 0.522 | 0.625 | 0.567 | 0.308 | 1.000 |
| Frame Drop | 0.500 | 0.364 | – | 0.719 | 0.456 | 0.406 | 0.825 | 0.475 | 0.333 | 0.884 | 0.507 | 0.688 | 0.422 | 0.297 | 1.000 |
| Frame Replace | 1.250 | 0.343 | – | 0.830 | 0.476 | 0.406 | 0.812 | 0.467 | 0.333 | 1.031 | 0.536 | 0.688 | 0.433 | 0.308 | 1.000 |
| Frame Repeat | 0.500 | 0.372 | – | 0.689 | 0.466 | 0.406 | 0.838 | 0.464 | 0.333 | 0.853 | 0.516 | 0.688 | 0.411 | 0.282 | 1.000 |
| Temporal Jitter | 0.500 | 0.399 | – | 0.881 | 0.481 | 0.406 | 0.850 | 0.473 | 0.333 | 1.039 | 0.523 | 0.688 | 0.478 | 0.299 | 1.000 |
| Jpeg Artifact | 1.250 | 0.308 | – | 0.800 | 0.479 | 0.375 | 0.787 | 0.468 | 0.333 | 0.899 | 0.518 | 0.625 | 0.478 | 0.311 | 1.000 |
| Bit Error | 1.000 | 0.353 | – | 0.796 | 0.476 | 0.406 | 0.988 | 0.477 | 0.333 | 1.078 | 0.534 | 0.688 | 0.433 | 0.305 | 1.000 |
| H265 Artifacts | 0.500 | 0.343 | – | 0.617 | 0.461 | 0.344 | 0.900 | 0.470 | 0.333 | 0.992 | 0.524 | 0.688 | 0.433 | 0.297 | 1.000 |
| Random Block | 0.500 | 0.371 | – | 0.898 | 0.488 | 0.375 | 0.850 | 0.456 | 0.500 | 0.977 | 0.524 | 0.688 | 0.433 | 0.307 | 1.000 |
| Target Block | 1.500 | 0.334 | – | 0.813 | 0.470 | 0.406 | 0.863 | 0.471 | 0.333 | 1.054 | 0.539 | 0.688 | 0.467 | 0.305 | 1.000 |
| Rolling Shutter | 1.000 | 0.356 | – | 0.796 | 0.476 | 0.219 | 0.812 | 0.467 | 0.333 | 1.023 | 0.523 | 0.688 | 0.500 | 0.295 | 1.000 |
| Resolution Degrade | 1.000 | 0.367 | – | 0.740 | 0.474 | 0.344 | 0.863 | 0.479 | 0.333 | 1.047 | 0.519 | 0.750 | 0.589 | 0.318 | 1.000 |
| Stretch Squish | 1.500 | 0.385 | – | 0.668 | 0.461 | 0.312 | 0.750 | 0.454 | 0.333 | 1.054 | 0.516 | 0.750 | 0.489 | 0.302 | 1.000 |
| Edge Sawtooth | 1.500 | 0.370 | – | 0.838 | 0.478 | 0.375 | 1.000 | 0.480 | 0.500 | 1.062 | 0.521 | 0.688 | 0.400 | 0.291 | 1.000 |
| Elastic | 0.750 | 0.334 | – | 0.847 | 0.483 | 0.375 | 0.938 | 0.476 | 0.333 | 1.070 | 0.530 | 0.688 | 0.456 | 0.316 | 1.000 |
| Color Quantized | 1.000 | 0.350 | – | 0.757 | 0.478 | 0.312 | 1.000 | 0.490 | 0.333 | 1.000 | 0.524 | 0.688 | 0.522 | 0.305 | 1.000 |
| Bright Transform | 0.750 | 0.318 | – | 0.804 | 0.478 | 0.406 | 0.988 | 0.478 | 0.500 | 0.876 | 0.516 | 0.750 | 0.533 | 0.312 | 1.000 |
| Contrast | 1.250 | 0.338 | – | 0.915 | 0.484 | 0.438 | 1.000 | 0.490 | 0.333 | 1.085 | 0.526 | 0.688 | 0.500 | 0.314 | 1.000 |
| Color Shift | 1.250 | 0.382 | – | 0.855 | 0.485 | 0.375 | 0.938 | 0.475 | 0.500 | 1.062 | 0.530 | 0.688 | 0.489 | 0.315 | 1.000 |
| Flicker | 0.750 | 0.343 | – | 0.855 | 0.485 | 0.406 | 0.838 | 0.479 | 0.333 | 1.039 | 0.530 | 0.625 | 0.511 | 0.304 | 1.000 |
| Overexposure | 1.000 | 0.359 | – | 0.838 | 0.477 | 0.375 | 0.762 | 0.466 | 0.333 | 0.922 | 0.530 | 0.750 | 0.556 | 0.314 | 1.000 |
| Underexposure | 0.250 | 0.355 | – | 0.898 | 0.488 | 0.438 | 0.938 | 0.458 | 0.500 | 1.008 | 0.529 | 0.562 | 0.511 | 0.322 | 1.000 |
| Rainy | 1.250 | 0.343 | – | 0.745 | 0.471 | 0.344 | 0.863 | 0.462 | 0.333 | 1.054 | 0.530 | 0.750 | 0.556 | 0.316 | 1.000 |
| Foggy | 1.000 | 0.339 | – | 0.885 | 0.478 | 0.438 | 0.975 | 0.476 | 0.500 | 1.078 | 0.532 | 0.688 | 0.556 | 0.313 | 1.000 |
| Snow | 1.500 | 0.352 | – | 0.779 | 0.480 | 0.375 | 0.975 | 0.493 | 0.333 | 1.202 | 0.524 | 0.688 | 0.511 | 0.312 | 1.000 |
| Frost | 1.500 | 0.357 | – | 0.868 | 0.482 | 0.500 | 0.800 | 0.464 | 0.333 | 0.992 | 0.519 | 0.625 | 0.500 | 0.319 | 1.000 |
| Reflect | 0.750 | 0.344 | – | 0.762 | 0.475 | 0.281 | 0.950 | 0.486 | 0.333 | 1.000 | 0.523 | 0.625 | 0.411 | 0.292 | 1.000 |
| Shadow | 1.250 | 0.339 | – | 0.830 | 0.475 | 0.375 | 1.000 | 0.467 | 0.500 | 1.039 | 0.527 | 0.688 | 0.489 | 0.314 | 1.000 |

| Noise Type | K GPT | K SBERT | K Acc | L GPT | L SBERT | L Acc | M GPT | M SBERT | M Acc | N GPT | N SBERT | N Acc | O GPT | O SBERT | O Acc |
|---|---|---|---|---|---|---|---|---|---|---|---|---|---|---|---|
| Clean | 0.453 | 0.332 | 0.667 | 1.455 | 0.420 | 1.000 | 0.810 | 0.515 | 0.571 | 0.926 | 0.338 | 0.600 | 0.692 | 0.334 | 0.000 |
| Gaussian | 0.547 | 0.353 | 0.667 | 1.167 | 0.394 | 1.000 | 0.816 | 0.509 | 0.500 | 0.821 | 0.337 | 0.600 | 0.923 | 0.343 | 0.000 |
| Impulse | 0.299 | 0.325 | 0.667 | 0.985 | 0.395 | 1.000 | 0.796 | 0.513 | 0.500 | 0.726 | 0.307 | 0.600 | 0.692 | 0.340 | 0.000 |
| Speckle | 0.504 | 0.330 | 0.667 | 1.045 | 0.374 | 1.000 | 0.810 | 0.513 | 0.571 | 0.768 | 0.323 | 0.600 | 0.923 | 0.343 | 0.000 |
| Poisson | 0.504 | 0.342 | 0.667 | 1.288 | 0.400 | 1.000 | 0.762 | 0.507 | 0.429 | 0.842 | 0.326 | 0.600 | 1.154 | 0.357 | 0.000 |
| Gaussian Blur | 0.372 | 0.327 | 0.667 | 1.197 | 0.401 | 1.000 | 0.762 | 0.515 | 0.429 | 0.705 | 0.309 | 0.600 | 0.846 | 0.336 | 0.000 |
| Motion Blur | 0.387 | 0.330 | 0.667 | 1.091 | 0.380 | 1.000 | 0.850 | 0.512 | 0.571 | 0.768 | 0.307 | 0.600 | 0.923 | 0.347 | 0.000 |
| Defocus Blur | 0.504 | 0.345 | 0.667 | 1.333 | 0.409 | 1.000 | 0.878 | 0.519 | 0.500 | 0.832 | 0.328 | 0.600 | 0.769 | 0.367 | 0.000 |
| Glass Blur | 0.533 | 0.341 | 0.667 | 1.288 | 0.407 | 1.000 | 0.884 | 0.512 | 0.429 | 0.726 | 0.308 | 0.600 | 1.000 | 0.347 | 0.000 |
| Zoom Blur | 0.577 | 0.349 | 0.667 | 1.136 | 0.412 | 1.000 | 0.796 | 0.514 | 0.357 | 0.779 | 0.317 | 0.600 | 0.923 | 0.345 | 0.000 |
| Frame Drop | 0.401 | 0.329 | 0.667 | 1.227 | 0.395 | 1.000 | 0.735 | 0.511 | 0.500 | 0.589 | 0.297 | 0.600 | 0.846 | 0.394 | 0.000 |
| Frame Replace | 0.526 | 0.342 | 0.667 | 1.500 | 0.421 | 1.000 | 0.830 | 0.516 | 0.571 | 0.979 | 0.327 | 0.600 | 0.846 | 0.361 | 0.000 |
| Frame Repeat | 0.365 | 0.312 | 0.667 | 1.182 | 0.383 | 0.667 | 0.844 | 0.507 | 0.429 | 0.611 | 0.297 | 0.600 | 0.692 | 0.352 | 0.000 |
| Temporal Jitter | 0.511 | 0.332 | 0.667 | 1.227 | 0.406 | 1.000 | 0.748 | 0.507 | 0.500 | 0.800 | 0.314 | 0.600 | 0.769 | 0.347 | 0.000 |
| Jpeg Artifact | 0.511 | 0.344 | 0.667 | 1.076 | 0.393 | 1.000 | 0.789 | 0.508 | 0.500 | 0.737 | 0.336 | 0.600 | 0.769 | 0.379 | 0.000 |
| Bit Error | 0.394 | 0.344 | 0.667 | 1.394 | 0.413 | 1.000 | 0.796 | 0.516 | 0.500 | 0.895 | 0.337 | 0.600 | 1.000 | 0.345 | 0.000 |
| H265 Artifacts | 0.504 | 0.331 | 0.667 | 1.182 | 0.392 | 1.000 | 0.796 | 0.511 | 0.500 | 0.568 | 0.304 | 0.400 | 0.769 | 0.317 | 0.000 |
| Random Block | 0.445 | 0.334 | 0.667 | 1.258 | 0.404 | 1.000 | 0.884 | 0.517 | 0.571 | 0.884 | 0.315 | 0.600 | 0.846 | 0.347 | 0.000 |
| Target Block | 0.504 | 0.339 | 0.667 | 1.364 | 0.416 | 1.000 | 0.816 | 0.516 | 0.500 | 0.821 | 0.337 | 0.600 | 1.077 | 0.324 | 0.000 |
| Rolling Shutter | 0.540 | 0.341 | 0.667 | 1.318 | 0.410 | 1.000 | 0.728 | 0.515 | 0.429 | 0.705 | 0.329 | 0.600 | 1.077 | 0.373 | 0.000 |
| Resolution Degrade | 0.613 | 0.351 | 0.667 | 1.182 | 0.387 | 1.000 | 0.816 | 0.512 | 0.429 | 0.768 | 0.320 | 0.600 | 1.000 | 0.350 | 0.000 |
| Stretch Squish | 0.423 | 0.332 | 0.667 | 1.091 | 0.393 | 1.000 | 0.762 | 0.516 | 0.500 | 0.779 | 0.312 | 0.600 | 0.923 | 0.356 | 0.000 |
| Edge Sawtooth | 0.504 | 0.345 | 0.667 | 1.136 | 0.388 | 1.000 | 0.823 | 0.521 | 0.500 | 0.863 | 0.331 | 0.600 | 0.923 | 0.351 | 0.000 |
| Elastic | 0.555 | 0.344 | 0.667 | 1.470 | 0.430 | 1.000 | 0.741 | 0.513 | 0.500 | 0.979 | 0.335 | 0.600 | 0.769 | 0.344 | 0.000 |
| Color Quantized | 0.547 | 0.345 | 0.667 | 1.318 | 0.405 | 1.000 | 0.810 | 0.514 | 0.500 | 0.747 | 0.329 | 0.600 | 1.077 | 0.339 | 0.000 |
| Bright Transform | 0.562 | 0.341 | 0.667 | 1.242 | 0.401 | 1.000 | 0.898 | 0.519 | 0.643 | 0.811 | 0.338 | 0.600 | 0.846 | 0.346 | 0.000 |
| Contrast | 0.540 | 0.345 | 0.667 | 1.167 | 0.404 | 1.000 | 0.905 | 0.512 | 0.429 | 0.758 | 0.334 | 0.600 | 0.923 | 0.334 | 0.000 |
| Color Shift | 0.569 | 0.344 | 0.667 | 1.379 | 0.409 | 1.000 | 0.803 | 0.515 | 0.571 | 0.895 | 0.321 | 0.600 | 0.846 | 0.325 | 0.000 |
| Flicker | 0.540 | 0.337 | 0.667 | 1.303 | 0.413 | 1.000 | 0.782 | 0.512 | 0.429 | 0.832 | 0.329 | 0.600 | 0.846 | 0.334 | 0.000 |
| Overexposure | 0.591 | 0.347 | 0.667 | 1.318 | 0.400 | 1.000 | 0.871 | 0.516 | 0.571 | 0.937 | 0.336 | 0.600 | 1.000 | 0.333 | 0.000 |
| Underexposure | 0.518 | 0.340 | 0.333 | 1.288 | 0.407 | 1.000 | 0.776 | 0.507 | 0.571 | 0.842 | 0.326 | 0.600 | 1.077 | 0.370 | 0.000 |
| Rainy | 0.569 | 0.347 | 0.667 | 1.348 | 0.409 | 1.000 | 0.850 | 0.517 | 0.571 | 0.905 | 0.334 | 0.600 | 0.769 | 0.351 | 0.000 |
| Foggy | 0.474 | 0.339 | 0.667 | 1.379 | 0.407 | 1.000 | 0.864 | 0.524 | 0.500 | 0.789 | 0.325 | 0.600 | 0.923 | 0.335 | 0.000 |
| Snow | 0.533 | 0.343 | 0.667 | 1.348 | 0.411 | 1.000 | 0.844 | 0.517 | 0.643 | 0.789 | 0.337 | 0.600 | 0.769 | 0.321 | 0.000 |
| Frost | 0.562 | 0.343 | 0.667 | 1.242 | 0.409 | 1.000 | 0.741 | 0.519 | 0.500 | 0.768 | 0.327 | 0.600 | 0.846 | 0.312 | 0.000 |
| Reflect | 0.423 | 0.336 | 0.333 | 1.273 | 0.400 | 1.000 | 0.830 | 0.516 | 0.500 | 0.779 | 0.327 | 0.600 | 1.000 | 0.344 | 0.000 |
| Shadow | 0.489 | 0.344 | 0.667 | 1.333 | 0.397 | 1.000 | 0.762 | 0.519 | 0.500 | 0.853 | 0.342 | 0.600 | 0.692 | 0.344 | 0.000 |

| Noise Type | P GPT | P SBERT | P Acc | Q GPT | Q SBERT | Q Acc | R GPT | R SBERT | R Acc | S GPT | S SBERT | S Acc | T GPT | T SBERT | T Acc |
|---|---|---|---|---|---|---|---|---|---|---|---|---|---|---|---|
| Clean | 0.855 | 0.410 | 0.263 | 1.231 | 0.409 | 1.000 | 0.698 | 0.536 | 0.400 | 1.236 | 0.453 | 0.714 | 0.870 | 0.429 | 0.507 |
| Gaussian | 0.763 | 0.412 | 0.211 | 1.121 | 0.408 | 1.000 | 0.685 | 0.529 | 0.200 | 1.236 | 0.451 | 0.714 | 0.848 | 0.429 | 0.500 |
| Impulse | 0.763 | 0.398 | 0.158 | 1.011 | 0.404 | 1.000 | 0.642 | 0.523 | 0.000 | 1.330 | 0.461 | 0.571 | 0.770 | 0.418 | 0.461 |
| Speckle | 0.763 | 0.410 | 0.211 | 1.154 | 0.402 | 1.000 | 0.722 | 0.538 | 0.000 | 1.311 | 0.462 | 0.714 | 0.852 | 0.426 | 0.480 |
| Poisson | 0.801 | 0.411 | 0.263 | 1.055 | 0.404 | 1.000 | 0.642 | 0.534 | 0.000 | 1.311 | 0.456 | 0.571 | 0.842 | 0.426 | 0.480 |
| Gaussian Blur | 0.726 | 0.402 | 0.211 | 1.044 | 0.411 | 1.000 | 0.741 | 0.531 | 0.200 | 1.321 | 0.462 | 0.714 | 0.792 | 0.421 | 0.461 |
| Motion Blur | 0.715 | 0.397 | 0.211 | 1.077 | 0.403 | 1.000 | 0.716 | 0.533 | 0.000 | 1.198 | 0.450 | 0.571 | 0.784 | 0.416 | 0.474 |
| Defocus Blur | 0.844 | 0.412 | 0.158 | 1.198 | 0.411 | 1.000 | 0.790 | 0.550 | 0.000 | 1.292 | 0.452 | 0.714 | 0.880 | 0.430 | 0.461 |
| Glass Blur | 0.833 | 0.408 | 0.263 | 1.187 | 0.413 | 1.000 | 0.796 | 0.539 | 0.000 | 1.208 | 0.446 | 0.714 | 0.864 | 0.426 | 0.500 |
| Zoom Blur | 0.747 | 0.402 | 0.263 | 1.198 | 0.403 | 1.000 | 0.673 | 0.531 | 0.000 | 1.189 | 0.454 | 0.714 | 0.838 | 0.425 | 0.441 |
| Frame Drop | 0.720 | 0.395 | 0.211 | 1.077 | 0.401 | 1.000 | 0.735 | 0.530 | 0.400 | 1.142 | 0.456 | 0.571 | 0.765 | 0.417 | 0.487 |
| Frame Replace | 0.855 | 0.419 | 0.211 | 1.231 | 0.413 | 1.000 | 0.722 | 0.540 | 0.000 | 1.208 | 0.452 | 0.571 | 0.881 | 0.431 | 0.487 |
| Frame Repeat | 0.699 | 0.394 | 0.211 | 0.967 | 0.392 | 1.000 | 0.679 | 0.522 | 0.000 | 1.075 | 0.452 | 0.571 | 0.721 | 0.410 | 0.454 |
| Temporal Jitter | 0.694 | 0.400 | 0.263 | 1.220 | 0.415 | 1.000 | 0.716 | 0.539 | 0.200 | 1.189 | 0.453 | 0.571 | 0.822 | 0.425 | 0.480 |
| Jpeg Artifact | 0.806 | 0.414 | 0.263 | 1.176 | 0.417 | 1.000 | 0.716 | 0.535 | 0.200 | 1.330 | 0.455 | 0.714 | 0.818 | 0.427 | 0.487 |
| Bit Error | 0.780 | 0.406 | 0.316 | 1.220 | 0.418 | 1.000 | 0.784 | 0.539 | 0.200 | 1.236 | 0.458 | 0.714 | 0.852 | 0.429 | 0.487 |
| H265 Artifacts | 0.661 | 0.395 | 0.211 | 1.099 | 0.396 | 1.000 | 0.747 | 0.530 | 0.000 | 1.113 | 0.451 | 0.714 | 0.765 | 0.418 | 0.454 |
| Random Block | 0.780 | 0.410 | 0.263 | 1.319 | 0.420 | 1.000 | 0.784 | 0.535 | 0.000 | 1.302 | 0.456 | 0.714 | 0.849 | 0.426 | 0.474 |
| Target Block | 0.742 | 0.405 | 0.211 | 1.143 | 0.406 | 1.000 | 0.660 | 0.536 | 0.000 | 1.226 | 0.457 | 0.714 | 0.834 | 0.427 | 0.474 |
| Rolling Shutter | 0.860 | 0.414 | 0.158 | 1.187 | 0.423 | 1.000 | 0.698 | 0.529 | 0.000 | 1.160 | 0.453 | 0.714 | 0.821 | 0.425 | 0.434 |
| Resolution Degrade | 0.780 | 0.406 | 0.158 | 1.176 | 0.410 | 1.000 | 0.809 | 0.541 | 0.000 | 1.217 | 0.453 | 0.571 | 0.851 | 0.426 | 0.474 |
| Stretch Squish | 0.672 | 0.397 | 0.211 | 1.121 | 0.404 | 1.000 | 0.765 | 0.529 | 0.000 | 1.189 | 0.451 | 0.571 | 0.793 | 0.418 | 0.461 |
| Edge Sawtooth | 0.833 | 0.415 | 0.158 | 1.154 | 0.423 | 1.000 | 0.759 | 0.545 | 0.200 | 1.264 | 0.461 | 0.714 | 0.858 | 0.428 | 0.480 |
| Elastic | 0.849 | 0.408 | 0.211 | 1.275 | 0.422 | 1.000 | 0.796 | 0.545 | 0.200 | 1.274 | 0.459 | 0.714 | 0.882 | 0.432 | 0.480 |
| Color Quantized | 0.871 | 0.413 | 0.211 | 1.264 | 0.411 | 1.000 | 0.759 | 0.544 | 0.200 | 1.340 | 0.466 | 0.714 | 0.862 | 0.429 | 0.461 |
| Bright Transform | 0.839 | 0.411 | 0.211 | 1.231 | 0.412 | 1.000 | 0.691 | 0.541 | 0.200 | 1.274 | 0.458 | 0.714 | 0.850 | 0.428 | 0.513 |
| Contrast | 0.849 | 0.413 | 0.211 | 1.198 | 0.410 | 1.000 | 0.753 | 0.550 | 0.000 | 1.340 | 0.464 | 0.714 | 0.889 | 0.432 | 0.493 |
| Color Shift | 0.758 | 0.406 | 0.211 | 1.275 | 0.423 | 1.000 | 0.753 | 0.535 | 0.000 | 1.217 | 0.458 | 0.714 | 0.873 | 0.429 | 0.493 |
| Flicker | 0.742 | 0.411 | 0.158 | 1.286 | 0.412 | 1.000 | 0.691 | 0.538 | 0.200 | 1.292 | 0.458 | 0.714 | 0.857 | 0.430 | 0.461 |
| Overexposure | 0.823 | 0.410 | 0.263 | 1.143 | 0.411 | 1.000 | 0.741 | 0.539 | 0.200 | 1.349 | 0.450 | 0.571 | 0.846 | 0.428 | 0.493 |
| Underexposure | 0.849 | 0.420 | 0.211 | 1.198 | 0.414 | 1.000 | 0.753 | 0.540 | 0.000 | 1.245 | 0.453 | 0.714 | 0.880 | 0.431 | 0.487 |
| Rainy | 0.833 | 0.413 | 0.263 | 1.121 | 0.410 | 1.000 | 0.710 | 0.541 | 0.000 | 1.283 | 0.457 | 0.714 | 0.852 | 0.429 | 0.487 |
| Foggy | 0.699 | 0.403 | 0.211 | 1.121 | 0.414 | 1.000 | 0.759 | 0.540 | 0.200 | 1.349 | 0.461 | 0.714 | 0.871 | 0.430 | 0.513 |
| Snow | 0.952 | 0.416 | 0.211 | 1.154 | 0.413 | 1.000 | 0.710 | 0.543 | 0.000 | 1.274 | 0.458 | 0.714 | 0.877 | 0.431 | 0.474 |
| Frost | 0.747 | 0.401 | 0.263 | 1.176 | 0.414 | 1.000 | 0.710 | 0.540 | 0.200 | 1.264 | 0.448 | 0.714 | 0.827 | 0.427 | 0.513 |
| Reflect | 0.812 | 0.419 | 0.263 | 1.209 | 0.412 | 1.000 | 0.722 | 0.537 | 0.200 | 1.236 | 0.451 | 0.714 | 0.832 | 0.426 | 0.454 |
| Shadow | 0.823 | 0.409 | 0.211 | 1.154 | 0.401 | 1.000 | 0.716 | 0.540 | 0.000 | 1.340 | 0.463 | 0.714 | 0.862 | 0.430 | 0.467 |

Note: A: Advertisements, B: Algorithm & Models, C: Autos & Vehicles, D: Business & Industrial, E: Computers & Electronics, F: Fairy Tale, G: Films & TV Shows, H: Finance, I: Food & Drink, J: Games, K: Humor, L: Instruction Video (how to), M: Knowledge, N: News, O: Others, P: People, Q: Pets & Animals, R: Science, S: Sports, T: Overall

Table 4: GPT-4o performance across different video types

| Noise Type | A GPT | A SBERT | A Acc | B GPT | B SBERT | B Acc | C GPT | C SBERT | C Acc | D GPT | D SBERT | D Acc | E GPT | E SBERT | E Acc | F GPT | F SBERT | F Acc | G GPT | G SBERT | G Acc | H GPT | H SBERT | H Acc | J GPT | J SBERT | J Acc |
|---|---|---|---|---|---|---|---|---|---|---|---|---|---|---|---|---|---|---|---|---|---|---|---|---|---|---|---|
| Clean | .577 | 0.508 | – | 2.125 | 0.406 | 0.667 | .955 | 0.523 | 1.000 | .855 | 0.577 | 0.722 | .865 | 0.508 | 0.600 | .250 | 0.433 | 1.000 | .553 | 0.516 | 0.438 | .850 | 0.548 | 0.833 | .567 | 0.441 | 0.000 |
| Gaussian | .545 | 0.512 | – | 1.750 | 0.379 | 0.667 | .795 | 0.496 | 0.333 | .709 | 0.597 | 0.722 | .644 | 0.501 | 0.800 | 1.000 | 0.582 | 0.000 | .517 | 0.517 | 0.719 | .800 | 0.536 | 0.667 | .356 | 0.444 | 1.000 |
| Impulse | .244 | 0.469 | – | 2.125 | 0.400 | 0.667 | .348 | 0.475 | 0.462 | .364 | 0.544 | 0.611 | .519 | 0.513 | 0.600 | 1.000 | 0.302 | 0.000 | .289 | 0.540 | 0.625 | .887 | 0.532 | 0.500 | .278 | 0.406 | 1.000 |
| Speckle | .512 | 0.509 | – | 2.000 | 0.400 | 0.667 | .768 | 0.523 | 0.462 | .536 | 0.588 | 0.611 | .683 | 0.512 | 1.000 | 1.000 | 0.409 | 0.500 | .643 | 0.505 | 0.594 | .850 | 0.569 | 0.500 | .389 | 0.429 | 1.000 |
| Poisson | .439 | 0.495 | – | 2.500 | 0.392 | 0.333 | .768 | 0.518 | 0.462 | .418 | 0.577 | 0.500 | .606 | 0.513 | 0.600 | .250 | 0.406 | 1.000 | .477 | 0.505 | 0.719 | .837 | 0.543 | 0.500 | .267 | 0.419 | 1.000 |
| Gaussian Blur | .228 | 0.474 | – | 2.125 | 0.427 | 0.667 | .661 | 0.505 | 0.538 | .291 | 0.586 | 0.667 | .442 | 0.463 | 0.600 | .500 | 0.431 | 0.500 | .383 | 0.533 | 0.625 | .938 | 0.585 | 0.688 | .233 | 0.404 | 1.000 |
| Motion Blur | .171 | 0.475 | – | 1.625 | 0.462 | 1.000 | .607 | 0.498 | 0.308 | .355 | 0.559 | 0.333 | .452 | 0.520 | 0.800 | 1.000 | 0.388 | 0.000 | .549 | 0.510 | 0.594 | .712 | 0.538 | 0.812 | .267 | 0.416 | 1.000 |
| Defocus Blur | .512 | 0.494 | – | 2.125 | 0.406 | 1.000 | .732 | 0.512 | 0.462 | .536 | 0.588 | 0.667 | .577 | 0.486 | 1.000 | .250 | 0.458 | 0.750 | .596 | 0.520 | 0.656 | .788 | 0.553 | 0.750 | .533 | 0.433 | 1.000 |
| Glass Blur | .447 | 0.495 | – | 1.875 | 0.401 | 1.000 | .759 | 0.491 | 0.462 | .573 | 0.587 | 0.500 | .721 | 0.503 | 1.000 | .250 | 0.445 | 1.000 | .472 | 0.517 | 0.750 | .738 | 0.568 | 0.625 | .389 | 0.445 | 1.000 |
| Zoom Blur | .512 | 0.495 | – | 2.000 | 0.430 | 1.000 | .723 | 0.511 | 0.538 | .573 | 0.575 | 0.500 | .337 | 0.478 | 0.600 | .250 | 0.426 | 1.000 | .213 | 0.495 | 0.750 | .500 | 0.557 | 0.688 | .089 | 0.411 | 1.000 |
| Frame Drop | .967 | 0.443 | – | 1.125 | 0.354 | 0.333 | .321 | 0.439 | 0.308 | .209 | 0.523 | 0.556 | 1.144 | 0.407 | 1.000 | .000 | 0.440 | 0.000 | .277 | 0.536 | 0.688 | .925 | 0.557 | 0.688 | .533 | 0.446 | 0.000 |
| Frame Replace | .602 | 0.524 | – | 2.000 | 0.430 | 0.500 | .348 | 0.440 | 0.231 | .155 | 0.544 | 0.611 | .606 | 0.466 | 0.600 | .250 | 0.363 | 1.000 | .532 | 0.524 | 0.625 | .850 | 0.559 | 0.500 | .167 | 0.372 | 1.000 |
| Frame Repeat | .927 | 0.420 | – | 1.500 | 0.436 | 0.333 | .714 | 0.495 | 0.462 | .591 | 0.572 | 0.611 | .769 | 0.529 | 1.000 | .250 | 0.407 | 0.750 | .532 | 0.530 | 0.781 | 2.075 | 0.544 | 0.833 | .411 | 0.396 | 1.000 |
| Temporal Jitter | .520 | 0.494 | – | 1.875 | 0.432 | 0.667 | .500 | 0.500 | 0.385 | .591 | 0.583 | 0.611 | .779 | 0.516 | 0.722 | .500 | 0.419 | 0.000 | .549 | 0.526 | 0.719 | 1.581 | 0.558 | 0.750 | .406 | 0.405 | 1.000 |
| Jpeg Artifact | .439 | 0.503 | – | 2.250 | 0.422 | 0.667 | .634 | 0.500 | 0.462 | .709 | 0.600 | 0.722 | .327 | 0.487 | 0.600 | .750 | 0.418 | 0.500 | .579 | 0.533 | 0.719 | 1.812 | 0.541 | 0.688 | .400 | 0.417 | 1.000 |
| Bit Error | .537 | 0.512 | – | 2.250 | 0.434 | 0.667 | .446 | 0.498 | 0.462 | .291 | 0.551 | 0.389 | .760 | 0.467 | 0.800 | .250 | 0.415 | 1.000 | .409 | 0.523 | 0.750 | .962 | 0.555 | 0.812 | .144 | 0.405 | 1.000 |
| H265 Artifacts | .089 | 0.454 | – | 2.000 | 0.400 | 0.667 | .571 | 0.507 | 0.385 | .545 | 0.609 | 0.833 | .545 | 0.492 | 1.000 | .250 | 0.427 | 1.000 | .540 | 0.523 | 0.750 | .825 | 0.561 | 0.625 | .422 | 0.470 | 1.000 |
| Random Block | .472 | 0.500 | – | 2.125 | 0.415 | 0.500 | .509 | 0.495 | 0.462 | .518 | 0.583 | 0.611 | .567 | 0.467 | 0.500 | .250 | 0.429 | 1.000 | .528 | 0.531 | 0.688 | .812 | 0.564 | 0.750 | .156 | 0.438 | 1.000 |
| Target Block | .528 | 0.509 | – | 2.125 | 0.400 | 0.167 | .509 | 0.512 | 0.538 | .627 | 0.552 | 0.611 | .904 | 0.493 | 1.000 | .250 | 0.427 | 0.500 | .319 | 0.459 | 0.344 | .800 | 0.552 | 0.625 | .211 | 0.415 | 1.000 |
| Rolling Shutter | .488 | 0.490 | – | 1.750 | 0.479 | 0.500 | .402 | 0.469 | 0.462 | .200 | 0.536 | 0.389 | .356 | 0.467 | 0.500 | .500 | 0.411 | 1.000 | .387 | 0.539 | 0.656 | .725 | 0.566 | 0.750 | .311 | 0.433 | 1.000 |
| Resolution Degrade | .463 | 0.476 | – | 2.000 | 0.504 | 0.667 | .705 | 0.509 | 0.538 | .627 | 0.585 | 0.611 | .817 | 0.522 | 1.000 | .260 | 0.059 | 1.000 | .570 | 0.531 | 0.656 | .887 | 0.552 | 0.625 | .433 | 0.443 | 1.000 |
| Color Quantized | .553 | 0.516 | – | 1.750 | 0.361 | 1.000 | .705 | 0.538 | 0.538 | .618 | 0.611 | 0.611 | .865 | 0.513 | 1.000 | .750 | 0.411 | 1.000 | .757 | 0.502 | 0.031 | 1.913 | 0.567 | 0.812 | .356 | 0.448 | 1.000 |
| Bright Transform | .626 | 0.516 | – | 1.750 | 0.480 | 0.667 | .670 | 0.509 | 0.538 | .627 | 0.543 | 0.667 | .817 | 0.524 | 1.000 | .500 | 0.430 | 0.500 | .638 | 0.549 | 0.688 | .925 | 0.569 | 0.625 | .411 | 0.446 | 0.000 |
| Stretch Squish | .350 | 0.434 | – | 1.750 | 0.003 | 0.500 | .545 | 0.482 | 0.231 | .491 | 0.589 | 0.833 | .769 | 0.521 | 0.667 | .500 | 0.428 | 0.000 | .583 | 0.535 | 0.688 | 1.988 | 0.553 | 0.625 | .489 | 0.446 | 0.000 |
| Edge Sawtooth | .431 | 0.510 | – | 2.250 | 0.498 | 0.667 | .732 | 0.527 | 0.462 | .909 | 0.613 | 0.611 | .817 | 0.517 | 1.000 | .500 | 0.426 | 0.000 | .583 | 0.530 | 0.750 | 1.938 | 0.567 | 0.833 | .456 | 0.443 | 0.000 |
| Elastic | .569 | 0.522 | – | 1.750 | 0.414 | 0.667 | .812 | 0.510 | 0.462 | .727 | 0.583 | 0.611 | .827 | 0.533 | 1.000 | .500 | 0.422 | 1.000 | .591 | 0.528 | 0.656 | 1.825 | 0.575 | 0.625 | .511 | 0.459 | 0.000 |
| Contrast | .650 | 0.527 | – | 1.875 | 0.422 | 0.667 | .795 | 0.502 | 0.462 | .645 | 0.590 | 0.778 | .788 | 0.524 | 1.000 | .250 | 0.415 | 1.000 | .506 | 0.528 | 0.688 | .950 | 0.572 | 0.812 | .433 | 0.461 | 0.000 |
| Color Shift | .569 | 0.506 | – | 1.875 | 0.414 | 0.667 | .723 | 0.509 | 0.538 | .736 | 0.622 | 0.667 | .769 | 0.512 | 0.667 | .250 | 0.423 | 0.500 | .583 | 0.531 | 0.750 | .581 | 0.555 | 0.750 | .378 | 0.450 | 0.000 |
| Flicker | .577 | 0.509 | – | 2.000 | 0.423 | 1.000 | .759 | 0.517 | 0.538 | .609 | 0.576 | 0.500 | .817 | 0.522 | 1.000 | .500 | 0.430 | 0.500 | .583 | 0.525 | 0.625 | 1.713 | 0.573 | 0.750 | .478 | 0.453 | 0.000 |
| Overexposure | .472 | 0.509 | – | 2.250 | 0.396 | 0.667 | .759 | 0.513 | 0.538 | .673 | 0.591 | 0.556 | .769 | 0.533 | 1.000 | .250 | 0.442 | 1.000 | .600 | 0.535 | 0.688 | .635 | 0.572 | 0.688 | .433 | 0.440 | 0.000 |
| Underexposure | .642 | 0.523 | – | 1.875 | 0.404 | 0.500 | .705 | 0.511 | 0.462 | .664 | 0.583 | 0.611 | .817 | 0.509 | 1.000 | .250 | 0.421 | 1.000 | .583 | 0.528 | 0.688 | 2.050 | 0.574 | 0.812 | .378 | 0.450 | 0.000 |
| Rainy | .480 | 0.506 | – | 2.000 | 0.423 | 0.500 | .723 | 0.509 | 0.385 | .745 | 0.622 | 0.667 | .769 | 0.512 | 0.800 | .250 | 0.423 | 0.500 | .583 | 0.531 | 0.688 | .925 | 0.574 | 0.750 | .556 | 0.457 | 0.000 |
| Foggy | .618 | 0.506 | – | 2.250 | 0.407 | 0.333 | .759 | 0.505 | 0.538 | .673 | 0.591 | 0.556 | .817 | 0.522 | 0.667 | .500 | 0.430 | 0.000 | .583 | 0.531 | 0.688 | 1.975 | 0.560 | 0.812 | .433 | 0.450 | 0.000 |
| Snow | .675 | 0.514 | – | 2.250 | 0.464 | 0.500 | .705 | 0.515 | 0.462 | .691 | 0.585 | 0.667 | .808 | 0.527 | 1.000 | .500 | 0.418 | 0.750 | .600 | 0.535 | 0.688 | 1.950 | 0.574 | 0.812 | .411 | 0.458 | 0.000 |
| Frost | .488 | 0.508 | – | 1.875 | 0.397 | 0.500 | .705 | 0.511 | 0.385 | .600 | 0.576 | 0.500 | .827 | 0.528 | 0.800 | .250 | 0.423 | 0.500 | .485 | 0.533 | 0.594 | 2.025 | 0.560 | 0.833 | .644 | 0.441 | 0.000 |
| Reflect | .537 | 0.514 | – | 1.875 | 0.454 | 0.667 | .812 | 0.511 | 0.308 | .600 | 0.604 | 0.667 | .750 | 0.531 | 0.600 | .250 | – | – | .613 | 0.524 | 0.656 | 1.938 | 0.565 | 0.750 | .489 | 0.454 | 1.000 |
| Shadow | .642 | 0.508 | – | 1.875 | 0.454 | – | .750 | 0.515 | 0.538 | .827 | – | 0.667 | .750 | 0.531 | 0.667 | .250 | 0.433 | – | .613 | 0.524 | 0.656 | 2.074 | 0.565 | 0.833 | .489 | 0.454 | 1.000 |

| Noise Type | K GPT | K SBERT | K Acc | L GPT | L SBERT | L Acc | M GPT | M SBERT | M Acc | N GPT | N SBERT | N Acc | O GPT | O SBERT | O Acc | P GPT | P SBERT | P Acc | Q GPT | Q SBERT | Q Acc | R GPT | R SBERT | R Acc | S GPT | S SBERT | S Acc | T GPT | T SBERT | T Acc |
|---|---|---|---|---|---|---|---|---|---|---|---|---|---|---|---|---|---|---|---|---|---|---|---|---|---|---|---|---|---|---|
| Clean | .350 | 0.468 | 1.000 | 2.242 | 0.590 | 1.000 | .769 | 0.574 | 0.786 | .905 | 0.518 | 0.600 | .400 | 0.444 | 0.000 | 1.629 | 0.495 | 0.526 | 2.088 | 0.513 | 0.543 | 1.932 | 0.611 | 0.612 | 1.689 | 0.526 | 0.600 | 2.019 | 0.740 | 0.546 |
| Gaussian | 1.007 | 0.436 | 1.000 | 2.136 | 0.579 | 1.000 | .694 | 0.571 | 0.857 | .789 | 0.521 | 0.800 | .444 | 0.478 | 0.000 | 1.500 | 0.459 | 0.684 | 1.901 | 0.478 | 0.684 | 1.710 | 0.612 | 0.612 | 1.943 | 0.580 | 0.600 | 2.000 | 0.650 | 0.658 |
| Impulse | .759 | 0.399 | 1.000 | 1.833 | 0.571 | 1.000 | .367 | 0.550 | 0.714 | .568 | 0.491 | 0.600 | .367 | 0.459 | 0.000 | 1.285 | 0.472 | 0.579 | 1.692 | 0.484 | 0.579 | 1.790 | 0.612 | 0.612 | 2.000 | 0.581 | 0.612 | 2.861 | 0.527 | 0.612 |
| Speckle | .109 | 0.434 | 1.000 | 2.136 | 0.571 | 1.000 | .612 | 0.560 | 0.786 | .958 | 0.537 | 1.000 | .462 | 0.462 | 0.000 | 1.597 | 0.484 | 0.842 | 1.890 | 0.484 | 0.842 | 1.790 | 0.612 | 0.612 | 2.019 | 0.600 | 0.600 | 2.861 | 0.533 | 0.612 |
| Poisson | .927 | 0.428 | 1.000 | 1.970 | 0.584 | 1.000 | 1.313 | 0.561 | 0.786 | .535 | 0.492 | 0.600 | .449 | 0.471 | 1.000 | 1.495 | 0.472 | 0.684 | 1.813 | 0.501 | 0.684 | 1.877 | 0.603 | 0.623 | 2.104 | 0.591 | 0.614 | 2.104 | 0.599 | 0.599 |
| Gaussian Blur | .876 | 0.432 | 1.000 | 1.727 | 0.585 | 1.000 | .626 | 0.534 | 0.786 | .411 | 0.385 | 0.600 | .443 | 0.455 | 0.000 | 1.385 | 0.492 | 0.632 | 1.626 | 0.459 | 0.632 | 1.728 | 0.598 | 0.603 | 2.094 | 0.599 | 0.614 | 2.861 | 0.510 | 0.664 |
| Motion Blur | .978 | 0.414 | 1.000 | 1.939 | 0.595 | 1.000 | .694 | 0.572 | 0.786 | .979 | 0.531 | 1.000 | .470 | 0.424 | 0.000 | 1.527 | 0.494 | 0.632 | 1.846 | 0.494 | 0.684 | 1.704 | 0.617 | 0.617 | 1.774 | 0.574 | 0.591 | 1.821 | 0.502 | 0.579 |
| Defocus Blur | .204 | 0.417 | 0.667 | 2.076 | 0.595 | 1.000 | .626 | 0.577 | 0.786 | .926 | 0.526 | 1.000 | .428 | 0.470 | 0.000 | 1.489 | 0.492 | 0.632 | 1.824 | 0.507 | 0.684 | 1.815 | 0.619 | 0.619 | 1.962 | 0.622 | 0.622 | 2.861 | 0.529 | 0.625 |
| Glass Blur | .029 | 0.417 | 1.000 | 2.182 | 0.582 | 1.000 | .769 | 0.577 | 0.786 | .737 | 0.526 | 0.400 | .428 | 0.426 | 0.000 | 1.507 | 0.492 | 0.579 | 1.791 | 0.508 | 0.579 | 1.820 | 0.582 | 0.582 | 1.934 | 0.609 | 0.629 | 2.861 | 0.528 | 0.658 |
| Zoom Blur | .745 | 0.401 | 1.000 | 1.682 | 0.578 | 1.000 | .531 | 0.486 | 0.786 | .653 | 0.492 | 1.000 | .364 | 0.364 | 0.000 | 1.091 | 0.426 | 0.579 | 1.264 | 0.502 | 0.579 | 1.783 | 0.571 | 0.571 | 1.783 | 0.571 | 0.571 | 1.629 | 0.528 | 0.625 |
| Frame Drop | 1.102 | 0.443 | 0.333 | 2.318 | 0.575 | 1.000 | .687 | 0.575 | 0.786 | .874 | 0.529 | 0.600 | .402 | 0.437 | 0.000 | 1.640 | 0.496 | 0.789 | 2.011 | 0.519 | 0.789 | 1.981 | 0.631 | 0.631 | 2.123 | 0.599 | 0.571 | 1.246 | 0.486 | 0.599 |
| Frame Replace | .672 | 0.378 | 1.000 | 1.591 | 0.477 | 1.000 | .565 | 0.591 | 0.786 | .923 | 0.519 | 0.600 | .371 | 0.398 | 0.000 | 1.156 | 0.431 | 0.526 | 1.659 | 0.526 | 0.526 | 1.778 | 0.606 | 0.606 | 1.406 | 0.599 | 0.571 | 2.861 | 0.475 | 0.487 |
| Frame Repeat | .190 | 0.432 | 0.667 | 2.242 | 0.598 | 0.667 | .524 | 0.571 | 0.786 | .358 | 0.449 | 0.600 | .371 | 0.444 | 0.000 | 1.000 | 0.485 | 0.684 | 1.802 | 0.504 | 0.684 | 1.526 | 0.625 | 0.638 | 1.991 | 0.571 | 0.400 | 2.861 | 0.522 | 0.605 |
| Temporal Jitter | .978 | 0.410 | 0.667 | 1.924 | 0.580 | 1.000 | .565 | 0.583 | 0.643 | .958 | 0.539 | 0.800 | .308 | 0.425 | 0.000 | 1.742 | 0.466 | 0.737 | 1.978 | 0.507 | 0.737 | 1.895 | 0.636 | 0.625 | 2.068 | 0.596 | 0.571 | 1.676 | 0.528 | 0.638 |
| Jpeg Artifact | .168 | 0.454 | 1.000 | 2.273 | 0.581 | 1.000 | 1.728 | 0.540 | 0.786 | .958 | 0.535 | 0.600 | .441 | 0.432 | 1.000 | 1.382 | 0.502 | 0.737 | 2.099 | 0.502 | 0.737 | 2.043 | 0.611 | 0.611 | 2.104 | 0.578 | 0.574 | 1.722 | 0.574 | 0.605 |
| Bit Error | .109 | 0.442 | 1.000 | 1.939 | 0.568 | 1.000 | .626 | 0.565 | 0.786 | .726 | 0.490 | 0.800 | .441 | 0.422 | 1.000 | 1.516 | 0.472 | 0.684 | 1.703 | 0.501 | 0.789 | .765 | 0.632 | 0.624 | 2.028 | 0.574 | 0.571 | 1.433 | 0.574 | 0.605 |
| H265 Artifacts | 1.007 | 0.441 | 1.000 | 1.909 | 0.554 | 1.000 | .531 | 0.578 | 0.786 | .832 | 0.506 | 0.800 | .437 | 0.402 | 0.000 | 1.516 | 0.507 | 0.789 | 1.846 | 0.515 | 0.789 | 1.815 | 0.619 | 0.610 | 1.962 | 0.595 | 0.595 | 1.577 | 0.529 | 0.730 |
| Random Block | .964 | 0.430 | 1.000 | 2.242 | 0.567 | 1.000 | .687 | 0.577 | 0.786 | .905 | 0.541 | 0.600 | .507 | 0.437 | 0.000 | 1.154 | 0.456 | 0.632 | 1.714 | 0.531 | 0.632 | 1.870 | 0.617 | 0.617 | 2.028 | 0.592 | 0.592 | 1.635 | 0.527 | 0.625 |
| Target Block | .277 | 0.446 | 1.000 | 2.242 | 0.593 | 1.000 | .748 | 0.589 | 0.929 | 2.032 | 0.540 | 1.000 | .448 | 0.431 | 0.000 | .591 | 0.488 | 0.789 | 1.901 | 0.522 | 0.789 | 2.031 | 0.627 | 0.627 | 2.170 | 0.599 | 0.599 | 1.750 | 0.541 | 0.704 |
| Rolling Shutter | .015 | 0.391 | 1.000 | 2.136 | 0.586 | 1.000 | .667 | 0.580 | 0.714 | .768 | 0.501 | 0.800 | .448 | 0.402 | 0.071 | 1.462 | 0.494 | 0.789 | 1.923 | 0.474 | 0.789 | 1.963 | 0.630 | 0.571 | 1.934 | 0.571 | 0.714 | 1.566 | 0.571 | 0.625 |
| Resolution Degrade | .044 | 0.412 | 1.000 | 2.030 | 0.595 | 1.000 | .646 | 0.572 | 0.786 | .154 | 0.474 | 0.600 | .071 | 0.433 | 0.000 | .989 | 0.336 | 0.000 | 1.341 | 0.474 | 0.474 | .988 | 0.361 | 0.571 | .453 | 0.221 | 0.453 | 1.577 | 0.221 | 0.388 |
| Color Quantized | .745 | 0.451 | 1.000 | 1.682 | 0.611 | 1.000 | .531 | 0.571 | 0.786 | .653 | 0.537 | 0.800 | .415 | 0.402 | 0.000 | .579 | 0.489 | 0.579 | .579 | 0.579 | 0.579 | 2.049 | 0.629 | 0.571 | 2.094 | 0.583 | 0.571 | 1.729 | 0.393 | 0.599 |
| Bright Transform | .241 | 0.451 | 1.000 | 2.227 | 0.601 | 0.667 | .755 | 0.583 | 0.786 | .937 | 0.534 | 0.600 | .451 | 0.423 | 0.000 | .522 | 0.472 | 0.789 | 2.088 | 0.523 | 0.789 | 1.821 | 0.638 | 0.571 | 2.066 | 0.599 | 0.571 | 1.739 | 0.736 | 0.678 |
| Stretch Squish | .394 | 0.452 | 1.000 | 2.212 | 0.610 | 0.667 | .701 | 0.585 | 0.714 | .884 | 0.529 | 0.400 | .433 | 0.451 | 0.000 | 1.645 | 0.487 | 0.632 | 1.989 | 0.507 | 0.632 | 2.019 | 0.606 | 0.606 | 2.113 | 0.586 | 0.571 | 1.753 | 0.536 | 0.664 |
| Edge Sawtooth | .168 | 0.453 | 1.000 | 2.212 | 0.604 | 0.667 | .714 | 0.585 | 0.786 | 1.884 | 0.516 | 0.600 | .446 | 0.433 | 0.000 | 1.575 | 0.491 | 0.737 | 1.864 | 0.528 | 0.526 | 1.870 | 0.616 | 0.616 | 2.038 | 0.593 | 0.593 | 1.695 | 0.541 | 0.671 |
| Elastic | 1.153 | 0.450 | 1.000 | 1.924 | 0.608 | 0.333 | .667 | 0.574 | 0.714 | 2.000 | 0.522 | 0.800 | .454 | 0.446 | 0.000 | 1.710 | 0.491 | 0.842 | 2.176 | 0.491 | 0.842 | 1.864 | 0.622 | 0.621 | 2.047 | 0.599 | 0.571 | 1.676 | 0.537 | 0.684 |
| Contrast | .036 | 0.451 | 1.000 | 2.242 | 0.605 | 1.000 | .673 | 0.582 | 0.786 | .231 | 0.551 | 0.800 | .430 | 0.430 | 0.000 | .704 | 0.491 | 0.789 | 1.923 | 0.532 | 0.789 | 1.963 | 0.627 | 0.627 | 2.179 | 0.613 | 0.714 | 2.861 | 0.539 | 0.618 |
| Color Shift | .241 | 0.444 | 0.667 | 2.242 | 0.605 | 1.000 | .741 | 0.583 | 0.786 | 1.462 | 0.539 | 0.400 | .430 | 0.415 | 1.000 | .629 | 0.520 | 0.789 | 2.033 | 0.520 | 0.789 | 1.889 | 0.568 | 0.571 | 2.104 | 0.568 | 0.429 | 1.706 | 0.536 | 0.632 |
| Flicker | 1.131 | 0.440 | 1.000 | 3.076 | 0.572 | 0.667 | .714 | 0.572 | 0.786 | 1.308 | 0.534 | 0.600 | .423 | 0.423 | 0.000 | .581 | 0.487 | 0.632 | 2.011 | 0.522 | 0.632 | 1.981 | 0.626 | 0.626 | 2.123 | 0.604 | 0.536 | 1.736 | 0.536 | 0.678 |
| Overexposure | .124 | 0.451 | 1.000 | 2.182 | 0.582 | 0.667 | .755 | 0.585 | 0.786 | .979 | 0.531 | 0.800 | .501 | 0.501 | 0.000 | .522 | 0.502 | 0.632 | 2.044 | 0.528 | 0.632 | 1.827 | 0.626 | 0.610 | 2.047 | 0.586 | 0.429 | 1.716 | 0.537 | 0.651 |
| Underexposure | .066 | 0.438 | 1.000 | 2.242 | 0.605 | 1.000 | .667 | 0.574 | 0.714 | 1.231 | 0.533 | 0.600 | .454 | 0.430 | 0.000 | .575 | 0.491 | 0.737 | 2.176 | 0.526 | 0.737 | 1.864 | 0.622 | 0.621 | 2.038 | 0.593 | 0.714 | 1.678 | 0.539 | 0.684 |
| Rainy | .066 | 0.428 | 1.000 | 2.182 | 0.627 | 0.667 | .673 | 0.582 | 0.714 | 2.053 | 0.539 | 0.400 | .462 | 0.430 | 0.000 | 1.704 | 0.491 | 0.842 | 1.923 | 0.526 | 0.842 | 1.963 | 0.621 | 0.626 | 2.179 | 0.613 | 0.714 | 1.762 | 0.539 | 0.618 |
| Foggy | 1.131 | 0.440 | 1.000 | 3.076 | 0.572 | 0.667 | .714 | 0.582 | 0.786 | .979 | 0.534 | 0.600 | .501 | 0.423 | 1.000 | .581 | 0.501 | 0.789 | 2.011 | 0.522 | 0.789 | 1.889 | 0.568 | 0.626 | 2.104 | 0.568 | 0.429 | 1.706 | 0.336 | 0.632 |
| Snow | .124 | 0.453 | 1.000 | 2.212 | 0.601 | 0.667 | .755 | 0.585 | 0.786 | 1.308 | 0.534 | 0.600 | .423 | 0.423 | 0.000 | .645 | 0.487 | 0.632 | 2.011 | 0.522 | 0.632 | 1.981 | 0.626 | 0.626 | 2.123 | 0.604 | 0.800 | 1.738 | 0.653 | 0.678 |
| Frost | .066 | 0.438 | 1.000 | 2.182 | 0.581 | 0.667 | .701 | 0.571 | 0.786 | 1.077 | 0.534 | 0.600 | .501 | 0.501 | 0.000 | .522 | 0.487 | 0.632 | 1.934 | 0.528 | 0.632 | 1.870 | 0.626 | 0.626 | 2.057 | 0.599 | 0.600 | 1.653 | 0.534 | 0.638 |
| Reflect | .204 | 0.459 | 1.000 | 2.242 | 0.591 | 1.000 | .782 | 0.585 | 0.857 | 1.947 | 0.531 | 0.800 | .438 | 0.438 | 0.000 | .538 | 0.489 | 0.632 | 2.033 | 0.511 | 0.632 | 1.938 | 0.599 | 0.599 | 2.028 | 0.570 | 0.600 | 1.719 | 0.533 | 0.618 |
| Shadow | 1.073 | 0.439 | 1.000 | 2.212 | 0.613 | 1.000 | .687 | 0.574 | 0.714 | 1.874 | 0.533 | 0.600 | .454 | 0.454 | 0.000 | .602 | 0.500 | 0.737 | 2.099 | 0.525 | 0.737 | 2.074 | 0.626 | 0.626 | 2.075 | 0.592 | 0.800 | 1.731 | 0.538 | 0.678 |

Note: A: Advertisements, B: Algorithm & Models, C: Autos & Vehicles, D: Business & Industrial, E: Computers & Electronics, F: Fairy Tale, G: Films & TV Shows, H: Finance, I: Food & Drink, J: Games, K: Humor, L: Instruction Video (how to), M: Knowledge, N: News, O: Others, P: People, Q: Pets & Animals, R: Science, S: Sports, T: Overall

Table 5: LLAMA-VID performance across different video types

| Noise Type | A GPT | A SBERT | A Acc | B GPT | B SBERT | B Acc | C GPT | C SBERT | C Acc | D GPT | D SBERT | D Acc | E GPT | E SBERT | E Acc | F GPT | F SBERT | F Acc | G GPT | G SBERT | G Acc | H GPT | H SBERT | H Acc | J GPT | J SBERT | J Acc |
|---|---|---|---|---|---|---|---|---|---|---|---|---|---|---|---|---|---|---|---|---|---|---|---|---|---|---|---|
| Clean | 0.992 | 0.417 | 0.167 | 0.625 | 0.341 | 0.328 | 1.134 | 0.440 | 0.923 | 1.88 | 0.363 | 0.222 | 0.927 | 0.359 | 0.278 | 0.500 | 0.478 | — | 0.770 | 0.481 | 0.281 | 1.038 | 0.486 | 0.333 | 0.756 | 0.544 | 0.875 | 1.000 |
| Gaussian | 0.854 | 0.416 | 0.333 | 0.625 | 0.328 | | 1.143 | 0.443 | 0.923 | 0.900 | 0.367 | 0.222 | 0.827 | 0.348 | 0.278 | 0.500 | 0.486 | | 0.728 | 0.481 | 0.281 | 0.850 | 0.472 | 0.333 | 0.578 | 0.548 | 0.875 | 1.000 |
| Impulse | 0.772 | 0.397 | | 0.125 | 0.345 | | 1.134 | 0.448 | 0.923 | 0.764 | 0.354 | 0.222 | 0.764 | 0.337 | 0.278 | 0.750 | 0.482 | | 0.651 | 0.486 | 0.281 | 0.938 | 0.471 | 0.333 | 0.525 | 0.525 | 0.750 | 1.000 |
| Speckle | 0.837 | 0.407 | | 0.750 | 0.331 | | 1.152 | 0.441 | 0.923 | 0.909 | 0.367 | 0.222 | 0.808 | 0.356 | 0.278 | 0.750 | 0.496 | | 0.753 | 0.472 | 0.281 | 0.912 | 0.477 | 0.333 | 0.544 | 0.544 | 0.812 | 1.000 |
| Poisson | 0.780 | 0.403 | | 0.205 | 0.439 | | 0.736 | 0.344 | 0.222 | 0.740 | 0.347 | 0.278 | 0.482 | | 0.706 | 0.471 | 0.281 | 0.912 | 0.473 | 0.333 | 0.600 | 0.522 | 0.812 | 1.000 | | | | |
| Gaussian Blur | 0.610 | 0.391 | | 1.071 | | | 0.709 | 0.335 | 0.709 | 0.367 | 0.342 | 0.278 | 0.479 | | 0.626 | 0.477 | 0.281 | 1.038 | 0.485 | 0.333 | 0.600 | 0.538 | 0.812 | 1.000 | | | |
| Motion Blur | 0.911 | 0.415 | | 1.000 | | | 1.125 | 0.440 | 0.936 | 0.918 | 0.367 | 0.222 | 0.721 | 0.345 | 0.278 | 0.478 | | 0.745 | 0.477 | 0.281 | 0.900 | 0.456 | 0.333 | 0.478 | 0.518 | 0.812 | 1.000 |
| Defocus Blur | 0.829 | 0.405 | | 0.750 | 0.321 | | 1.250 | 0.443 | 0.923 | 0.682 | 0.352 | 0.222 | 0.702 | 0.352 | 0.278 | 0.502 | | 0.723 | 0.482 | 0.281 | 1.012 | 0.479 | 0.333 | 0.733 | 0.540 | 0.875 | 1.000 |
| Glass Blur | 0.837 | 0.399 | | 1.125 | 0.334 | | 1.205 | 0.447 | 1.000 | 0.606 | 0.360 | 0.278 | 0.606 | 0.338 | 0.278 | 0.486 | | 0.638 | 0.474 | 0.281 | 0.912 | 0.470 | 0.333 | 0.678 | 0.530 | 0.875 | 1.000 |
| Zoom Blur | 0.740 | 0.395 | | 1.125 | 0.322 | | 1.018 | 0.434 | 0.923 | 0.712 | 0.358 | 0.222 | 0.712 | 0.352 | 0.278 | 0.491 | | 0.638 | 0.462 | 0.281 | 0.713 | 0.451 | 0.333 | 0.478 | 0.520 | 0.875 | 1.000 |
| Frame Drop | 0.593 | 0.421 | | 0.625 | 0.334 | | 0.938 | 0.412 | 0.923 | 0.942 | 0.360 | 0.278 | 0.596 | 0.336 | 0.278 | 0.500 | | 0.809 | 0.484 | 0.281 | 0.992 | 0.451 | 0.333 | 0.756 | 0.521 | 0.875 | 1.000 |
| Frame Replace | 0.593 | 0.364 | 0.333 | 0.875 | 0.350 | | 1.116 | 0.437 | 0.923 | 0.718 | 0.368 | 0.278 | 0.740 | 0.365 | 0.278 | 0.500 | | 0.570 | 0.463 | 0.281 | 0.812 | 0.472 | 0.333 | 0.456 | 0.538 | 0.875 | 1.000 |
| Frame Repeat | 0.821 | 0.401 | | 0.750 | 0.322 | | 1.036 | 0.412 | 0.923 | 0.900 | 0.361 | 0.222 | 0.913 | 0.355 | 0.278 | 0.497 | | 0.719 | 0.463 | 0.281 | 0.875 | 0.471 | 0.333 | 0.667 | 0.540 | 0.875 | 1.000 |
| Temporal Jitter | 0.846 | 0.406 | | 1.000 | 0.331 | | 1.134 | 0.437 | 0.923 | 0.800 | 0.368 | 0.222 | 0.846 | 0.360 | 0.278 | 0.493 | | 0.694 | 0.478 | 0.281 | 0.975 | 0.482 | 0.333 | 0.756 | 0.539 | 0.875 | 1.000 |
| Jpeg Artifact | 0.740 | 0.404 | | 0.875 | 0.348 | | 1.134 | 0.451 | 0.923 | 0.900 | 0.366 | 0.222 | 0.913 | 0.357 | 0.278 | 0.496 | | 0.762 | 0.481 | 0.281 | 1.025 | 0.469 | 0.333 | 0.667 | 0.546 | 0.875 | 1.000 |
| Bit Error | 0.691 | 0.386 | | 0.500 | 0.342 | | 1.134 | 0.444 | 0.923 | 0.845 | 0.360 | 0.222 | 0.702 | 0.339 | 0.278 | 0.484 | | 0.664 | 0.466 | 0.281 | 1.050 | 0.469 | 0.333 | 0.633 | 0.525 | 0.875 | 1.000 |
| H265 Artifacts | 0.813 | 0.405 | | 0.875 | 0.322 | | 1.045 | 0.441 | 0.923 | 0.882 | 0.354 | 0.222 | 0.837 | 0.353 | 0.278 | 0.478 | | 0.766 | 0.480 | 0.281 | 0.863 | 0.477 | 0.333 | 0.667 | 0.531 | 0.875 | 1.000 |
| Random Block | 0.602 | 0.380 | | 1.500 | 0.370 | | 1.098 | 0.440 | 0.923 | 0.918 | 0.371 | 0.222 | 0.894 | 0.364 | 0.278 | 0.489 | | 0.791 | 0.478 | 0.281 | 0.925 | 0.471 | 0.333 | 0.611 | 0.532 | 0.875 | 1.000 |
| Target Block | 0.886 | 0.413 | | 0.875 | 0.353 | | 1.170 | 0.436 | 0.923 | 0.936 | 0.366 | 0.222 | 0.740 | 0.341 | 0.278 | 0.487 | | 0.689 | 0.478 | 0.281 | 0.850 | 0.467 | 0.333 | 0.611 | 0.544 | 0.875 | 1.000 |
| Rolling Shutter | 0.919 | 0.401 | | 1.125 | 0.351 | | 1.152 | 0.447 | 0.923 | 0.782 | 0.358 | 0.222 | 0.654 | 0.346 | 0.278 | 0.503 | | 0.766 | 0.485 | 0.281 | 0.925 | 0.467 | 0.333 | 0.678 | 0.543 | 0.875 | 1.000 |
| Resolution Degrade | 0.911 | 0.413 | | 0.625 | 0.342 | | 1.152 | 0.444 | 0.923 | 0.791 | 0.363 | 0.278 | 0.769 | 0.352 | 0.278 | 0.476 | | 0.723 | 0.483 | 0.281 | 1.113 | 0.498 | 0.333 | 0.711 | 0.540 | 0.875 | 1.000 |
| Stretch Squish | 0.984 | 0.412 | | 1.125 | 0.321 | | 1.188 | 0.447 | 0.923 | 0.909 | 0.365 | 0.222 | 0.779 | 0.359 | 0.278 | 0.494 | | 0.787 | 0.481 | 0.281 | 1.038 | 0.489 | 0.333 | 0.767 | 0.543 | 0.875 | 1.000 |
| Edge Sawtooth | 0.935 | 0.422 | 0.333 | 1.000 | 0.326 | | 1.196 | 0.444 | 0.923 | 0.918 | 0.366 | 0.222 | 0.873 | 0.364 | 0.278 | 0.499 | | 0.757 | 0.480 | 0.281 | 1.012 | 0.489 | 0.333 | 0.711 | 0.539 | 0.875 | 1.000 |
| Elastic | 0.943 | 0.416 | | 0.625 | 0.344 | | 1.134 | 0.447 | 0.923 | 0.955 | 0.367 | 0.222 | 0.885 | 0.363 | 0.278 | 0.478 | | 0.732 | 0.478 | 0.281 | 1.012 | 0.489 | 0.333 | 0.789 | 0.532 | 0.875 | 1.000 |
| Color Quantized | 1.057 | 0.420 | | 1.000 | 0.337 | | 1.134 | 0.436 | 0.923 | 0.936 | 0.361 | 0.222 | 0.740 | 0.346 | 0.278 | 0.489 | | 0.745 | 0.481 | 0.281 | 1.000 | 0.474 | 0.333 | 0.711 | 0.548 | 0.875 | 1.000 |
| Bright Transform | 0.911 | 0.411 | | 1.000 | 0.375 | | 1.125 | 0.436 | 0.923 | 0.891 | 0.361 | 0.222 | 0.856 | 0.355 | 0.278 | 0.497 | | 0.800 | 0.479 | 0.281 | 1.000 | 0.482 | 0.333 | 0.744 | 0.553 | 0.875 | 1.000 |
| Contrast | 0.976 | 0.413 | | 1.000 | 0.326 | | 1.232 | 0.453 | 0.923 | 0.845 | 0.360 | 0.222 | 0.827 | 0.356 | 0.278 | 0.492 | | 0.723 | 0.477 | 0.281 | 0.938 | 0.475 | 0.333 | 0.711 | 0.545 | 0.875 | 1.000 |
| Color Shift | 0.862 | 0.399 | | 0.750 | 0.325 | | 1.250 | 0.442 | 0.923 | 0.845 | 0.357 | 0.222 | 0.865 | 0.360 | 0.278 | 0.502 | | 0.762 | 0.488 | 0.281 | 1.038 | 0.469 | 0.333 | 0.689 | 0.547 | 0.875 | 1.000 |
| Flicker | 0.894 | 0.402 | | 1.000 | 0.321 | | 1.143 | 0.436 | 0.923 | 0.909 | 0.357 | 0.222 | 0.817 | 0.357 | 0.278 | 0.496 | | 0.719 | 0.483 | 0.281 | 0.988 | 0.481 | 0.333 | 0.700 | 0.550 | 0.875 | 1.000 |
| Overexposure | 0.797 | 0.408 | | 0.750 | 0.338 | | 1.107 | 0.440 | 0.923 | 0.827 | 0.363 | 0.222 | 0.875 | 0.351 | 0.278 | 0.502 | | 0.621 | 0.480 | 0.281 | 0.988 | 0.481 | 0.333 | 0.600 | 0.550 | 0.875 | 1.000 |
| Underexposure | 0.911 | 0.410 | | 0.750 | 0.333 | | 1.152 | 0.446 | 0.923 | 0.918 | 0.364 | 0.222 | 0.750 | 0.363 | 0.278 | 0.494 | | 0.779 | 0.486 | 0.281 | 0.925 | 0.475 | 0.333 | 0.667 | 0.550 | 0.875 | 1.000 |
| Shadow | 0.886 | 0.409 | | 0.625 | 0.338 | | 1.161 | 0.448 | 0.923 | 0.882 | 0.363 | 0.222 | 0.779 | 0.359 | 0.278 | 0.485 | | 0.723 | 0.482 | 0.281 | 1.012 | 0.486 | 0.333 | 0.667 | 0.546 | 0.875 | 1.000 |

| Noise Type | K GPT | K SBERT | K Acc | L GPT | L SBERT | M GPT | M SBERT | M Acc | N GPT | N SBERT | N Acc | O GPT | O SBERT | P GPT | P SBERT | P Acc | Q GPT | Q SBERT | R GPT | R SBERT | S GPT | S SBERT | S Acc | T GPT | T SBERT | T Acc |
|---|---|---|---|---|---|---|---|---|---|---|---|---|---|---|---|---|---|---|---|---|---|---|---|---|---|---|
| Clean | 0.569 | 0.347 | 0.667 | 0.288 | 0.409 | 1.000 | 0.755 | 0.510 | 0.726 | 0.319 | 0.286 | 0.600 | 0.846 | 0.365 | 0.000 | 0.790 | 0.405 | 0.211 | 1.077 | 0.407 | 0.877 | 0.546 | 0.283 | 0.457 | 0.571 | 0.407 | 0.435 | 0.447 |
| Gaussian | 0.526 | 0.336 | 0.667 | 0.273 | 0.407 | 1.000 | 0.721 | 0.509 | 0.674 | 0.312 | 0.214 | 0.846 | 0.322 | 0.780 | 0.403 | 1.165 | 0.412 | 0.883 | 0.549 | 0.368 | 0.463 | 0.571 | 0.874 | 0.433 | 0.421 |
| Impulse | 0.409 | 0.336 | 0.667 | 0.273 | 0.394 | 1.000 | 0.619 | 0.504 | 0.642 | 0.297 | 0.214 | 0.615 | 0.316 | 0.683 | 0.391 | 1.088 | 0.403 | 0.858 | 0.541 | 0.292 | 0.462 | 0.571 | 0.810 | 0.434 | 0.414 |
| Speckle | 0.460 | 0.343 | 0.667 | 0.364 | 0.415 | 1.000 | 0.741 | 0.510 | 0.589 | 0.306 | 0.214 | 0.846 | 0.336 | 0.731 | 0.400 | 1.055 | 0.406 | 0.802 | 0.547 | 0.311 | 0.468 | 0.571 | 0.854 | 0.434 | 0.421 |
| Poisson | 0.562 | 0.334 | 0.667 | 0.182 | 0.406 | 1.000 | 0.755 | 0.504 | 0.663 | 0.303 | 0.214 | 0.615 | 0.342 | 0.747 | 0.399 | 1.077 | 0.406 | 0.914 | 0.547 | 0.311 | 0.450 | 0.571 | 0.858 | 0.424 | 0.421 |
| Gaussian Blur | 0.460 | 0.337 | 0.667 | 0.379 | 0.391 | 1.000 | 0.741 | 0.509 | 0.674 | 0.315 | 0.286 | 0.846 | 0.331 | 0.758 | 0.404 | 1.209 | 0.399 | 0.852 | 0.545 | 0.208 | 0.454 | 0.571 | 0.832 | 0.434 | 0.441 |
| Motion Blur | 0.409 | 0.330 | 0.667 | 0.182 | 0.388 | 1.000 | 0.694 | 0.511 | 0.726 | 0.323 | 0.214 | 0.769 | 0.333 | 0.640 | 0.390 | 0.989 | 0.404 | 0.815 | 0.546 | 0.264 | 0.464 | 0.571 | 0.772 | 0.419 | 0.441 |
| Defocus Blur | 0.467 | 0.341 | 0.667 | 0.364 | 0.403 | 1.000 | 0.748 | 0.517 | 0.726 | 0.305 | 0.214 | 0.769 | 0.333 | 0.774 | 0.397 | 1.132 | 0.408 | 0.827 | 0.551 | 0.330 | 0.456 | 0.571 | 0.885 | 0.432 | 0.434 |
| Glass Blur | 0.401 | 0.338 | 0.667 | 0.258 | 0.398 | 1.000 | 0.694 | 0.515 | 0.695 | 0.306 | 0.286 | 0.923 | 0.319 | 0.677 | 0.397 | 1.055 | 0.417 | 0.827 | 0.545 | 0.208 | 0.456 | 0.571 | 0.876 | 0.433 | 0.428 |
| Zoom Blur | 0.401 | 0.321 | 0.667 | 0.273 | 0.398 | 1.000 | 0.748 | 0.502 | 0.726 | 0.316 | 0.286 | 0.769 | 0.308 | 0.704 | 0.405 | 1.022 | 0.405 | 0.926 | 0.548 | 0.170 | 0.459 | 0.571 | 0.814 | 0.427 | 0.428 |
| Frame Drop | 0.606 | 0.353 | 0.667 | 0.273 | 0.420 | 1.000 | 0.578 | 0.508 | 0.737 | 0.316 | 0.286 | 0.692 | 0.343 | 0.704 | 0.393 | 1.088 | 0.413 | 0.815 | 0.551 | 0.226 | 0.457 | 0.571 | 0.765 | 0.435 | 0.414 |
| Frame Replace | 0.321 | 0.313 | 0.667 | 0.182 | 0.411 | 1.000 | 0.796 | 0.510 | 0.285 | 0.307 | 0.769 | 0.304 | 0.629 | 0.379 | 1.077 | 0.396 | 0.784 | 0.535 | 0.953 | 0.436 | 0.571 | 0.724 | 0.410 | 0.434 |
| Frame Repeat | 0.526 | 0.332 | 0.667 | 0.182 | 0.411 | 1.000 | 0.728 | 0.513 | 0.737 | 0.320 | 0.214 | 0.615 | 0.308 | 0.683 | 0.394 | 1.154 | 0.408 | 0.870 | 0.548 | 0.292 | 0.458 | 0.571 | 0.865 | 0.430 | 0.434 |
| Temporal Jitter | 0.518 | 0.344 | 0.667 | 0.303 | 0.408 | 1.000 | 0.803 | 0.513 | 0.674 | 0.319 | 0.214 | 0.923 | 0.338 | 0.763 | 0.400 | 1.121 | 0.416 | 0.858 | 0.549 | 0.292 | 0.458 | 0.571 | 0.855 | 0.435 | 0.441 |
| Jpeg Artifact | 0.533 | 0.344 | 0.667 | 0.379 | 0.388 | 1.000 | 0.803 | 0.512 | 0.632 | 0.314 | 0.286 | 0.846 | 0.334 | 0.769 | 0.400 | 1.110 | 0.403 | 0.852 | 0.544 | 0.274 | 0.455 | 0.571 | 0.904 | 0.432 | 0.441 |
| Bit Error | 0.358 | 0.338 | 0.667 | 0.182 | 0.403 | 1.000 | 0.810 | 0.517 | 0.695 | 0.311 | 0.214 | 0.769 | 0.332 | 0.634 | 0.387 | 1.121 | 0.408 | 0.852 | 0.556 | 0.302 | 0.469 | 0.571 | 0.806 | 0.422 | 0.428 |
| H265 Artifacts | 0.540 | 0.338 | 0.667 | 0.258 | 0.403 | 1.000 | 0.844 | 0.512 | 0.726 | 0.312 | 0.286 | 0.846 | 0.332 | 0.629 | 0.397 | 1.110 | 0.405 | 0.778 | 0.548 | 0.208 | 0.455 | 0.571 | 0.831 | 0.428 | 0.428 |
| Random Block | 0.343 | 0.328 | 0.667 | 0.136 | 0.394 | 1.000 | 0.646 | 0.517 | 0.621 | 0.300 | 0.214 | 0.615 | 0.335 | 0.774 | 0.397 | 1.000 | 0.417 | 0.815 | 0.548 | 0.179 | 0.451 | 0.571 | 0.824 | 0.434 | 0.428 |
| Target Block | 0.504 | 0.343 | 0.667 | 0.152 | 0.393 | 1.000 | 0.810 | 0.509 | 0.716 | 0.315 | 0.214 | 0.785 | 0.323 | 0.785 | 0.401 | 1.066 | 0.413 | 0.784 | 0.551 | 0.217 | 0.457 | 0.571 | 0.857 | 0.432 | 0.434 |
| Rolling Shutter | 0.518 | 0.343 | 0.667 | 0.273 | 0.404 | 1.000 | 0.837 | 0.514 | 0.695 | 0.323 | 0.286 | 0.769 | 0.337 | 0.828 | 0.407 | 1.099 | 0.406 | 0.784 | 0.547 | 0.200 | 0.463 | 0.571 | 0.824 | 0.434 | 0.434 |
| Resolution Degrade | 0.526 | 0.343 | 0.667 | 0.242 | 0.407 | 1.000 | 0.707 | 0.510 | 0.692 | 0.310 | 0.286 | 0.806 | 0.325 | 0.785 | 0.404 | 1.132 | 0.407 | 0.790 | 0.554 | 0.236 | 0.463 | 0.571 | 0.893 | 0.435 | 0.441 |
| Stretch Squish | 0.511 | 0.336 | 0.667 | 0.379 | 0.411 | 1.000 | 0.769 | 0.511 | 0.589 | 0.323 | 0.286 | 0.846 | 0.328 | 0.817 | 0.409 | 1.099 | 0.407 | 0.790 | 0.546 | 0.292 | 0.463 | 0.571 | 0.902 | 0.435 | 0.441 |
| Edge Sawtooth | 0.562 | 0.346 | 0.667 | 0.091 | 0.408 | 1.000 | 0.707 | 0.513 | 0.747 | 0.324 | 0.214 | 0.692 | 0.347 | 0.849 | 0.402 | 1.077 | 0.413 | 0.728 | 0.541 | 0.302 | 0.466 | 0.571 | 0.909 | 0.433 | 0.441 |
| Elastic | 0.460 | 0.344 | 0.667 | 0.212 | 0.398 | 1.000 | 0.803 | 0.515 | 0.716 | 0.313 | 0.286 | 0.800 | 0.339 | 0.774 | 0.402 | 1.011 | 0.403 | 0.901 | 0.550 | 0.217 | 0.457 | 0.571 | 0.908 | 0.434 | 0.434 |
| Color Quantized | 0.504 | 0.339 | 0.667 | 0.318 | 0.409 | 1.000 | 0.789 | 0.516 | 0.663 | 0.321 | 0.214 | 0.923 | 0.344 | 0.774 | 0.416 | 1.088 | 0.412 | 0.833 | 0.550 | 0.236 | 0.459 | 0.571 | 0.881 | 0.441 | 0.441 |
| Bright Transform | 0.460 | 0.336 | 0.667 | 0.348 | 0.419 | 1.000 | 0.701 | 0.513 | 0.663 | 0.316 | 0.286 | 0.923 | 0.351 | 0.790 | 0.403 | 1.055 | 0.403 | 0.796 | 0.547 | 0.255 | 0.461 | 0.571 | 0.871 | 0.434 | 0.441 |
| Contrast | 0.431 | 0.344 | 0.667 | 0.379 | 0.410 | 1.000 | 0.884 | 0.516 | 0.695 | 0.321 | 0.286 | 0.923 | 0.351 | 0.860 | 0.404 | 1.055 | 0.405 | 0.735 | 0.549 | 0.274 | 0.466 | 0.571 | 0.904 | 0.436 | 0.434 |
| Color Shift | 0.489 | 0.342 | 0.667 | 0.303 | 0.410 | 1.000 | 0.687 | 0.509 | 0.663 | 0.315 | 0.286 | 0.769 | 0.335 | 0.672 | 0.396 | 1.099 | 0.399 | 0.821 | 0.539 | 0.274 | 0.466 | 0.571 | 0.821 | 0.429 | 0.428 |
| Reflect | 0.489 | 0.338 | 0.667 | 0.303 | 0.407 | 1.000 | 0.803 | 0.516 | 0.663 | 0.321 | 0.286 | 0.769 | 0.364 | 0.742 | 0.396 | 1.088 | 0.413 | 0.802 | 0.549 | 0.283 | 0.465 | 0.571 | 0.878 | 0.435 | 0.441 |
| Shadow | 0.562 | 0.342 | 0.667 | 0.455 | 0.423 | 1.000 | 0.741 | 0.514 | 0.653 | 0.314 | 0.286 | 0.753 | 0.327 | 0.753 | 0.400 | 1.088 | 0.411 | 0.802 | 0.548 | 0.283 | 0.468 | 0.571 | 0.879 | 0.441 | 0.441 |

Note: A: Advertisements, B: Algorithm & Models, C: Autos & Vehicles, D: Business & Industrial, E: Computers & Electronics, F: Fairy Tale, G: Films & TV Shows, H: Finance, I: Food & Drink, J: Games, K: Humor, L: Instruction Video (how to), M: Knowledge, N: News, O: Others, P: People, Q: Pets & Animals, R: Science, S: Sports, T: Overall

Table 6: PLLaVA 13B performance across different video types

| Noise Type | A GPT | A SBERT | A Acc | B GPT | B SBERT | B Acc | C GPT | C SBERT | C Acc | D GPT | D SBERT | D Acc | E GPT | E SBERT | E Acc | F GPT | F SBERT | F Acc | G GPT | G SBERT | G Acc | H GPT | H SBERT | H Acc | I GPT | I SBERT | I Acc | J GPT | J SBERT | J Acc |
|---|---|---|---|---|---|---|---|---|---|---|---|---|---|---|---|---|---|---|---|---|---|---|---|---|---|---|---|---|---|---|
| Clean | 0.984 | 0.414 | 0.167 | 1.125 | 0.345 | – | 0.973 | 0.435 | 0.385 | 0.832 | 0.344 | 0.143 | 0.371 | 0.353 | 0.000 | 1.096 | 0.343 | 0.000 | 0.719 | 0.469 | 0.500 | 1.163 | 0.470 | 0.500 | 0.800 | 0.188 | 0.250 | 0.756 | 0.326 | 1.000 |
| Gaussian | 0.675 | 0.394 | 0.167 | 0.875 | 0.418 | – | 0.782 | 0.425 | 0.385 | 0.789 | 0.323 | 0.143 | 0.353 | 0.338 | 0.000 | 0.856 | 0.247 | 0.000 | 0.536 | 0.455 | 0.500 | 1.212 | 0.475 | 0.500 | 0.922 | 0.490 | 0.250 | 0.633 | 0.313 | 1.000 |
| Impulse | 0.504 | 0.322 | 0.000 | 1.125 | 0.379 | – | 0.875 | 0.423 | 0.385 | 0.621 | 0.310 | 0.071 | 0.338 | 0.344 | 0.000 | 0.702 | 0.259 | 0.000 | 0.626 | 0.445 | 0.500 | 1.038 | 0.463 | 0.500 | 0.704 | 0.492 | 0.188 | 0.722 | 0.319 | 1.000 |
| Speckle | 0.715 | 0.391 | 0.167 | 0.750 | 0.408 | – | 1.036 | 0.433 | 0.385 | 0.737 | 0.324 | 0.071 | 0.344 | 0.342 | 0.000 | 0.798 | 0.287 | 0.000 | 0.604 | 0.451 | 0.500 | 1.363 | 0.482 | 0.500 | 0.822 | 0.481 | 0.188 | 0.644 | 0.308 | 1.000 |
| Poisson | 0.724 | 0.382 | 0.167 | 0.375 | 0.398 | – | 1.000 | 0.430 | 0.385 | 0.663 | 0.275 | 0.143 | 0.342 | 0.348 | 0.000 | 0.856 | 0.275 | 0.000 | 0.451 | 0.427 | 0.500 | 0.875 | 0.449 | 0.500 | 0.690 | 0.463 | 0.125 | 0.589 | 0.301 | 1.000 |
| Gaussian Blur | 0.520 | 0.349 | 0.167 | 1.000 | 0.372 | – | 0.768 | 0.404 | 0.308 | 0.347 | 0.290 | 0.143 | 0.326 | 0.326 | 0.000 | 0.673 | 0.333 | 0.000 | 0.485 | 0.456 | 0.500 | 1.200 | 0.461 | 0.500 | 0.721 | 0.455 | 0.125 | 0.567 | 0.303 | 1.000 |
| Motion Blur | 0.382 | 0.338 | 0.167 | 0.500 | 0.287 | – | 0.946 | 0.384 | 0.308 | 0.505 | 0.328 | 0.143 | 0.345 | 0.345 | 0.000 | 0.609 | 0.232 | 0.000 | 0.728 | 0.450 | 0.500 | 1.000 | 0.451 | 0.500 | 0.891 | 0.496 | 0.250 | 0.544 | 0.303 | 1.000 |
| Defocus Blur | 0.846 | 0.405 | 0.167 | 0.500 | 0.303 | – | 0.938 | 0.419 | 0.385 | 0.853 | 0.342 | 0.143 | 0.352 | 0.329 | 0.000 | 0.827 | 0.333 | 0.000 | 0.732 | 0.456 | 0.500 | 1.175 | 0.482 | 0.500 | 0.845 | 0.485 | 0.125 | 0.633 | 0.311 | 1.000 |
| Glass Blur | 0.813 | 0.399 | 0.167 | 0.375 | 0.299 | – | 0.866 | 0.397 | 0.385 | 0.684 | 0.327 | 0.071 | 0.329 | 0.345 | 0.000 | 0.769 | 0.294 | 0.000 | 0.523 | 0.457 | 0.333 | 1.163 | 0.462 | 0.500 | 0.628 | 0.457 | 0.250 | 0.567 | 0.305 | 1.000 |
| Zoom Blur | 0.585 | 0.361 | 0.167 | 0.500 | 0.299 | – | 0.875 | 0.397 | 0.308 | 0.505 | 0.320 | 0.143 | 0.367 | 0.352 | 0.000 | 0.625 | 0.286 | 0.000 | 0.455 | 0.429 | 0.333 | 0.887 | 0.472 | 0.167 | 0.600 | 0.491 | 0.125 | 0.600 | 0.305 | 1.000 |
| Frame Drop | 0.374 | 0.339 | 0.167 | 1.000 | 0.323 | – | 0.714 | 0.308 | 0.385 | 0.295 | 0.287 | 0.071 | 0.367 | 0.333 | 0.000 | 0.644 | 0.286 | 0.000 | 0.416 | 0.416 | 0.500 | 0.887 | 0.432 | 0.333 | 0.589 | 0.484 | 0.250 | 0.533 | 0.312 | 1.000 |
| Frame Replace | 0.902 | 0.407 | 0.167 | 0.875 | 0.299 | – | 1.071 | 0.431 | 0.231 | 0.884 | 0.348 | 0.143 | 0.356 | 0.367 | 0.000 | 0.933 | 0.279 | 0.000 | 0.455 | 0.473 | 0.333 | 1.288 | 0.452 | 0.333 | 1.000 | 0.484 | 0.250 | 0.722 | 0.281 | 1.000 |
| Frame Repeat | 0.228 | 0.310 | 0.000 | 1.000 | 0.303 | – | 0.714 | 0.391 | 0.231 | 0.474 | 0.314 | 0.071 | 0.324 | 0.356 | 0.000 | 0.536 | 0.271 | 0.000 | 0.438 | 0.451 | 0.333 | 1.075 | 0.433 | 0.333 | 0.891 | 0.484 | 0.250 | 0.500 | 0.314 | 1.000 |
| Temporal Jitter | 0.561 | 0.382 | 0.000 | 0.750 | 0.296 | – | 1.062 | 0.434 | 0.385 | 0.579 | 0.331 | 0.143 | 0.359 | 0.359 | 0.000 | 0.769 | 0.204 | 0.000 | 0.749 | 0.451 | 0.452 | 1.225 | 0.470 | 0.500 | 0.876 | 0.478 | 0.219 | 0.667 | 0.314 | 1.000 |
| Jpeg Artifact | 0.683 | 0.389 | 0.167 | 0.625 | 0.296 | – | 0.991 | 0.417 | 0.385 | 0.832 | 0.336 | 0.071 | 0.336 | 0.358 | 0.000 | 0.750 | 0.226 | 0.000 | 0.638 | 0.452 | 0.500 | 1.113 | 0.478 | 0.500 | 0.535 | 0.453 | 0.094 | 0.700 | 0.314 | 1.000 |
| Bit Error | 0.756 | 0.389 | 0.167 | 0.750 | 0.304 | – | 1.027 | 0.423 | 0.385 | 0.726 | 0.318 | 0.143 | 0.359 | 0.367 | 0.000 | 0.635 | 0.317 | 0.000 | 0.481 | 0.446 | 0.500 | 0.975 | 0.453 | 0.500 | 0.822 | 0.476 | 0.125 | 0.489 | 0.286 | 1.000 |
| H265 Artifacts | 0.472 | 0.385 | 0.167 | 0.625 | 0.322 | – | 0.598 | 0.386 | 0.231 | 0.663 | 0.317 | 0.071 | 0.304 | 0.358 | 0.000 | 0.673 | 0.317 | 0.000 | 0.702 | 0.446 | 0.333 | 1.087 | 0.462 | 0.500 | 0.822 | 0.480 | 0.312 | 0.800 | 0.309 | 1.000 |
| Random Block | 0.634 | 0.383 | 0.167 | 0.500 | 0.326 | – | 1.045 | 0.423 | 0.385 | 0.505 | 0.290 | 0.143 | 0.348 | 0.363 | 0.000 | 0.909 | 0.279 | 0.000 | 0.740 | 0.447 | 0.500 | 1.062 | 0.462 | 0.500 | 0.868 | 0.488 | 0.250 | 0.733 | 0.322 | 1.000 |
| Target Block | 0.675 | 0.390 | 0.167 | 0.750 | 0.319 | – | 0.982 | 0.419 | 0.385 | 0.684 | 0.318 | 0.071 | 0.367 | 0.361 | 0.000 | 0.933 | 0.274 | 0.000 | 0.757 | 0.457 | 0.500 | 1.212 | 0.466 | 0.500 | 0.868 | 0.487 | 0.312 | 0.689 | 0.306 | 1.000 |
| Rolling Shutter | 0.528 | 0.366 | 0.167 | 0.375 | 0.279 | – | 0.875 | 0.410 | 0.308 | 0.505 | 0.324 | 0.071 | 0.334 | 0.358 | 0.000 | 0.740 | 0.234 | 0.000 | 0.655 | 0.447 | 0.500 | 1.150 | 0.467 | 0.500 | 0.900 | 0.470 | 0.250 | 0.656 | 0.304 | 1.000 |
| Resolution Degrade | 0.764 | 0.388 | 0.167 | 0.750 | 0.296 | – | 1.071 | 0.400 | 0.462 | 0.545 | 0.302 | 0.143 | 0.329 | 0.367 | 0.000 | 0.760 | 0.321 | 0.000 | 0.757 | 0.458 | 0.500 | 1.087 | 0.458 | 0.500 | 0.977 | 0.491 | 0.250 | 0.700 | 0.317 | 1.000 |
| Stretch Squish | 0.512 | 0.362 | 0.167 | 0.625 | 0.319 | – | 1.080 | 0.426 | 0.385 | 0.909 | 0.345 | 0.143 | 0.365 | 0.334 | 0.000 | 0.683 | 0.308 | 0.000 | 0.617 | 0.443 | 0.333 | 1.137 | 0.455 | 0.333 | 0.922 | 0.487 | 0.250 | 0.722 | 0.315 | 1.000 |
| Edge Sawtooth | 0.805 | 0.397 | 0.167 | 0.750 | 0.311 | – | 0.427 | 0.426 | 0.385 | 0.909 | 0.346 | 0.143 | 0.363 | 0.355 | 0.000 | 0.865 | 0.280 | 0.000 | 0.787 | 0.458 | 0.333 | 1.238 | 0.480 | 0.500 | 0.977 | 0.487 | 0.250 | 0.689 | 0.315 | 1.000 |
| Elastic | 0.780 | 0.390 | 0.167 | 1.125 | 0.326 | – | 0.982 | 0.425 | 0.385 | 0.836 | 0.349 | 0.143 | 0.358 | 0.363 | 0.000 | 0.894 | 0.279 | 0.000 | 0.757 | 0.457 | 0.333 | 1.288 | 0.480 | 0.500 | 0.822 | 0.476 | 0.312 | 0.711 | 0.318 | 1.000 |
| Color Quantized | 0.772 | 0.394 | 0.167 | 0.625 | 0.329 | – | 1.107 | 0.432 | 0.385 | 0.827 | 0.343 | 0.071 | 0.361 | 0.361 | 0.000 | 0.933 | 0.243 | 0.000 | 0.728 | 0.464 | 0.500 | 1.350 | 0.471 | 0.500 | 0.868 | 0.488 | 0.250 | 0.656 | 0.314 | 1.000 |
| Bright Transform | 1.033 | 0.416 | 0.167 | 1.125 | 0.344 | – | 1.098 | 0.432 | 0.385 | 0.955 | 0.351 | 0.143 | 0.362 | 0.357 | 0.000 | 0.981 | 0.257 | 0.000 | 0.719 | 0.456 | 0.500 | 1.200 | 0.464 | 0.500 | 0.915 | 0.485 | 0.188 | 0.722 | 0.317 | 1.000 |
| Contrast | 0.805 | 0.402 | 0.167 | 0.875 | 0.316 | – | 1.009 | 0.428 | 0.385 | 0.864 | 0.349 | 0.071 | 0.357 | 0.368 | 0.000 | 0.942 | 0.242 | 0.000 | 0.715 | 0.456 | 0.500 | 1.225 | 0.472 | 0.500 | 0.946 | 0.488 | 0.250 | 0.700 | 0.296 | 1.000 |
| Color Shift | 0.943 | 0.416 | 0.167 | 1.125 | 0.329 | – | 1.134 | 0.436 | 0.385 | 0.900 | 0.353 | 0.143 | 0.361 | 0.367 | 0.000 | 0.990 | 0.259 | 0.000 | 0.753 | 0.478 | 0.500 | 1.238 | 0.478 | 0.500 | 0.899 | 0.487 | 0.250 | 0.711 | 0.314 | 1.000 |
| Flicker | 0.797 | 0.403 | 0.167 | 1.250 | 0.329 | – | 1.027 | 0.420 | 0.385 | 0.791 | 0.347 | 0.143 | 0.363 | 0.361 | 0.000 | 1.077 | 0.313 | 0.000 | 0.702 | 0.455 | 0.500 | 1.312 | 0.482 | 0.500 | 0.922 | 0.494 | 0.188 | 0.756 | 0.315 | 1.000 |
| Overexposure | 0.846 | 0.403 | 0.167 | 0.750 | 0.326 | – | 1.161 | 0.431 | 0.462 | 0.955 | 0.354 | 0.000 | 0.360 | 0.360 | 0.000 | 0.990 | 0.270 | 0.000 | 0.783 | 0.470 | 0.500 | 1.212 | 0.464 | 0.500 | 1.031 | 0.507 | 0.125 | 0.711 | 0.316 | 1.000 |
| Underexposure | 0.854 | 0.419 | 0.167 | 0.375 | 0.350 | – | 0.991 | 0.435 | 0.385 | 0.855 | 0.337 | 0.000 | 0.363 | 0.363 | 0.000 | 0.865 | 0.268 | 0.000 | 0.655 | 0.461 | 0.500 | 1.238 | 0.474 | 0.500 | 0.837 | 0.486 | 0.188 | 0.656 | 0.324 | 1.000 |
| Rainy | 0.992 | 0.405 | 0.167 | 1.250 | 0.313 | – | 1.036 | 0.431 | 0.385 | 0.873 | 0.342 | 0.143 | 0.363 | 0.363 | 0.000 | 0.904 | 0.281 | 0.000 | 0.736 | 0.461 | 0.500 | 1.188 | 0.467 | 0.500 | 0.868 | 0.496 | 0.250 | 0.600 | 0.315 | 1.000 |
| Snow | 0.675 | 0.327 | 0.167 | 0.375 | 0.321 | – | 1.107 | 0.435 | 0.385 | 0.945 | 0.355 | 0.000 | 0.366 | 0.366 | 0.000 | 0.894 | 0.281 | 0.000 | 0.843 | 0.463 | 0.500 | 1.300 | 0.474 | 0.500 | 0.915 | 0.491 | 0.250 | 0.722 | 0.325 | 1.000 |
| Frost | | | | | | | | | | | | | | | | | | | | | | | | | | | | | | |
| Reflect | 0.613 | 0.349 | 0.167 | | | | | | | | | | | | | | | | | | | | | | | | | | | |
| Shadow | 0.886 | 0.405 | 0.167 | | | | | | | | | | | | | | | | | | | | | | | | | | | |

| Noise Type | K GPT | K SBERT | K Acc | L GPT | L SBERT | L Acc | M GPT | M SBERT | M Acc | N GPT | N SBERT | N Acc | O GPT | O SBERT | O Acc | P GPT | P SBERT | P Acc | Q GPT | Q SBERT | Q Acc | R GPT | R SBERT | R Acc | S GPT | S SBERT | S Acc | T GPT | T SBERT | T Acc |
|---|---|---|---|---|---|---|---|---|---|---|---|---|---|---|---|---|---|---|---|---|---|---|---|---|---|---|---|---|---|---|
| Clean | 0.693 | 0.368 | 0.0000 | 0.591 | 0.434 | 0.0000 | 0.496 | 0.487 | – | 0.816 | 0.496 | 0.1430 | 0.342 | 0.331 | 0.0000 | 0.919 | 0.412 | 0.1050 | 0.978 | 0.395 | 0.0530 | 0.802 | 0.532 | 0.0000 | 1.179 | 0.423 | 0.0000 | 0.948 | 0.429 | 0.178 |
| Gaussian | 0.504 | 0.338 | 0.0001 | 1.424 | 0.418 | 0.0001 | 0.487 | 0.478 | – | 0.789 | 0.487 | 0.1430 | 0.338 | 0.338 | 0.0000 | 0.817 | 0.410 | 0.389 | 0.967 | 0.381 | 0.0530 | 0.525 | 0.499 | 0.0000 | 1.104 | 0.431 | 0.0000 | 0.832 | 0.418 | 0.158 |
| Impulse | 0.358 | 0.322 | 0.0001 | 1.136 | 0.379 | 0.3330 | 0.592 | 0.478 | – | 0.621 | 0.478 | 0.0710 | 0.349 | 0.338 | 0.0000 | 0.726 | 0.389 | 0.393 | 1.000 | 0.393 | 0.0530 | 0.704 | 0.510 | 0.0000 | 0.887 | 0.409 | 0.0000 | 0.708 | 0.397 | 0.105 |
| Speckle | 0.438 | 0.340 | 0.0001 | 1.394 | 0.408 | 0.0000 | 0.486 | 0.485 | – | 0.701 | 0.486 | 0.0710 | 0.349 | 0.349 | 0.0000 | 0.747 | 0.396 | 0.396 | 0.901 | 0.381 | 0.0530 | 0.741 | 0.515 | 0.0000 | 1.057 | 0.416 | 0.0000 | 0.797 | 0.412 | 0.132 |
| Poisson | 0.380 | 0.325 | 0.0001 | 0.379 | 0.416 | 0.0001 | 0.471 | 0.485 | – | 0.612 | 0.485 | 0.1430 | 0.307 | 0.297 | 0.0000 | 0.747 | 0.318 | 0.396 | 0.912 | 0.381 | 0.0530 | 0.642 | 0.510 | 0.0000 | 0.962 | 0.399 | 0.0000 | 0.761 | 0.410 | 0.145 |
| Gaussian Blur | 0.372 | 0.324 | 0.0001 | 1.288 | 0.398 | 0.3330 | 0.491 | 0.467 | – | 0.707 | 0.491 | 0.1430 | 0.318 | 0.286 | 0.0000 | 0.651 | 0.375 | 0.370 | 0.923 | 0.412 | 0.0000 | 0.642 | 0.501 | 0.0000 | 0.764 | 0.399 | 0.0000 | 0.642 | 0.393 | 0.112 |
| Motion Blur | 0.336 | 0.319 | 0.0001 | 1.091 | 0.397 | 0.3330 | 0.485 | 0.499 | – | 0.612 | 0.490 | 0.1430 | 0.286 | 0.333 | 0.0000 | 0.812 | 0.403 | 0.403 | 1.055 | 0.402 | 0.0000 | 0.736 | 0.522 | 0.0000 | 0.736 | 0.390 | 0.0000 | 0.607 | 0.383 | 0.086 |
| Defocus Blur | 0.526 | 0.347 | 0.0001 | 0.242 | 0.402 | 0.3330 | 0.491 | 0.497 | – | 0.707 | 0.468 | 0.1430 | 0.333 | 0.356 | 0.0000 | 0.823 | 0.403 | 0.382 | 0.978 | 0.412 | 0.0000 | 0.815 | 0.522 | 0.0000 | 1.104 | 0.418 | 0.0000 | 0.846 | 0.418 | 0.164 |
| Glass Blur | 0.591 | 0.328 | 0.0001 | 0.227 | 0.385 | 0.3330 | 0.468 | 0.493 | – | 0.646 | 0.493 | 0.0710 | 0.313 | 0.356 | 0.0000 | 0.597 | 0.403 | 0.403 | 1.000 | 0.381 | 0.0000 | 0.710 | 0.528 | 0.0000 | 0.858 | 0.392 | 0.0000 | 0.579 | 0.417 | 0.164 |
| Zoom Blur | 0.394 | 0.328 | 0.0001 | 0.409 | 0.385 | 0.3330 | 0.468 | 0.493 | – | 0.782 | 0.495 | 0.0710 | 0.357 | 0.313 | 0.221 | 0.559 | 0.403 | 0.403 | 0.736 | 0.381 | 0.0530 | 0.630 | 0.507 | 0.0000 | 0.792 | 0.392 | 0.0000 | 0.667 | 0.394 | 0.092 |
| Frame Drop | 0.270 | 0.317 | 0.0001 | 0.227 | 0.385 | 0.3330 | 0.501 | 0.501 | – | 0.646 | 0.483 | 0.0710 | 0.357 | 0.357 | 0.0530 | 0.283 | 0.382 | 0.382 | 1.000 | 0.411 | 0.0530 | 0.630 | 0.506 | 0.0000 | 1.198 | 0.420 | 0.0000 | 0.667 | 0.390 | 0.118 |
| Frame Replace | 0.635 | 0.361 | 0.0001 | 1.545 | 0.425 | 0.0000 | 0.782 | 0.497 | – | 0.769 | 0.495 | 0.0710 | 0.324 | 0.324 | 0.0000 | 0.855 | 0.374 | 0.374 | 0.692 | 0.411 | 0.0530 | 0.852 | 0.530 | 0.0000 | 0.538 | 0.420 | 0.0000 | 0.939 | 0.426 | 0.171 |
| Frame Repeat | 0.307 | 0.309 | 0.0001 | 0.167 | 0.397 | 0.0000 | 0.673 | 0.470 | – | 0.694 | 0.491 | 0.0710 | 0.261 | 0.261 | 0.0530 | 0.581 | 0.374 | 0.374 | 0.692 | 0.402 | 0.0530 | 0.500 | 0.530 | 0.0000 | 0.567 | 0.406 | 0.0000 | 0.567 | 0.413 | 0.105 |
| Temporal Jitter | 0.511 | 0.346 | 0.0001 | 1.152 | 0.397 | 0.0000 | 0.755 | 0.484 | – | 0.701 | 0.492 | 0.1430 | 0.324 | 0.354 | 0.0000 | 0.742 | 0.408 | 0.384 | 0.923 | 0.413 | 0.0530 | 0.840 | 0.530 | 0.0000 | 0.972 | 0.414 | 0.0000 | 0.776 | 0.413 | 0.145 |
| Jpeg Artifact | 0.526 | 0.349 | 0.0001 | 1.591 | 0.432 | 0.0001 | 0.680 | 0.488 | – | 0.612 | 0.491 | 0.0710 | 0.354 | 0.354 | 0.0530 | 0.785 | 0.408 | 0.408 | 0.923 | 0.411 | 0.0530 | 0.772 | 0.514 | 0.0000 | 0.991 | 0.420 | 0.0000 | 0.797 | 0.415 | 0.151 |
| Bit Error | 0.577 | 0.355 | 0.0001 | 0.591 | 0.404 | 0.3330 | 0.755 | 0.493 | – | 0.707 | 0.497 | 0.0710 | 0.307 | 0.334 | 0.0000 | 0.538 | 0.375 | 0.375 | 0.736 | 0.407 | 0.0000 | 0.660 | 0.507 | 0.0000 | 1.075 | 0.414 | 0.0000 | 0.856 | 0.416 | 0.145 |
| H265 Artifacts | 0.562 | 0.344 | 0.0001 | 0.273 | 0.427 | 0.3330 | 0.707 | 0.497 | – | 0.626 | 0.494 | 0.0710 | 0.353 | 0.353 | 0.0000 | 0.672 | 0.389 | 0.389 | 1.044 | 0.402 | 0.0000 | 0.815 | 0.519 | 0.0000 | 1.019 | 0.420 | 0.381 | 0.519 | 0.413 | 0.086 |
| Random Block | 0.467 | 0.336 | 0.0001 | 0.667 | 0.404 | 0.0000 | 0.816 | 0.485 | – | 0.707 | 0.490 | 0.1430 | 0.334 | 0.321 | 0.0000 | 0.817 | 0.406 | 0.398 | 1.077 | 0.387 | 0.0000 | 0.698 | 0.518 | 0.0000 | 1.047 | 0.420 | 0.0000 | 0.807 | 0.413 | 0.158 |
| Target Block | 0.453 | 0.351 | 0.0001 | 0.227 | 0.390 | 0.3330 | 0.619 | 0.493 | – | 0.748 | 0.495 | 0.0710 | 0.353 | 0.345 | 0.0000 | 0.731 | 0.401 | 0.401 | 1.044 | 0.391 | 0.0530 | 0.778 | 0.524 | 0.0000 | 1.085 | 0.401 | 0.0000 | 0.834 | 0.415 | 0.164 |
| Rolling Shutter | 0.380 | 0.325 | 0.0001 | 0.364 | 0.403 | 0.0000 | 0.748 | 0.494 | – | 0.769 | 0.494 | 0.0710 | 0.294 | 0.322 | 0.0000 | 0.683 | 0.406 | 0.406 | 0.868 | 0.412 | 0.0530 | 0.698 | 0.519 | 0.0000 | 0.915 | 0.399 | 0.0000 | 0.692 | 0.407 | 0.125 |
| Resolution Degrade | 0.584 | 0.344 | 0.0001 | 0.545 | 0.423 | 0.3330 | 0.769 | 0.494 | – | 0.796 | 0.493 | 0.1430 | 0.341 | 0.341 | 0.0000 | 0.844 | 0.419 | 0.419 | 0.967 | 0.407 | 0.0530 | 0.722 | 0.510 | 0.0000 | 0.972 | 0.410 | 0.0000 | 0.899 | 0.423 | 0.138 |
| Stretch Squish | 0.650 | 0.349 | 0.0001 | 0.409 | 0.423 | 0.0000 | 0.837 | 0.494 | – | 0.803 | 0.495 | 0.1430 | 0.345 | 0.345 | 0.0000 | 0.973 | 0.416 | 0.416 | 1.033 | 0.402 | 0.0530 | 0.759 | 0.525 | 0.0000 | 1.132 | 0.427 | 0.0000 | 0.889 | 0.422 | 0.158 |
| Edge Sawtooth | 0.591 | 0.351 | 0.0001 | 0.500 | 0.423 | 0.3330 | 0.803 | 0.503 | – | 0.735 | 0.491 | 0.1430 | 0.351 | 0.322 | 0.0000 | 0.887 | 0.410 | 0.410 | 1.055 | 0.407 | 0.0530 | 0.747 | 0.520 | 0.0000 | 1.160 | 0.421 | 0.0000 | 0.873 | 0.422 | 0.158 |
| Elastic | 0.759 | 0.371 | 0.0001 | 0.485 | 0.423 | 0.0000 | 0.735 | 0.498 | – | 0.769 | 0.492 | 0.0710 | 0.359 | 0.374 | 0.0000 | 0.846 | 0.419 | 0.419 | 1.077 | 0.408 | 0.0530 | 0.753 | 0.518 | 0.0000 | 1.132 | 0.424 | 0.0000 | 0.877 | 0.421 | 0.178 |
| Color Quantized | 0.679 | 0.360 | 0.0001 | 0.530 | 0.424 | 0.0000 | 0.707 | 0.492 | – | 0.803 | 0.492 | 0.1430 | 0.342 | 0.359 | 0.0000 | 0.946 | 0.416 | 0.416 | 0.978 | 0.400 | 0.0530 | 0.728 | 0.524 | 0.0000 | 1.160 | 0.419 | 0.0000 | 0.908 | 0.422 | 0.158 |
| Bright Transform | 0.591 | 0.353 | 0.0001 | 0.561 | 0.427 | 0.3330 | 0.891 | 0.507 | – | 0.803 | 0.498 | 0.1430 | 0.330 | 0.330 | 0.0000 | 0.887 | 0.410 | 0.410 | 1.121 | 0.405 | 0.0530 | 0.741 | 0.518 | 0.0000 | 1.085 | 0.416 | 0.0000 | 0.877 | 0.421 | 0.158 |
| Contrast | 0.657 | 0.353 | 0.0001 | 0.576 | 0.422 | 0.0000 | 0.891 | 0.498 | – | 0.714 | 0.494 | 0.1430 | 0.348 | 0.348 | 0.0000 | 0.946 | 0.412 | 0.412 | 1.066 | 0.400 | 0.0530 | 0.753 | 0.513 | 0.0000 | 1.142 | 0.417 | 0.0000 | 0.903 | 0.423 | 0.178 |
| Color Shift | 0.759 | 0.371 | 0.0001 | 0.530 | 0.429 | 0.3330 | 0.803 | 0.492 | – | 0.707 | 0.498 | 0.0710 | 0.365 | 0.299 | 0.0530 | 1.077 | 0.408 | 0.408 | 1.000 | 0.405 | 0.0530 | 0.728 | 0.524 | 0.0000 | 1.160 | 0.419 | 0.0000 | 0.873 | 0.421 | 0.158 |
| Flicker | 0.620 | 0.360 | 0.0001 | 0.485 | 0.431 | 0.0000 | 0.735 | 0.492 | – | 0.803 | 0.491 | 0.0710 | 0.299 | 0.330 | 0.0000 | 0.849 | 0.419 | 0.419 | 1.088 | 0.400 | 0.0530 | 0.728 | 0.518 | 0.0000 | 1.132 | 0.420 | 0.0000 | 0.880 | 0.423 | 0.178 |
| Overexposure | 0.628 | 0.358 | 0.0001 | 0.439 | 0.425 | 0.0000 | 0.784 | 0.493 | – | 0.803 | 0.503 | 0.1430 | 0.353 | 0.357 | 0.0000 | 0.925 | 0.427 | 0.427 | 1.088 | 0.408 | 0.0530 | 0.759 | 0.524 | 0.0000 | 1.085 | 0.429 | 0.0000 | 0.904 | 0.422 | 0.171 |
| Underexposure | 0.628 | 0.362 | 0.0001 | 0.425 | 0.410 | 0.0000 | 0.748 | 0.493 | – | 0.735 | 0.501 | 0.0710 | 0.340 | 0.402 | 0.0000 | 0.817 | 0.410 | 0.410 | 1.000 | 0.400 | 0.0530 | 0.784 | 0.520 | 0.0000 | 1.142 | 0.416 | 0.0000 | 0.903 | 0.422 | 0.158 |
| Rainy | 0.577 | 0.327 | 0.0001 | 0.364 | 0.418 | 0.0000 | 0.891 | 0.487 | – | 0.707 | 0.498 | 0.1430 | 0.344 | 0.337 | 0.0000 | 0.747 | 0.392 | 0.399 | 0.978 | 0.402 | 0.0530 | 0.667 | 0.512 | 0.0000 | 1.123 | 0.421 | 0.0000 | 0.812 | 0.426 | 0.184 |
| Foggy | 0.613 | 0.349 | 0.0001 | 0.419 | 0.397 | 0.3330 | 0.776 | 0.487 | – | 0.803 | 0.498 | 0.0710 | 0.352 | 0.392 | 0.0000 | 0.846 | 0.392 | 0.392 | 0.989 | 0.402 | 0.0530 | 0.716 | 0.512 | 0.0000 | 1.142 | 0.419 | 0.0000 | 0.874 | 0.420 | 0.118 |
| Shadow | 0.686 | 0.352 | 0.0000 | 0.621 | 0.429 | 0.0000 | 0.782 | 0.498 | – | 0.782 | 0.498 | 0.1430 | 0.352 | 0.352 | 0.0000 | 0.866 | 0.413 | 0.413 | 1.066 | 0.415 | 0.0530 | 0.747 | 0.522 | 0.0000 | 1.132 | 0.424 | 0.0000 | 0.917 | 0.425 | 0.158 |

Note: A: Advertisements, B: Algorithm & Models, C: Autos & Vehicles, D: Business & Industrial, E: Computers & Electronics, F: Fairy Tale, G: Films & TV Shows, H: Finance, I: Food & Drink, J: Games, K: Humor, L: Instruction Video (how to), M: Knowledge, N: News, O: Others, P: People, Q: Pets & Animals, R: Science, S: Sports, T: Overall

Table 7: PLLaVA 7B performance across different video types

**Top section (categories A–J)**

| Noise Type | A GPT | A SBERT | A Acc | B GPT | B SBERT | B Acc | C GPT | C SBERT | C Acc | D GPT | D SBERT | D Acc | E GPT | E SBERT | E Acc | F GPT | F SBERT | F Acc | G GPT | G SBERT | G Acc | H GPT | H SBERT | H Acc | I GPT | I SBERT | I Acc | J GPT | J SBERT | J Acc |
|---|---|---|---|---|---|---|---|---|---|---|---|---|---|---|---|---|---|---|---|---|---|---|---|---|---|---|---|---|---|---|
| Clean | 0.569 | 0.372 | — | 0.750 | 0.413 | 0.000 | 0.946 | 0.406 | 0.000 | 0.632 | 0.322 | 0.000 | 0.385 | 0.341 | 0.000 | 0.625 | 0.250 | 0.000 | 0.890 | 0.417 | 0.000 | 0.938 | 0.449 | 0.031 | 0.553 | 0.456 | 0.333 | 0.467 | 0.298 | 0.000 |
| Gaussian | 0.374 | 0.344 | 0.000 | 0.375 | 0.376 | 0.000 | 0.670 | 0.386 | 0.000 | 0.621 | 0.317 | 0.000 | 0.320 | 0.324 | 0.000 | 0.000 | 0.373 | 0.000 | 0.692 | 0.340 | 0.000 | 0.875 | 0.446 | 0.333 | 0.504 | 0.458 | 0.333 | 0.389 | 0.291 | 0.000 |
| Impulse | 0.211 | 0.317 | 0.000 | 0.125 | 0.324 | 0.000 | 0.408 | 0.386 | 0.000 | 0.263 | 0.314 | 0.000 | 0.283 | 0.302 | 0.000 | 0.000 | 0.335 | 0.000 | 0.495 | 0.352 | 0.000 | 0.762 | 0.422 | 0.167 | 0.234 | 0.417 | 0.167 | 0.278 | 0.266 | 0.000 |
| Speckle | 0.236 | 0.338 | 0.000 | 0.000 | 0.374 | 0.000 | 0.589 | 0.370 | 0.000 | 0.516 | 0.314 | 0.000 | 0.312 | 0.305 | 0.000 | 0.250 | 0.306 | 0.000 | 0.484 | 0.393 | 0.000 | 0.787 | 0.438 | 0.000 | 0.256 | 0.447 | 0.333 | 0.289 | 0.294 | 0.000 |
| Poisson | 0.333 | 0.324 | 0.000 | 0.750 | 0.340 | 0.000 | 0.571 | 0.379 | 0.000 | 0.526 | 0.307 | 0.000 | 0.290 | 0.302 | 0.000 | 0.250 | 0.318 | 0.000 | 0.758 | 0.381 | 0.000 | 0.812 | 0.434 | 0.000 | 0.504 | 0.454 | 0.333 | 0.411 | 0.285 | 0.000 |
| Gaussian Blur | 0.244 | 0.333 | 0.000 | 0.750 | 0.267 | 0.000 | 0.509 | 0.379 | 0.000 | 0.316 | 0.307 | 0.000 | 0.305 | 0.312 | 0.000 | 0.250 | 0.306 | 0.000 | 0.626 | 0.404 | 0.000 | 0.613 | 0.432 | 0.000 | 0.426 | 0.414 | 0.167 | 0.222 | 0.281 | 0.000 |
| Motion Blur | 0.496 | 0.366 | 0.000 | 0.375 | 0.263 | 0.000 | 0.679 | 0.381 | 0.000 | 0.537 | 0.323 | 0.000 | 0.288 | 0.327 | 0.000 | 0.250 | 0.308 | 0.000 | 0.549 | 0.365 | 0.000 | 0.838 | 0.457 | 0.000 | 0.411 | 0.412 | 0.167 | 0.400 | 0.267 | 0.000 |
| Defocus Blur | 0.203 | 0.332 | 0.000 | 0.125 | 0.264 | 0.000 | 0.795 | 0.354 | 0.000 | 0.505 | 0.271 | 0.000 | 0.273 | 0.317 | 0.000 | 0.289 | 0.308 | 0.000 | 0.956 | 0.403 | 0.000 | 0.963 | 0.454 | 0.000 | 0.597 | 0.462 | 0.167 | 0.433 | 0.267 | 0.000 |
| Glass Blur | 0.431 | 0.365 | 0.000 | 0.500 | 0.284 | 0.000 | 0.589 | 0.371 | 0.000 | 0.453 | 0.301 | 0.000 | 0.292 | 0.291 | 0.000 | 0.250 | 0.296 | 0.000 | 0.527 | 0.403 | 0.000 | 0.700 | 0.423 | 0.000 | 0.451 | 0.451 | 0.333 | 0.300 | 0.267 | 0.000 |
| Zoom Blur | 0.341 | 0.336 | 0.000 | 0.750 | 0.302 | 0.000 | 0.446 | 0.407 | 0.000 | 0.211 | 0.291 | 0.000 | 0.278 | 0.326 | 0.000 | 0.288 | 0.324 | 0.000 | 0.484 | 0.378 | 0.000 | 0.487 | 0.372 | 0.000 | 0.170 | 0.419 | 0.333 | 0.222 | 0.241 | 0.000 |
| Frame Drop | 0.138 | 0.273 | 0.000 | 0.375 | 0.329 | 0.000 | 0.518 | 0.346 | 0.000 | 0.232 | 0.274 | 0.000 | 0.274 | 0.280 | 0.000 | 0.226 | 0.316 | 0.000 | 0.571 | 0.414 | 0.000 | 0.700 | 0.442 | 0.031 | 0.187 | 0.456 | 0.333 | 0.533 | 0.309 | 0.000 |
| Frame Replace | 0.472 | 0.373 | 0.000 | 0.500 | 0.330 | 0.000 | 0.732 | 0.379 | 0.000 | 0.547 | 0.310 | 0.000 | 0.305 | 0.326 | 0.000 | 0.304 | 0.306 | 0.000 | 0.392 | 0.402 | 0.000 | 0.700 | 0.395 | 0.000 | 0.319 | 0.429 | 0.167 | 0.300 | 0.241 | 0.000 |
| Frame Repeat | 0.203 | 0.308 | 0.000 | 0.500 | 0.385 | 0.000 | 0.750 | 0.383 | 0.000 | 0.558 | 0.303 | 0.000 | 0.265 | 0.331 | 0.000 | 0.371 | 0.273 | 0.000 | 0.758 | 0.401 | 0.000 | 0.688 | 0.421 | 0.000 | 0.409 | 0.458 | 0.167 | 0.411 | 0.299 | 0.000 |
| Temporal Jitter | 0.423 | 0.325 | 0.000 | 0.375 | 0.392 | 0.000 | 0.875 | 0.388 | 0.000 | 0.432 | 0.314 | 0.000 | 0.304 | 0.301 | 0.000 | 0.350 | 0.277 | 0.000 | 0.670 | 0.376 | 0.000 | 0.925 | 0.449 | 0.000 | 0.400 | 0.457 | 0.167 | 0.378 | 0.287 | 0.000 |
| Jpeg Artifact | 0.407 | 0.362 | 0.000 | 0.875 | 0.344 | 0.000 | 0.741 | 0.383 | 0.000 | 0.326 | 0.327 | 0.000 | 0.307 | 0.309 | 0.000 | 0.371 | 0.284 | 0.000 | 0.725 | 0.402 | 0.000 | 0.700 | 0.452 | 0.000 | 0.226 | 0.431 | 0.167 | 0.389 | 0.258 | 0.000 |
| Bit Error | 0.480 | 0.315 | 0.000 | 0.250 | 0.389 | 0.000 | 0.723 | 0.377 | 0.000 | 0.558 | 0.321 | 0.000 | 0.265 | 0.331 | 0.000 | 0.419 | 0.263 | 0.000 | 0.670 | 0.401 | 0.000 | 0.912 | 0.437 | 0.000 | 0.383 | 0.437 | 0.167 | 0.356 | 0.286 | 0.000 |
| H265 Artifacts | 0.276 | 0.355 | 0.000 | 0.375 | 0.379 | 0.000 | 0.804 | 0.360 | 0.000 | 0.505 | 0.322 | 0.000 | 0.307 | 0.330 | 0.000 | 0.543 | 0.298 | 0.000 | 0.593 | 0.404 | 0.000 | 0.838 | 0.431 | 0.000 | 0.400 | 0.455 | 0.167 | 0.356 | 0.292 | 0.000 |
| Random Block | 0.341 | 0.358 | 0.000 | 0.250 | 0.336 | 0.000 | 0.812 | 0.403 | 0.000 | 0.421 | 0.317 | 0.000 | 0.312 | 0.307 | 0.000 | 0.489 | 0.293 | 0.000 | 0.725 | 0.402 | 0.000 | 0.975 | 0.460 | 0.000 | 0.302 | 0.456 | 0.167 | 0.389 | 0.277 | 0.000 |
| Target Block | 0.520 | 0.355 | 0.000 | 0.375 | 0.387 | 0.000 | 0.804 | 0.382 | 0.000 | 0.505 | 0.303 | 0.000 | 0.273 | 0.329 | 0.000 | 0.543 | 0.302 | 0.000 | 0.593 | 0.387 | 0.000 | 0.950 | 0.447 | 0.000 | 0.506 | 0.454 | 0.333 | 0.433 | 0.297 | 0.000 |
| Rolling Shutter | 0.293 | 0.334 | 0.000 | 0.250 | 0.392 | 0.000 | 0.812 | 0.395 | 0.000 | 0.421 | 0.306 | 0.000 | 0.262 | 0.330 | 0.000 | 0.392 | 0.293 | 0.000 | 0.780 | 0.416 | 0.000 | 0.887 | 0.457 | 0.000 | 0.468 | 0.458 | 0.167 | 0.389 | 0.289 | 0.000 |
| Resolution Degrade | 0.520 | 0.374 | 0.000 | 0.500 | 0.355 | 0.000 | 0.777 | 0.377 | 0.000 | 0.211 | 0.291 | 0.000 | 0.278 | 0.344 | 0.000 | 0.258 | 0.307 | 0.000 | 0.519 | 0.417 | 0.000 | 0.963 | 0.449 | 0.000 | 0.519 | 0.458 | 0.167 | 0.433 | 0.297 | 0.000 |
| Stretch Squish | 0.512 | 0.358 | 0.000 | 0.375 | 0.387 | 0.000 | 0.688 | 0.390 | 0.000 | 0.249 | 0.324 | 0.000 | 0.305 | 0.321 | 0.000 | 0.000 | 0.500 | 0.000 | 0.451 | 0.413 | 0.000 | 0.667 | 0.447 | 0.000 | 0.451 | 0.453 | 0.167 | 0.489 | 0.312 | 0.000 |
| Edge Sawtooth | 0.423 | 0.366 | 0.000 | 0.500 | 0.392 | 0.000 | 0.884 | 0.397 | 0.000 | 0.673 | 0.327 | 0.000 | 0.310 | 0.324 | 0.000 | 0.750 | 0.289 | 0.000 | 0.434 | 0.400 | 0.000 | 0.887 | 0.451 | 0.000 | 0.455 | 0.461 | 0.500 | 0.511 | 0.302 | 0.000 |
| Elastic | 0.455 | 0.361 | 0.000 | 0.375 | 0.382 | 0.000 | 0.723 | 0.383 | 0.000 | 0.545 | 0.315 | 0.000 | 0.326 | 0.330 | 0.000 | 0.548 | 0.327 | 0.000 | 0.447 | 0.409 | 0.000 | 0.875 | 0.456 | 0.000 | 0.451 | 0.458 | 0.167 | 0.500 | 0.302 | 0.000 |
| Color Quantized | 0.537 | 0.376 | 0.000 | 0.625 | 0.383 | 0.000 | 0.759 | 0.403 | 0.000 | 0.573 | 0.325 | 0.000 | 0.333 | 0.325 | 0.000 | 0.600 | 0.298 | 0.000 | 0.421 | 0.419 | 0.167 | 0.875 | 0.455 | 0.000 | 0.421 | 0.455 | 0.167 | 0.433 | 0.283 | 0.000 |
| Bright Transform | 0.480 | 0.352 | 0.000 | 0.375 | 0.380 | 0.000 | 0.804 | 0.308 | 0.000 | 0.673 | 0.322 | 0.000 | 0.309 | 0.324 | 0.000 | 0.625 | 0.258 | 0.000 | 0.409 | 0.401 | 0.000 | 0.887 | 0.447 | 0.000 | 0.409 | 0.453 | 0.167 | 0.467 | 0.306 | 0.000 |
| Contrast | 0.480 | 0.365 | 0.000 | 0.625 | 0.383 | 0.000 | 0.777 | 0.390 | 0.000 | 0.645 | 0.328 | 0.000 | 0.343 | 0.321 | 0.000 | 0.250 | 0.302 | 0.000 | 0.460 | 0.416 | 0.000 | 0.912 | 0.460 | 0.000 | 0.502 | 0.451 | 0.167 | 0.511 | 0.305 | 0.000 |
| Color Shift | 0.504 | 0.362 | 0.000 | 0.625 | 0.380 | 0.000 | 0.955 | 0.397 | 0.000 | 0.664 | 0.320 | 0.000 | 0.321 | 0.325 | 0.000 | 0.529 | 0.293 | 0.000 | 0.451 | 0.413 | 0.000 | 0.963 | 0.453 | 0.031 | 0.451 | 0.453 | 0.167 | 0.456 | 0.303 | 0.000 |
| Flicker | 0.512 | 0.359 | 0.000 | 0.500 | 0.382 | 0.000 | — | — | — | — | — | — | 0.344 | 0.330 | 0.000 | 0.491 | 0.307 | 0.000 | 0.434 | 0.410 | 0.000 | 0.887 | 0.449 | 0.000 | 0.455 | 0.461 | 0.167 | 0.489 | 0.312 | 0.000 |
| Overexposure | 0.480 | 0.370 | 0.000 | 0.625 | 0.385 | 0.000 | — | — | — | — | — | — | 0.324 | 0.321 | 0.000 | 0.548 | 0.309 | 0.000 | 0.399 | 0.400 | 0.000 | 0.938 | 0.451 | 0.000 | 0.449 | 0.458 | 0.500 | 0.500 | 0.302 | 0.000 |
| Underexposure | 0.439 | 0.355 | 0.000 | 0.125 | 0.314 | 0.000 | 0.759 | 0.383 | 0.000 | 0.573 | 0.315 | 0.000 | 0.330 | 0.326 | 0.000 | 0.548 | 0.327 | 0.000 | 0.447 | 0.409 | 0.000 | 0.875 | 0.456 | 0.000 | 0.421 | 0.455 | 0.167 | 0.433 | 0.283 | 0.000 |
| Rainy | 0.585 | 0.373 | 0.000 | 0.500 | 0.393 | 0.000 | 0.786 | 0.403 | 0.000 | 0.600 | 0.321 | 0.000 | 0.333 | 0.333 | 0.000 | 0.625 | 0.298 | 0.000 | 0.438 | 0.419 | 0.000 | 0.887 | 0.458 | 0.000 | 0.438 | 0.455 | 0.167 | 0.511 | 0.301 | 0.000 |
| Foggy | 0.321 | 0.331 | 0.000 | 0.625 | 0.380 | 0.000 | 0.804 | 0.383 | 0.000 | 0.673 | 0.322 | 0.000 | 0.297 | 0.324 | 0.000 | 0.548 | 0.298 | 0.000 | 0.421 | 0.401 | 0.000 | 0.875 | 0.447 | 0.000 | 0.421 | 0.455 | 0.167 | 0.433 | 0.283 | 0.000 |
| Snow | 0.328 | 0.336 | 0.000 | 0.250 | 0.381 | 0.000 | 0.777 | 0.403 | 0.000 | 0.673 | 0.306 | 0.000 | 0.297 | 0.325 | 0.000 | 0.500 | 0.258 | 0.000 | 0.409 | 0.409 | 0.000 | 0.887 | 0.453 | 0.000 | 0.409 | 0.453 | 0.167 | 0.533 | 0.300 | 0.000 |
| Frost | 0.343 | 0.322 | 0.000 | 0.500 | 0.401 | 0.000 | 0.812 | 0.401 | 0.000 | 0.558 | 0.313 | 0.000 | 0.301 | 0.324 | 0.000 | 0.250 | 0.300 | 0.000 | 0.460 | 0.409 | 0.000 | 0.688 | 0.453 | 0.000 | 0.533 | 0.453 | 0.333 | 0.489 | 0.294 | 0.000 |
| Reflect | 0.350 | 0.334 | 0.000 | 0.750 | 0.372 | 0.000 | 0.812 | 0.578 | 0.000 | 0.558 | 0.325 | 0.000 | 0.321 | 0.321 | 0.000 | 0.250 | 0.293 | 0.000 | 0.802 | 0.409 | 0.000 | 0.912 | 0.449 | 0.000 | 0.605 | 0.453 | 0.167 | 0.544 | 0.289 | 0.000 |
| Shadow | 0.387 | 0.333 | 0.000 | 0.625 | 0.395 | 0.000 | 0.955 | 0.544 | 0.000 | 0.664 | 0.335 | 0.000 | 0.300 | 0.325 | 0.000 | 1.000 | 0.266 | 0.000 | 0.912 | 0.412 | 0.000 | 1.000 | 0.453 | 0.000 | 0.597 | 0.452 | 0.167 | 0.489 | 0.294 | 0.000 |

**Bottom section (categories K–T)**

| Noise Type | K GPT | K SBERT | K Acc | L GPT | L SBERT | L Acc | M GPT | M SBERT | M Acc | N GPT | N SBERT | N Acc | O GPT | O SBERT | O Acc | P GPT | P SBERT | P Acc | Q GPT | Q SBERT | Q Acc | R GPT | R SBERT | R Acc | S GPT | S SBERT | S Acc | T GPT | T SBERT | T Acc |
|---|---|---|---|---|---|---|---|---|---|---|---|---|---|---|---|---|---|---|---|---|---|---|---|---|---|---|---|---|---|---|
| Clean | 0.460 | 0.342 | 0.000 | 0.348 | 1.076 | 0.000 | 0.551 | 0.454 | 0.438 | 0.632 | 0.332 | 0.000 | 0.387 | 0.311 | 0.000 | 0.774 | 0.393 | 0.000 | 0.890 | 0.390 | 0.000 | 0.498 | 0.896 | 0.000 | 0.562 | 0.390 | 0.000 | 0.687 | 0.396 | 0.059 |
| Gaussian | 0.336 | 0.322 | 0.333 | 1.076 | 0.375 | 0.000 | 0.408 | 0.438 | 0.000 | 0.621 | 0.317 | 0.000 | 0.320 | 0.311 | 0.000 | 0.478 | 0.372 | 0.000 | 0.692 | 0.374 | 0.000 | 0.489 | 0.642 | 0.000 | 0.370 | 0.378 | 0.000 | 0.507 | 0.381 | 0.053 |
| Impulse | 0.161 | 0.300 | 0.333 | 0.924 | 0.355 | 0.000 | 0.272 | 0.428 | 0.000 | 0.263 | 0.314 | 0.000 | 0.321 | 0.320 | 0.000 | 0.366 | 0.344 | 0.000 | 0.495 | 0.352 | 0.000 | 0.473 | 0.387 | 0.000 | 0.346 | 0.351 | 0.033 | 0.315 | 0.356 | 0.033 |
| Speckle | 0.190 | 0.310 | 0.333 | 0.955 | 0.374 | 0.000 | 0.439 | 0.424 | 0.000 | 0.516 | 0.314 | 0.000 | 0.339 | 0.290 | 0.000 | 0.075 | 0.368 | 0.000 | 0.484 | 0.362 | 0.000 | 0.480 | 0.500 | 0.000 | 0.385 | 0.359 | 0.000 | 0.375 | 0.375 | 0.053 |
| Poisson | 0.226 | 0.316 | 0.333 | 0.384 | 0.392 | 0.000 | 0.503 | 0.442 | 0.000 | 0.526 | 0.307 | 0.000 | 0.290 | 0.304 | 0.000 | 0.581 | 0.372 | 0.000 | 0.758 | 0.356 | 0.000 | 0.477 | 0.481 | 0.000 | 0.396 | 0.350 | 0.046 | 0.481 | 0.359 | 0.046 |
| Gaussian Blur | 0.139 | 0.301 | 0.333 | 0.864 | 0.389 | 0.000 | 0.497 | 0.430 | 0.000 | 0.316 | 0.307 | 0.000 | 0.290 | 0.298 | 0.000 | 0.312 | 0.352 | 0.000 | 0.626 | 0.356 | 0.000 | 0.492 | 0.396 | 0.000 | 0.390 | 0.361 | 0.026 | 0.401 | 0.364 | 0.026 |
| Motion Blur | 0.387 | 0.332 | 0.333 | 1.076 | 0.364 | 0.000 | 0.456 | 0.448 | 0.000 | 0.537 | 0.302 | 0.000 | 0.311 | 0.265 | 0.000 | 0.511 | 0.384 | 0.000 | 0.824 | 0.394 | 0.000 | 0.480 | 0.330 | 0.000 | 0.632 | 0.355 | 0.039 | 0.557 | 0.355 | 0.039 |
| Defocus Blur | 0.131 | 0.294 | 0.333 | 0.864 | 0.340 | 0.000 | 0.415 | 0.446 | 0.000 | 0.505 | 0.271 | 0.000 | 0.312 | 0.273 | 0.200 | 0.543 | 0.384 | 0.000 | 0.956 | 0.358 | 0.000 | 0.499 | 0.632 | 0.000 | 0.494 | 0.388 | 0.059 | 0.584 | 0.387 | 0.059 |
| Glass Blur | 0.343 | 0.329 | 0.333 | 0.403 | 0.379 | 0.000 | 0.401 | 0.444 | 0.000 | 0.306 | 0.302 | 0.000 | 0.305 | 0.305 | 0.000 | 0.419 | 0.362 | 0.000 | 0.549 | 0.386 | 0.000 | 0.489 | 0.379 | 0.000 | 0.451 | 0.387 | 0.053 | 0.515 | 0.358 | 0.053 |
| Zoom Blur | 0.161 | 0.301 | 0.333 | 0.773 | 0.403 | 0.000 | 0.415 | 0.428 | 0.000 | 0.211 | 0.291 | 0.000 | 0.306 | 0.316 | 0.000 | 0.615 | 0.384 | 0.000 | 0.484 | 0.357 | 0.000 | 0.479 | 0.500 | 0.000 | 0.379 | 0.362 | 0.046 | 0.415 | 0.362 | 0.046 |
| Frame Drop | 0.080 | 0.258 | 0.333 | 0.606 | 0.377 | 0.000 | 0.401 | 0.407 | 0.000 | 0.211 | 0.249 | 0.000 | 0.370 | 0.262 | 0.000 | 0.258 | 0.304 | 0.000 | 0.484 | 0.405 | 0.000 | 0.447 | 0.311 | 0.000 | 0.370 | 0.316 | 0.039 | 0.295 | 0.319 | 0.039 |
| Frame Replace | 0.423 | 0.333 | 0.333 | 1.379 | 0.403 | 0.000 | 0.558 | 0.458 | 0.000 | 0.653 | 0.324 | 0.000 | 0.332 | 0.305 | 0.000 | 0.645 | 0.389 | 0.000 | 1.033 | 0.405 | 0.000 | 0.495 | 0.830 | 0.000 | 0.420 | 0.407 | 0.059 | 0.654 | 0.397 | 0.059 |
| Frame Repeat | 0.182 | 0.277 | 0.333 | 0.894 | 0.385 | 0.000 | 0.490 | 0.441 | 0.000 | 0.274 | 0.310 | 0.000 | 0.332 | 0.296 | 0.000 | 0.392 | 0.346 | 0.000 | 0.571 | 0.379 | 0.000 | 0.459 | 0.491 | 0.000 | 0.362 | 0.387 | 0.046 | 0.387 | 0.345 | 0.046 |
| Temporal Jitter | 0.263 | 0.317 | 0.333 | 1.121 | 0.392 | 0.000 | 0.448 | 0.447 | 0.000 | 0.547 | 0.310 | 0.000 | 0.296 | 0.307 | 0.000 | 0.500 | 0.363 | 0.000 | 0.868 | 0.379 | 0.000 | 0.480 | 0.679 | 0.000 | 0.383 | 0.380 | 0.046 | 0.551 | 0.372 | 0.046 |
| Jpeg Artifact | 0.350 | 0.325 | 0.333 | 1.197 | 0.395 | 0.000 | 0.451 | 0.438 | 0.000 | 0.432 | 0.303 | 0.000 | 0.304 | 0.275 | 0.000 | 0.473 | 0.371 | 0.000 | 0.758 | 0.383 | 0.000 | 0.497 | 0.623 | 0.000 | 0.401 | 0.383 | 0.046 | 0.496 | 0.381 | 0.046 |
| Bit Error | 0.299 | 0.329 | 0.333 | 0.788 | 0.389 | 0.000 | 0.497 | 0.451 | 0.000 | 0.558 | 0.307 | 0.000 | 0.307 | 0.305 | 0.000 | 0.371 | 0.354 | 0.000 | 0.725 | 0.391 | 0.000 | 0.482 | 0.651 | 0.000 | 0.481 | 0.355 | 0.059 | 0.564 | 0.385 | 0.059 |
| H265 Artifacts | 0.139 | 0.291 | 0.333 | 1.106 | 0.344 | 0.000 | 0.401 | 0.466 | 0.000 | 0.516 | 0.317 | 0.000 | 0.305 | 0.307 | 0.000 | 0.489 | 0.362 | 0.000 | 0.670 | 0.391 | 0.000 | 0.497 | 0.245 | 0.000 | 0.390 | 0.355 | 0.053 | 0.523 | 0.358 | 0.053 |
| Random Block | 0.277 | 0.322 | 0.333 | 1.182 | 0.389 | 0.000 | 0.444 | 0.449 | 0.000 | 0.421 | 0.303 | 0.000 | 0.316 | 0.305 | 0.000 | 0.543 | 0.362 | 0.000 | 0.725 | 0.380 | 0.000 | 0.494 | 0.755 | 0.000 | 0.515 | 0.379 | 0.059 | 0.549 | 0.387 | 0.059 |
| Target Block | 0.263 | 0.323 | 0.333 | 0.045 | 0.379 | 0.000 | 0.422 | 0.450 | 0.000 | 0.505 | 0.303 | 0.000 | 0.296 | 0.296 | 0.000 | 0.543 | 0.362 | 0.000 | 0.780 | 0.385 | 0.000 | 0.496 | 0.528 | 0.000 | 0.385 | 0.385 | 0.046 | 0.564 | 0.382 | 0.046 |
| Rolling Shutter | 0.365 | 0.333 | 0.333 | 0.591 | 0.336 | 0.000 | 0.456 | 0.449 | 0.000 | 0.695 | 0.317 | 0.000 | 0.310 | 0.279 | 0.000 | 0.452 | 0.380 | 0.000 | 0.835 | 0.380 | 0.000 | 0.500 | 0.462 | 0.000 | 0.402 | 0.363 | 0.046 | 0.441 | 0.373 | 0.046 |
| Resolution Degrade | 0.409 | 0.332 | 0.333 | 0.490 | 0.387 | 0.000 | 0.469 | 0.449 | 0.000 | 0.769 | 0.323 | 0.000 | 0.310 | 0.305 | 0.000 | 0.624 | 0.395 | 0.000 | 0.857 | 0.392 | 0.000 | 0.475 | 0.698 | 0.000 | 0.519 | 0.397 | 0.046 | 0.611 | 0.390 | 0.046 |
| Stretch Squish | 0.277 | 0.312 | 0.333 | 0.435 | 0.463 | 0.000 | 0.442 | 0.445 | 0.000 | 0.526 | 0.313 | 0.000 | 0.325 | 0.310 | 0.000 | 0.597 | 0.388 | 0.000 | 0.879 | 0.384 | 0.000 | 0.488 | 0.585 | 0.000 | 0.463 | 0.394 | 0.053 | 0.592 | 0.382 | 0.053 |
| Edge Sawtooth | 0.453 | 0.334 | 0.333 | 0.435 | 0.334 | 0.000 | 0.445 | 0.436 | 0.000 | 0.558 | 0.317 | 0.000 | 0.299 | 0.325 | 0.000 | 0.608 | 0.395 | 0.000 | 0.956 | 0.375 | 0.000 | 0.491 | 0.698 | 0.000 | 0.375 | 0.363 | 0.046 | 0.564 | 0.373 | 0.046 |
| Elastic | 0.333 | 0.333 | 0.333 | 0.591 | 0.392 | 0.000 | 0.449 | 0.449 | 0.000 | 0.611 | 0.315 | 0.000 | 0.286 | 0.299 | 0.000 | 0.661 | 0.384 | 0.000 | 0.846 | 0.390 | 0.000 | 0.493 | 0.462 | 0.000 | 0.397 | 0.390 | 0.046 | 0.637 | 0.390 | 0.046 |
| Color Quantized | 0.358 | 0.336 | 0.333 | 0.061 | 0.488 | 0.000 | 0.497 | 0.451 | 0.000 | 0.526 | 0.323 | 0.000 | 0.305 | 0.312 | 0.000 | 0.522 | 0.395 | 0.000 | 0.835 | 0.384 | 0.000 | 0.494 | 0.764 | 0.000 | 0.394 | 0.394 | 0.053 | 0.592 | 0.391 | 0.053 |
| Bright Transform | 0.380 | 0.332 | 0.333 | 0.167 | 0.463 | 0.000 | 0.442 | 0.436 | 0.000 | 0.663 | 0.312 | 0.000 | 0.299 | 0.292 | 0.000 | 0.645 | 0.388 | 0.000 | 0.879 | 0.377 | 0.000 | 0.493 | 0.745 | 0.000 | 0.400 | 0.390 | 0.046 | 0.618 | 0.391 | 0.046 |
| Contrast | 0.343 | 0.342 | 0.333 | 1.000 | 0.392 | 0.000 | 0.431 | 0.462 | 0.000 | 0.558 | 0.312 | 0.000 | 0.312 | 0.286 | 0.000 | 0.522 | 0.384 | 0.000 | 0.956 | 0.384 | 0.000 | 0.487 | 0.632 | 0.000 | 0.394 | 0.388 | 0.053 | 0.597 | 0.394 | 0.053 |
| Color Shift | 0.328 | 0.338 | 0.333 | 0.061 | 0.397 | 0.000 | 0.436 | 0.451 | 0.000 | 0.621 | 0.313 | 0.000 | 0.299 | 0.305 | 0.000 | 0.608 | 0.392 | 0.000 | 0.714 | 0.377 | 0.000 | 0.487 | 0.745 | 0.000 | 0.388 | 0.390 | 0.046 | 0.584 | 0.389 | 0.046 |
| Flicker | 0.423 | 0.346 | 0.333 | 0.348 | 0.400 | 0.000 | 0.453 | 0.441 | 0.000 | 0.526 | 0.322 | 0.000 | 0.316 | 0.301 | 0.000 | 0.667 | 0.379 | 0.000 | 0.846 | 0.390 | 0.000 | 0.488 | 0.632 | 0.000 | 0.404 | 0.397 | 0.053 | 0.597 | 0.391 | 0.053 |
| Overexposure | 0.328 | 0.330 | 0.333 | 1.015 | 0.469 | 0.000 | 0.462 | 0.436 | 0.000 | 0.663 | 0.324 | 0.000 | 0.282 | 0.286 | 0.000 | 0.554 | 0.388 | 0.000 | 0.846 | 0.382 | 0.000 | 0.490 | 0.858 | 0.000 | 0.396 | 0.390 | 0.046 | 0.575 | 0.390 | 0.046 |
| Underexposure | 0.372 | 0.335 | 0.333 | 0.273 | 0.573 | 0.000 | 0.469 | 0.453 | 0.000 | 0.769 | 0.306 | 0.000 | 0.266 | 0.292 | 0.000 | 0.522 | 0.374 | 0.000 | 0.835 | 0.377 | 0.000 | 0.469 | 0.783 | 0.000 | 0.404 | 0.389 | 0.053 | 0.573 | 0.389 | 0.053 |
| Rainy | 0.387 | 0.389 | 0.333 | 0.242 | 0.573 | 0.000 | 0.441 | 0.441 | 0.000 | 0.769 | 0.324 | 0.000 | 0.305 | 0.301 | 0.000 | 0.624 | 0.391 | 0.000 | 0.846 | 0.390 | 0.000 | 0.488 | 0.755 | 0.000 | 0.386 | 0.396 | 0.039 | 0.596 | 0.391 | 0.053 |
| Foggy | 0.321 | 0.331 | 0.333 | 1.348 | 0.404 | 0.000 | 0.503 | 0.456 | 0.000 | 0.538 | 0.306 | 0.000 | 0.316 | 0.282 | 0.000 | 0.667 | 0.379 | 0.000 | 0.868 | 0.382 | 0.000 | 0.469 | 0.755 | 0.000 | 0.376 | 0.404 | 0.053 | 0.603 | 0.387 | 0.053 |
| Snow | 0.328 | 0.336 | 0.333 | 0.197 | 0.573 | 0.000 | 0.435 | 0.441 | 0.000 | 0.568 | 0.313 | 0.000 | 0.297 | 0.301 | 0.000 | 0.581 | 0.374 | 0.000 | 0.824 | 0.381 | 0.000 | 0.488 | 0.755 | 0.000 | 0.381 | 0.386 | 0.053 | 0.573 | 0.392 | 0.053 |
| Frost | 0.343 | 0.322 | 0.333 | 1.106 | 0.561 | 0.000 | 0.442 | 0.443 | 0.000 | 0.558 | 0.315 | 0.000 | 0.301 | 0.297 | 0.000 | 0.581 | 0.374 | 0.000 | 0.824 | 0.378 | 0.000 | 0.489 | 0.755 | 0.000 | 0.401 | 0.573 | 0.053 | 0.561 | 0.384 | 0.053 |
| Reflect | 0.350 | 0.334 | 0.333 | 0.970 | 0.303 | 0.000 | 0.578 | 0.456 | 0.000 | 0.558 | 0.313 | 0.000 | 0.301 | 0.300 | 0.000 | 0.640 | 0.381 | 0.000 | 0.802 | 0.378 | 0.000 | 0.492 | 0.783 | 0.000 | 0.398 | 0.561 | 0.059 | 0.598 | 0.388 | 0.059 |
| Shadow | 0.387 | 0.333 | 0.333 | 0.303 | 0.395 | 0.000 | 0.544 | 0.449 | 0.000 | 0.716 | 0.335 | 0.000 | 0.300 | 0.381 | 0.000 | 0.613 | 0.391 | 0.000 | 0.912 | 0.381 | 0.000 | 0.486 | 0.755 | 0.000 | 0.391 | 0.646 | 0.053 | 0.646 | 0.390 | 0.053 |

Note: A: Advertisements, B: Algorithm & Models, C: Autos & Vehicles, D: Business & Industrial, E: Computers & Electronics, F: Fairy Tale, G: Films & TV Shows, H: Finance, I: Food & Drink, J: Games, K: Humor, L: Instruction Video (how to), M: Knowledge, N: News, O: Others, P: People, Q: Pets & Animals, R: Science, S: Sports, T: Overall

Table 8: Qwen2.5-VL 3B performance across different video types

| Noise Type | A | | | B | | | C | | | D | | | E | | | F | | | G | | | H | | | I | | | J | | |
|---|---|---|---|---|---|---|---|---|---|---|---|---|---|---|---|---|---|---|---|---|---|---|---|---|---|---|---|---|---|---|
| | GPT | SBERT | Acc | GPT | SBERT | Acc | GPT | SBERT | Acc | GPT | SBERT | Acc | GPT | SBERT | Acc | GPT | SBERT | Acc | GPT | SBERT | Acc | GPT | SBERT | Acc | GPT | SBERT | Acc | GPT | SBERT | Acc |
| Clean | 1.000 | 0.414 | 0.167 | 1.375 | 0.384 | – | 1.384 | 0.433 | 0.615 | 0.964 | 0.361 | 0.111 | 1.154 | 0.367 | 0.500 | 1.500 | 0.316 | – | 0.851 | 0.470 | 0.375 | 1.288 | 0.481 | 0.333 | 1.062 | 0.510 | 0.500 | 0.789 | 0.332 | 1.000 |
| Gaussian | 0.667 | 0.377 | 0.167 | 1.500 | 0.376 | – | 1.062 | 0.417 | 0.538 | 0.827 | 0.371 | 0.167 | 0.981 | 0.376 | 0.500 | 0.750 | 0.265 | – | 0.638 | 0.445 | 0.250 | 1.212 | 0.463 | 0.000 | 0.915 | 0.511 | 0.375 | 0.544 | 0.303 | 1.000 |
| Impulse | 0.439 | 0.345 | 0.167 | 1.500 | 0.338 | – | 0.723 | 0.381 | 0.462 | 0.618 | 0.335 | 0.056 | 0.529 | 0.329 | 0.500 | 0.250 | 0.245 | – | 0.562 | 0.434 | 0.281 | 0.938 | 0.451 | 0.000 | 0.860 | 0.470 | 0.438 | 0.311 | 0.260 | 1.000 |
| Speckle | 0.740 | 0.369 | 0.167 | 1.500 | 0.384 | – | 0.920 | 0.413 | 0.308 | 0.745 | 0.353 | 0.000 | 0.846 | 0.355 | 0.500 | 0.750 | 0.308 | – | 0.668 | 0.445 | 0.281 | 1.262 | 0.470 | 0.167 | 0.760 | 0.490 | 0.375 | 0.556 | 0.303 | 1.000 |
| Poisson | 0.683 | 0.372 | 0.167 | 1.625 | 0.385 | – | 1.080 | 0.428 | 0.538 | 0.809 | 0.351 | 0.167 | 0.885 | 0.357 | 0.500 | 0.750 | 0.285 | – | 0.711 | 0.449 | 0.281 | 1.225 | 0.483 | 0.000 | 0.899 | 0.509 | 0.562 | 0.500 | 0.296 | 1.000 |
| Gaussian Blur | 0.667 | 0.380 | 0.167 | 1.250 | 0.385 | – | 0.973 | 0.403 | 0.385 | 0.655 | 0.336 | 0.167 | 0.692 | 0.346 | 0.500 | 0.750 | 0.295 | – | 0.728 | 0.443 | 0.188 | 1.363 | 0.477 | 0.333 | 0.961 | 0.490 | 0.375 | 0.544 | 0.317 | 1.000 |
| Motion Blur | 0.537 | 0.357 | 0.167 | 1.125 | 0.347 | – | 0.964 | 0.399 | 0.308 | 0.545 | 0.340 | 0.167 | 0.702 | 0.336 | 0.500 | 0.750 | 0.292 | – | 0.664 | 0.437 | 0.219 | 1.025 | 0.452 | 0.167 | 0.760 | 0.472 | 0.312 | 0.533 | 0.302 | 1.000 |
| Defocus Blur | 0.764 | 0.395 | 0.000 | 1.500 | 0.367 | – | 1.036 | 0.413 | 0.462 | 0.836 | 0.370 | 0.000 | 0.760 | 0.359 | 0.500 | 0.750 | 0.297 | – | 0.753 | 0.453 | 0.219 | 1.312 | 0.475 | 0.333 | 0.938 | 0.511 | 0.312 | 0.467 | 0.308 | 1.000 |
| Glass Blur | 0.707 | 0.386 | 0.167 | 1.875 | 0.386 | – | 1.152 | 0.429 | 0.615 | 0.891 | 0.359 | 0.056 | 0.808 | 0.354 | 0.500 | 0.750 | 0.321 | – | 0.817 | 0.461 | 0.281 | 1.413 | 0.489 | 0.333 | 0.930 | 0.499 | 0.500 | 0.400 | 0.298 | 1.000 |
| Zoom Blur | 0.691 | 0.386 | 0.167 | 1.375 | 0.376 | – | 1.250 | 0.423 | 0.615 | 0.718 | 0.346 | 0.222 | 0.760 | 0.358 | 0.500 | 1.250 | 0.340 | – | 0.719 | 0.453 | 0.250 | 1.212 | 0.460 | 0.167 | 0.930 | 0.487 | 0.438 | 0.433 | 0.298 | 1.000 |
| Frame Drop | 0.602 | 0.340 | 0.167 | 1.750 | 0.353 | – | 0.875 | 0.379 | 0.385 | 0.436 | 0.322 | 0.167 | 0.692 | 0.309 | 0.500 | 0.000 | 0.291 | – | 0.579 | 0.381 | 0.250 | 0.938 | 0.449 | 0.000 | 0.915 | 0.459 | 0.312 | 0.589 | 0.305 | 0.000 |
| Frame Replace | 0.935 | 0.412 | 0.000 | 1.750 | 0.373 | – | 1.339 | 0.433 | 0.538 | 0.964 | 0.352 | 0.111 | 1.019 | 0.367 | 0.500 | 1.000 | 0.362 | – | 0.838 | 0.467 | 0.281 | 1.275 | 0.487 | 0.167 | 0.969 | 0.506 | 0.375 | 0.856 | 0.340 | 1.000 |
| Frame Repeat | 0.276 | 0.277 | 0.167 | 1.000 | 0.271 | – | 0.857 | 0.372 | 0.308 | 0.509 | 0.292 | 0.056 | 0.740 | 0.319 | 0.500 | 0.000 | 0.275 | – | 0.468 | 0.352 | 0.281 | 0.812 | 0.408 | 0.167 | 0.713 | 0.447 | 0.312 | 0.344 | 0.248 | 0.000 |
| Temporal Jitter | 0.829 | 0.390 | 0.167 | 1.500 | 0.379 | – | 1.205 | 0.431 | 0.462 | 0.845 | 0.340 | 0.167 | 0.933 | 0.348 | 0.500 | 0.750 | 0.321 | – | 0.766 | 0.452 | 0.312 | 1.188 | 0.478 | 0.333 | 1.078 | 0.510 | 0.438 | 0.689 | 0.321 | 1.000 |
| Jpeg Artifact | 0.951 | 0.404 | 0.167 | 1.625 | 0.403 | – | 1.071 | 0.427 | 0.538 | 0.855 | 0.368 | 0.111 | 1.087 | 0.373 | 0.500 | 0.750 | 0.317 | – | 0.749 | 0.454 | 0.281 | 1.175 | 0.466 | 0.333 | 0.891 | 0.498 | 0.312 | 0.833 | 0.326 | 1.000 |
| Bit Error | 0.951 | 0.405 | 0.167 | 1.500 | 0.391 | – | 1.339 | 0.432 | 0.615 | 0.973 | 0.361 | 0.111 | 1.038 | 0.370 | 0.500 | 1.000 | 0.311 | – | 0.855 | 0.465 | 0.406 | 1.325 | 0.487 | 0.333 | 1.000 | 0.509 | 0.500 | 0.700 | 0.328 | 1.000 |
| H265 Artifacts | 0.650 | 0.384 | 0.000 | 1.750 | 0.395 | – | 1.018 | 0.409 | 0.538 | 0.727 | 0.360 | 0.111 | 0.808 | 0.343 | 0.500 | 0.750 | 0.293 | – | 0.749 | 0.453 | 0.281 | 1.413 | 0.460 | 0.333 | 0.860 | 0.498 | 0.562 | 0.400 | 0.289 | 1.000 |
| Random Block | 0.780 | 0.402 | 0.167 | 1.250 | 0.344 | – | 1.321 | 0.429 | 0.538 | 1.000 | 0.373 | 0.056 | 1.029 | 0.368 | 0.500 | 1.250 | 0.334 | – | 0.813 | 0.465 | 0.375 | 1.225 | 0.483 | 0.333 | 1.000 | 0.500 | 0.438 | 0.644 | 0.317 | 1.000 |
| Target Block | 0.951 | 0.405 | 0.167 | 1.500 | 0.379 | – | 1.143 | 0.412 | 0.462 | 0.927 | 0.358 | 0.111 | 0.990 | 0.362 | 0.500 | 1.000 | 0.341 | – | 0.817 | 0.464 | 0.344 | 1.038 | 0.435 | 0.333 | 1.039 | 0.515 | 0.438 | 0.767 | 0.323 | 1.000 |
| Rolling Shutter | 0.707 | 0.380 | 0.167 | 1.125 | 0.362 | – | 1.205 | 0.425 | 0.692 | 0.709 | 0.349 | 0.111 | 0.942 | 0.355 | 0.500 | 0.750 | 0.307 | – | 0.877 | 0.463 | 0.281 | 1.325 | 0.464 | 0.167 | 0.853 | 0.494 | 0.438 | 0.633 | 0.319 | 1.000 |
| Resolution Degrade | 0.797 | 0.383 | 0.167 | 1.375 | 0.382 | – | 1.232 | 0.441 | 0.538 | 0.600 | 0.354 | 0.111 | 0.683 | 0.342 | 0.500 | 0.750 | 0.308 | – | 0.668 | 0.451 | 0.281 | 1.150 | 0.456 | 0.500 | 0.969 | 0.507 | 0.500 | 0.522 | 0.305 | 1.000 |
| Stretch Squish | 0.691 | 0.375 | 0.167 | 1.625 | 0.403 | – | 1.062 | 0.412 | 0.385 | 0.655 | 0.344 | 0.167 | 0.615 | 0.330 | 0.500 | 0.750 | 0.362 | – | 0.719 | 0.446 | 0.188 | 1.087 | 0.457 | 0.333 | 0.961 | 0.503 | 0.500 | 0.422 | 0.303 | 1.000 |
| Edge Sawtooth | 0.935 | 0.411 | 0.167 | 1.625 | 0.405 | – | 1.179 | 0.442 | 0.615 | 0.864 | 0.348 | 0.056 | 0.865 | 0.349 | 0.500 | 0.750 | 0.336 | – | 0.740 | 0.453 | 0.312 | 1.312 | 0.475 | 0.333 | 0.961 | 0.509 | 0.500 | 0.556 | 0.309 | 1.000 |
| Elastic | 1.057 | 0.409 | 0.167 | 1.375 | 0.398 | – | 1.161 | 0.419 | 0.615 | 0.627 | 0.348 | 0.111 | 1.048 | 0.369 | 0.500 | 0.750 | 0.319 | – | 0.838 | 0.471 | 0.375 | 1.375 | 0.466 | 0.333 | 0.946 | 0.502 | 0.438 | 0.667 | 0.332 | 1.000 |
| Color Quantized | 1.033 | 0.420 | 0.167 | 1.500 | 0.391 | – | 1.393 | 0.451 | 0.692 | 1.073 | 0.362 | 0.167 | 1.048 | 0.353 | 0.500 | 0.750 | 0.296 | – | 0.834 | 0.475 | 0.312 | 1.262 | 0.474 | 0.167 | 1.000 | 0.518 | 0.438 | 0.922 | 0.333 | 1.000 |
| Bright Transform | 0.919 | 0.402 | 0.167 | 1.250 | 0.363 | – | 1.232 | 0.433 | 0.615 | 0.818 | 0.357 | 0.167 | 1.077 | 0.365 | 0.500 | 0.750 | 0.310 | – | 0.809 | 0.466 | 0.312 | 1.238 | 0.475 | 0.167 | 0.977 | 0.508 | 0.438 | 0.833 | 0.334 | 1.000 |
| Contrast | 0.878 | 0.400 | 0.167 | 1.000 | 0.369 | – | 1.286 | 0.423 | 0.538 | 0.964 | 0.358 | 0.056 | 1.077 | 0.362 | 0.500 | 0.750 | 0.312 | – | 0.894 | 0.466 | 0.344 | 1.288 | 0.479 | 0.333 | 0.992 | 0.508 | 0.438 | 0.822 | 0.330 | 1.000 |
| Color Shift | 0.919 | 0.407 | 0.167 | 0.875 | 0.374 | – | 1.330 | 0.436 | 0.692 | 0.891 | 0.355 | 0.167 | 1.000 | 0.357 | 0.500 | 1.000 | 0.311 | – | 0.804 | 0.458 | 0.312 | 1.212 | 0.465 | 0.167 | 0.915 | 0.501 | 0.375 | 0.700 | 0.328 | 1.000 |
| Flicker | 0.911 | 0.398 | 0.167 | 1.250 | 0.382 | – | 1.312 | 0.417 | 0.615 | 0.955 | 0.360 | 0.167 | 1.096 | 0.369 | 0.500 | 0.750 | 0.309 | – | 0.885 | 0.463 | 0.375 | 1.337 | 0.474 | 0.167 | 1.070 | 0.521 | 0.438 | 0.633 | 0.314 | 1.000 |
| Overexposure | 0.862 | 0.398 | 0.167 | 1.500 | 0.362 | – | 1.321 | 0.434 | 0.538 | 0.882 | 0.359 | 0.167 | 1.048 | 0.356 | 0.500 | 0.750 | 0.269 | – | 0.830 | 0.458 | 0.344 | 1.275 | 0.476 | 0.167 | 1.016 | 0.507 | 0.438 | 0.767 | 0.324 | 1.000 |
| Underexposure | 0.919 | 0.410 | 0.000 | 1.250 | 0.372 | – | 1.223 | 0.428 | 0.462 | 0.973 | 0.367 | 0.167 | 1.096 | 0.364 | 0.500 | 0.750 | 0.328 | – | 0.787 | 0.466 | 0.375 | 1.288 | 0.468 | 0.167 | 0.961 | 0.515 | 0.438 | 0.756 | 0.330 | 1.000 |
| Rainy | 0.951 | 0.409 | 0.167 | 1.375 | 0.355 | – | 1.268 | 0.432 | 0.615 | 0.864 | 0.362 | 0.222 | 1.221 | 0.375 | 0.500 | 1.250 | 0.327 | – | 0.809 | 0.471 | 0.312 | 1.325 | 0.469 | 0.333 | 1.000 | 0.507 | 0.438 | 0.667 | 0.340 | 1.000 |
| Foggy | 0.902 | 0.407 | 0.167 | 1.250 | 0.366 | – | 1.384 | 0.440 | 0.615 | 0.936 | 0.372 | 0.111 | 1.183 | 0.377 | 0.500 | 1.000 | 0.339 | – | 0.774 | 0.470 | 0.344 | 1.163 | 0.476 | 0.167 | 1.062 | 0.507 | 0.500 | 0.678 | 0.323 | 1.000 |
| Snow | 1.041 | 0.410 | 0.167 | 1.375 | 0.363 | – | 1.473 | 0.446 | 0.615 | 0.827 | 0.359 | 0.167 | 1.106 | 0.369 | 0.500 | 0.750 | 0.295 | – | 0.889 | 0.464 | 0.375 | 1.275 | 0.467 | 0.333 | 1.039 | 0.513 | 0.375 | 0.789 | 0.336 | 1.000 |
| Frost | 0.878 | 0.404 | 0.167 | 1.375 | 0.369 | – | 1.312 | 0.429 | 0.538 | 0.964 | 0.355 | 0.111 | 1.019 | 0.367 | 0.500 | 1.000 | 0.317 | – | 0.843 | 0.461 | 0.438 | 1.275 | 0.479 | 0.167 | 0.977 | 0.513 | 0.375 | 0.789 | 0.335 | 1.000 |
| Reflect | 0.902 | 0.395 | 0.167 | 1.125 | 0.377 | – | 1.223 | 0.430 | 0.615 | 0.927 | 0.358 | 0.167 | 0.913 | 0.362 | 0.500 | 1.500 | 0.338 | – | 0.851 | 0.472 | 0.344 | 1.300 | 0.487 | 0.167 | 1.078 | 0.518 | 0.438 | 0.756 | 0.319 | 1.000 |
| Shadow | 1.016 | 0.416 | 0.167 | 1.250 | 0.374 | – | 1.330 | 0.434 | 0.538 | 0.936 | 0.369 | 0.167 | 1.077 | 0.359 | 0.500 | 0.750 | 0.313 | – | 0.851 | 0.475 | 0.312 | 1.225 | 0.476 | 0.333 | 0.953 | 0.514 | 0.438 | 0.833 | 0.333 | 1.000 |

| Noise Type | K | | | L | | | M | | | N | | | O | | | P | | | Q | | | R | | | S | | | T | | |
|---|---|---|---|---|---|---|---|---|---|---|---|---|---|---|---|---|---|---|---|---|---|---|---|---|---|---|---|---|---|---|
| | GPT | SBERT | Acc | GPT | SBERT | Acc | GPT | SBERT | Acc | GPT | SBERT | Acc | GPT | SBERT | Acc | GPT | SBERT | Acc | GPT | SBERT | Acc | GPT | SBERT | Acc | GPT | SBERT | Acc | GPT | SBERT | Acc |
| Clean | 0.591 | 0.351 | 0.667 | 1.773 | 0.412 | 1.000 | 1.014 | 0.511 | 0.214 | 1.179 | 0.351 | 0.600 | 0.923 | 0.345 | 0.000 | 1.011 | 0.409 | 0.105 | 1.187 | 0.418 | 1.000 | 1.068 | 0.550 | 0.200 | 1.660 | 0.459 | 0.571 | 1.075 | 0.434 | 0.355 |
| Gaussian | 0.453 | 0.335 | 0.667 | 1.379 | 0.398 | 1.000 | 0.803 | 0.497 | 0.286 | 0.874 | 0.333 | 0.600 | 1.077 | 0.366 | 0.000 | 0.737 | 0.406 | 0.053 | 1.022 | 0.402 | 1.000 | 0.951 | 0.556 | 0.200 | 1.415 | 0.441 | 0.286 | 0.863 | 0.422 | 0.289 |
| Impulse | 0.358 | 0.310 | 0.667 | 1.167 | 0.370 | 0.667 | 0.769 | 0.487 | 0.286 | 0.611 | 0.297 | 0.400 | 0.692 | 0.318 | 0.000 | 0.565 | 0.366 | 0.053 | 0.802 | 0.351 | 0.000 | 0.778 | 0.522 | 0.200 | 0.943 | 0.405 | 0.286 | 0.664 | 0.391 | 0.263 |
| Speckle | 0.453 | 0.317 | 0.333 | 1.152 | 0.366 | 1.000 | 0.850 | 0.501 | 0.357 | 0.747 | 0.314 | 0.200 | 0.923 | 0.362 | 0.000 | 0.780 | 0.397 | 0.053 | 0.791 | 0.376 | 0.000 | 0.883 | 0.536 | 0.200 | 1.226 | 0.426 | 0.286 | 0.811 | 0.411 | 0.243 |
| Poisson | 0.372 | 0.320 | 0.333 | 1.227 | 0.362 | 1.000 | 0.850 | 0.498 | 0.357 | 0.842 | 0.322 | 0.600 | 0.846 | 0.345 | 0.000 | 0.677 | 0.394 | 0.053 | 1.000 | 0.388 | 0.000 | 1.037 | 0.549 | 0.200 | 1.283 | 0.432 | 0.286 | 0.849 | 0.415 | 0.309 |
| Gaussian Blur | 0.365 | 0.316 | 0.667 | 1.106 | 0.378 | 0.667 | 0.844 | 0.502 | 0.429 | 0.958 | 0.331 | 0.400 | 0.923 | 0.385 | 0.000 | 0.656 | 0.380 | 0.158 | 1.033 | 0.395 | 0.000 | 0.802 | 0.543 | 0.200 | 1.245 | 0.426 | 0.286 | 0.815 | 0.411 | 0.283 |
| Motion Blur | 0.350 | 0.316 | 0.667 | 1.152 | 0.371 | 0.667 | 0.925 | 0.498 | 0.429 | 0.695 | 0.312 | 0.400 | 0.923 | 0.373 | 0.000 | 0.715 | 0.392 | 0.105 | 1.088 | 0.396 | 0.000 | 0.877 | 0.538 | 0.200 | 1.151 | 0.434 | 0.286 | 0.769 | 0.405 | 0.263 |
| Defocus Blur | 0.460 | 0.339 | 0.667 | 1.591 | 0.401 | 1.000 | 0.850 | 0.500 | 0.357 | 0.821 | 0.322 | 0.600 | 0.923 | 0.330 | 0.000 | 0.801 | 0.399 | 0.105 | 1.099 | 0.393 | 0.000 | 0.907 | 0.548 | 0.200 | 1.142 | 0.424 | 0.429 | 0.871 | 0.421 | 0.270 |
| Glass Blur | 0.474 | 0.335 | 0.667 | 1.561 | 0.396 | 1.000 | 0.844 | 0.499 | 0.357 | 0.674 | 0.315 | 0.400 | 1.077 | 0.358 | 0.000 | 0.801 | 0.394 | 0.105 | 1.110 | 0.406 | 0.000 | 1.019 | 0.550 | 0.200 | 1.311 | 0.434 | 0.429 | 0.901 | 0.421 | 0.322 |
| Zoom Blur | 0.423 | 0.328 | 0.667 | 1.470 | 0.421 | 1.000 | 0.857 | 0.503 | 0.286 | 0.842 | 0.328 | 0.600 | 1.000 | 0.383 | 0.000 | 0.828 | 0.400 | 0.105 | 1.110 | 0.396 | 0.000 | 1.031 | 0.547 | 0.200 | 1.170 | 0.415 | 0.286 | 0.873 | 0.417 | 0.316 |
| Frame Drop | 0.314 | 0.274 | 0.000 | 1.030 | 0.352 | 0.667 | 0.830 | 0.485 | 0.357 | 0.747 | 0.297 | 0.200 | 0.462 | 0.237 | 0.000 | 0.796 | 0.371 | 0.158 | 0.956 | 0.372 | 0.000 | 0.772 | 0.508 | 0.000 | 1.387 | 0.424 | 0.286 | 0.753 | 0.381 | 0.237 |
| Frame Replace | 0.496 | 0.343 | 0.667 | 1.758 | 0.421 | 1.000 | 0.905 | 0.506 | 0.214 | 1.179 | 0.342 | 0.400 | 0.923 | 0.318 | 0.000 | 0.973 | 0.410 | 0.053 | 1.077 | 0.414 | 1.000 | 1.049 | 0.549 | 0.200 | 1.623 | 0.450 | 0.286 | 1.030 | 0.431 | 0.276 |
| Frame Repeat | 0.292 | 0.280 | 0.000 | 1.394 | 0.353 | 0.333 | 0.748 | 0.470 | 0.286 | 0.558 | 0.282 | 0.000 | 0.462 | 0.307 | 0.000 | 0.747 | 0.351 | 0.053 | 0.923 | 0.387 | 0.000 | 0.778 | 0.473 | 0.200 | 0.934 | 0.395 | 0.286 | 0.660 | 0.360 | 0.204 |
| Temporal Jitter | 0.467 | 0.320 | 0.667 | 1.621 | 0.411 | 1.000 | 0.728 | 0.498 | 0.214 | 1.042 | 0.338 | 0.600 | 0.846 | 0.326 | 0.000 | 0.876 | 0.389 | 0.105 | 1.165 | 0.387 | 0.000 | 0.901 | 0.534 | 0.200 | 1.415 | 0.439 | 0.571 | 0.936 | 0.418 | 0.322 |
| Jpeg Artifact | 0.504 | 0.338 | 0.667 | 1.470 | 0.406 | 1.000 | 0.844 | 0.493 | 0.286 | 1.084 | 0.327 | 0.600 | 1.077 | 0.382 | 0.000 | 0.812 | 0.406 | 0.053 | 1.033 | 0.399 | 1.000 | 1.068 | 0.557 | 0.200 | 1.368 | 0.443 | 0.286 | 0.946 | 0.425 | 0.296 |
| Bit Error | 0.526 | 0.337 | 0.667 | 1.682 | 0.408 | 1.000 | 0.959 | 0.511 | 0.286 | 1.211 | 0.349 | 0.600 | 0.846 | 0.357 | 0.000 | 0.995 | 0.423 | 0.053 | 1.154 | 0.421 | 1.000 | 1.068 | 0.545 | 0.200 | 1.689 | 0.462 | 0.571 | 1.046 | 0.433 | 0.362 |
| H265 Artifacts | 0.387 | 0.336 | 0.667 | 1.212 | 0.379 | 1.000 | 0.864 | 0.494 | 0.429 | 0.505 | 0.267 | 0.400 | 0.923 | 0.363 | 0.000 | 0.634 | 0.385 | 0.105 | 1.077 | 0.394 | 1.000 | 0.877 | 0.550 | 0.200 | 0.906 | 0.419 | 0.286 | 0.793 | 0.411 | 0.329 |
| Random Block | 0.496 | 0.331 | 0.667 | 1.788 | 0.417 | 1.000 | 0.871 | 0.512 | 0.214 | 1.105 | 0.339 | 0.600 | 1.077 | 0.384 | 0.000 | 0.903 | 0.415 | 0.105 | 1.132 | 0.408 | 1.000 | 1.093 | 0.549 | 0.200 | 1.594 | 0.459 | 0.286 | 1.002 | 0.430 | 0.322 |
| Target Block | 0.533 | 0.350 | 0.667 | 1.515 | 0.390 | 0.667 | 0.952 | 0.504 | 0.214 | 1.147 | 0.352 | 0.600 | 0.846 | 0.352 | 0.000 | 0.876 | 0.402 | 0.105 | 1.055 | 0.402 | 1.000 | 1.062 | 0.540 | 0.200 | 1.481 | 0.445 | 0.571 | 0.983 | 0.425 | 0.322 |
| Rolling Shutter | 0.438 | 0.328 | 0.667 | 1.561 | 0.389 | 0.667 | 0.878 | 0.495 | 0.071 | 0.895 | 0.322 | 0.400 | 1.077 | 0.363 | 0.000 | 0.796 | 0.394 | 0.053 | 1.121 | 0.417 | 1.000 | 0.994 | 0.551 | 0.200 | 1.406 | 0.435 | 0.286 | 0.921 | 0.419 | 0.283 |
| Resolution Degrade | 0.445 | 0.322 | 0.667 | 1.409 | 0.391 | 1.000 | 0.871 | 0.503 | 0.357 | 0.674 | 0.290 | 0.600 | 0.769 | 0.340 | 0.000 | 0.694 | 0.397 | 0.105 | 0.978 | 0.411 | 0.000 | 0.907 | 0.549 | 0.200 | 1.311 | 0.435 | 0.429 | 0.835 | 0.417 | 0.336 |
| Stretch Squish | 0.409 | 0.329 | 0.667 | 1.227 | 0.362 | 0.667 | 0.789 | 0.504 | 0.357 | 0.758 | 0.306 | 0.600 | 1.154 | 0.388 | 0.000 | 0.667 | 0.401 | 0.105 | 0.956 | 0.386 | 0.000 | 0.895 | 0.542 | 0.200 | 1.179 | 0.439 | 0.286 | 0.798 | 0.412 | 0.289 |
| Edge Sawtooth | 0.496 | 0.337 | 0.667 | 1.606 | 0.404 | 1.000 | 0.932 | 0.494 | 0.214 | 0.968 | 0.331 | 0.600 | 1.077 | 0.371 | 0.000 | 0.823 | 0.407 | 0.053 | 1.044 | 0.418 | 0.000 | 1.093 | 0.548 | 0.200 | 1.349 | 0.442 | 0.429 | 0.944 | 0.424 | 0.316 |
| Elastic | 0.533 | 0.337 | 0.667 | 1.682 | 0.412 | 1.000 | 0.884 | 0.512 | 0.214 | 1.126 | 0.350 | 0.600 | 0.923 | 0.342 | 0.000 | 0.860 | 0.400 | 0.105 | 1.198 | 0.423 | 1.000 | 1.080 | 0.553 | 0.200 | 1.509 | 0.444 | 0.571 | 0.990 | 0.429 | 0.349 |
| Color Quantized | 0.533 | 0.334 | 0.667 | 1.939 | 0.423 | 1.000 | 1.068 | 0.512 | 0.357 | 1.200 | 0.340 | 0.600 | 1.000 | 0.358 | 0.000 | 1.000 | 0.420 | 0.105 | 1.176 | 0.404 | 1.000 | 1.049 | 0.552 | 0.000 | 1.623 | 0.457 | 0.571 | 1.078 | 0.434 | 0.349 |
| Bright Transform | 0.562 | 0.342 | 0.667 | 1.727 | 0.391 | 1.000 | 0.993 | 0.514 | 0.143 | 1.200 | 0.356 | 0.400 | 1.000 | 0.380 | 0.000 | 0.978 | 0.410 | 0.053 | 1.220 | 0.415 | 1.000 | 1.074 | 0.556 | 0.000 | 1.613 | 0.450 | 0.429 | 1.030 | 0.431 | 0.303 |
| Contrast | 0.613 | 0.341 | 0.667 | 1.818 | 0.420 | 1.000 | 0.878 | 0.504 | 0.071 | 1.032 | 0.342 | 0.600 | 1.077 | 0.379 | 0.000 | 0.995 | 0.422 | 0.053 | 1.154 | 0.409 | 0.000 | 1.037 | 0.537 | 0.200 | 1.792 | 0.469 | 0.429 | 1.046 | 0.430 | 0.296 |
| Color Shift | 0.599 | 0.346 | 0.333 | 1.652 | 0.408 | 1.000 | 0.871 | 0.494 | 0.071 | 1.158 | 0.339 | 0.200 | 1.154 | 0.400 | 0.000 | 0.935 | 0.409 | 0.105 | 1.165 | 0.409 | 1.000 | 1.056 | 0.553 | 0.000 | 1.717 | 0.456 | 0.429 | 1.011 | 0.427 | 0.289 |
| Flicker | 0.496 | 0.333 | 0.667 | 1.652 | 0.411 | 1.000 | 0.898 | 0.498 | 0.214 | 0.947 | 0.343 | 0.400 | 1.000 | 0.362 | 0.000 | 0.903 | 0.402 | 0.158 | 1.209 | 0.418 | 1.000 | 1.099 | 0.553 | 0.000 | 1.566 | 0.449 | 0.286 | 1.019 | 0.427 | 0.329 |
| Overexposure | 0.460 | 0.333 | 0.333 | 1.712 | 0.414 | 1.000 | 1.075 | 0.505 | 0.286 | 1.211 | 0.356 | 0.200 | 0.923 | 0.343 | 0.000 | 0.882 | 0.409 | 0.105 | 1.099 | 0.406 | 1.000 | 1.019 | 0.549 | 0.200 | 1.500 | 0.436 | 0.429 | 1.012 | 0.426 | 0.316 |
| Underexposure | 0.620 | 0.351 | 0.667 | 1.818 | 0.423 | 1.000 | 1.068 | 0.503 | 0.286 | 1.084 | 0.340 | 0.600 | 1.000 | 0.390 | 0.000 | 0.952 | 0.415 | 0.105 | 1.165 | 0.415 | 1.000 | 1.068 | 0.542 | 0.000 | 1.472 | 0.447 | 0.286 | 1.028 | 0.431 | 0.316 |
| Rainy | 0.511 | 0.338 | 0.667 | 1.803 | 0.414 | 1.000 | 0.959 | 0.511 | 0.286 | 1.053 | 0.345 | 0.600 | 0.923 | 0.376 | 0.000 | 0.903 | 0.405 | 0.053 | 0.967 | 0.390 | 1.000 | 0.957 | 0.537 | 0.200 | 1.642 | 0.461 | 0.429 | 1.006 | 0.430 | 0.342 |
| Foggy | 0.460 | 0.326 | 0.667 | 1.606 | 0.413 | 1.000 | 0.898 | 0.508 | 0.214 | 1.105 | 0.345 | 0.600 | 0.615 | 0.344 | 0.000 | 0.984 | 0.415 | 0.105 | 1.165 | 0.406 | 0.000 | 0.969 | 0.544 | 0.200 | 1.604 | 0.458 | 0.429 | 1.006 | 0.431 | 0.329 |
| Snow | 0.562 | 0.340 | 0.667 | 1.727 | 0.408 | 1.000 | 1.027 | 0.511 | 0.214 | 1.137 | 0.331 | 0.600 | 0.923 | 0.362 | 0.000 | 0.903 | 0.408 | 0.053 | 1.099 | 0.410 | 1.000 | 1.105 | 0.552 | 0.200 | 1.708 | 0.470 | 0.571 | 1.061 | 0.431 | 0.342 |
| Frost | 0.526 | 0.342 | 0.667 | 1.636 | 0.399 | 1.000 | 0.912 | 0.508 | 0.214 | 1.032 | 0.336 | 0.600 | 0.923 | 0.347 | 0.000 | 0.882 | 0.401 | 0.105 | 1.022 | 0.387 | 1.000 | 1.037 | 0.545 | 0.200 | 1.509 | 0.446 | 0.429 | 0.995 | 0.426 | 0.336 |
| Reflect | 0.562 | 0.344 | 0.333 | 1.636 | 0.404 | 1.000 | 0.932 | 0.514 | 0.214 | 1.158 | 0.362 | 0.200 | 0.923 | 0.358 | 0.000 | 0.930 | 0.416 | 0.053 | 1.132 | 0.415 | 1.000 | 1.006 | 0.550 | 0.000 | 1.585 | 0.454 | 0.571 | 1.012 | 0.433 | 0.309 |
| Shadow | 0.569 | 0.339 | 0.667 | 1.742 | 0.415 | 1.000 | 0.918 | 0.503 | 0.214 | 1.221 | 0.344 | 0.600 | 1.000 | 0.382 | 0.000 | 0.978 | 0.405 | 0.053 | 1.220 | 0.416 | 1.000 | 1.086 | 0.549 | 0.200 | 1.604 | 0.463 | 0.571 | 1.048 | 0.433 | 0.329 |

Note: A: Advertisements, B: Algorithm & Models, C: Autos & Vehicles, D: Business & Industrial, E: Computers & Electronics, F: Fairy Tale, G: Films & TV Shows, H: Finance, I: Food & Drink, J: Games, K: Humor, L: Instruction Video (how to), M: Knowledge, N: News, O: Others, P: People, Q: Pets & Animals, R: Science, S: Sports, T: Overall

Table 9: Video-ChatGPT performance across different video types

| Noise Type | A GPT | A SBERT | A Acc | B GPT | B SBERT | B Acc | C GPT | C SBERT | C Acc | D GPT | D SBERT | D Acc | E GPT | E SBERT | E Acc | F GPT | F SBERT | F Acc | G GPT | G SBERT | G Acc | H GPT | H SBERT | H Acc | I GPT | I SBERT | I Acc | J GPT | J SBERT | J Acc |
|---|---|---|---|---|---|---|---|---|---|---|---|---|---|---|---|---|---|---|---|---|---|---|---|---|---|---|---|---|---|---|
| Clean | 0.675 | 0.385 | 0.333 | 0.625 | 0.289 | — | 0.071 | 0.431 | 0.769 | 0.351 | 0.343 | 0.500 | 0.354 | — | — | 0.769 | 0.618 | 0.384 | 0.745 | 0.470 | 0.500 | 0.912 | 0.522 | 0.750 | 0.533 | 0.319 | 1.000 |
| Gaussian | 0.626 | 0.366 | 0.333 | 0.375 | 0.289 | — | 0.946 | 0.432 | 0.846 | 0.343 | 0.750 | 0.339 | 0.355 | 0.750 | 0.627 | 0.355 | 0.600 | 0.466 | 0.375 | 0.787 | 0.454 | 0.750 | 0.853 | 0.508 | 0.688 | 0.433 | 0.300 | 1.000 |
| Impulse | 0.431 | 0.359 | 0.333 | 0.500 | 0.363 | — | 0.964 | 0.425 | 0.923 | 0.336 | 0.222 | 0.341 | 0.365 | 0.0000 | 0.654 | 0.340 | 0.532 | 0.469 | 0.436 | 0.725 | 0.463 | 0.688 | 0.356 | 0.287 | 1.000 |
| Speckle | 0.545 | 0.370 | 0.333 | 0.375 | 0.291 | — | 0.911 | 0.424 | 0.846 | 0.347 | 0.750 | 0.500 | 0.419 | 0.167 | 0.609 | 0.363 | 0.681 | 0.452 | 0.463 | 0.887 | 0.445 | 0.750 | 0.516 | 0.319 | 1.000 |
| Poisson | 0.569 | 0.357 | 0.333 | 0.250 | 0.268 | — | 0.973 | 0.424 | 0.846 | 0.345 | 1.077 | 0.295 | 0.411 | 0.663 | 0.337 | 0.604 | 0.459 | 0.445 | 0.775 | 0.459 | 0.750 | 0.505 | 0.294 | 1.000 |
| Gaussian Blur | 0.545 | 0.362 | 0.333 | 0.375 | 0.270 | — | 0.946 | 0.434 | 0.846 | 0.338 | 0.853 | 0.357 | 0.409 | 0.674 | 0.636 | 0.371 | 0.574 | 0.452 | 0.457 | 0.825 | 0.469 | 0.688 | 0.524 | 0.296 | 1.000 |
| Motion Blur | 0.593 | 0.374 | 0.333 | 0.375 | 0.280 | — | 0.884 | 0.429 | 0.769 | 0.348 | 0.789 | 0.294 | 0.388 | 0.745 | 0.642 | 0.361 | 0.681 | 0.461 | 0.469 | 0.800 | 0.457 | 0.688 | 0.526 | 0.308 | 1.000 |
| Defocus Blur | 0.602 | 0.381 | 0.333 | 0.875 | 0.273 | — | 0.884 | 0.433 | 0.846 | 0.356 | 0.800 | 0.312 | 0.369 | 0.718 | 0.655 | 0.371 | 0.749 | 0.473 | 0.472 | 0.825 | 0.469 | 0.750 | 0.529 | 0.317 | 1.000 |
| Glass Blur | 0.691 | 0.385 | 0.333 | 0.875 | 0.292 | — | 0.866 | 0.434 | 0.846 | 0.346 | 0.663 | 0.231 | 0.392 | 0.712 | 0.655 | 0.918 | 0.536 | 0.463 | 0.473 | 0.750 | 0.449 | 0.750 | 0.516 | 0.318 | 1.000 |
| Zoom Blur | 0.569 | 0.373 | 0.333 | 0.375 | 0.276 | — | 1.054 | 0.399 | 0.509 | 0.330 | 0.625 | 0.346 | 0.337 | 0.625 | 0.655 | 0.316 | 0.489 | 0.452 | 0.472 | 0.750 | 0.444 | 0.688 | 0.411 | 0.321 | 1.000 |
| Frame Drop | 0.504 | 0.350 | 0.350 | 0.500 | 0.276 | — | 1.000 | 0.435 | 0.769 | 0.359 | 0.913 | 0.312 | 0.389 | 0.779 | 0.718 | 0.376 | 0.740 | 0.463 | 0.449 | 0.837 | 0.442 | 0.688 | 0.589 | 0.288 | 1.000 |
| Frame Replace | 0.683 | 0.381 | 0.333 | 0.500 | 0.290 | — | 1.054 | 0.428 | 0.923 | 0.361 | 0.837 | 0.283 | 0.347 | 0.773 | 0.584 | 0.376 | 0.475 | 0.467 | 0.863 | 0.467 | 0.750 | 0.422 | 0.323 | 1.000 |
| Frame Repeat | 0.561 | 0.362 | 0.333 | 0.250 | 0.277 | — | 1.054 | 0.426 | 0.923 | 0.353 | 0.788 | 0.308 | 0.367 | 0.740 | 0.627 | 0.347 | 0.600 | 0.460 | 0.458 | 0.838 | 0.458 | 0.750 | 0.400 | 0.296 | 1.000 |
| Temporal Jitter | 0.610 | 0.362 | 0.333 | 0.375 | 0.295 | — | 0.929 | 0.442 | 0.769 | 0.359 | 0.808 | 0.296 | 0.358 | 0.747 | 0.545 | 0.309 | 0.677 | 0.468 | 0.451 | 0.838 | 0.451 | 0.750 | 0.400 | 0.300 | 1.000 |
| Jpeg Artifact | 0.618 | 0.376 | 0.333 | 0.375 | 0.273 | — | 0.902 | 0.435 | 0.692 | 0.358 | 0.827 | 0.315 | 0.373 | 0.705 | 0.709 | 0.373 | 0.783 | 0.480 | 0.469 | 0.900 | 0.454 | 0.750 | 0.625 | 0.309 | 1.000 |
| Bit Error | 0.561 | 0.378 | 0.333 | 0.750 | 0.294 | — | 1.054 | 0.432 | 0.923 | 0.346 | 0.768 | 0.342 | 0.318 | 0.691 | 0.627 | 0.318 | 0.689 | 0.473 | 0.455 | 0.775 | 0.455 | 0.625 | 0.522 | 0.320 | 1.000 |
| H265 Artifacts | 0.520 | 0.367 | 0.500 | 0.875 | 0.287 | — | 0.982 | 0.428 | 0.769 | 0.358 | 0.885 | 0.346 | 0.400 | 0.750 | 0.764 | 0.318 | 0.753 | 0.473 | 0.454 | 0.838 | 0.500 | 0.688 | 0.533 | 0.314 | 1.000 |
| Random Block | 0.618 | 0.378 | 0.333 | 0.500 | 0.288 | — | 1.000 | 0.432 | 0.846 | 0.347 | 0.853 | 0.390 | 0.324 | 0.500 | 0.655 | 0.345 | 0.745 | 0.464 | 0.463 | 0.950 | 0.406 | 0.812 | 0.544 | 0.313 | 1.000 |
| Rolling Shutter | 0.545 | 0.373 | 0.333 | 0.375 | 0.251 | 0.250 | 0.964 | 0.435 | 0.923 | 0.352 | 0.768 | 0.300 | 0.346 | 0.721 | 0.700 | 0.308 | 0.672 | 0.475 | 0.457 | 0.787 | 0.406 | 0.750 | 0.533 | 0.315 | 1.000 |
| Resolution Degrade | 0.488 | 0.366 | 0.333 | 0.625 | 0.270 | — | 0.991 | 0.428 | 0.846 | 0.362 | 0.779 | 0.779 | 0.349 | 0.779 | 0.764 | 0.400 | 0.719 | 0.477 | 0.464 | 0.838 | 0.469 | 0.750 | 0.533 | 0.321 | 1.000 |
| Stretch Squish | 0.642 | 0.377 | 0.500 | 0.375 | 0.288 | 0.270 | 1.018 | 0.443 | 0.923 | 0.352 | 0.798 | 0.373 | 0.320 | 0.750 | 0.700 | 0.345 | 0.711 | 0.475 | 0.475 | 0.950 | 0.463 | 0.750 | 0.522 | 0.320 | 1.000 |
| Edge Sawtooth | 0.667 | 0.382 | 0.333 | 0.250 | 0.283 | — | 1.027 | 0.434 | 0.769 | 0.358 | 0.636 | 0.392 | 0.353 | 0.750 | 0.664 | 0.369 | 0.719 | 0.482 | 0.462 | 0.787 | 0.438 | 0.750 | 0.656 | 0.318 | 1.000 |
| Elastic | 0.683 | 0.387 | 0.333 | 0.375 | 0.279 | — | 1.054 | 0.428 | 0.846 | 0.349 | 0.779 | 0.279 | 0.375 | 0.721 | 0.700 | 0.320 | 0.732 | 0.472 | 0.466 | 0.984 | 0.453 | 0.750 | 0.533 | 0.325 | 1.000 |
| Color Quantized | 0.626 | 0.379 | 0.333 | 0.625 | 0.280 | — | 1.009 | 0.443 | 0.769 | 0.352 | 0.779 | 0.740 | 0.375 | 0.749 | 0.636 | 0.375 | 0.736 | 0.474 | 0.449 | 0.953 | 0.466 | 0.625 | 0.644 | 0.318 | 1.000 |
| Bright Transform | 0.675 | 0.389 | 0.500 | 0.375 | 0.280 | — | 1.071 | 0.434 | 0.846 | 0.350 | 0.808 | 0.582 | 0.333 | 0.677 | 0.718 | 0.321 | 0.677 | 0.482 | 0.465 | 0.992 | 0.449 | 0.562 | 0.544 | 0.314 | 1.000 |
| Contrast | 0.675 | 0.368 | 0.333 | 0.375 | 0.281 | — | 0.982 | 0.431 | 0.692 | 0.355 | 0.779 | 0.231 | 0.333 | 0.732 | 0.582 | 0.352 | 0.732 | 0.474 | 0.461 | 0.907 | 0.465 | 0.625 | 0.544 | 0.314 | 1.000 |
| Color Shift | 0.593 | 0.372 | 0.333 | 0.375 | 0.268 | — | 1.018 | 0.435 | 0.846 | 0.355 | 0.779 | 0.740 | 0.344 | 0.500 | 0.655 | 0.352 | 0.677 | 0.477 | 0.469 | 0.812 | 0.466 | 0.625 | 0.433 | 0.310 | 1.000 |
| Flicker | 0.691 | 0.380 | 0.333 | 0.375 | 0.292 | — | 1.009 | 0.428 | 0.923 | 0.349 | 0.798 | 0.582 | 0.350 | 0.500 | 0.718 | 0.321 | 0.719 | 0.473 | 0.473 | 0.850 | 0.466 | 0.625 | 0.522 | 0.322 | 1.000 |
| Overexposure | 0.618 | 0.383 | 0.333 | 0.375 | 0.275 | — | 1.045 | 0.443 | 0.846 | 0.353 | 0.702 | 0.279 | 0.352 | 0.500 | 0.627 | 0.314 | 0.643 | 0.472 | 0.472 | 0.800 | 0.438 | 0.625 | 0.544 | 0.310 | 1.000 |
| Underexposure | 0.659 | 0.382 | 0.333 | 0.500 | 0.267 | — | 1.071 | 0.434 | 0.769 | 0.347 | 0.702 | 0.279 | 0.360 | 0.753 | 0.709 | 0.310 | 0.745 | 0.478 | 0.473 | 0.775 | 0.375 | 0.625 | 0.611 | 0.323 | 1.000 |
| Rainy | 0.650 | 0.382 | 0.333 | 0.375 | 0.259 | — | 1.000 | 0.431 | 0.846 | 0.353 | 0.798 | 0.702 | 0.321 | 0.500 | 0.627 | 0.360 | 0.689 | 0.472 | 0.465 | 0.850 | 0.465 | 0.750 | 0.544 | 0.315 | 1.000 |
| Foggy | 0.561 | 0.383 | 0.333 | 0.375 | 0.288 | — | 0.982 | 0.435 | 0.769 | 0.344 | 0.673 | 0.231 | 0.310 | 0.655 | 0.709 | 0.310 | 0.655 | 0.478 | 0.466 | 0.838 | 0.406 | 0.750 | 0.478 | 0.311 | 1.000 |
| Snow | 0.659 | 0.383 | 0.333 | 0.625 | 0.267 | — | 1.018 | 0.435 | 0.923 | 0.352 | 0.709 | 0.077 | 0.328 | 0.711 | 0.709 | 0.344 | 0.711 | 0.525 | 0.469 | 0.787 | 0.438 | 0.750 | 0.511 | 0.310 | 1.000 |
| Frost | 0.520 | 0.364 | 0.333 | 0.625 | 0.288 | — | 1.009 | 0.431 | 0.846 | 0.351 | 0.788 | 0.278 | 0.355 | 0.753 | 0.753 | 0.476 | 0.711 | 0.468 | 0.406 | 0.863 | 0.472 | 0.750 | 0.533 | 0.311 | 1.000 |
| Reflect | 0.585 | 0.384 | 0.333 | 0.875 | 0.279 | — | 0.973 | 0.431 | 0.846 | 0.350 | 0.700 | 0.167 | 0.344 | 0.753 | 0.700 | 0.472 | 0.472 | 0.863 | 0.472 | 0.750 | 0.511 | 0.311 | 1.000 |
| Shadow | 0.707 | 0.384 | 0.333 | 0.500 | 0.291 | — | 0.946 | 0.438 | 0.846 | 0.363 | 0.753 | 0.528 | 0.311 | 1.000 | 1.031 | 0.528 | 0.750 | 0.533 | 0.311 | 1.000 |

| Noise Type | K GPT | K SBERT | K Acc | L GPT | L SBERT | L Acc | M GPT | M SBERT | M Acc | N GPT | N SBERT | N Acc | O GPT | O SBERT | O Acc | P GPT | P SBERT | P Acc | Q GPT | Q SBERT | Q Acc | R GPT | R SBERT | R Acc | S GPT | S SBERT | S Acc | T GPT | T SBERT | T Acc |
|---|---|---|---|---|---|---|---|---|---|---|---|---|---|---|---|---|---|---|---|---|---|---|---|---|---|---|---|---|---|---|
| Clean | 0.518 | 0.352 | 0.667 | 1.106 | 0.407 | 1.000 | 0.639 | 0.506 | 1.000 | 0.768 | 0.737 | 0.429 | 0.375 | 1.154 | 0.672 | 0.401 | 0.400 | 1.077 | 0.211 | 0.535 | 1.000 | 0.586 | 0.466 | 0.292 | 0.535 | 0.000 | 0.571 | 0.783 | 0.427 | 0.467 |
| Gaussian | 0.401 | 0.319 | 0.667 | 1.045 | 0.403 | 1.000 | 0.639 | 0.496 | 1.000 | 0.737 | 0.558 | 0.429 | 0.399 | 0.543 | 0.375 | 0.399 | 1.110 | 0.263 | 0.521 | 0.574 | 0.962 | 0.433 | 0.571 | 0.706 | 0.412 | 0.434 |
| Impulse | 0.314 | 0.318 | 0.667 | 1.000 | 0.363 | 1.000 | 0.496 | 0.505 | 1.000 | 0.558 | 0.286 | 0.365 | 0.473 | 0.372 | 0.211 | 0.934 | 0.211 | 0.508 | 0.514 | 1.075 | 0.439 | 0.571 | 0.632 | 0.402 | 0.434 |
| Speckle | 0.482 | 0.339 | 0.667 | 1.030 | 0.387 | 1.000 | 0.505 | 0.505 | 1.000 | 0.716 | 0.286 | 0.419 | 0.613 | 0.391 | 1.099 | 0.391 | 0.211 | 0.527 | 0.586 | 1.075 | 0.444 | 0.571 | 0.723 | 0.418 | 0.461 |
| Poisson | 0.358 | 0.318 | 0.667 | 0.879 | 0.360 | 1.000 | 0.612 | 0.499 | 1.000 | 0.674 | 0.357 | 0.478 | 0.411 | 0.263 | 0.967 | 0.380 | 0.443 | 0.523 | 0.611 | 0.028 | 0.571 | 0.657 | 0.405 | 0.434 |
| Gaussian Blur | 0.431 | 0.327 | 0.667 | 1.061 | 0.389 | 1.000 | 0.605 | 0.500 | 1.000 | 0.684 | 0.357 | 0.532 | 0.388 | 0.263 | 1.066 | 0.396 | 0.380 | 0.524 | 0.574 | 0.414 | 0.571 | 0.700 | 0.412 | 0.447 |
| Motion Blur | 0.467 | 0.329 | 0.667 | 1.076 | 0.401 | 1.000 | 0.571 | 0.495 | 1.000 | 0.642 | 0.294 | 0.409 | 0.388 | 0.677 | 0.383 | 0.399 | 1.121 | 0.263 | 0.526 | 0.556 | 0.868 | 0.440 | 0.571 | 0.702 | 0.413 | 0.434 |
| Defocus Blur | 0.606 | 0.339 | 0.667 | 1.091 | 0.403 | 1.000 | 0.639 | 0.510 | 1.000 | 0.800 | 0.357 | 0.369 | 0.597 | 0.399 | 1.088 | 0.211 | 0.516 | 0.623 | 1.070 | 0.457 | 0.571 | 0.773 | 0.423 | 0.474 |
| Glass Blur | 0.482 | 0.340 | 0.667 | 1.167 | 0.420 | 1.000 | 0.667 | 0.518 | 1.000 | 0.663 | 0.357 | 0.392 | 0.452 | 0.394 | 1.044 | 0.316 | 0.531 | 0.580 | 1.170 | 0.456 | 0.571 | 0.695 | 0.415 | 0.454 |
| Zoom Blur | 0.489 | 0.335 | 0.667 | 1.030 | 0.390 | 1.000 | 0.626 | 0.491 | 1.000 | 0.611 | 0.357 | 0.389 | 0.672 | 0.263 | 0.934 | 0.279 | 0.531 | 0.630 | 0.840 | 0.085 | 0.571 | 0.634 | 0.396 | 0.441 |
| Frame Drop | 0.314 | 0.312 | 0.667 | 0.833 | 0.335 | 1.000 | 0.680 | 0.502 | 1.000 | 0.779 | 0.308 | 0.370 | 0.672 | 0.364 | 1.220 | 0.415 | 0.623 | 0.417 | 0.571 | 0.804 | 0.396 | 0.467 |
| Frame Replace | 0.518 | 0.342 | 0.667 | 1.182 | 0.410 | 1.000 | 0.673 | 0.511 | 1.000 | 0.895 | 0.429 | 0.394 | 0.683 | 0.211 | 0.956 | 0.385 | 0.623 | 0.460 | 0.571 | 1.160 | 0.464 | 0.571 | 0.787 | 0.426 | 0.467 |
| Frame Repeat | 0.613 | 0.343 | 0.667 | 1.015 | 0.393 | 1.000 | 0.721 | 0.511 | 1.000 | 0.747 | 0.429 | 0.394 | 0.586 | 0.391 | 0.989 | 0.263 | 0.513 | 0.617 | 0.160 | 0.571 | 0.770 | 0.418 | 0.454 |
| Temporal Jitter | 0.365 | 0.325 | 0.667 | 1.076 | 0.366 | 1.000 | 0.619 | 0.513 | 1.000 | 0.705 | 0.357 | 0.296 | 0.621 | 0.307 | 1.176 | 0.412 | 0.528 | 0.028 | 0.571 | 0.701 | 0.407 | 0.467 |
| Jpeg Artifact | 0.474 | 0.335 | 0.667 | 1.167 | 0.397 | 1.000 | 0.667 | 0.513 | 1.000 | 0.705 | 0.357 | 0.393 | 1.242 | 0.263 | 0.574 | 0.457 | 0.113 | 0.571 | 0.746 | 0.422 | 0.474 |
| Bit Error | 0.445 | 0.336 | 0.667 | 1.167 | 0.408 | 1.000 | 0.680 | 0.505 | 1.000 | 0.853 | 0.357 | 0.307 | 0.645 | 0.390 | 1.066 | 0.386 | 0.574 | 0.460 | 0.255 | 0.571 | 0.795 | 0.427 | 0.474 |
| H265 Artifacts | 0.372 | 0.317 | 0.667 | 1.121 | 0.411 | 1.000 | 0.571 | 0.501 | 1.000 | 0.768 | 0.357 | 0.393 | 0.575 | 0.386 | 1.088 | 0.263 | 0.452 | 0.458 | 0.057 | 0.571 | 0.726 | 0.417 | 0.434 |
| Random Block | 0.401 | 0.342 | 0.667 | 1.318 | 0.420 | 1.000 | 0.673 | 0.510 | 1.000 | 0.811 | 0.357 | 0.312 | 0.624 | 0.388 | 1.099 | 0.526 | 0.463 | 1.113 | 0.571 | 0.751 | 0.420 | 0.447 |
| Rolling Shutter | 0.409 | 0.318 | 0.667 | 1.030 | 0.394 | 1.000 | 0.626 | 0.509 | 1.000 | 0.853 | 0.357 | 0.319 | 0.710 | 0.373 | 1.110 | 0.263 | 0.456 | 0.673 | 0.066 | 0.571 | 0.743 | 0.423 | 0.467 |
| Resolution Degrade | 0.526 | 0.347 | 0.667 | 1.182 | 0.405 | 1.000 | 0.673 | 0.507 | 1.000 | 0.905 | 0.286 | 0.308 | 0.683 | 0.402 | 1.033 | 0.316 | 0.453 | 0.580 | 0.085 | 0.571 | 0.634 | 0.415 | 0.396 |
| Stretch Squish | 0.482 | 0.345 | 0.667 | 1.258 | 0.405 | 1.000 | 0.639 | 0.498 | 1.000 | 0.621 | 0.357 | 0.394 | 0.699 | 0.358 | 0.989 | 0.415 | 0.535 | 0.630 | 0.160 | 0.571 | 0.787 | 0.425 | 0.467 |
| Edge Sawtooth | 0.489 | 0.332 | 0.667 | 1.136 | 0.415 | 1.000 | 0.687 | 0.505 | 1.000 | 0.705 | 0.429 | 0.398 | 0.570 | 0.393 | 1.242 | 0.211 | 0.513 | 0.593 | 0.028 | 0.571 | 0.746 | 0.422 | 0.474 |
| Elastic | 0.562 | 0.343 | 0.667 | 1.152 | 0.409 | 1.000 | 0.653 | 0.510 | 1.000 | 0.895 | 0.357 | 0.320 | 0.694 | 0.407 | 1.099 | 0.211 | 0.465 | 0.589 | 0.264 | 0.740 | 0.786 | 0.426 | 0.474 |
| Color Quantized | 0.474 | 0.343 | 0.667 | 1.273 | 0.420 | 1.000 | 0.741 | 0.509 | 1.000 | 0.811 | 0.357 | 0.318 | 0.608 | 0.395 | 1.099 | 0.211 | 0.464 | 0.673 | 0.264 | 0.740 | 0.789 | 0.425 | 0.441 |
| Bright Transform | 0.555 | 0.341 | 0.667 | 1.182 | 0.428 | 1.000 | 0.673 | 0.513 | 1.000 | 0.853 | 0.429 | 0.325 | 0.634 | 0.391 | 1.088 | 0.263 | 0.457 | 0.648 | 0.179 | 0.740 | 0.772 | 0.427 | 0.454 |
| Contrast | 0.482 | 0.350 | 0.667 | 1.167 | 0.425 | 1.000 | 0.626 | 0.503 | 1.000 | 0.874 | 0.429 | 0.319 | 0.683 | 0.404 | 1.253 | 0.316 | 0.464 | 0.630 | 0.179 | 0.740 | 0.782 | 0.426 | 0.428 |
| Color Shift | 0.401 | 0.347 | 0.667 | 1.227 | 0.421 | 1.000 | 0.735 | 0.502 | 1.000 | 0.842 | 0.357 | 0.309 | 0.720 | 0.398 | 1.066 | 0.316 | 0.531 | 0.580 | 0.283 | 0.571 | 0.770 | 0.425 | 0.474 |
| Flicker | 0.467 | 0.334 | 0.667 | 1.167 | 0.397 | 1.000 | 0.626 | 0.508 | 1.000 | 0.705 | 0.357 | 0.307 | 0.634 | 0.386 | 1.044 | 0.211 | 0.447 | 0.531 | 0.113 | 0.571 | 0.764 | 0.422 | 0.447 |
| Overexposure | 0.518 | 0.340 | 0.667 | 1.000 | 0.391 | 1.000 | 0.578 | 0.510 | 1.000 | 0.789 | 0.429 | 0.313 | 0.769 | 0.386 | 1.187 | 0.211 | 0.452 | 0.623 | 0.160 | 0.571 | 0.788 | 0.423 | 0.414 |
| Underexposure | 0.496 | 0.346 | 0.667 | 1.015 | 0.406 | 1.000 | 0.626 | 0.510 | 1.000 | 0.758 | 0.357 | 0.399 | 0.624 | 0.402 | 1.011 | 0.211 | 0.450 | 0.593 | 0.113 | 0.571 | 0.772 | 0.422 | 0.414 |
| Rainy | 0.445 | 0.333 | 0.667 | 1.000 | 0.402 | 1.000 | 0.619 | 0.506 | 1.000 | 0.768 | 0.429 | 0.314 | 0.846 | 0.393 | 1.022 | 0.211 | 0.462 | 0.543 | 0.236 | 0.571 | 0.752 | 0.423 | 0.428 |
| Foggy | 0.496 | 0.334 | 0.667 | 1.061 | 0.402 | 1.000 | 0.578 | 0.510 | 1.000 | 0.726 | 0.357 | 0.311 | 0.923 | 0.379 | 1.187 | 0.211 | 0.451 | 0.605 | 0.160 | 0.571 | 0.752 | 0.422 | 0.414 |
| Snow | 0.518 | 0.337 | 0.667 | 1.015 | 0.406 | 1.000 | 0.701 | 0.509 | 1.000 | 0.705 | 0.429 | 0.313 | 0.624 | 0.399 | 1.055 | 0.211 | 0.462 | 0.586 | 0.057 | 0.571 | 0.794 | 0.425 | 0.461 |
| Frost | 0.453 | 0.337 | 0.667 | 1.136 | 0.400 | 1.000 | 0.694 | 0.507 | 1.000 | 0.674 | 0.286 | 0.311 | 0.731 | 0.450 | 1.044 | 0.211 | 0.450 | 0.524 | 0.123 | 0.571 | 0.772 | 0.419 | 0.434 |
| Reflect | 0.438 | 0.333 | 0.667 | 1.136 | 0.395 | 1.000 | 0.626 | 0.508 | 1.000 | 0.758 | 0.357 | 0.307 | 0.731 | 0.372 | 1.066 | 0.211 | 0.464 | 0.529 | 0.170 | 0.571 | 0.762 | 0.423 | 0.474 |
| Shadow | 0.496 | 0.341 | 0.667 | 1.136 | 0.400 | 1.000 | 0.863 | 0.508 | 1.000 | 0.863 | 0.429 | 0.312 | 0.780 | 0.388 | 1.176 | 0.316 | 0.456 | 1.226 | 0.571 | 0.802 | 0.493 |

Note: A: Advertisements, B: Algorithm & Models, C: Autos & Vehicles, D: Business & Industrial, E: Computers & Electronics, F: Fairy Tale, G: Films & TV Shows, H: Finance, I: Food & Drink, J: Games, K: Humor, L: Instruction Video (how to), M: Knowledge, N: News, O: Others, P: People, Q: Pets & Animals, R: Science, S: Sports, T: Overall

Table 10: Video-LLaVA performance across different video types

| Noise Type | A GPT | A SBERT | A Acc | B GPT | B SBERT | B Acc | C GPT | C SBERT | C Acc | D GPT | D SBERT | D Acc | E GPT | E SBERT | E Acc | F GPT | F SBERT | F Acc | G GPT | G SBERT | G Acc | H GPT | H SBERT | H Acc | I GPT | I SBERT | I Acc | J GPT | J SBERT | J Acc |
|---|---|---|---|---|---|---|---|---|---|---|---|---|---|---|---|---|---|---|---|---|---|---|---|---|---|---|---|---|---|---|
| Clean | .846 | .416 | — | .250 | .316 | — | 1.045 | .464 | — | 1.152 | .351 | — | .809 | .350 | — | 1.000 | .380 | — | .894 | .495 | .375 | 1.048 | .534 | .498 | 1.078 | .498 | .812 | .522 | .309 | 1.000 |
| Gaussian | .780 | .397 | .333 | .500 | .420 | — | .938 | .447 | — | .990 | .350 | — | .818 | .355 | — | .750 | .429 | — | .766 | .483 | .281 | .990 | .530 | .750 | 1.031 | .530 | .750 | .433 | .293 | 1.000 |
| Impulse | .732 | .385 | .167 | 1.167 | .375 | — | .893 | .439 | — | .855 | .354 | — | .737 | .369 | — | .250 | .421 | — | .685 | .468 | .312 | 1.050 | .476 | .812 | 1.050 | .516 | .290 | .489 | .301 | 1.000 |
| Speckle | .797 | .395 | .333 | .250 | .403 | — | .938 | .439 | — | .837 | .357 | — | .779 | .369 | — | .500 | .378 | — | .779 | .481 | .312 | 1.062 | .491 | .812 | 1.047 | .529 | .300 | .500 | .301 | 1.000 |
| Poisson | .789 | .399 | .333 | .375 | .402 | — | .964 | .442 | — | .882 | .352 | — | .791 | .364 | — | .500 | .359 | — | .766 | .476 | .344 | 1.025 | .522 | .750 | .938 | .521 | .299 | .422 | .297 | 1.000 |
| Gaussian Blur | .618 | .378 | .333 | .273 | .396 | — | .991 | .440 | — | .791 | .352 | — | .791 | .356 | — | .608 | .423 | — | .630 | .476 | .312 | .969 | .522 | .812 | .907 | .522 | .300 | .367 | .299 | 1.000 |
| Motion Blur | .813 | .394 | .167 | .500 | .406 | — | 1.134 | .456 | — | .791 | .355 | — | .547 | .364 | — | .306 | .436 | — | .889 | .493 | .375 | 1.025 | .543 | .750 | .930 | .524 | .300 | .422 | .300 | 1.000 |
| Defocus Blur | .715 | .404 | .333 | .273 | .411 | — | .920 | .443 | — | .718 | .357 | — | .737 | .359 | — | .484 | .362 | — | .323 | .465 | .312 | 1.100 | .532 | .750 | .837 | .543 | .285 | .478 | .300 | 1.000 |
| Glass Blur | .724 | .392 | .333 | 1.273 | .409 | — | .911 | .440 | — | .818 | .354 | — | .737 | .363 | — | .625 | .362 | — | .889 | .492 | .375 | .975 | .532 | .875 | 1.163 | .523 | .302 | .478 | .302 | 1.000 |
| Zoom Blur | .724 | .402 | .167 | .394 | .411 | — | .866 | .431 | — | .727 | .346 | — | .695 | .354 | — | .000 | .362 | — | .736 | .474 | .312 | 1.100 | .525 | .750 | 1.039 | .523 | .300 | .422 | .290 | 1.000 |
| Frame Drop | .626 | .390 | .333 | .394 | .412 | — | 1.214 | .461 | — | 1.106 | .349 | — | .758 | .380 | — | .750 | .419 | — | .745 | .477 | .281 | .975 | .484 | .875 | 1.186 | .525 | .296 | .489 | .290 | 1.000 |
| Frame Replace | .780 | .402 | .333 | 1.515 | .418 | — | .938 | .452 | — | .855 | .352 | — | .832 | .365 | — | .000 | .462 | — | .757 | .489 | .375 | 1.100 | .499 | .750 | 1.163 | .542 | .291 | .522 | .316 | 1.000 |
| Frame Repeat | .528 | .372 | .167 | .394 | .428 | — | 1.054 | .452 | — | .736 | .352 | — | .611 | .365 | — | .750 | .450 | — | .643 | .484 | .312 | 1.008 | .494 | .750 | 1.008 | .531 | .291 | .344 | .309 | 1.000 |
| Temporal Jitter | .732 | .395 | .167 | 1.515 | .415 | — | 1.036 | .458 | — | .845 | .355 | — | .768 | .377 | — | .250 | .388 | — | .728 | .482 | .312 | 1.150 | .491 | .812 | .977 | .534 | .294 | .500 | .309 | 1.000 |
| Jpeg Artifact | .732 | .404 | .167 | .273 | .427 | .667 | 1.107 | .458 | — | .845 | .355 | — | .811 | .371 | — | 1.000 | .383 | — | .791 | .477 | .344 | 1.100 | .500 | .812 | 1.000 | .539 | .294 | .556 | .307 | 1.000 |
| Bit Error | .772 | .397 | 1.000 | .288 | .410 | — | .955 | .450 | — | .736 | .357 | — | .768 | .364 | — | .774 | .397 | — | .664 | .483 | .344 | 1.062 | .469 | .812 | .953 | .523 | .304 | .411 | .300 | 1.000 |
| H265 Artifacts | .650 | .379 | .333 | 1.318 | .394 | — | 1.062 | .449 | — | .791 | .340 | — | .695 | .364 | — | .613 | .399 | — | .800 | .482 | .344 | 1.012 | .469 | .812 | 1.140 | .537 | .306 | .500 | .308 | 1.000 |
| Random Block | .748 | .399 | .167 | .394 | .432 | — | 1.080 | .450 | — | .909 | .349 | — | .737 | .357 | — | .484 | .362 | — | .800 | .489 | .312 | 1.038 | .494 | .812 | 1.085 | .546 | .306 | .444 | .304 | 1.000 |
| Target Block | .675 | .391 | .167 | .439 | .410 | — | 1.054 | .449 | — | .791 | .346 | — | .695 | .369 | — | .720 | .362 | — | .813 | .488 | .344 | 1.175 | .494 | .812 | 1.054 | .542 | .304 | .478 | .302 | 1.000 |
| Rolling Shutter | .496 | .382 | .333 | 1.318 | .427 | — | .107 | .447 | — | .736 | .358 | — | .874 | .373 | — | .737 | .385 | — | .634 | .472 | .312 | .863 | .480 | .750 | 1.023 | .520 | .812 | .478 | .304 | 1.000 |
| Resolution Degrade | .805 | .401 | .167 | .530 | .414 | — | 1.054 | .439 | — | .955 | .349 | — | .695 | .366 | — | .737 | .398 | — | .962 | .490 | .312 | 1.150 | .495 | .750 | 1.186 | .536 | .304 | .578 | .304 | 1.000 |
| Stretch Squish | .667 | .383 | .333 | 1.318 | .411 | — | 1.116 | .452 | — | .836 | .352 | — | .821 | .381 | — | .790 | .390 | — | .766 | .497 | .344 | 1.150 | .494 | .812 | 1.116 | .535 | .305 | .633 | .306 | 1.000 |
| Edge Sawtooth | .886 | .403 | .167 | .500 | .412 | — | .161 | .447 | — | .855 | .354 | — | .853 | .367 | — | .710 | .400 | — | .813 | .481 | .344 | 1.062 | .480 | .812 | 1.062 | .531 | .306 | .511 | .306 | 1.000 |
| Elastic | .732 | .411 | .333 | .606 | .414 | — | 1.018 | .437 | — | .718 | .344 | — | .821 | .379 | — | .817 | .400 | — | .855 | .497 | .344 | 1.087 | .494 | .750 | 1.147 | .538 | .305 | .467 | .306 | 1.000 |
| Color Quantized | .789 | .411 | .333 | .500 | .415 | — | 1.125 | .460 | — | .286 | .346 | — | .853 | .377 | — | .710 | .403 | — | .843 | .493 | .344 | 1.150 | .480 | .750 | 1.062 | .536 | .306 | .633 | .306 | 1.000 |
| Bright Transform | .829 | .413 | .333 | .364 | .406 | — | 1.080 | .455 | — | .782 | .353 | — | .905 | .374 | — | .720 | .406 | — | .851 | .492 | .344 | 1.100 | .495 | .812 | 1.054 | .531 | .307 | .467 | .305 | 1.000 |
| Contrast | .886 | .414 | .167 | .576 | .414 | — | 1.080 | .443 | — | .891 | .344 | — | .853 | .367 | — | .790 | .401 | — | .843 | .492 | .344 | 1.062 | .497 | .875 | 1.093 | .538 | .313 | .467 | .303 | 1.000 |
| Color Shift | .854 | .404 | .167 | .561 | .428 | — | .179 | .455 | — | .918 | .356 | — | .915 | .337 | — | .753 | .413 | — | .830 | .493 | .344 | 1.100 | .502 | .688 | 1.063 | .531 | .311 | .533 | .309 | 1.000 |
| Flicker | .837 | .407 | .333 | .485 | .426 | — | .179 | .443 | — | .782 | .353 | — | .905 | .362 | — | .699 | .409 | — | .843 | .493 | .375 | 1.087 | .497 | .812 | 1.093 | .536 | .307 | .511 | .303 | 1.000 |
| Overexposure | .748 | .402 | .333 | .470 | .426 | — | 1.045 | .452 | — | .762 | .353 | — | .853 | .360 | — | .715 | .409 | — | .830 | .489 | .312 | 1.050 | .496 | .812 | 1.124 | .539 | .312 | .478 | .301 | 1.000 |
| Underexposure | .797 | .413 | .333 | .606 | .424 | — | 1.062 | .452 | — | .864 | .359 | — | .842 | .371 | — | .763 | .403 | — | .851 | .489 | .375 | 1.200 | .489 | .750 | 1.093 | .536 | .295 | .511 | .307 | 1.000 |
| Rainy | .772 | .407 | .333 | .439 | .433 | — | 1.045 | .446 | — | .891 | .353 | — | .926 | .375 | — | .677 | .411 | — | .834 | .488 | .344 | 1.050 | .485 | .750 | 1.023 | .533 | .293 | .500 | .295 | 1.000 |
| Foggy | .780 | .405 | .333 | .470 | .425 | — | 1.080 | .449 | — | .823 | .353 | — | .863 | .370 | — | .704 | .400 | — | .809 | .485 | .312 | 1.075 | .504 | .688 | 1.047 | .534 | .301 | .456 | .311 | 1.000 |
| Snow | .789 | — | — | — | — | — | — | — | — | .809 | .353 | — | .853 | .388 | — | .758 | .373 | — | — | — | — | 1.087 | .495 | .750 | 1.062 | .539 | — | .467 | .301 | 1.000 |

Note: A: Advertisements, B: Algorithm & Models, C: Autos & Vehicles, D: Business & Industrial, E: Computers & Electronics, F: Fairy Tale, G: Films & TV Shows, H: Finance, I: Food & Drink, J: Games, K: Humor, L: Instruction Video (how to), M: Knowledge, N: News, O: Others, P: People, Q: Pets & Animals, R: Science, S: Sports, T: Overall

| Noise Type | K GPT | K SBERT | K Acc | L GPT | L SBERT | L Acc | M GPT | M SBERT | M Acc | N GPT | N SBERT | N Acc | O GPT | O SBERT | O Acc | Q GPT | Q SBERT | Q Acc | R GPT | R SBERT | R Acc | S GPT | S SBERT | S Acc | T GPT | T SBERT | T Acc |
|---|---|---|---|---|---|---|---|---|---|---|---|---|---|---|---|---|---|---|---|---|---|---|---|---|---|---|---|
| Clean | .620 | .354 | — | .606 | .442 | — | 1.000 | .510 | — | .916 | .341 | — | .600 | .380 | — | 1.110 | .404 | — | .685 | .553 | — | .208 | .454 | — | .772 | .454 | .467 |
| Gaussian | .445 | .338 | — | .515 | .420 | — | 1.000 | .512 | — | .811 | .315 | — | .600 | .390 | — | 1.121 | .408 | — | .685 | .540 | — | .217 | .458 | — | .845 | .430 | .454 |
| Impulse | .387 | .325 | — | 1.167 | .375 | — | 1.000 | .503 | — | .737 | .309 | — | .600 | .355 | — | 1.088 | .386 | — | .568 | .526 | — | .170 | .460 | — | .769 | .418 | .467 |
| Speckle | .474 | .336 | — | .379 | .403 | — | 1.000 | .504 | — | .779 | .314 | — | .600 | .385 | — | 1.077 | .407 | — | .691 | .541 | — | .094 | .460 | — | .827 | .429 | .447 |
| Poisson | .387 | .322 | — | .273 | .402 | — | 1.000 | .508 | — | .735 | .318 | — | .600 | .367 | — | 1.055 | .406 | — | .642 | .542 | — | .132 | .457 | — | .800 | .426 | .467 |
| Gaussian Blur | .431 | .334 | — | .273 | .394 | — | 1.000 | .505 | — | .705 | .304 | — | .600 | .380 | — | 1.088 | .389 | — | .623 | .533 | — | .189 | .458 | — | .774 | .423 | .454 |
| Motion Blur | .124 | .339 | — | .394 | .406 | — | 1.000 | .509 | — | .811 | .301 | — | .600 | .401 | — | .989 | .416 | — | .728 | .547 | — | .236 | .469 | — | .874 | .435 | .474 |
| Defocus Blur | .518 | .334 | — | .500 | .411 | — | 1.000 | .512 | — | .737 | .318 | — | .600 | .357 | — | 1.055 | .413 | — | .617 | .540 | — | .151 | .458 | — | .612 | .425 | .454 |
| Glass Blur | .467 | .341 | — | .273 | .394 | — | 1.000 | .511 | — | .758 | .316 | — | .600 | .364 | — | .967 | .390 | — | .741 | .537 | — | .198 | .457 | — | .786 | .425 | .441 |
| Zoom Blur | .467 | .337 | — | .394 | .412 | — | 1.000 | .507 | — | .695 | .307 | — | .600 | .381 | — | 1.088 | .416 | — | .586 | .550 | — | .264 | .467 | — | .906 | .441 | .414 |
| Frame Drop | .350 | .337 | — | .515 | .418 | — | 1.000 | .514 | — | .832 | .336 | — | .600 | .343 | — | 1.099 | .420 | — | .667 | .530 | — | .972 | .445 | — | .758 | .418 | .421 |
| Frame Replace | .613 | .357 | — | .394 | .415 | — | .667 | .514 | — | .768 | .314 | — | .600 | .372 | — | 1.253 | .412 | — | .710 | .551 | — | .104 | .451 | — | .832 | .432 | .428 |
| Frame Repeat | .350 | .329 | — | .515 | .415 | — | 1.000 | .509 | — | .811 | .311 | — | .600 | .360 | — | 1.165 | .410 | — | .753 | .555 | — | .142 | .448 | — | .866 | .430 | .454 |
| Temporal Jitter | .467 | .339 | — | .273 | .410 | — | 1.000 | .505 | — | .695 | .323 | — | .600 | .390 | — | 1.198 | .413 | — | .667 | .551 | — | .198 | .456 | — | .759 | .434 | .447 |
| Jpeg Artifact | .533 | .344 | — | .561 | .427 | — | 1.000 | .506 | — | .821 | .322 | — | .600 | .373 | — | 1.210 | .409 | — | .735 | .551 | — | .349 | .440 | — | .862 | .434 | .461 |
| Bit Error | .518 | .322 | — | .318 | .406 | — | 1.000 | .516 | — | .863 | .316 | — | .600 | .369 | — | .912 | .382 | — | .685 | .544 | — | .179 | .468 | — | .843 | .435 | .461 |
| H265 Artifacts | .453 | .343 | — | .530 | .414 | — | 1.000 | .511 | — | .821 | .326 | — | .600 | .362 | — | 1.231 | .409 | — | .648 | .553 | — | .255 | .458 | — | .877 | .432 | .454 |
| Random Block | .504 | .331 | — | .500 | .420 | — | 1.000 | .502 | — | .905 | .322 | — | .600 | .364 | — | 1.088 | .412 | — | .716 | .549 | — | .123 | .451 | — | .915 | .434 | .467 |
| Target Block | .467 | .337 | — | .273 | .398 | — | 1.000 | .511 | — | 1.011 | .337 | — | .600 | .381 | — | .967 | .417 | — | .735 | .547 | — | .160 | .453 | — | .771 | .436 | .480 |
| Rolling Shutter | .664 | .349 | — | .530 | .414 | — | 1.000 | .514 | — | .853 | .314 | — | .600 | .372 | — | 1.044 | .390 | — | .636 | .539 | — | .274 | .456 | — | .886 | .435 | .480 |
| Resolution Degrade | .599 | .353 | — | .606 | .417 | — | 1.000 | .508 | — | .853 | .316 | — | .600 | .360 | — | 1.165 | .419 | — | .648 | .551 | — | .160 | .452 | — | .784 | .436 | .474 |
| Stretch Squish | .569 | .350 | — | .576 | .446 | — | 1.000 | .513 | — | .905 | .335 | — | .600 | .385 | — | 1.209 | .405 | — | .809 | .551 | — | .292 | .469 | — | .875 | .437 | .480 |
| Edge Sawtooth | .569 | .357 | — | .561 | .428 | — | 1.000 | .511 | — | .842 | .335 | — | .600 | .377 | — | 1.253 | .417 | — | .648 | .546 | — | .198 | .453 | — | .898 | .438 | .467 |
| Elastic | .460 | .340 | — | .439 | .428 | — | 1.000 | .515 | — | .821 | .314 | — | .600 | .360 | — | 1.132 | .414 | — | .725 | .549 | — | .170 | .465 | — | .889 | .433 | .461 |
| Color Quantized | .504 | .343 | — | .485 | .426 | — | 1.000 | .508 | — | .863 | .324 | — | .600 | .371 | — | 1.143 | .419 | — | .543 | .543 | — | .104 | .452 | — | .857 | .435 | .467 |
| Shadow | .650 | .355 | — | .470 | .438 | — | 1.000 | .506 | — | .853 | .328 | — | .600 | .388 | — | 1.077 | .402 | — | .660 | .549 | — | .217 | .455 | — | .789 | .436 | .461 |

Table 11: VideoChat2-HD performance across different video types

| Noise Type | A GPT | A SBERT | A Acc | B GPT | B SBERT | C GPT | C SBERT | C Acc | D GPT | D SBERT | D Acc | E GPT | E SBERT | E Acc | F GPT | F SBERT | F Acc | G GPT | G SBERT | G Acc | H GPT | H SBERT | H Acc | J GPT | J SBERT | J Acc |
|---|---|---|---|---|---|---|---|---|---|---|---|---|---|---|---|---|---|---|---|---|---|---|---|---|---|---|
| Clean | 1.195 | 0.423 | 0.167 | 1.125 | 0.459 | 1.214 | 0.472 | 0.846 | 1.321 | 0.351 | 0.389 | — | 0.400 | 0.846 | 1.210 | 0.438 | — | 1.051 | 0.517 | 0.438 | 1.312 | 0.512 | 0.500 | 1.350 | 0.336 | 1.000 |
| Gaussian | 0.919 | 0.406 | 0.667 | 1.125 | 0.423 | 1.241 | 0.456 | 0.692 | 0.773 | 0.361 | 0.389 | 0.904 | 0.384 | 0.904 | 0.930 | 0.435 | — | 0.877 | 0.491 | 0.469 | 1.217 | 0.507 | 0.667 | 1.217 | 0.323 | 1.000 |
| Impulse | 0.732 | 0.382 | 0.667 | 1.250 | 0.394 | 1.241 | 0.456 | 0.923 | 0.809 | 0.362 | 0.389 | 0.779 | 0.367 | 0.773 | 0.935 | 0.443 | — | 1.004 | 0.503 | 0.531 | 1.238 | 0.493 | 0.667 | 1.175 | 0.329 | 1.000 |
| Speckle | 0.894 | 0.403 | 0.333 | 0.500 | 0.418 | 1.045 | 0.456 | 0.923 | 0.809 | 0.365 | 0.444 | 0.837 | 0.374 | 0.891 | 1.016 | 0.443 | — | 0.902 | 0.494 | 0.531 | 1.163 | 0.487 | 0.500 | 1.163 | 0.317 | 1.000 |
| Poisson | 0.829 | 0.401 | 0.167 | 1.125 | 0.432 | 1.250 | 0.452 | 0.846 | 0.891 | 0.371 | 0.500 | 0.856 | 0.374 | 0.891 | 0.701 | 0.425 | — | 0.945 | 0.488 | 0.438 | 1.113 | 0.494 | 0.500 | 1.194 | 0.327 | 1.000 |
| Gaussian Blur | 0.724 | 0.390 | 0.167 | 1.125 | 0.424 | 1.179 | 0.449 | 0.846 | 0.809 | 0.358 | 0.500 | 0.904 | 0.367 | 0.782 | 0.919 | 0.435 | — | 0.902 | 0.493 | 0.438 | 1.264 | 0.504 | 0.500 | 1.264 | 0.325 | 1.000 |
| Motion Blur | 0.715 | 0.385 | 0.167 | 1.250 | 0.445 | 1.170 | 0.461 | 0.846 | 0.818 | 0.370 | 0.333 | 0.913 | 0.374 | 0.952 | 0.984 | 0.446 | — | 0.974 | 0.493 | 0.469 | 1.113 | 0.498 | 0.500 | 1.202 | 0.325 | 1.000 |
| Defocus Blur | 0.959 | 0.413 | 0.167 | 1.125 | 0.445 | 1.223 | 0.464 | 0.769 | 0.891 | 0.382 | 0.500 | 0.981 | 0.378 | 0.913 | 1.134 | 0.435 | — | 0.974 | 0.508 | 0.469 | 1.200 | 0.493 | 0.500 | 1.300 | 0.344 | 1.000 |
| Glass Blur | 1.098 | 0.419 | 0.333 | 1.125 | 0.424 | 1.304 | 0.468 | 0.846 | 0.782 | 0.355 | 0.556 | 0.962 | 0.378 | 0.962 | 1.231 | 0.435 | — | 1.047 | 0.517 | 0.531 | 1.295 | 0.504 | 0.500 | 1.295 | 0.340 | 1.000 |
| Zoom Blur | 0.911 | 0.398 | 0.333 | 0.500 | 0.333 | 1.188 | 0.466 | 0.769 | 0.782 | 0.360 | 0.556 | 0.962 | 0.376 | 0.962 | 0.923 | 0.421 | — | 1.081 | 0.502 | 0.531 | 1.150 | 0.505 | 0.500 | 1.300 | 0.337 | 1.000 |
| Frame Drop | 0.789 | 0.388 | 0.333 | 1.250 | 0.361 | 1.152 | 0.442 | 0.769 | 0.791 | 0.355 | 0.500 | 0.885 | 0.373 | 0.885 | 1.059 | 0.404 | — | 1.000 | 0.504 | 0.500 | 1.217 | 0.480 | 0.500 | 1.217 | 0.321 | 1.000 |
| Frame Replace | 1.065 | 0.428 | 0.167 | 1.000 | 0.452 | 1.259 | 0.473 | 0.846 | 1.009 | 0.383 | 0.444 | 0.962 | 0.391 | 0.962 | 1.077 | 0.438 | — | 0.923 | 0.485 | 0.531 | 1.333 | 0.514 | 0.667 | 1.333 | 0.321 | 1.000 |
| Frame Repeat | 0.618 | 0.365 | 0.000 | 0.750 | 0.439 | 1.152 | 0.442 | 0.385 | 0.764 | 0.355 | 0.740 | 1.048 | 0.373 | 1.048 | 0.855 | 0.397 | — | 0.949 | 0.489 | 0.531 | 1.109 | 0.468 | 0.333 | 1.109 | 0.314 | 1.000 |
| Temporal Jitter | 1.106 | 0.417 | 0.333 | 1.000 | 0.430 | 1.286 | 0.468 | 0.769 | 0.936 | 0.376 | 0.500 | 0.885 | 0.377 | 0.885 | 1.113 | 0.418 | — | 1.026 | 0.497 | 0.500 | 1.250 | 0.506 | 0.500 | 1.357 | 0.317 | 1.000 |
| Jpeg Artifact | 0.894 | 0.409 | 0.333 | 0.333 | 0.439 | 1.214 | 0.460 | 0.769 | 0.991 | 0.368 | 0.827 | 1.019 | 0.378 | 1.019 | 1.154 | 0.422 | — | 0.911 | 0.493 | 0.406 | 1.300 | 0.513 | 0.500 | 1.300 | 0.333 | 1.000 |
| Bit Error | 0.976 | 0.392 | 0.167 | 0.500 | 0.371 | 1.214 | 0.472 | 0.846 | 0.991 | 0.382 | 0.846 | 0.827 | 0.385 | 0.827 | 1.091 | 0.425 | — | 0.945 | 0.512 | 0.531 | 1.312 | 0.513 | 0.500 | 1.341 | 0.315 | 1.000 |
| H265 Artifacts | 0.846 | 0.414 | 0.167 | 0.875 | 0.347 | 1.170 | 0.467 | 0.692 | 0.909 | 0.369 | 0.952 | 1.087 | 0.379 | 1.019 | 1.161 | 0.433 | — | 0.945 | 0.511 | 0.531 | 1.062 | 0.511 | 0.500 | 1.264 | 0.341 | 1.000 |
| Random Block | 0.927 | 0.408 | 0.167 | 1.000 | 0.342 | 1.250 | 0.453 | 0.846 | 0.864 | 0.381 | 0.945 | 1.000 | 0.381 | 0.945 | 0.930 | 0.438 | — | 1.072 | 0.511 | 0.531 | 1.375 | 0.515 | 0.500 | 1.264 | 0.341 | 1.000 |
| Target Block | 0.967 | 0.400 | 0.333 | 0.875 | 0.332 | 1.152 | 0.453 | 0.846 | 0.945 | 0.374 | 0.865 | 1.087 | 0.374 | 1.087 | 1.231 | 0.438 | — | 1.089 | 0.506 | 0.438 | 1.375 | 0.511 | 0.500 | 1.326 | 0.332 | 1.000 |
| Rolling Shutter | 0.886 | 0.409 | 0.167 | 0.750 | 0.332 | 1.054 | 0.456 | 0.692 | 0.864 | 0.365 | 0.846 | 0.952 | 0.381 | 0.952 | 0.984 | 0.438 | — | 1.055 | 0.503 | 0.438 | 1.087 | 0.515 | 0.875 | 1.194 | 0.332 | 1.000 |
| Resolution Degrade | 0.976 | 0.399 | 0.333 | 1.000 | 0.344 | 1.098 | 0.443 | 0.692 | 0.800 | 0.362 | 0.278 | 0.846 | 0.374 | 0.846 | 1.059 | 0.415 | — | 0.996 | 0.495 | 0.438 | 1.238 | 0.493 | 0.500 | 1.256 | 0.322 | 1.000 |
| Stretch Squish | 0.992 | 0.421 | 0.333 | 0.500 | 0.359 | 1.232 | 0.472 | 0.846 | 0.873 | 0.373 | 0.333 | 0.923 | 0.381 | 0.923 | 0.923 | 0.441 | — | 1.051 | 0.516 | 0.469 | 1.375 | 0.515 | 0.500 | 1.567 | 0.340 | 1.000 |
| Edge Sawtooth | 1.057 | 0.424 | 0.167 | 1.000 | 0.381 | 1.205 | 0.466 | 0.769 | 0.845 | 0.375 | 0.333 | 0.923 | 0.380 | 0.923 | 0.923 | 0.438 | — | 1.064 | 0.504 | 0.500 | 1.363 | 0.511 | 0.875 | 1.256 | 0.336 | 1.000 |
| Elastic | 1.138 | 0.421 | 0.333 | 1.000 | 0.340 | 1.241 | 0.465 | 0.846 | 0.873 | 0.375 | 0.444 | 0.389 | 0.396 | 0.389 | 1.154 | 0.438 | — | 1.000 | 0.504 | 0.562 | 1.312 | 0.524 | 0.500 | 1.256 | 0.334 | 1.000 |
| Color Quantized | 1.146 | 0.426 | 0.167 | 0.875 | 0.369 | 1.196 | 0.468 | 0.923 | 0.955 | 0.372 | 0.500 | 1.067 | 0.383 | 1.067 | 1.059 | 0.438 | — | 1.047 | 0.509 | 0.469 | 1.312 | 0.516 | 0.500 | 1.256 | 0.337 | 1.000 |
| Bright Transform | 1.049 | 0.415 | 0.167 | 1.000 | 0.338 | 1.196 | 0.468 | 0.846 | 0.873 | 0.375 | 0.441 | 1.067 | 0.385 | 1.154 | 1.059 | 0.438 | — | 1.072 | 0.511 | 0.469 | 1.350 | 0.509 | 0.500 | 1.800 | 0.338 | 1.000 |
| Contrast | 1.081 | 0.420 | 0.167 | 0.750 | 0.344 | 1.214 | 0.468 | 0.769 | 0.955 | 0.375 | 0.500 | 0.990 | 0.380 | 0.990 | 1.113 | 0.438 | — | 1.000 | 0.504 | 0.438 | 1.288 | 0.510 | 0.500 | 1.264 | 0.337 | 1.000 |
| Color Shift | 1.041 | 0.407 | 0.167 | 1.000 | 0.369 | 1.179 | 0.465 | 0.846 | 0.973 | 0.375 | 0.333 | 1.010 | 0.387 | 1.010 | 1.091 | 0.438 | — | 1.013 | 0.509 | 0.469 | 1.275 | 0.513 | 0.500 | 1.275 | 0.346 | 1.000 |
| Flicker | 0.919 | 0.416 | 0.167 | 0.375 | 0.387 | 1.411 | 0.468 | 0.769 | 0.927 | 0.375 | 1.110 | 1.038 | 0.384 | 1.038 | 1.043 | 0.438 | — | 1.085 | 0.498 | 0.438 | 1.337 | 0.513 | 0.500 | 1.256 | 0.336 | 1.000 |
| Overexposure | 1.073 | 0.424 | 0.167 | 1.000 | 0.366 | 1.241 | 0.465 | 0.923 | 0.927 | 0.376 | 0.333 | 1.010 | 0.383 | 1.010 | 0.769 | 0.438 | — | 1.030 | 0.506 | 0.469 | 1.200 | 0.508 | 0.500 | 1.256 | 0.340 | 1.000 |
| Underexposure | 1.098 | 0.417 | 0.167 | 0.625 | 0.367 | 1.312 | 0.469 | 0.769 | 0.936 | 0.383 | 0.955 | 1.010 | 0.383 | 1.010 | 1.000 | 0.439 | — | 1.000 | 0.509 | 0.438 | 1.275 | 0.508 | 0.875 | 1.326 | 0.343 | 1.000 |
| Rainy | 1.057 | 0.410 | 0.167 | 0.625 | 0.396 | 1.232 | 0.467 | 0.923 | 0.936 | 0.378 | 0.964 | 1.048 | 0.382 | 1.010 | 1.091 | 0.438 | — | 1.000 | 0.496 | 0.500 | 1.212 | 0.508 | 0.875 | 1.178 | 0.335 | 1.000 |
| Foggy | 0.943 | 0.413 | 0.167 | 1.125 | 0.376 | 1.214 | 0.467 | 0.846 | 0.955 | 0.380 | 0.556 | 1.048 | 0.377 | 0.913 | 1.113 | 0.420 | — | 1.013 | 0.502 | 0.469 | 1.337 | 0.502 | 0.875 | 1.271 | 0.335 | 1.000 |
| Frost | 0.992 | 0.417 | 0.167 | 1.125 | 0.359 | 1.214 | 0.467 | 0.923 | 0.964 | 0.377 | 0.444 | 0.913 | 0.378 | 0.913 | 0.253 | 0.438 | — | 1.085 | 0.515 | 0.500 | 1.250 | 0.515 | 0.875 | 1.248 | 0.338 | 1.000 |
| Shadow | 1.187 | 0.426 | — | — | — | 1.268 | 0.471 | 0.846 | 1.036 | 0.385 | 0.444 | — | — | — | — | — | — | — | — | — | — | — | — | — | — | — |

| Noise Type | K GPT | K SBERT | L GPT | L SBERT | L Acc | M GPT | M SBERT | N GPT | N SBERT | N Acc | O GPT | O SBERT | O Acc | P GPT | P SBERT | P Acc | Q GPT | Q SBERT | Q Acc | R GPT | R SBERT | R Acc | S GPT | S SBERT | S Acc | T GPT | T SBERT | T Acc |
|---|---|---|---|---|---|---|---|---|---|---|---|---|---|---|---|---|---|---|---|---|---|---|---|---|---|---|---|---|
| Clean | 0.715 | 0.369 | 0.606 | 1.000 | 0.755 | 0.532 | 1.095 | 0.571 | 0.800 | 0.846 | 0.400 | 1.210 | 0.438 | 0.421 | 1.560 | 0.438 | 1.056 | 0.470 | 1.000 | 0.577 | 0.600 | 0.330 | 0.470 | 0.553 | 1.110 | 0.458 | 0.553 |
| Gaussian | 0.606 | 0.359 | 0.303 | 0.423 | 0.741 | 0.528 | 0.821 | 0.500 | 0.821 | 0.904 | 0.428 | 0.930 | 0.414 | 0.421 | 1.110 | 0.415 | 0.840 | 0.467 | 1.000 | 0.564 | 0.400 | 0.245 | 0.467 | 0.546 | 0.944 | 0.444 | 0.546 |
| Impulse | 0.620 | 0.346 | 0.394 | 0.424 | 0.762 | 0.524 | 0.779 | 0.643 | 0.905 | 0.800 | 0.388 | 0.935 | 0.409 | 0.409 | 1.022 | 0.409 | 0.903 | 0.468 | 1.000 | 0.565 | 0.600 | 0.151 | 0.452 | 0.553 | 0.903 | 0.435 | 0.553 |
| Speckle | 0.657 | 0.356 | 0.409 | 0.418 | 0.728 | 0.533 | 0.905 | 0.643 | 0.811 | 0.800 | 0.436 | 1.016 | 0.423 | 0.474 | 1.286 | 0.418 | 0.864 | 0.468 | 1.000 | 0.565 | 0.600 | 0.311 | 0.468 | 0.618 | 0.963 | 0.447 | 0.618 |
| Poisson | 0.518 | 0.347 | 0.258 | 0.399 | 0.701 | 0.521 | 0.811 | 0.500 | 0.811 | 0.800 | 0.404 | 0.919 | 0.410 | 0.474 | 1.297 | 0.418 | 0.784 | 0.467 | 1.000 | 0.562 | 0.600 | 0.208 | 0.463 | 0.553 | 0.916 | 0.438 | 0.553 |
| Gaussian Blur | 0.628 | 0.341 | 0.470 | 0.432 | 0.741 | 0.518 | 0.916 | 0.500 | 0.853 | 0.800 | 0.386 | 0.957 | 0.412 | 0.474 | 1.275 | 0.415 | 0.901 | 0.470 | 1.000 | 0.561 | 0.600 | 0.255 | 0.463 | 0.546 | 0.958 | 0.440 | 0.546 |
| Motion Blur | 0.591 | 0.345 | 0.333 | 0.424 | 0.782 | 0.522 | 1.095 | 0.643 | 1.042 | 0.800 | 0.407 | 0.984 | 0.426 | 0.421 | 1.231 | 0.415 | 0.951 | 0.470 | 1.000 | 0.581 | 0.600 | 0.217 | 0.474 | 0.539 | 1.055 | 0.440 | 0.539 |
| Defocus Blur | 0.759 | 0.354 | 0.561 | 0.445 | 0.782 | 0.533 | 1.095 | 0.571 | 1.077 | 1.000 | 0.388 | 1.134 | 0.420 | 0.474 | 1.462 | 0.424 | 1.056 | 0.466 | 1.000 | 0.573 | 0.600 | 0.292 | 0.474 | 0.559 | 1.091 | 0.453 | 0.559 |
| Glass Blur | 0.701 | 0.361 | 0.530 | 0.424 | 0.741 | 0.526 | 1.042 | 0.571 | 0.923 | 1.000 | 0.314 | 1.077 | 0.420 | 0.368 | 1.231 | 0.419 | 0.923 | 0.468 | 1.000 | 0.565 | 0.600 | 0.217 | 0.474 | 0.553 | 1.011 | 0.454 | 0.559 |
| Zoom Blur | 0.737 | 0.358 | 0.409 | 0.424 | 0.721 | 0.519 | 0.853 | 0.571 | 1.021 | 0.800 | 0.382 | 1.077 | 0.417 | 0.368 | 1.132 | 0.420 | 0.932 | 0.468 | 1.000 | 0.565 | 0.600 | 0.292 | 0.466 | 0.553 | 0.986 | 0.433 | 0.553 |
| Frame Drop | 0.569 | 0.358 | 0.530 | 0.452 | 0.667 | 0.519 | 0.821 | 0.571 | 0.979 | 0.923 | 0.410 | 1.059 | 0.404 | 0.368 | 1.231 | 0.419 | 1.056 | 0.468 | 1.000 | 0.565 | 0.600 | 0.217 | 0.468 | 0.566 | 0.986 | 0.433 | 0.566 |
| Frame Replace | 0.686 | 0.355 | 0.530 | 0.452 | 0.844 | 0.533 | 1.137 | 0.571 | 1.000 | 0.800 | 0.417 | 1.077 | 0.420 | 0.474 | 1.549 | 0.439 | 0.951 | 0.478 | 1.000 | 0.576 | 0.600 | 0.094 | 0.454 | 0.592 | 1.109 | 0.433 | 0.592 |
| Frame Repeat | 0.526 | 0.350 | 0.197 | 0.439 | 0.333 | 0.528 | 0.937 | 0.500 | 0.821 | 0.800 | 0.405 | 1.113 | 0.397 | 0.526 | 1.132 | 0.425 | 0.802 | 0.451 | 1.000 | 0.556 | 0.800 | 0.123 | 0.430 | 0.480 | 0.859 | 0.430 | 0.480 |
| Temporal Jitter | 0.664 | 0.360 | 0.379 | 0.439 | 0.816 | 0.528 | 0.916 | 0.643 | 1.063 | 0.800 | 0.390 | 1.113 | 0.421 | 0.474 | 1.363 | 0.425 | 0.907 | 0.459 | 1.000 | 0.573 | 0.800 | 0.330 | 0.459 | 0.579 | 1.047 | 0.448 | 0.579 |
| Jpeg Artifact | 0.715 | 0.355 | 0.591 | 0.441 | 1.000 | 0.538 | 1.154 | 0.643 | 1.105 | 0.800 | 0.425 | 1.091 | 0.418 | 0.526 | 1.527 | 0.421 | 1.018 | 0.470 | 1.000 | 0.575 | 0.600 | 0.245 | 0.470 | 0.566 | 1.018 | 0.448 | 0.566 |
| Bit Error | 0.730 | 0.352 | 0.591 | 0.453 | 0.823 | 0.530 | 1.105 | 0.571 | 0.926 | 0.800 | 0.408 | 1.161 | 0.425 | 0.526 | 1.209 | 0.439 | 0.815 | 0.477 | 1.000 | 0.559 | 0.600 | 0.321 | 0.477 | 0.605 | 1.082 | 0.456 | 0.605 |
| H265 Artifacts | 0.701 | 0.357 | 0.576 | 0.453 | 0.789 | 0.529 | 0.853 | 0.329 | 1.032 | 0.800 | 0.402 | 0.930 | 0.426 | 0.368 | 1.330 | 0.415 | 0.901 | 0.470 | 1.000 | 0.576 | 0.800 | 0.245 | 0.470 | 0.612 | 0.971 | 0.438 | 0.612 |
| Random Block | 0.664 | 0.357 | 0.576 | 0.453 | 0.782 | 0.531 | 1.032 | 0.571 | 1.021 | 0.600 | 0.416 | 0.984 | 0.415 | 0.632 | 1.495 | 0.418 | 1.046 | 0.470 | 1.000 | 0.561 | 0.600 | 0.245 | 0.470 | 0.599 | 1.078 | 0.440 | 0.599 |
| Target Block | 0.759 | 0.355 | 0.530 | 0.456 | 1.000 | 0.531 | 1.021 | 0.571 | 1.105 | 0.800 | 0.411 | 1.231 | 0.419 | 0.526 | 1.407 | 0.433 | 1.011 | 0.474 | 1.000 | 0.562 | 0.800 | 0.415 | 0.474 | 0.592 | 1.046 | 0.449 | 0.592 |
| Rolling Shutter | 0.766 | 0.353 | 0.455 | 0.445 | 0.871 | 0.530 | 1.105 | 0.571 | 1.105 | 0.800 | 0.401 | 1.059 | 0.419 | 0.368 | 1.549 | 0.418 | 0.957 | 0.474 | 1.000 | 0.572 | 0.600 | 0.358 | 0.459 | 0.579 | 1.011 | 0.449 | 0.579 |
| Resolution Degrade | 0.745 | 0.365 | 0.394 | 0.451 | 0.707 | 0.524 | 1.021 | 0.429 | 0.923 | 1.000 | 0.398 | 0.923 | 0.404 | 0.474 | 1.385 | 0.435 | 1.056 | 0.466 | 1.000 | 0.561 | 0.600 | 0.292 | 0.466 | 0.546 | 0.987 | 0.433 | 0.546 |
| Stretch Squish | 0.679 | 0.352 | 0.455 | 0.451 | 0.796 | 0.531 | 1.105 | 0.571 | 1.077 | 0.800 | 0.377 | 1.016 | 0.418 | 0.526 | 1.407 | 0.433 | 0.877 | 0.459 | 1.000 | 0.562 | 0.600 | 0.264 | 0.459 | 0.592 | 0.987 | 0.442 | 0.592 |
| Edge Sawtooth | 0.774 | 0.365 | 0.561 | 0.455 | 0.721 | 0.533 | 1.105 | 0.571 | 1.032 | 0.800 | 0.414 | 1.151 | 0.431 | 0.421 | 1.527 | 0.435 | 0.944 | 0.478 | 1.000 | 0.572 | 0.600 | 0.462 | 0.485 | 0.553 | 1.079 | 0.449 | 0.553 |
| Elastic | 0.788 | 0.369 | 0.561 | 0.450 | 1.000 | 0.531 | 0.979 | 0.643 | 1.105 | 0.800 | 0.401 | 1.097 | 0.426 | 0.579 | 1.385 | 0.429 | 1.091 | 0.466 | 1.000 | 0.571 | 0.600 | 0.387 | 0.478 | 0.572 | 1.096 | 0.454 | 0.572 |
| Color Quantized | 0.708 | 0.353 | 0.545 | 0.436 | 0.694 | 0.538 | 1.000 | 0.571 | 1.032 | 0.800 | 0.376 | 1.177 | 0.423 | 0.526 | 1.429 | 0.442 | 0.889 | 0.465 | 1.000 | 0.577 | 0.600 | 0.472 | 0.465 | 0.586 | 1.069 | 0.456 | 0.586 |
| Bright Transform | 0.672 | 0.358 | 0.545 | 0.436 | 0.830 | 0.536 | 1.063 | 0.571 | 1.074 | 0.800 | 0.388 | 1.129 | 0.431 | 0.526 | 1.418 | 0.438 | 0.926 | 0.474 | 1.000 | 0.573 | 0.600 | 0.377 | 0.474 | 0.605 | 1.055 | 0.456 | 0.605 |
| Contrast | 0.693 | 0.364 | 0.409 | 0.451 | 0.844 | 0.529 | 1.074 | 0.643 | 1.000 | 0.800 | 0.396 | 1.124 | 0.421 | 0.474 | 1.440 | 0.431 | 0.864 | 0.467 | 1.000 | 0.562 | 0.600 | 0.321 | 0.467 | 0.572 | 1.039 | 0.451 | 0.572 |
| Color Shift | 0.635 | 0.358 | 0.451 | 0.434 | 0.810 | 0.531 | 0.958 | 0.329 | 0.769 | 0.800 | 0.393 | 1.065 | 0.433 | 0.474 | 1.385 | 0.427 | 0.870 | 0.468 | 1.000 | 0.580 | 0.600 | 0.321 | 0.480 | 0.592 | 1.074 | 0.490 | 0.592 |
| Flicker | 0.766 | 0.356 | 0.439 | 0.450 | 1.000 | 0.535 | 0.923 | 0.357 | 1.126 | 0.800 | 0.392 | 1.156 | 0.424 | 0.474 | 1.527 | 0.435 | 0.914 | 0.482 | 1.000 | 0.571 | 0.600 | 0.321 | 0.488 | 0.572 | 1.090 | 0.458 | 0.572 |
| Overexposure | 0.679 | 0.357 | 0.455 | 0.450 | 0.957 | 0.533 | 0.914 | 0.357 | 0.832 | 0.800 | 0.405 | 1.086 | 0.430 | 0.526 | 1.462 | 0.433 | 0.907 | 0.488 | 1.000 | 0.573 | 0.600 | 0.491 | 0.488 | 0.553 | 1.063 | 0.456 | 0.586 |
| Underexposure | 0.723 | 0.357 | 0.606 | 0.464 | 1.000 | 0.538 | 1.116 | 0.429 | 0.989 | 0.800 | 0.381 | 1.092 | 0.424 | 0.474 | 1.407 | 0.435 | 1.025 | 0.474 | 1.000 | 0.571 | 0.600 | 0.443 | 0.474 | 0.592 | 1.063 | 0.458 | 0.592 |
| Rainy | 0.723 | 0.353 | 0.712 | 0.474 | 1.000 | 0.538 | 1.063 | 0.643 | 0.989 | 0.600 | 0.386 | 1.113 | 0.426 | 0.526 | 1.440 | 0.435 | 0.901 | 0.467 | 1.000 | 0.571 | 0.600 | 0.443 | 0.485 | 0.586 | 1.069 | 0.456 | 0.586 |
| Foggy | 0.701 | 0.358 | 0.652 | 0.457 | 1.000 | 0.531 | 0.832 | 0.357 | 0.958 | 0.800 | 0.357 | 1.091 | 0.433 | 0.474 | 1.418 | 0.435 | 0.864 | 0.467 | 1.000 | 0.567 | 0.600 | 0.396 | 0.486 | 0.605 | 1.069 | 0.453 | 0.605 |
| Snow | 0.686 | 0.356 | 0.515 | 0.457 | 1.000 | 0.538 | 0.989 | 0.571 | 0.769 | 0.800 | 0.401 | 1.043 | 0.424 | 0.474 | 1.407 | 0.430 | 0.870 | 0.474 | 1.000 | 0.571 | 0.600 | 0.443 | 0.474 | 0.599 | 1.037 | 0.451 | 0.599 |
| Frost | 0.723 | 0.364 | 0.561 | 0.452 | 1.000 | 0.525 | 0.989 | 0.643 | 1.063 | 0.800 | 0.405 | 1.091 | 0.419 | 0.526 | 1.418 | 0.435 | 0.901 | 0.456 | 1.000 | 0.568 | 0.600 | 0.349 | 0.473 | 0.599 | 1.048 | 0.451 | 0.599 |
| Shadow | 0.730 | 0.362 | 0.576 | 0.456 | 1.000 | 0.529 | 0.823 | 0.643 | 0.989 | 0.800 | 0.402 | 1.253 | 0.438 | 0.474 | 1.516 | 0.436 | 1.006 | 0.475 | 1.000 | 0.579 | 0.600 | 0.368 | 0.475 | 0.592 | 1.104 | 0.453 | 0.592 |

Note: A: Advertisements, B: Algorithm & Models, C: Autos & Vehicles, D: Business & Industrial, E: Computers & Electronics, F: Fairy Tale, G: Films & TV Shows, H: Finance, I: Food & Drink, J: Games, K: Humor, L: Instruction Video (how to), M: Knowledge, N: News, O: Others, P: People, Q: Pets & Animals, R: Science, S: Sports, T: Overall

Table 12: Chat-UniVi-v1.5 performance across different question types

| Noise Type | CP GPT | CP SBERT | CP Acc | FP-S GPT | FP-S SBERT | FP-S Acc | FP-C GPT | FP-C SBERT | FP-C Acc | HL GPT | HL SBERT | HL Acc | Mean GPT | Mean SBERT | Mean Acc | LR GPT | LR SBERT | LR Acc | AR GPT | AR SBERT | AR Acc | RR GPT | RR SBERT | RR Acc | CSR GPT | CSR SBERT | CSR Acc | TR GPT | TR SBERT | TR Acc | Mean2 GPT | Mean2 SBERT | Mean2 Acc | Overall GPT | Overall SBERT | Overall Acc |
|---|---|---|---|---|---|---|---|---|---|---|---|---|---|---|---|---|---|---|---|---|---|---|---|---|---|---|---|---|---|---|---|---|---|---|---|---|
| Clean | 1.080 | 0.405 | 0.462 | 0.860 | 0.413 | 0.543 | 0.770 | 0.453 | 0.200 | 0.870 | 0.151 | 0.000 | 0.900 | 0.408 | 0.500 | 0.500 | 0.491 | 0.222 | 1.190 | 0.487 | 0.429 | 0.920 | 0.550 | 0.333 | 0.600 | 0.419 | 0.000 | 0.850 | 0.441 | 0.480 | 0.860 | 0.475 | 0.386 | 0.900 | 0.430 | 0.461 |
| Bit Error | 1.000 | 0.408 | 0.231 | 0.890 | 0.413 | 0.529 | 0.770 | 0.458 | 0.200 | 0.890 | 0.152 | 0.000 | 0.900 | 0.410 | 0.457 | 0.560 | 0.501 | 0.222 | 0.960 | 0.473 | 0.286 | 0.960 | 0.559 | 0.333 | 0.740 | 0.407 | 0.000 | 0.760 | 0.448 | 0.480 | 0.820 | 0.477 | 0.386 | 0.880 | 0.431 | 0.434 |
| Bright Transform | 0.930 | 0.397 | 0.231 | 0.840 | 0.412 | 0.529 | 0.800 | 0.460 | 0.200 | 0.870 | 0.155 | 0.000 | 0.860 | 0.407 | 0.457 | 0.520 | 0.495 | 0.222 | 1.080 | 0.481 | 0.429 | 0.930 | 0.556 | 0.333 | 0.640 | 0.417 | 0.000 | 0.780 | 0.445 | 0.520 | 0.830 | 0.477 | 0.400 | 0.870 | 0.429 | 0.441 |
| Color Quantized | 1.070 | 0.404 | 0.308 | 0.830 | 0.412 | 0.529 | 0.770 | 0.457 | 0.200 | 0.760 | 0.156 | 0.000 | 0.880 | 0.409 | 0.467 | 0.360 | 0.482 | 0.222 | 1.050 | 0.471 | 0.571 | 0.910 | 0.557 | 0.367 | 0.800 | 0.422 | 0.000 | 0.800 | 0.442 | 0.440 | 0.820 | 0.472 | 0.400 | 0.870 | 0.429 | 0.447 |
| Color Shift | 1.030 | 0.404 | 0.385 | 0.880 | 0.415 | 0.529 | 0.750 | 0.460 | 0.133 | 0.770 | 0.148 | 0.000 | 0.890 | 0.410 | 0.478 | 0.510 | 0.502 | 0.222 | 1.070 | 0.483 | 0.429 | 0.950 | 0.556 | 0.333 | 0.770 | 0.415 | 0.000 | 0.810 | 0.449 | 0.440 | 0.850 | 0.479 | 0.371 | 0.890 | 0.432 | 0.441 |
| Contrast | 1.080 | 0.413 | 0.385 | 0.840 | 0.409 | 0.529 | 0.740 | 0.457 | 0.200 | 0.920 | 0.144 | 0.000 | 0.890 | 0.407 | 0.478 | 0.470 | 0.488 | 0.222 | 1.040 | 0.485 | 0.429 | 0.890 | 0.555 | 0.367 | 0.680 | 0.405 | 0.000 | 0.780 | 0.456 | 0.480 | 0.810 | 0.478 | 0.400 | 0.870 | 0.429 | 0.447 |
| Defocus Blur | 1.020 | 0.401 | 0.385 | 0.860 | 0.411 | 0.529 | 0.790 | 0.455 | 0.200 | 1.030 | 0.166 | 0.000 | 0.900 | 0.406 | 0.478 | 0.560 | 0.501 | 0.222 | 0.950 | 0.466 | 0.429 | 0.880 | 0.550 | 0.333 | 0.700 | 0.408 | 0.000 | 0.700 | 0.448 | 0.440 | 0.780 | 0.473 | 0.371 | 0.870 | 0.427 | 0.441 |
| Edge Sawtooth | 0.940 | 0.394 | 0.308 | 0.840 | 0.411 | 0.529 | 0.720 | 0.457 | 0.200 | 0.790 | 0.163 | 0.000 | 0.850 | 0.406 | 0.457 | 0.420 | 0.493 | 0.222 | 1.000 | 0.480 | 0.429 | 0.860 | 0.558 | 0.333 | 0.650 | 0.420 | 0.000 | 0.770 | 0.442 | 0.440 | 0.780 | 0.476 | 0.371 | 0.840 | 0.429 | 0.428 |
| Elastic | 1.060 | 0.413 | 0.538 | 0.830 | 0.414 | 0.543 | 0.710 | 0.458 | 0.200 | 0.810 | 0.155 | 0.000 | 0.870 | 0.410 | 0.500 | 0.440 | 0.496 | 0.333 | 0.990 | 0.473 | 0.429 | 0.930 | 0.554 | 0.367 | 0.640 | 0.410 | 0.000 | 0.790 | 0.449 | 0.480 | 0.800 | 0.475 | 0.414 | 0.870 | 0.431 | 0.474 |
| Flicker | 1.150 | 0.404 | 0.462 | 0.860 | 0.412 | 0.543 | 0.750 | 0.455 | 0.200 | 0.760 | 0.151 | 0.000 | 0.910 | 0.407 | 0.489 | 0.580 | 0.499 | 0.333 | 0.990 | 0.467 | 0.429 | 0.860 | 0.554 | 0.333 | 0.860 | 0.416 | 0.000 | 0.810 | 0.445 | 0.480 | 0.830 | 0.473 | 0.414 | 0.890 | 0.429 | 0.467 |
| Foggy | 1.010 | 0.404 | 0.385 | 0.870 | 0.415 | 0.529 | 0.810 | 0.463 | 0.133 | 0.630 | 0.158 | 0.000 | 0.880 | 0.411 | 0.478 | 0.480 | 0.498 | 0.222 | 1.070 | 0.480 | 0.571 | 0.910 | 0.556 | 0.367 | 0.840 | 0.429 | 0.000 | 0.760 | 0.449 | 0.440 | 0.830 | 0.479 | 0.400 | 0.870 | 0.433 | 0.454 |
| Frame Drop | 0.800 | 0.370 | 0.308 | 0.720 | 0.396 | 0.471 | 0.680 | 0.449 | 0.133 | 0.600 | 0.147 | 0.000 | 0.740 | 0.390 | 0.424 | 0.410 | 0.486 | 0.222 | 0.860 | 0.449 | 0.286 | 0.870 | 0.552 | 0.400 | 0.410 | 0.368 | 0.000 | 0.800 | 0.446 | 0.440 | 0.730 | 0.462 | 0.386 | 0.750 | 0.413 | 0.414 |
| Frame Repeat | 0.820 | 0.371 | 0.385 | 0.710 | 0.392 | 0.486 | 0.660 | 0.450 | 0.200 | 0.660 | 0.147 | 0.000 | 0.730 | 0.388 | 0.435 | 0.500 | 0.488 | 0.222 | 0.940 | 0.461 | 0.286 | 0.890 | 0.542 | 0.367 | 0.680 | 0.389 | 0.000 | 0.750 | 0.434 | 0.480 | 0.780 | 0.462 | 0.386 | 0.750 | 0.412 | 0.421 |
| Frame Replace | 1.040 | 0.404 | 0.308 | 0.920 | 0.418 | 0.543 | 0.810 | 0.456 | 0.200 | 0.850 | 0.152 | 0.000 | 0.930 | 0.412 | 0.467 | 0.460 | 0.491 | 0.222 | 1.100 | 0.480 | 0.429 | 0.970 | 0.555 | 0.367 | 0.740 | 0.428 | 0.000 | 0.830 | 0.447 | 0.400 | 0.860 | 0.478 | 0.371 | 0.920 | 0.432 | 0.434 |
| Frost | 0.890 | 0.391 | 0.308 | 0.830 | 0.410 | 0.529 | 0.760 | 0.462 | 0.200 | 0.900 | 0.143 | 0.000 | 0.840 | 0.404 | 0.467 | 0.410 | 0.490 | 0.222 | 0.960 | 0.474 | 0.429 | 0.930 | 0.560 | 0.367 | 0.590 | 0.398 | 0.000 | 0.780 | 0.441 | 0.400 | 0.780 | 0.472 | 0.371 | 0.840 | 0.425 | 0.434 |
| Gaussian | 0.990 | 0.402 | 0.308 | 0.830 | 0.410 | 0.514 | 0.780 | 0.465 | 0.200 | 0.740 | 0.156 | 0.000 | 0.860 | 0.407 | 0.446 | 0.510 | 0.498 | 0.222 | 0.990 | 0.479 | 0.286 | 0.990 | 0.561 | 0.333 | 0.860 | 0.412 | 0.000 | 0.830 | 0.448 | 0.400 | 0.860 | 0.478 | 0.343 | 0.860 | 0.430 | 0.408 |
| Gaussian Blur | 0.870 | 0.387 | 0.308 | 0.760 | 0.402 | 0.514 | 0.750 | 0.448 | 0.200 | 0.890 | 0.151 | 0.000 | 0.800 | 0.398 | 0.446 | 0.480 | 0.500 | 0.222 | 0.860 | 0.456 | 0.286 | 0.740 | 0.534 | 0.367 | 0.490 | 0.390 | 0.000 | 0.740 | 0.442 | 0.440 | 0.710 | 0.464 | 0.386 | 0.780 | 0.419 | 0.428 |
| Glass Blur | 1.020 | 0.406 | 0.308 | 0.830 | 0.412 | 0.529 | 0.780 | 0.462 | 0.133 | 1.030 | 0.155 | 0.000 | 0.850 | 0.408 | 0.457 | 0.480 | 0.497 | 0.222 | 1.050 | 0.479 | 0.286 | 0.800 | 0.544 | 0.400 | 0.800 | 0.424 | 0.000 | 0.800 | 0.448 | 0.480 | 0.830 | 0.477 | 0.400 | 0.840 | 0.430 | 0.441 |
| H265 Artifacts | 0.880 | 0.383 | 0.154 | 0.750 | 0.397 | 0.514 | 0.610 | 0.435 | 0.067 | 1.020 | 0.169 | 0.000 | 0.780 | 0.392 | 0.424 | 0.500 | 0.491 | 0.222 | 0.800 | 0.456 | 0.429 | 0.760 | 0.544 | 0.333 | 0.600 | 0.406 | 0.000 | 0.740 | 0.443 | 0.440 | 0.720 | 0.467 | 0.371 | 0.760 | 0.415 | 0.408 |
| Impulse | 0.900 | 0.390 | 0.308 | 0.760 | 0.403 | 0.514 | 0.700 | 0.445 | 0.067 | 0.840 | 0.155 | 0.000 | 0.790 | 0.398 | 0.435 | 0.430 | 0.491 | 0.222 | 0.910 | 0.462 | 0.429 | 0.760 | 0.544 | 0.367 | 0.420 | 0.384 | 0.000 | 0.710 | 0.443 | 0.480 | 0.700 | 0.466 | 0.400 | 0.780 | 0.420 | 0.428 |
| Jpeg Artifact | 0.940 | 0.398 | 0.231 | 0.860 | 0.411 | 0.514 | 0.770 | 0.468 | 0.133 | 1.060 | 0.162 | 0.000 | 0.850 | 0.408 | 0.446 | 0.500 | 0.502 | 0.222 | 0.980 | 0.468 | 0.286 | 0.870 | 0.561 | 0.400 | 0.600 | 0.403 | 0.000 | 0.800 | 0.444 | 0.800 | 0.740 | 0.474 | 0.386 | 0.840 | 0.428 | 0.428 |
| Motion Blur | 0.810 | 0.379 | 0.231 | 0.770 | 0.398 | 0.529 | 0.760 | 0.451 | 0.133 | 1.060 | 0.154 | 0.000 | 0.800 | 0.394 | 0.435 | 0.440 | 0.496 | 0.222 | 0.830 | 0.454 | 0.286 | 0.770 | 0.538 | 0.333 | 0.530 | 0.399 | 0.000 | 0.720 | 0.441 | 0.400 | 0.700 | 0.465 | 0.343 | 0.780 | 0.417 | 0.401 |
| Overexposure | 1.050 | 0.399 | 0.462 | 0.860 | 0.417 | 0.543 | 0.820 | 0.456 | 0.200 | 0.840 | 0.157 | 0.000 | 0.900 | 0.410 | 0.500 | 0.450 | 0.499 | 0.222 | 0.940 | 0.472 | 0.286 | 0.860 | 0.548 | 0.333 | 0.750 | 0.411 | 0.000 | 0.770 | 0.441 | 0.480 | 0.780 | 0.472 | 0.371 | 0.870 | 0.429 | 0.454 |
| Poisson | 0.990 | 0.385 | 0.308 | 0.790 | 0.412 | 0.529 | 0.730 | 0.461 | 0.200 | 1.020 | 0.147 | 0.000 | 0.800 | 0.406 | 0.446 | 0.530 | 0.497 | 0.222 | 0.940 | 0.470 | 0.286 | 0.880 | 0.558 | 0.333 | 0.780 | 0.413 | 0.000 | 0.790 | 0.446 | 0.386 | 0.810 | 0.474 | 0.386 | 0.880 | 0.428 | 0.441 |
| Rainy | 1.010 | 0.403 | 0.308 | 0.830 | 0.412 | 0.543 | 0.750 | 0.460 | 0.133 | 0.870 | 0.147 | 0.000 | 0.870 | 0.408 | 0.467 | 0.470 | 0.494 | 0.222 | 0.960 | 0.480 | 0.286 | 1.010 | 0.562 | 0.367 | 0.670 | 0.422 | 0.000 | 0.790 | 0.448 | 0.440 | 0.810 | 0.479 | 0.371 | 0.860 | 0.430 | 0.434 |
| Random Block | 1.030 | 0.401 | 0.308 | 0.760 | 0.407 | 0.557 | 0.700 | 0.466 | 0.067 | 0.850 | 0.155 | 0.000 | 0.790 | 0.406 | 0.489 | 0.510 | 0.498 | 0.222 | 0.870 | 0.469 | 0.429 | 0.760 | 0.560 | 0.333 | 0.420 | 0.384 | 0.000 | 0.790 | 0.445 | 0.520 | 0.700 | 0.476 | 0.400 | 0.780 | 0.420 | 0.461 |
| Reflect | 1.060 | 0.408 | 0.308 | 0.830 | 0.413 | 0.529 | 0.770 | 0.455 | 0.200 | 0.920 | 0.152 | 0.000 | 0.880 | 0.409 | 0.457 | 0.430 | 0.494 | 0.222 | 0.990 | 0.481 | 0.429 | 0.860 | 0.559 | 0.367 | 0.630 | 0.408 | 0.000 | 0.780 | 0.448 | 0.400 | 0.780 | 0.477 | 0.371 | 0.860 | 0.431 | 0.428 |
| Resolution Degrade | 1.000 | 0.401 | 0.385 | 0.810 | 0.409 | 0.543 | 0.830 | 0.462 | 0.133 | 0.920 | 0.160 | 0.000 | 0.860 | 0.407 | 0.467 | 0.460 | 0.495 | 0.222 | 0.990 | 0.473 | 0.286 | 0.770 | 0.554 | 0.333 | 0.670 | 0.426 | 0.000 | 0.790 | 0.438 | 0.440 | 0.780 | 0.473 | 0.357 | 0.840 | 0.427 | 0.428 |
| Rolling Shutter | 0.930 | 0.398 | 0.231 | 0.790 | 0.403 | 0.514 | 0.660 | 0.448 | 0.200 | 0.800 | 0.147 | 0.000 | 0.800 | 0.399 | 0.446 | 0.490 | 0.482 | 0.222 | 1.080 | 0.476 | 0.286 | 0.950 | 0.559 | 0.367 | 0.630 | 0.411 | 0.000 | 0.760 | 0.439 | 0.400 | 0.820 | 0.472 | 0.357 | 0.820 | 0.422 | 0.414 |
| Shadow | 0.980 | 0.403 | 0.385 | 0.830 | 0.411 | 0.514 | 0.770 | 0.464 | 0.200 | 0.810 | 0.158 | 0.000 | 0.860 | 0.409 | 0.467 | 0.460 | 0.496 | 0.222 | 1.050 | 0.479 | 0.286 | 0.890 | 0.565 | 0.367 | 0.690 | 0.419 | 0.000 | 0.790 | 0.444 | 0.440 | 0.820 | 0.478 | 0.371 | 0.860 | 0.431 | 0.434 |
| Snow | 1.020 | 0.408 | 0.538 | 0.850 | 0.413 | 0.586 | 0.780 | 0.462 | 0.200 | 0.630 | 0.144 | 0.000 | 0.860 | 0.409 | 0.533 | 0.480 | 0.498 | 0.222 | 1.040 | 0.476 | 0.429 | 0.960 | 0.557 | 0.333 | 0.700 | 0.406 | 0.000 | 0.740 | 0.449 | 0.520 | 0.820 | 0.477 | 0.386 | 0.860 | 0.430 | 0.474 |
| Speckle | 0.920 | 0.394 | 0.308 | 0.840 | 0.407 | 0.557 | 0.700 | 0.448 | 0.200 | 0.970 | 0.154 | 0.000 | 0.850 | 0.402 | 0.478 | 0.510 | 0.491 | 0.222 | 0.920 | 0.468 | 0.429 | 0.980 | 0.556 | 0.333 | 0.600 | 0.395 | 0.000 | 0.780 | 0.440 | 0.480 | 0.780 | 0.469 | 0.386 | 0.840 | 0.424 | 0.447 |
| Stretch Squish | 0.900 | 0.388 | 0.308 | 0.750 | 0.400 | 0.529 | 0.660 | 0.456 | 0.200 | 0.940 | 0.159 | 0.000 | 0.800 | 0.398 | 0.446 | 0.460 | 0.487 | 0.222 | 0.850 | 0.453 | 0.429 | 0.770 | 0.547 | 0.367 | 0.580 | 0.397 | 0.000 | 0.740 | 0.442 | 0.480 | 0.740 | 0.464 | 0.400 | 0.790 | 0.419 | 0.434 |
| Target Block | 0.990 | 0.399 | 0.385 | 0.840 | 0.406 | 0.500 | 0.870 | 0.472 | 0.133 | 0.870 | 0.152 | 0.000 | 0.870 | 0.405 | 0.446 | 0.520 | 0.498 | 0.222 | 1.020 | 0.482 | 0.429 | 0.900 | 0.551 | 0.333 | 0.640 | 0.411 | 0.000 | 0.820 | 0.453 | 0.440 | 0.830 | 0.479 | 0.371 | 0.870 | 0.429 | 0.421 |
| Temporal Jitter | 0.990 | 0.385 | 0.308 | 0.810 | 0.408 | 0.514 | 0.700 | 0.455 | 0.200 | 0.740 | 0.154 | 0.000 | 0.840 | 0.404 | 0.467 | 0.420 | 0.491 | 0.222 | 0.930 | 0.474 | 0.286 | 0.790 | 0.541 | 0.367 | 0.620 | 0.396 | 0.000 | 0.830 | 0.447 | 0.440 | 0.770 | 0.470 | 0.371 | 0.820 | 0.425 | 0.434 |
| Underexposure | 1.040 | 0.414 | 0.385 | 0.860 | 0.417 | 0.514 | 0.840 | 0.464 | 0.200 | 0.840 | 0.146 | 0.000 | 0.890 | 0.413 | 0.467 | 0.410 | 0.489 | 0.222 | 1.120 | 0.475 | 0.429 | 0.940 | 0.551 | 0.300 | 0.810 | 0.418 | 0.000 | 0.830 | 0.453 | 0.440 | 0.860 | 0.476 | 0.357 | 0.890 | 0.433 | 0.428 |
| Zoom Blur | 0.880 | 0.394 | 0.231 | 0.760 | 0.406 | 0.514 | 0.890 | 0.451 | 0.067 | 0.890 | 0.147 | 0.000 | 0.790 | 0.401 | 0.435 | 0.570 | 0.503 | 0.333 | 0.930 | 0.470 | 0.571 | 0.810 | 0.546 | 0.367 | 0.690 | 0.411 | 0.000 | 0.770 | 0.446 | 0.440 | 0.770 | 0.473 | 0.414 | 0.790 | 0.424 | 0.434 |

Table 13: Chat-UniVi performance across different question types

| Noise Type | CP GPT | CP SBERT | CP Acc | FP-S GPT | FP-S SBERT | FP-S Acc | FP-C GPT | FP-C SBERT | FP-C Acc | HL GPT | HL SBERT | HL Acc | Mean GPT | Mean SBERT | Mean Acc | LR GPT | LR SBERT | LR Acc | AR GPT | AR SBERT | AR Acc | RR GPT | RR SBERT | RR Acc | CSR GPT | CSR SBERT | CSR Acc | TR GPT | TR SBERT | TR Acc | Mean2 GPT | Mean2 SBERT | Mean2 Acc | Overall GPT | Overall SBERT | Overall Acc |
|---|---|---|---|---|---|---|---|---|---|---|---|---|---|---|---|---|---|---|---|---|---|---|---|---|---|---|---|---|---|---|---|---|---|---|---|---|
| Clean | 1.110 | 0.404 | 0.538 | 0.870 | 0.413 | 0.600 | 0.710 | 0.457 | 0.200 | 0.270 | 0.142 | 0.000 | 0.870 | 0.407 | 0.533 | 0.520 | 0.492 | 0.556 | 1.040 | 0.477 | 0.571 | 0.890 | 0.559 | 0.367 | 0.640 | 0.406 | 0.000 | 0.830 | 0.440 | 0.560 | 0.830 | 0.473 | 0.486 | 0.870 | 0.429 | 0.507 |
| Bit Error | 0.960 | 0.398 | 0.385 | 0.880 | 0.417 | 0.586 | 0.620 | 0.450 | 0.180 | 0.180 | 0.134 | 0.000 | 0.840 | 0.407 | 0.522 | 0.600 | 0.491 | 0.556 | 1.060 | 0.478 | 0.571 | 1.010 | 0.567 | 0.367 | 0.650 | 0.413 | 0.000 | 0.850 | 0.444 | 0.400 | 0.850 | 0.474 | 0.476 | 0.850 | 0.429 | 0.487 |
| Bright Transform | 1.000 | 0.394 | 0.385 | 0.860 | 0.414 | 0.614 | 0.720 | 0.455 | 0.290 | 0.290 | 0.144 | 0.333 | 0.850 | 0.406 | 0.533 | 0.510 | 0.479 | 0.778 | 1.010 | 0.477 | 0.571 | 0.970 | 0.564 | 0.367 | 0.790 | 0.421 | 0.000 | 0.840 | 0.440 | 0.560 | 0.840 | 0.474 | 0.514 | 0.840 | 0.428 | 0.513 |
| Color Quantized | 0.970 | 0.400 | 0.308 | 0.850 | 0.414 | 0.600 | 0.660 | 0.455 | 0.190 | 0.190 | 0.140 | 0.333 | 0.820 | 0.407 | 0.522 | 0.560 | 0.490 | 0.556 | 1.160 | 0.489 | 0.286 | 1.010 | 0.554 | 0.300 | 0.810 | 0.413 | 0.000 | 0.900 | 0.443 | 0.480 | 0.900 | 0.476 | 0.400 | 0.900 | 0.429 | 0.461 |
| Color Shift | 1.030 | 0.403 | 0.385 | 0.860 | 0.414 | 0.614 | 0.710 | 0.460 | 0.230 | 0.230 | 0.143 | 0.333 | 0.850 | 0.408 | 0.522 | 0.560 | 0.486 | 0.778 | 1.000 | 0.477 | 0.429 | 1.000 | 0.562 | 0.400 | 0.630 | 0.407 | 0.000 | 0.870 | 0.441 | 0.440 | 0.870 | 0.474 | 0.457 | 0.870 | 0.430 | 0.493 |
| Contrast | 1.040 | 0.404 | 0.385 | 0.880 | 0.416 | 0.643 | 0.700 | 0.459 | 0.230 | 0.230 | 0.143 | 0.333 | 0.870 | 0.410 | 0.565 | 0.620 | 0.490 | 0.556 | 1.110 | 0.487 | 0.429 | 1.100 | 0.565 | 0.400 | 0.740 | 0.424 | 0.000 | 0.900 | 0.441 | 0.360 | 0.900 | 0.478 | 0.414 | 0.890 | 0.432 | 0.493 |
| Defocus Blur | 1.050 | 0.406 | 0.308 | 0.860 | 0.413 | 0.571 | 0.770 | 0.463 | 0.270 | 0.270 | 0.138 | 0.000 | 0.860 | 0.408 | 0.478 | 0.530 | 0.491 | 0.556 | 1.120 | 0.483 | 0.429 | 1.090 | 0.567 | 0.400 | 0.840 | 0.424 | 0.000 | 0.790 | 0.440 | 0.400 | 0.790 | 0.477 | 0.443 | 0.880 | 0.432 | 0.461 |
| Edge Sawtooth | 0.990 | 0.392 | 0.385 | 0.850 | 0.413 | 0.629 | 0.720 | 0.460 | 0.290 | 0.290 | 0.142 | 0.333 | 0.840 | 0.406 | 0.533 | 0.540 | 0.486 | 0.667 | 1.000 | 0.472 | 0.143 | 0.690 | 0.571 | 0.367 | 0.690 | 0.414 | 0.000 | 0.810 | 0.443 | 0.440 | 0.850 | 0.475 | 0.414 | 0.860 | 0.428 | 0.480 |
| Elastic | 1.030 | 0.408 | 0.385 | 0.890 | 0.416 | 0.629 | 0.730 | 0.459 | 0.110 | 0.110 | 0.135 | 0.067 | 0.870 | 0.411 | 0.533 | 0.420 | 0.477 | 0.667 | 1.090 | 0.491 | 0.143 | 1.050 | 0.562 | 0.400 | 0.810 | 0.425 | 0.000 | 0.890 | 0.441 | 0.440 | 0.890 | 0.476 | 0.429 | 0.880 | 0.432 | 0.480 |
| Flicker | 1.020 | 0.405 | 0.462 | 0.840 | 0.415 | 0.514 | 0.650 | 0.455 | 0.290 | 0.290 | 0.140 | 0.200 | 0.840 | 0.408 | 0.478 | 0.510 | 0.491 | 0.667 | 1.010 | 0.483 | 0.571 | 1.030 | 0.563 | 0.333 | 0.680 | 0.414 | 0.000 | 0.870 | 0.441 | 0.440 | 0.870 | 0.477 | 0.443 | 0.870 | 0.430 | 0.461 |
| Foggy | 0.950 | 0.396 | 0.385 | 0.880 | 0.416 | 0.614 | 0.700 | 0.452 | 0.340 | 0.340 | 0.150 | 0.200 | 0.860 | 0.408 | 0.543 | 0.520 | 0.486 | 0.667 | 1.060 | 0.482 | 0.429 | 1.060 | 0.562 | 0.367 | 0.650 | 0.405 | 0.000 | 0.830 | 0.444 | 0.520 | 0.870 | 0.475 | 0.486 | 0.870 | 0.430 | 0.513 |
| Frame Drop | 0.850 | 0.374 | 0.385 | 0.750 | 0.404 | 0.571 | 0.590 | 0.444 | 0.230 | 0.230 | 0.136 | 0.133 | 0.740 | 0.395 | 0.522 | 0.500 | 0.488 | 0.778 | 0.990 | 0.463 | 0.286 | 0.910 | 0.554 | 0.400 | 0.570 | 0.396 | 0.000 | 0.790 | 0.437 | 0.480 | 0.790 | 0.466 | 0.471 | 0.770 | 0.418 | 0.487 |
| Frame Repeat | 0.770 | 0.362 | 0.308 | 0.690 | 0.396 | 0.557 | 0.530 | 0.431 | 0.350 | 0.350 | 0.142 | 0.267 | 0.680 | 0.386 | 0.500 | 0.560 | 0.483 | 0.667 | 0.930 | 0.454 | 0.286 | 0.950 | 0.550 | 0.400 | 0.470 | 0.387 | 0.000 | 0.750 | 0.435 | 0.360 | 0.790 | 0.462 | 0.414 | 0.720 | 0.410 | 0.454 |
| Frame Replace | 1.020 | 0.399 | 0.385 | 0.890 | 0.416 | 0.629 | 0.720 | 0.458 | 0.210 | 0.210 | 0.145 | 0.200 | 0.870 | 0.410 | 0.543 | 0.510 | 0.487 | 0.556 | 1.090 | 0.480 | 0.429 | 1.020 | 0.565 | 0.333 | 0.730 | 0.420 | 0.000 | 0.840 | 0.445 | 0.480 | 0.880 | 0.477 | 0.429 | 0.880 | 0.431 | 0.487 |
| Frost | 0.980 | 0.403 | 0.462 | 0.820 | 0.412 | 0.600 | 0.630 | 0.444 | 0.180 | 0.180 | 0.133 | 0.133 | 0.800 | 0.405 | 0.543 | 0.480 | 0.480 | 0.667 | 1.010 | 0.479 | 0.571 | 1.050 | 0.562 | 0.433 | 0.650 | 0.408 | 0.000 | 0.850 | 0.444 | 0.480 | 0.830 | 0.474 | 0.486 | 0.830 | 0.427 | 0.513 |
| Gaussian | 0.950 | 0.402 | 0.385 | 0.830 | 0.405 | 0.600 | 0.710 | 0.459 | 0.260 | 0.260 | 0.140 | 0.200 | 0.820 | 0.406 | 0.543 | 0.600 | 0.493 | 0.778 | 1.080 | 0.481 | 0.429 | 1.080 | 0.572 | 0.400 | 0.730 | 0.410 | 0.000 | 0.820 | 0.442 | 0.360 | 0.800 | 0.477 | 0.486 | 0.850 | 0.421 | 0.500 |
| Gaussian Blur | 0.870 | 0.388 | 0.385 | 0.790 | 0.409 | 0.571 | 0.640 | 0.447 | 0.180 | 0.180 | 0.137 | 0.133 | 0.770 | 0.398 | 0.500 | 0.470 | 0.490 | 0.556 | 1.020 | 0.472 | 0.429 | 0.920 | 0.557 | 0.367 | 0.630 | 0.403 | 0.000 | 0.810 | 0.440 | 0.360 | 0.800 | 0.470 | 0.400 | 0.790 | 0.421 | 0.461 |
| Glass Blur | 1.040 | 0.404 | 0.385 | 0.820 | 0.413 | 0.643 | 0.770 | 0.463 | 0.240 | 0.240 | 0.137 | 0.200 | 0.840 | 0.406 | 0.554 | 0.580 | 0.491 | 0.667 | 1.110 | 0.476 | 0.429 | 0.920 | 0.551 | 0.367 | 0.720 | 0.404 | 0.000 | 0.900 | 0.441 | 0.400 | 0.900 | 0.471 | 0.414 | 0.860 | 0.427 | 0.500 |
| H265 Artifacts | 0.870 | 0.388 | 0.385 | 0.780 | 0.403 | 0.557 | 0.630 | 0.444 | 0.230 | 0.230 | 0.144 | 0.133 | 0.760 | 0.397 | 0.489 | 0.580 | 0.488 | 0.667 | 0.760 | 0.445 | 0.286 | 0.860 | 0.553 | 0.367 | 0.780 | 0.408 | 0.000 | 0.770 | 0.434 | 0.400 | 0.770 | 0.463 | 0.414 | 0.770 | 0.418 | 0.454 |
| Impulse | 0.810 | 0.382 | 0.385 | 0.790 | 0.404 | 0.586 | 0.580 | 0.446 | 0.260 | 0.260 | 0.134 | 0.133 | 0.740 | 0.395 | 0.500 | 0.470 | 0.484 | 0.556 | 1.010 | 0.471 | 0.286 | 1.010 | 0.552 | 0.433 | 0.650 | 0.402 | 0.000 | 0.810 | 0.438 | 0.360 | 0.810 | 0.468 | 0.414 | 0.770 | 0.418 | 0.461 |
| Jpeg Artifact | 0.880 | 0.395 | 0.308 | 0.820 | 0.413 | 0.614 | 0.690 | 0.458 | 0.240 | 0.240 | 0.141 | 0.133 | 0.760 | 0.406 | 0.533 | 0.540 | 0.490 | 0.667 | 1.030 | 0.486 | 0.429 | 1.030 | 0.560 | 0.367 | 0.690 | 0.414 | 0.000 | 0.840 | 0.433 | 0.440 | 0.840 | 0.473 | 0.443 | 0.820 | 0.427 | 0.487 |
| Motion Blur | 0.820 | 0.378 | 0.395 | 0.790 | 0.401 | 0.600 | 0.670 | 0.443 | 0.290 | 0.290 | 0.144 | 0.000 | 0.760 | 0.393 | 0.500 | 0.580 | 0.488 | 0.667 | 0.890 | 0.467 | 0.429 | 0.890 | 0.551 | 0.400 | 0.600 | 0.393 | 0.000 | 0.810 | 0.440 | 0.360 | 0.780 | 0.466 | 0.443 | 0.780 | 0.416 | 0.474 |
| Overexposure | 1.010 | 0.400 | 0.385 | 0.860 | 0.411 | 0.629 | 0.700 | 0.454 | 0.260 | 0.260 | 0.148 | 0.333 | 0.850 | 0.406 | 0.565 | 0.400 | 0.480 | 0.667 | 0.940 | 0.477 | 0.429 | 1.080 | 0.564 | 0.367 | 0.530 | 0.406 | 0.000 | 0.860 | 0.445 | 0.360 | 0.850 | 0.473 | 0.429 | 0.850 | 0.428 | 0.493 |
| Poisson | 0.980 | 0.393 | 0.462 | 0.830 | 0.414 | 0.600 | 0.620 | 0.450 | 0.290 | 0.290 | 0.143 | 0.200 | 0.830 | 0.403 | 0.533 | 0.460 | 0.484 | 0.667 | 0.940 | 0.476 | 0.429 | 0.980 | 0.568 | 0.400 | 0.620 | 0.417 | 0.000 | 0.830 | 0.440 | 0.360 | 0.830 | 0.475 | 0.386 | 0.840 | 0.426 | 0.480 |
| Rainy | 1.030 | 0.402 | 0.385 | 0.860 | 0.414 | 0.600 | 0.640 | 0.458 | 0.260 | 0.260 | 0.143 | 0.133 | 0.850 | 0.408 | 0.533 | 0.470 | 0.481 | 0.667 | 1.040 | 0.477 | 0.429 | 1.030 | 0.553 | 0.367 | 0.630 | 0.420 | 0.000 | 0.800 | 0.441 | 0.440 | 0.820 | 0.472 | 0.443 | 0.850 | 0.429 | 0.487 |
| Random Block | 1.070 | 0.398 | 0.462 | 0.830 | 0.410 | 0.614 | 0.700 | 0.455 | 0.240 | 0.240 | 0.141 | 0.200 | 0.840 | 0.404 | 0.511 | 0.570 | 0.487 | 0.667 | 1.060 | 0.481 | 0.286 | 0.750 | 0.560 | 0.333 | 0.750 | 0.422 | 0.000 | 0.840 | 0.440 | 0.480 | 0.880 | 0.474 | 0.429 | 0.850 | 0.426 | 0.474 |
| Reflect | 0.920 | 0.396 | 0.385 | 0.830 | 0.409 | 0.586 | 0.670 | 0.451 | 0.370 | 0.370 | 0.139 | 0.200 | 0.810 | 0.404 | 0.511 | 0.450 | 0.484 | 0.556 | 1.110 | 0.484 | 0.429 | 0.980 | 0.566 | 0.267 | 0.640 | 0.403 | 0.000 | 0.860 | 0.440 | 0.400 | 0.830 | 0.474 | 0.386 | 0.830 | 0.426 | 0.454 |
| Resolution Degrade | 0.970 | 0.398 | 0.308 | 0.830 | 0.410 | 0.600 | 0.650 | 0.451 | 0.260 | 0.260 | 0.141 | 0.133 | 0.820 | 0.404 | 0.511 | 0.590 | 0.488 | 0.667 | 1.010 | 0.475 | 0.286 | 1.120 | 0.561 | 0.367 | 0.670 | 0.403 | 0.000 | 0.830 | 0.443 | 0.440 | 0.880 | 0.474 | 0.429 | 0.850 | 0.426 | 0.474 |
| Rolling Shutter | 0.990 | 0.403 | 0.308 | 0.870 | 0.410 | 0.614 | 0.700 | 0.455 | 0.260 | 0.260 | 0.139 | 0.000 | 0.840 | 0.402 | 0.511 | 0.600 | 0.492 | 0.556 | 1.040 | 0.489 | 0.286 | 1.020 | 0.563 | 0.333 | 0.670 | 0.419 | 0.000 | 0.840 | 0.432 | 0.400 | 0.830 | 0.473 | 0.343 | 0.830 | 0.426 | 0.467 |
| Shadow | 1.080 | 0.408 | 0.385 | 0.850 | 0.414 | 0.586 | 0.770 | 0.459 | 0.240 | 0.240 | 0.142 | 0.333 | 0.870 | 0.410 | 0.522 | 0.550 | 0.493 | 0.778 | 1.060 | 0.482 | 0.571 | 0.950 | 0.563 | 0.333 | 0.750 | 0.410 | 0.000 | 0.850 | 0.435 | 0.520 | 0.860 | 0.477 | 0.443 | 0.880 | 0.432 | 0.474 |
| Snow | 0.920 | 0.388 | 0.308 | 0.820 | 0.411 | 0.614 | 0.650 | 0.452 | 0.240 | 0.240 | 0.149 | 0.333 | 0.820 | 0.403 | 0.533 | 0.580 | 0.498 | 0.778 | 1.100 | 0.473 | 0.571 | 1.050 | 0.562 | 0.333 | 0.650 | 0.421 | 0.000 | 0.860 | 0.445 | 0.400 | 0.850 | 0.476 | 0.443 | 0.850 | 0.426 | 0.480 |
| Stretch Squish | 0.880 | 0.386 | 0.385 | 0.790 | 0.403 | 0.600 | 0.710 | 0.451 | 0.310 | 0.310 | 0.138 | 0.333 | 0.780 | 0.398 | 0.522 | 0.540 | 0.485 | 0.143 | 0.930 | 0.457 | 0.143 | 1.060 | 0.554 | 0.367 | 0.740 | 0.409 | 0.000 | 0.750 | 0.441 | 0.400 | 0.810 | 0.467 | 0.400 | 0.790 | 0.419 | 0.461 |
| Target Block | 0.920 | 0.387 | 0.385 | 0.790 | 0.412 | 0.557 | 0.680 | 0.452 | 0.210 | 0.210 | 0.143 | 0.000 | 0.780 | 0.401 | 0.511 | 0.560 | 0.494 | 0.556 | 0.990 | 0.477 | 0.429 | 1.100 | 0.561 | 0.333 | 0.740 | 0.412 | 0.000 | 0.800 | 0.444 | 0.440 | 0.840 | 0.471 | 0.400 | 0.830 | 0.427 | 0.474 |
| Temporal Jitter | 0.920 | 0.402 | 0.385 | 0.810 | 0.408 | 0.586 | 0.710 | 0.457 | 0.110 | 0.110 | 0.135 | 0.200 | 0.820 | 0.409 | 0.533 | 0.580 | 0.493 | 0.556 | 1.060 | 0.485 | 0.571 | 1.060 | 0.565 | 0.367 | 0.780 | 0.424 | 0.000 | 0.850 | 0.446 | 0.440 | 0.920 | 0.479 | 0.471 | 0.880 | 0.431 | 0.480 |
| Underexposure | 1.020 | 0.402 | 0.308 | 0.860 | 0.415 | 0.614 | 0.710 | 0.457 | 0.230 | 0.230 | 0.148 | 0.333 | 0.850 | 0.409 | 0.522 | 0.580 | 0.493 | 0.778 | 1.010 | 0.473 | 0.286 | 0.990 | 0.564 | 0.367 | 0.590 | 0.398 | 0.000 | 0.860 | 0.439 | 0.440 | 0.850 | 0.470 | 0.414 | 0.840 | 0.425 | 0.441 |
| Zoom Blur | 0.910 | 0.395 | | 0.840 | 0.409 | 0.543 | 0.720 | 0.452 | | | | | 0.820 | 0.404 | 0.467 | 0.570 | | | | | | | 0.557 | | | | | | | | | | | | | |

Table 14: GPT-4o performance across different question types

| Noise Type | CP GPT | CP SBERT | CP Acc | FP-S GPT | FP-S SBERT | FP-S Acc | FP-C GPT | FP-C SBERT | FP-C Acc | HL GPT | HL SBERT | HL Acc | Mean GPT | Mean SBERT | Mean Acc | LR GPT | LR SBERT | LR Acc | AR GPT | AR SBERT | AR Acc | RR GPT | RR SBERT | RR Acc | CSR GPT | CSR SBERT | CSR Acc | TR GPT | TR SBERT | TR Acc | Mean2 GPT | Mean2 SBERT | Mean2 Acc | Overall GPT | Overall SBERT | Overall Acc |
|---|---|---|---|---|---|---|---|---|---|---|---|---|---|---|---|---|---|---|---|---|---|---|---|---|---|---|---|---|---|---|---|---|---|---|---|---|
| Clean | 1.980 | 0.529 | 0.538 | 1.670 | 0.531 | 0.614 | 1.570 | 0.515 | 0.600 | 2.400 | 0.188 | 0.667 | 1.740 | 0.516 | 0.609 | 1.360 | 0.653 | 0.667 | 1.940 | 0.527 | 0.571 | 1.750 | 0.599 | 0.467 | 1.910 | 0.524 | 0.000 | 1.630 | 0.493 | 0.360 | 1.720 | 0.550 | 0.500 | 1.740 | 0.527 | 0.546 |
| Bit Error | 1.770 | 0.517 | 0.615 | 1.740 | 0.553 | 0.600 | 1.380 | 0.503 | 0.667 | 0.970 | 0.144 | 0.667 | 1.670 | 0.523 | 0.620 | 1.420 | 0.681 | 1.000 | 2.100 | 0.540 | 0.857 | 2.050 | 0.620 | 0.700 | 1.900 | 0.544 | 0.500 | 1.710 | 0.513 | 0.520 | 1.830 | 0.570 | 0.643 | 1.720 | 0.536 | 0.638 |
| Bright Transform | 1.820 | 0.520 | 0.923 | 1.690 | 0.550 | 0.643 | 1.540 | 0.521 | 0.600 | 1.180 | 0.133 | 1.000 | 1.680 | 0.524 | 0.685 | 1.420 | 0.681 | 0.889 | 2.170 | 0.546 | 0.714 | 1.860 | 0.634 | 0.600 | 1.880 | 0.556 | 0.500 | 1.670 | 0.514 | 0.520 | 1.800 | 0.574 | 0.643 | 1.730 | 0.540 | 0.664 |
| Color Quantized | 1.660 | 0.466 | 0.462 | 1.490 | 0.487 | 0.557 | 0.920 | 0.335 | 0.333 | 0.980 | 0.144 | 0.667 | 1.410 | 0.445 | 0.500 | 1.020 | 0.449 | 1.000 | 0.850 | 0.265 | 0.429 | 0.480 | 0.190 | 0.033 | 1.600 | 0.491 | 0.500 | 0.840 | 0.278 | 0.240 | 0.870 | 0.304 | 0.286 | 1.220 | 0.396 | 0.388 |
| Color Shift | 1.840 | 0.528 | 0.692 | 1.730 | 0.551 | 0.643 | 1.530 | 0.522 | 0.733 | 1.310 | 0.146 | 0.333 | 1.700 | 0.526 | 0.674 | 1.590 | 0.682 | 1.000 | 2.020 | 0.543 | 0.857 | 1.960 | 0.629 | 0.667 | 2.070 | 0.569 | 1.000 | 1.630 | 0.513 | 0.520 | 1.820 | 0.576 | 0.671 | 1.740 | 0.541 | 0.671 |
| Contrast | 1.890 | 0.520 | 0.692 | 1.760 | 0.548 | 0.729 | 1.460 | 0.512 | 0.733 | 0.920 | 0.134 | 1.000 | 1.720 | 0.521 | 0.717 | 1.720 | 0.700 | 0.778 | 2.030 | 0.546 | 1.000 | 1.950 | 0.622 | 0.533 | 1.880 | 0.534 | 1.000 | 1.710 | 0.512 | 0.480 | 1.820 | 0.569 | 0.643 | 1.750 | 0.536 | 0.678 |
| Defocus Blur | 1.740 | 0.518 | 0.846 | 1.670 | 0.548 | 0.643 | 1.670 | 0.507 | 0.600 | 0.890 | 0.134 | 1.000 | 1.620 | 0.521 | 0.674 | 1.220 | 0.639 | 0.889 | 1.990 | 0.538 | 0.571 | 1.990 | 0.624 | 0.633 | 1.680 | 0.544 | 0.500 | 1.680 | 0.502 | 0.500 | 1.700 | 0.557 | 0.657 | 1.660 | 0.532 | 0.658 |
| Edge Sawtooth | 1.810 | 0.530 | 0.846 | 1.700 | 0.549 | 0.657 | 1.580 | 0.514 | 0.600 | 0.890 | 0.136 | 1.000 | 1.660 | 0.524 | 0.685 | 1.420 | 0.680 | 0.778 | 2.080 | 0.540 | 1.000 | 2.050 | 0.628 | 0.567 | 1.950 | 0.557 | 0.500 | 1.950 | 0.516 | 0.520 | 1.830 | 0.573 | 0.629 | 1.720 | 0.539 | 0.658 |
| Elastic | 1.910 | 0.530 | 0.692 | 1.740 | 0.552 | 0.714 | 1.500 | 0.511 | 0.733 | 0.900 | 0.136 | 1.000 | 1.700 | 0.525 | 0.717 | 1.420 | 0.682 | 1.000 | 2.140 | 0.555 | 0.857 | 2.080 | 0.635 | 0.667 | 1.890 | 0.546 | 1.000 | 1.680 | 0.520 | 0.560 | 1.840 | 0.578 | 0.686 | 1.750 | 0.541 | 0.704 |
| Flicker | 1.820 | 0.516 | 0.846 | 1.680 | 0.548 | 0.671 | 1.480 | 0.515 | 0.667 | 0.970 | 0.145 | 0.667 | 1.650 | 0.521 | 0.685 | 1.350 | 0.669 | 0.778 | 2.040 | 0.557 | 0.714 | 2.040 | 0.633 | 0.633 | 1.790 | 0.550 | 0.500 | 1.690 | 0.512 | 0.520 | 1.780 | 0.574 | 0.586 | 1.690 | 0.537 | 0.651 |
| Foggy | 1.860 | 0.522 | 0.692 | 1.690 | 0.546 | 0.643 | 1.510 | 0.501 | 0.733 | 1.060 | 0.137 | 1.000 | 1.670 | 0.519 | 0.663 | 1.440 | 0.678 | 1.000 | 2.070 | 0.549 | 0.857 | 1.910 | 0.629 | 0.567 | 1.830 | 0.540 | 0.500 | 1.680 | 0.520 | 0.480 | 1.780 | 0.574 | 0.586 | 1.710 | 0.536 | 0.632 |
| Frame Drop | 1.340 | 0.457 | 0.692 | 1.190 | 0.497 | 0.543 | 0.970 | 0.460 | 0.667 | 0.810 | 0.143 | 0.667 | 1.170 | 0.471 | 0.565 | 1.920 | 0.571 | 0.667 | 1.470 | 0.486 | 0.571 | 1.640 | 0.598 | 0.533 | 1.540 | 0.543 | 0.500 | 1.430 | 0.472 | 0.760 | 1.400 | 0.522 | 0.643 | 1.250 | 0.486 | 0.599 |
| Frame Repeat | 1.300 | 0.457 | 0.538 | 1.090 | 0.469 | 0.386 | 1.610 | 0.455 | 0.467 | 1.610 | 0.145 | 1.000 | 1.150 | 0.454 | 0.435 | 1.150 | 0.592 | 0.556 | 1.630 | 0.505 | 0.429 | 1.540 | 0.569 | 0.567 | 1.440 | 0.514 | 0.500 | 1.370 | 0.487 | 0.560 | 1.390 | 0.525 | 0.557 | 1.220 | 0.475 | 0.487 |
| Frame Replace | 1.810 | 0.527 | 0.692 | 1.730 | 0.546 | 0.671 | 1.460 | 0.506 | 0.733 | 1.160 | 0.135 | 1.000 | 1.690 | 0.521 | 0.685 | 1.470 | 0.667 | 0.778 | 2.100 | 0.561 | 0.857 | 1.950 | 0.618 | 0.633 | 1.930 | 0.544 | 1.000 | 1.690 | 0.518 | 0.400 | 1.820 | 0.573 | 0.629 | 1.740 | 0.536 | 0.664 |
| Frost | 1.790 | 0.522 | 0.769 | 1.680 | 0.554 | 0.671 | 1.400 | 0.496 | 0.733 | 0.920 | 0.134 | 0.667 | 1.620 | 0.523 | 0.685 | 1.460 | 0.656 | 0.667 | 1.890 | 0.537 | 0.857 | 1.890 | 0.621 | 0.633 | 1.670 | 0.536 | 0.500 | 1.600 | 0.511 | 0.360 | 1.720 | 0.563 | 0.571 | 1.650 | 0.534 | 0.638 |
| Gaussian | 1.760 | 0.514 | 0.769 | 1.610 | 0.540 | 0.657 | 1.400 | 0.501 | 0.733 | 0.980 | 0.140 | 1.000 | 1.590 | 0.513 | 0.685 | 1.590 | 0.659 | 0.889 | 2.020 | 0.532 | 0.714 | 1.890 | 0.622 | 0.633 | 1.730 | 0.517 | 1.000 | 1.690 | 0.509 | 0.440 | 1.760 | 0.559 | 0.643 | 1.650 | 0.528 | 0.658 |
| Gaussian Blur | 1.570 | 0.492 | 0.769 | 1.450 | 0.517 | 0.629 | 1.250 | 0.492 | 0.600 | 0.980 | 0.136 | 1.000 | 1.430 | 0.495 | 0.641 | 1.120 | 0.673 | 1.000 | 1.880 | 0.513 | 0.667 | 1.930 | 0.603 | 0.733 | 1.620 | 0.518 | 0.500 | 1.520 | 0.492 | 0.640 | 1.620 | 0.544 | 0.686 | 1.490 | 0.510 | 0.664 |
| Glass Blur | 1.740 | 0.509 | 0.769 | 1.620 | 0.537 | 0.568 | 1.310 | 0.497 | 0.467 | 1.060 | 0.138 | 0.667 | 1.580 | 0.512 | 0.598 | 1.210 | 0.673 | 0.444 | 1.970 | 0.540 | 0.714 | 1.970 | 0.611 | 0.667 | 1.800 | 0.556 | 1.000 | 1.690 | 0.517 | 0.600 | 1.740 | 0.568 | 0.643 | 1.630 | 0.528 | 0.625 |
| H265 Artifacts | 1.530 | 0.482 | 0.769 | 1.380 | 0.515 | 0.543 | 1.190 | 0.480 | 0.533 | 0.740 | 0.127 | 0.667 | 1.360 | 0.490 | 0.565 | 1.220 | 0.640 | 0.667 | 1.810 | 0.520 | 1.000 | 1.760 | 0.602 | 0.700 | 1.530 | 0.510 | 0.500 | 1.520 | 0.495 | 0.640 | 1.590 | 0.547 | 0.686 | 1.430 | 0.507 | 0.605 |
| Impulse | 1.450 | 0.477 | 0.692 | 1.410 | 0.515 | 0.643 | 1.210 | 0.473 | 0.667 | 0.840 | 0.169 | 1.000 | 1.370 | 0.489 | 0.641 | 1.060 | 0.619 | 0.778 | 1.750 | 0.516 | 0.778 | 1.830 | 0.589 | 0.600 | 1.490 | 0.512 | 0.000 | 1.560 | 0.502 | 0.480 | 1.570 | 0.541 | 0.557 | 1.430 | 0.505 | 0.612 |
| Jpeg Artifact | 1.790 | 0.518 | 0.769 | 1.650 | 0.535 | 0.686 | 1.360 | 0.497 | 0.733 | 0.980 | 0.132 | 1.000 | 1.610 | 0.512 | 0.707 | 1.430 | 0.662 | 0.778 | 2.020 | 0.540 | 0.778 | 2.020 | 0.623 | 0.700 | 1.750 | 0.531 | 1.000 | 1.690 | 0.512 | 0.640 | 1.800 | 0.562 | 0.714 | 1.680 | 0.528 | 0.724 |
| Motion Blur | 1.480 | 0.477 | 0.846 | 1.350 | 0.504 | 0.557 | 1.320 | 0.492 | 0.600 | 1.030 | 0.147 | 0.333 | 1.360 | 0.484 | 0.587 | 1.080 | 0.654 | 0.667 | 1.670 | 0.514 | 0.667 | 1.750 | 0.606 | 0.567 | 1.590 | 0.522 | 0.500 | 1.460 | 0.488 | 0.520 | 1.500 | 0.544 | 0.571 | 1.410 | 0.503 | 0.579 |
| Overexposure | 1.770 | 0.518 | 0.769 | 1.700 | 0.552 | 0.700 | 1.410 | 0.508 | 0.667 | 0.890 | 0.125 | 0.667 | 1.640 | 0.523 | 0.696 | 1.420 | 0.671 | 0.778 | 1.990 | 0.545 | 0.778 | 1.960 | 0.620 | 0.633 | 1.790 | 0.530 | 0.500 | 1.550 | 0.511 | 0.520 | 1.730 | 0.567 | 0.657 | 1.680 | 0.537 | 0.684 |
| Poisson | 1.720 | 0.518 | 0.538 | 1.580 | 0.546 | 0.586 | 1.360 | 0.492 | 0.667 | 1.070 | 0.143 | 1.000 | 1.550 | 0.517 | 0.598 | 1.260 | 0.673 | 0.778 | 1.890 | 0.544 | 0.778 | 1.810 | 0.625 | 0.629 | 1.810 | 0.562 | 1.000 | 1.560 | 0.499 | 0.560 | 1.690 | 0.562 | 0.629 | 1.600 | 0.531 | 0.599 |
| Rainy | 1.820 | 0.526 | 0.769 | 1.720 | 0.554 | 0.657 | 1.430 | 0.510 | 0.667 | 0.810 | 0.129 | 1.000 | 1.670 | 0.527 | 0.685 | 1.470 | 0.673 | 0.778 | 2.010 | 0.530 | 0.857 | 1.880 | 0.622 | 0.533 | 1.860 | 0.564 | 0.500 | 1.690 | 0.520 | 0.320 | 1.780 | 0.571 | 0.543 | 1.710 | 0.539 | 0.618 |
| Random Block | 1.770 | 0.514 | 0.769 | 1.650 | 0.538 | 0.743 | 1.420 | 0.502 | 0.800 | 1.050 | 0.151 | 1.000 | 1.640 | 0.513 | 0.761 | 1.320 | 0.628 | 1.000 | 1.830 | 0.554 | 0.857 | 2.010 | 0.603 | 0.733 | 1.830 | 0.538 | 0.500 | 1.790 | 0.513 | 0.560 | 1.790 | 0.568 | 0.700 | 1.690 | 0.529 | 0.730 |
| Reflect | 1.830 | 0.522 | 0.769 | 1.710 | 0.545 | 0.614 | 1.450 | 0.517 | 0.600 | 1.060 | 0.140 | 1.000 | 1.670 | 0.520 | 0.620 | 1.420 | 0.676 | 0.667 | 2.020 | 0.536 | 0.714 | 1.970 | 0.614 | 0.667 | 1.850 | 0.564 | 0.500 | 1.690 | 0.508 | 0.480 | 1.800 | 0.565 | 0.614 | 1.720 | 0.533 | 0.618 |
| Resolution Degrade | 1.740 | 0.513 | 0.769 | 1.570 | 0.533 | 0.629 | 1.270 | 0.453 | 0.600 | 0.820 | 0.138 | 1.000 | 1.540 | 0.503 | 0.652 | 1.200 | 0.628 | 0.667 | 1.430 | 0.423 | 0.857 | 1.410 | 0.426 | 0.200 | 1.810 | 0.536 | 0.500 | 1.420 | 0.478 | 0.480 | 1.440 | 0.492 | 0.429 | 1.500 | 0.496 | 0.546 |
| Rolling Shutter | 1.710 | 0.510 | 0.615 | 1.550 | 0.533 | 0.714 | 1.270 | 0.499 | 0.667 | 1.020 | 0.140 | 1.000 | 1.520 | 0.509 | 0.576 | 1.120 | 0.674 | 0.556 | 1.920 | 0.529 | 0.857 | 2.020 | 0.628 | 0.733 | 1.720 | 0.555 | 0.500 | 1.690 | 0.513 | 0.480 | 1.690 | 0.566 | 0.629 | 1.580 | 0.526 | 0.612 |
| Shadow | 1.820 | 0.514 | 0.769 | 1.750 | 0.554 | 0.714 | 1.520 | 0.511 | 0.733 | 1.020 | 0.138 | 1.000 | 1.700 | 0.525 | 0.728 | 1.490 | 0.697 | 0.889 | 1.970 | 0.542 | 0.714 | 1.910 | 0.612 | 0.600 | 1.950 | 0.550 | 1.000 | 1.710 | 0.514 | 0.520 | 1.800 | 0.573 | 0.600 | 1.730 | 0.539 | 0.678 |
| Snow | 1.880 | 0.517 | 0.769 | 1.710 | 0.547 | 0.643 | 1.490 | 0.518 | 1.100 | 1.000 | 0.144 | 0.667 | 1.690 | 0.522 | 0.685 | 1.380 | 0.664 | 0.778 | 2.140 | 0.528 | 0.857 | 2.000 | 0.622 | 0.633 | 1.910 | 0.565 | 0.500 | 1.730 | 0.518 | 0.520 | 1.830 | 0.568 | 0.643 | 1.740 | 0.535 | 0.678 |
| Speckle | 1.850 | 0.527 | 0.615 | 1.610 | 0.546 | 0.614 | 1.380 | 0.497 | 0.600 | 0.920 | 0.146 | 1.000 | 1.600 | 0.519 | 0.609 | 1.270 | 0.663 | 0.667 | 2.010 | 0.534 | 1.000 | 1.900 | 0.610 | 0.633 | 1.900 | 0.518 | 0.500 | 1.590 | 0.502 | 0.520 | 1.750 | 0.564 | 0.643 | 1.640 | 0.533 | 0.612 |
| Stretch Squish | 1.570 | 0.497 | 0.846 | 1.470 | 0.513 | 0.571 | 1.190 | 0.496 | 0.790 | 0.790 | 0.145 | 1.000 | 1.430 | 0.495 | 0.630 | 1.010 | 0.644 | 0.333 | 1.910 | 0.530 | 0.857 | 1.800 | 0.613 | 0.600 | 1.700 | 0.527 | 1.000 | 1.510 | 0.508 | 0.560 | 1.600 | 0.555 | 0.586 | 1.490 | 0.514 | 0.632 |
| Target Block | 1.780 | 0.511 | 0.769 | 1.590 | 0.542 | 0.614 | 1.330 | 0.498 | 0.667 | 1.160 | 0.153 | 0.667 | 1.580 | 0.515 | 0.641 | 1.400 | 0.673 | 0.667 | 2.060 | 0.555 | 0.857 | 1.750 | 0.610 | 0.567 | 1.770 | 0.535 | 1.000 | 1.630 | 0.509 | 0.440 | 1.740 | 0.566 | 0.600 | 1.630 | 0.530 | 0.625 |
| Temporal Jitter | 1.740 | 0.506 | 0.692 | 1.540 | 0.527 | 0.543 | 1.310 | 0.489 | 0.600 | 0.950 | 0.136 | 1.000 | 1.520 | 0.505 | 0.565 | 1.250 | 0.648 | 0.889 | 1.980 | 0.532 | 0.857 | 1.890 | 0.618 | 0.600 | 1.740 | 0.564 | 1.000 | 1.680 | 0.510 | 0.640 | 1.720 | 0.560 | 0.657 | 1.590 | 0.522 | 0.605 |
| Underexposure | 1.880 | 0.523 | 0.769 | 1.760 | 0.553 | 0.657 | 1.560 | 0.505 | 0.800 | 0.980 | 0.153 | 1.000 | 1.720 | 0.524 | 0.696 | 1.420 | 0.676 | 0.667 | 2.080 | 0.555 | 1.000 | 1.990 | 0.621 | 0.633 | 1.890 | 0.558 | 1.000 | 1.750 | 0.516 | 0.520 | 1.840 | 0.576 | 0.657 | 1.760 | 0.539 | 0.684 |
| Zoom Blur | 1.730 | 0.520 | 0.692 | 1.620 | 0.536 | 0.600 | 1.370 | 0.504 | 0.667 | 1.020 | 0.136 | 0.667 | 1.580 | 0.513 | 0.609 | 1.270 | 0.687 | 0.556 | 1.820 | 0.519 | 0.857 | 1.960 | 0.610 | 0.633 | 1.850 | 0.558 | 0.000 | 1.710 | 0.514 | 0.480 | 1.720 | 0.564 | 0.571 | 1.620 | 0.528 | 0.599 |

Table 15: LLaMA-VID performance across different question types

| Noise Type | CP | | | FP-S | | | FP-C | | | HL | | | Mean | | | LR | | | AR | | | RR | | | CSR | | | TR | | | Mean2 | | | Overall | | |
|---|---|---|---|---|---|---|---|---|---|---|---|---|---|---|---|---|---|---|---|---|---|---|---|---|---|---|---|---|---|---|---|---|---|---|---|---|
| | GPT | SBERT | Acc | GPT | SBERT | Acc | GPT | SBERT | Acc | GPT | SBERT | Acc | GPT | SBERT | Acc | GPT | SBERT | Acc | GPT | SBERT | Acc | GPT | SBERT | Acc | GPT | SBERT | Acc | GPT | SBERT | Acc | GPT | SBERT | Acc | GPT | SBERT | Acc |
| Clean | 1.020 | 0.411 | 0.385 | 0.890 | 0.417 | 0.514 | 0.800 | 0.460 | 0.133 | 0.600 | 0.168 | 0.000 | 0.910 | 0.414 | 0.467 | 0.530 | 0.496 | 0.222 | 1.120 | 0.473 | 0.429 | 0.900 | 0.562 | 0.367 | 0.730 | 0.425 | 0.000 | 0.900 | 0.455 | 0.480 | 0.890 | 0.481 | 0.400 | 0.910 | 0.435 | 0.447 |
| Bit Error | 1.010 | 0.409 | 0.385 | 0.890 | 0.417 | 0.514 | 0.710 | 0.460 | 0.133 | 0.790 | 0.163 | 0.000 | 0.910 | 0.413 | 0.467 | 0.580 | 0.508 | 0.222 | 1.040 | 0.471 | 0.429 | 0.890 | 0.559 | 0.367 | 0.720 | 0.417 | 0.000 | 0.870 | 0.452 | 0.400 | 0.860 | 0.480 | 0.400 | 0.900 | 0.435 | 0.434 |
| Bright Transform | 1.040 | 0.413 | 0.385 | 0.870 | 0.419 | 0.500 | 0.800 | 0.458 | 0.133 | 0.610 | 0.167 | 0.000 | 0.900 | 0.416 | 0.457 | 0.580 | 0.507 | 0.222 | 1.040 | 0.467 | 0.429 | 0.920 | 0.557 | 0.367 | 0.770 | 0.426 | 0.000 | 0.900 | 0.456 | 0.520 | 0.880 | 0.480 | 0.414 | 0.910 | 0.436 | 0.447 |
| Color Quantized | 1.030 | 0.407 | 0.385 | 0.850 | 0.416 | 0.514 | 0.770 | 0.460 | 0.133 | 0.610 | 0.166 | 0.000 | 0.890 | 0.413 | 0.467 | 0.570 | 0.508 | 0.222 | 1.090 | 0.478 | 0.429 | 0.790 | 0.556 | 0.367 | 0.750 | 0.423 | 0.000 | 0.890 | 0.454 | 0.440 | 0.860 | 0.481 | 0.400 | 0.890 | 0.435 | 0.441 |
| Color Shift | 0.990 | 0.411 | 0.385 | 0.840 | 0.414 | 0.500 | 0.740 | 0.460 | 0.133 | 0.630 | 0.165 | 0.000 | 0.870 | 0.412 | 0.457 | 0.570 | 0.501 | 0.222 | 1.030 | 0.473 | 0.429 | 0.850 | 0.553 | 0.367 | 0.740 | 0.419 | 0.000 | 0.850 | 0.455 | 0.400 | 0.840 | 0.479 | 0.386 | 0.870 | 0.434 | 0.428 |
| Contrast | 0.980 | 0.406 | 0.385 | 0.870 | 0.417 | 0.514 | 0.790 | 0.460 | 0.133 | 0.650 | 0.170 | 0.000 | 0.890 | 0.414 | 0.467 | 0.530 | 0.499 | 0.222 | 1.070 | 0.472 | 0.429 | 1.020 | 0.562 | 0.367 | 0.680 | 0.412 | 0.000 | 0.890 | 0.454 | 0.440 | 0.890 | 0.480 | 0.400 | 0.900 | 0.435 | 0.441 |
| Defocus Blur | 1.010 | 0.409 | 0.385 | 0.830 | 0.413 | 0.514 | 0.780 | 0.458 | 0.133 | 0.660 | 0.164 | 0.000 | 0.880 | 0.411 | 0.467 | 0.660 | 0.510 | 0.222 | 1.090 | 0.475 | 0.286 | 0.900 | 0.557 | 0.367 | 0.810 | 0.416 | 0.000 | 0.810 | 0.452 | 0.440 | 0.880 | 0.479 | 0.371 | 0.880 | 0.433 | 0.434 |
| Edge Sawtooth | 1.020 | 0.406 | 0.385 | 0.860 | 0.418 | 0.500 | 0.780 | 0.452 | 0.133 | 0.580 | 0.164 | 0.000 | 0.890 | 0.413 | 0.457 | 0.600 | 0.503 | 0.222 | 1.030 | 0.471 | 0.429 | 0.860 | 0.553 | 0.367 | 0.750 | 0.427 | 0.000 | 0.880 | 0.456 | 0.440 | 0.860 | 0.480 | 0.386 | 0.890 | 0.434 | 0.434 |
| Elastic | 1.020 | 0.413 | 0.385 | 0.870 | 0.416 | 0.500 | 0.800 | 0.460 | 0.133 | 0.550 | 0.170 | 0.000 | 0.890 | 0.414 | 0.457 | 0.550 | 0.499 | 0.222 | 1.060 | 0.473 | 0.429 | 0.850 | 0.552 | 0.367 | 0.690 | 0.418 | 0.000 | 0.900 | 0.461 | 0.440 | 0.860 | 0.481 | 0.386 | 0.890 | 0.435 | 0.434 |
| Flicker | 1.100 | 0.415 | 0.385 | 0.860 | 0.415 | 0.514 | 0.810 | 0.460 | 0.133 | 0.630 | 0.164 | 0.000 | 0.890 | 0.414 | 0.457 | 0.540 | 0.502 | 0.222 | 1.030 | 0.469 | 0.429 | 0.890 | 0.559 | 0.367 | 0.690 | 0.410 | 0.000 | 0.890 | 0.459 | 0.440 | 0.860 | 0.480 | 0.386 | 0.890 | 0.435 | 0.441 |
| Foggy | 0.940 | 0.402 | 0.385 | 0.840 | 0.416 | 0.500 | 0.750 | 0.457 | 0.133 | 0.630 | 0.169 | 0.000 | 0.860 | 0.412 | 0.457 | 0.580 | 0.501 | 0.222 | 1.060 | 0.470 | 0.429 | 0.800 | 0.557 | 0.367 | 0.830 | 0.426 | 0.000 | 0.750 | 0.454 | 0.440 | 0.820 | 0.480 | 0.386 | 0.860 | 0.433 | 0.434 |
| Frame Drop | 0.810 | 0.382 | 0.231 | 0.730 | 0.402 | 0.500 | 0.590 | 0.442 | 0.133 | 0.500 | 0.162 | 0.000 | 0.730 | 0.395 | 0.435 | 0.570 | 0.503 | 0.222 | 1.010 | 0.463 | 0.286 | 0.870 | 0.557 | 0.367 | 0.590 | 0.399 | 0.000 | 0.840 | 0.449 | 0.440 | 0.820 | 0.474 | 0.371 | 0.760 | 0.420 | 0.414 |
| Frame Repeat | 0.810 | 0.370 | 0.385 | 0.680 | 0.389 | 0.500 | 0.510 | 0.435 | 0.133 | 0.520 | 0.154 | 0.000 | 0.690 | 0.384 | 0.457 | 0.580 | 0.492 | 0.222 | 0.840 | 0.456 | 0.429 | 0.860 | 0.545 | 0.367 | 0.520 | 0.383 | 0.000 | 0.740 | 0.448 | 0.440 | 0.750 | 0.466 | 0.386 | 0.720 | 0.410 | 0.434 |
| Frame Replace | 1.060 | 0.412 | 0.385 | 0.860 | 0.417 | 0.514 | 0.800 | 0.461 | 0.133 | 0.660 | 0.166 | 0.000 | 0.900 | 0.414 | 0.467 | 0.470 | 0.500 | 0.222 | 1.090 | 0.473 | 0.429 | 0.970 | 0.564 | 0.367 | 0.730 | 0.415 | 0.000 | 0.880 | 0.455 | 0.440 | 0.880 | 0.480 | 0.386 | 0.900 | 0.435 | 0.441 |
| Frost | 0.870 | 0.397 | 0.308 | 0.800 | 0.412 | 0.500 | 0.680 | 0.454 | 0.133 | 0.680 | 0.168 | 0.000 | 0.810 | 0.407 | 0.446 | 0.500 | 0.501 | 0.222 | 0.990 | 0.472 | 0.429 | 0.820 | 0.557 | 0.367 | 0.670 | 0.394 | 0.000 | 0.820 | 0.454 | 0.400 | 0.800 | 0.477 | 0.386 | 0.820 | 0.430 | 0.428 |
| Gaussian | 0.980 | 0.410 | 0.385 | 0.840 | 0.412 | 0.500 | 0.750 | 0.455 | 0.133 | 0.650 | 0.166 | 0.000 | 0.860 | 0.410 | 0.457 | 0.500 | 0.503 | 0.222 | 1.060 | 0.471 | 0.286 | 0.930 | 0.558 | 0.367 | 0.690 | 0.421 | 0.000 | 0.860 | 0.458 | 0.400 | 0.860 | 0.481 | 0.386 | 0.870 | 0.433 | 0.441 |
| Gaussian Blur | 0.920 | 0.393 | 0.385 | 0.790 | 0.406 | 0.500 | 0.700 | 0.455 | 0.133 | 0.710 | 0.164 | 0.000 | 0.810 | 0.403 | 0.457 | 0.630 | 0.490 | 0.222 | 0.960 | 0.459 | 0.429 | 0.820 | 0.555 | 0.367 | 0.670 | 0.407 | 0.000 | 0.900 | 0.453 | 0.480 | 0.840 | 0.472 | 0.400 | 0.830 | 0.425 | 0.428 |
| Glass Blur | 1.010 | 0.412 | 0.385 | 0.840 | 0.413 | 0.500 | 0.700 | 0.453 | 0.133 | 0.710 | 0.168 | 0.000 | 0.870 | 0.411 | 0.457 | 0.530 | 0.500 | 0.222 | 1.050 | 0.470 | 0.429 | 0.920 | 0.558 | 0.367 | 0.790 | 0.425 | 0.000 | 0.820 | 0.455 | 0.400 | 0.880 | 0.480 | 0.371 | 0.880 | 0.433 | 0.441 |
| H265 Artifacts | 0.930 | 0.390 | 0.308 | 0.780 | 0.404 | 0.500 | 0.620 | 0.450 | 0.133 | 0.740 | 0.161 | 0.000 | 0.800 | 0.399 | 0.446 | 0.600 | 0.504 | 0.222 | 0.790 | 0.452 | 0.429 | 0.830 | 0.548 | 0.367 | 0.850 | 0.418 | 0.000 | 0.810 | 0.451 | 0.520 | 0.790 | 0.473 | 0.414 | 0.810 | 0.422 | 0.441 |
| Impulse | 0.890 | 0.398 | 0.231 | 0.770 | 0.405 | 0.500 | 0.650 | 0.452 | 0.133 | 0.660 | 0.165 | 0.000 | 0.790 | 0.403 | 0.435 | 0.630 | 0.502 | 0.222 | 0.930 | 0.454 | 0.429 | 0.860 | 0.552 | 0.367 | 0.780 | 0.413 | 0.000 | 0.840 | 0.451 | 0.400 | 0.840 | 0.473 | 0.371 | 0.810 | 0.425 | 0.414 |
| Jpeg Artifact | 0.970 | 0.411 | 0.308 | 0.810 | 0.413 | 0.514 | 0.760 | 0.457 | 0.133 | 0.680 | 0.165 | 0.000 | 0.850 | 0.411 | 0.446 | 0.510 | 0.504 | 0.222 | 0.960 | 0.461 | 0.429 | 0.960 | 0.556 | 0.367 | 0.620 | 0.413 | 0.000 | 0.890 | 0.454 | 0.400 | 0.830 | 0.476 | 0.371 | 0.860 | 0.432 | 0.428 |
| Motion Blur | 0.830 | 0.387 | 0.308 | 0.750 | 0.400 | 0.500 | 0.620 | 0.444 | 0.133 | 0.650 | 0.170 | 0.000 | 0.760 | 0.397 | 0.446 | 0.570 | 0.494 | 0.222 | 0.890 | 0.448 | 0.429 | 0.730 | 0.547 | 0.367 | 0.690 | 0.395 | 0.000 | 0.770 | 0.449 | 0.520 | 0.770 | 0.467 | 0.414 | 0.770 | 0.419 | 0.441 |
| Overexposure | 0.990 | 0.408 | 0.385 | 0.850 | 0.416 | 0.500 | 0.800 | 0.457 | 0.133 | 0.660 | 0.164 | 0.000 | 0.890 | 0.413 | 0.457 | 0.550 | 0.499 | 0.222 | 1.060 | 0.472 | 0.429 | 0.800 | 0.558 | 0.367 | 0.800 | 0.431 | 0.000 | 0.840 | 0.460 | 0.440 | 0.840 | 0.483 | 0.386 | 0.880 | 0.435 | 0.434 |
| Poisson | 0.990 | 0.409 | 0.308 | 0.850 | 0.412 | 0.500 | 0.760 | 0.459 | 0.133 | 0.680 | 0.167 | 0.000 | 0.860 | 0.410 | 0.446 | 0.560 | 0.502 | 0.222 | 1.030 | 0.463 | 0.429 | 0.860 | 0.561 | 0.367 | 0.720 | 0.419 | 0.000 | 0.890 | 0.454 | 0.400 | 0.850 | 0.478 | 0.371 | 0.860 | 0.432 | 0.421 |
| Rainy | 0.980 | 0.407 | 0.385 | 0.840 | 0.414 | 0.500 | 0.680 | 0.456 | 0.133 | 0.650 | 0.164 | 0.000 | 0.860 | 0.411 | 0.457 | 0.550 | 0.498 | 0.222 | 1.100 | 0.474 | 0.429 | 0.890 | 0.558 | 0.367 | 0.800 | 0.429 | 0.000 | 0.870 | 0.455 | 0.480 | 0.870 | 0.480 | 0.400 | 0.870 | 0.433 | 0.441 |
| Random Block | 0.970 | 0.407 | 0.385 | 0.810 | 0.411 | 0.500 | 0.760 | 0.465 | 0.133 | 0.550 | 0.167 | 0.000 | 0.840 | 0.410 | 0.457 | 0.590 | 0.502 | 0.222 | 1.020 | 0.472 | 0.429 | 0.910 | 0.555 | 0.367 | 0.700 | 0.408 | 0.000 | 0.860 | 0.455 | 0.400 | 0.850 | 0.478 | 0.371 | 0.860 | 0.432 | 0.428 |
| Reflect | 1.020 | 0.414 | 0.385 | 0.840 | 0.415 | 0.514 | 0.730 | 0.457 | 0.133 | 0.610 | 0.169 | 0.000 | 0.880 | 0.413 | 0.467 | 0.570 | 0.504 | 0.222 | 1.060 | 0.474 | 0.429 | 0.900 | 0.560 | 0.367 | 0.830 | 0.425 | 0.000 | 0.880 | 0.455 | 0.440 | 0.870 | 0.482 | 0.386 | 0.880 | 0.435 | 0.441 |
| Resolution Degrade | 0.960 | 0.406 | 0.385 | 0.810 | 0.411 | 0.500 | 0.740 | 0.459 | 0.133 | 0.580 | 0.167 | 0.000 | 0.850 | 0.409 | 0.457 | 0.530 | 0.496 | 0.222 | 1.020 | 0.475 | 0.429 | 0.910 | 0.554 | 0.367 | 0.680 | 0.421 | 0.000 | 0.870 | 0.458 | 0.440 | 0.850 | 0.480 | 0.386 | 0.860 | 0.432 | 0.434 |
| Rolling Shutter | 0.910 | 0.397 | 0.385 | 0.860 | 0.417 | 0.500 | 0.730 | 0.452 | 0.133 | 0.610 | 0.168 | 0.000 | 0.830 | 0.406 | 0.457 | 0.710 | 0.508 | 0.222 | 0.940 | 0.460 | 0.429 | 0.800 | 0.552 | 0.367 | 0.730 | 0.418 | 0.000 | 0.920 | 0.458 | 0.400 | 0.860 | 0.478 | 0.371 | 0.850 | 0.429 | 0.428 |
| Shadow | 0.960 | 0.407 | 0.385 | 0.870 | 0.417 | 0.500 | 0.760 | 0.462 | 0.133 | 0.560 | 0.162 | 0.000 | 0.870 | 0.413 | 0.457 | 0.580 | 0.503 | 0.222 | 1.070 | 0.474 | 0.429 | 0.880 | 0.558 | 0.367 | 0.690 | 0.419 | 0.000 | 0.870 | 0.457 | 0.480 | 0.870 | 0.482 | 0.400 | 0.860 | 0.436 | 0.441 |
| Snow | 1.070 | 0.415 | 0.385 | 0.830 | 0.415 | 0.500 | 0.780 | 0.463 | 0.133 | 0.560 | 0.165 | 0.000 | 0.900 | 0.415 | 0.457 | 0.500 | 0.503 | 0.222 | 1.070 | 0.470 | 0.429 | 0.980 | 0.562 | 0.367 | 0.890 | 0.421 | 0.000 | 0.930 | 0.458 | 0.440 | 0.890 | 0.483 | 0.386 | 0.900 | 0.436 | 0.434 |
| Speckle | 0.950 | 0.412 | 0.308 | 0.760 | 0.407 | 0.500 | 0.730 | 0.455 | 0.133 | 0.530 | 0.163 | 0.000 | 0.840 | 0.412 | 0.446 | 0.560 | 0.508 | 0.222 | 1.020 | 0.469 | 0.429 | 0.890 | 0.558 | 0.367 | 0.670 | 0.443 | 0.000 | 0.870 | 0.454 | 0.400 | 0.870 | 0.483 | 0.371 | 0.850 | 0.434 | 0.421 |
| Stretch Squish | 0.910 | 0.400 | 0.385 | 0.810 | 0.409 | 0.500 | 0.680 | 0.457 | 0.133 | 0.770 | 0.170 | 0.000 | 0.810 | 0.405 | 0.457 | 0.510 | 0.493 | 0.222 | 1.020 | 0.470 | 0.429 | 0.820 | 0.556 | 0.367 | 0.620 | 0.408 | 0.000 | 0.810 | 0.453 | 0.440 | 0.800 | 0.476 | 0.386 | 0.820 | 0.428 | 0.434 |
| Target Block | 0.950 | 0.407 | 0.308 | 0.810 | 0.414 | 0.500 | 0.710 | 0.461 | 0.133 | 0.660 | 0.158 | 0.000 | 0.840 | 0.408 | 0.446 | 0.620 | 0.509 | 0.222 | 0.960 | 0.484 | 0.429 | 0.860 | 0.561 | 0.367 | 0.670 | 0.417 | 0.000 | 0.840 | 0.453 | 0.480 | 0.850 | 0.483 | 0.386 | 0.830 | 0.432 | 0.428 |
| Temporal Jitter | 0.970 | 0.397 | 0.308 | 0.880 | 0.416 | 0.514 | 0.800 | 0.452 | 0.133 | 0.600 | 0.165 | 0.000 | 0.810 | 0.407 | 0.457 | 0.640 | 0.505 | 0.222 | 1.020 | 0.465 | 0.429 | 0.830 | 0.556 | 0.367 | 0.740 | 0.415 | 0.000 | 0.890 | 0.455 | 0.480 | 0.890 | 0.479 | 0.400 | 0.860 | 0.430 | 0.434 |
| Underexposure | 1.060 | 0.414 | 0.385 | 0.880 | 0.416 | 0.514 | 0.800 | 0.456 | 0.133 | 0.630 | 0.160 | 0.000 | 0.910 | 0.414 | 0.467 | 0.570 | 0.503 | 0.222 | 1.070 | 0.475 | 0.429 | 0.890 | 0.557 | 0.367 | 0.860 | 0.421 | 0.000 | 0.860 | 0.455 | 0.400 | 0.870 | 0.480 | 0.371 | 0.910 | 0.435 | 0.434 |
| Zoom Blur | 0.920 | 0.396 | 0.385 | 0.790 | 0.408 | 0.500 | 0.650 | 0.453 | 0.133 | 0.770 | 0.168 | 0.000 | 0.810 | 0.404 | 0.457 | 0.600 | 0.506 | 0.222 | 0.910 | 0.456 | 0.429 | 0.730 | 0.552 | 0.367 | 0.680 | 0.419 | 0.000 | 0.790 | 0.458 | 0.440 | 0.790 | 0.477 | 0.386 | 0.810 | 0.427 | 0.434 |

Table 16: PLLaVA 13B performance across different question types

| Noise Type | CP | | | FP-S | | | FP-C | | | HL | | | Mean | | | LR | | | AR | | | RR | | | CSR | | | TR | | | Mean2 | | Overall | | |
|---|---|---|---|---|---|---|---|---|---|---|---|---|---|---|---|---|---|---|---|---|---|---|---|---|---|---|---|---|---|---|---|---|---|---|---|
| | GPT | SBERT | Acc | GPT | SBERT | Acc | GPT | SBERT | Acc | GPT | SBERT | Acc | GPT | SBERT | Acc | GPT | SBERT | Acc | GPT | SBERT | Acc | GPT | SBERT | Acc | GPT | SBERT | Acc | GPT | SBERT | Acc | SBERT | Acc | GPT | SBERT | Acc |
| Clean | 1.140 | 0.421 | 0.308 | 0.940 | 0.412 | 0.214 | 0.740 | 0.451 | 0.000 | 0.940 | 0.145 | 0.000 | 0.550 | 0.410 | 0.207 | 0.550 | 0.480 | 0.000 | 1.220 | 0.491 | 0.429 | 1.020 | 0.547 | 0.167 | 0.670 | 0.425 | 0.000 | 0.870 | 0.419 | 0.000 | 0.467 | 0.114 | 0.950 | 0.429 | 0.178 |
| Bit Error | 1.060 | 0.408 | 0.154 | 0.810 | 0.398 | 0.214 | 0.700 | 0.441 | 0.000 | 0.890 | 0.145 | 0.000 | 0.570 | 0.397 | 0.185 | 0.570 | 0.478 | 0.000 | 1.160 | 0.480 | 0.286 | 1.020 | 0.537 | 0.100 | 0.720 | 0.394 | 0.000 | 0.620 | 0.397 | 0.000 | 0.453 | 0.071 | 0.860 | 0.416 | 0.145 |
| Bright Transform | 1.040 | 0.406 | 0.231 | 0.850 | 0.405 | 0.200 | 0.660 | 0.451 | 0.000 | 0.840 | 0.139 | 0.000 | 0.580 | 0.403 | 0.185 | 0.580 | 0.480 | 0.000 | 1.120 | 0.477 | 0.286 | 1.110 | 0.538 | 0.133 | 0.860 | 0.424 | 0.000 | 0.590 | 0.407 | 0.000 | 0.458 | 0.100 | 0.870 | 0.421 | 0.158 |
| Color Quantized | 1.080 | 0.408 | 0.231 | 0.870 | 0.405 | 0.200 | 0.690 | 0.445 | 0.000 | 0.770 | 0.142 | 0.000 | 0.580 | 0.402 | 0.185 | 0.580 | 0.491 | 0.000 | 1.160 | 0.489 | 0.429 | 1.030 | 0.541 | 0.133 | 0.830 | 0.424 | 0.000 | 0.620 | 0.406 | 0.000 | 0.463 | 0.100 | 0.890 | 0.422 | 0.158 |
| Color Shift | 1.070 | 0.413 | 0.308 | 0.880 | 0.406 | 0.186 | 0.780 | 0.447 | 0.000 | 0.810 | 0.143 | 0.000 | 0.650 | 0.404 | 0.185 | 0.650 | 0.484 | 0.000 | 1.120 | 0.479 | 0.429 | 0.980 | 0.542 | 0.133 | 0.850 | 0.421 | 0.000 | 0.660 | 0.415 | 0.000 | 0.462 | 0.100 | 0.910 | 0.423 | 0.158 |
| Contrast | 1.130 | 0.417 | 0.308 | 0.910 | 0.410 | 0.214 | 0.700 | 0.454 | 0.000 | 0.820 | 0.139 | 0.000 | 0.580 | 0.408 | 0.207 | 0.580 | 0.485 | 0.000 | 1.240 | 0.489 | 0.429 | 1.060 | 0.547 | 0.167 | 0.860 | 0.428 | 0.000 | 0.650 | 0.417 | 0.000 | 0.468 | 0.114 | 0.920 | 0.427 | 0.178 |
| Defocus Blur | 1.020 | 0.406 | 0.308 | 0.810 | 0.397 | 0.214 | 0.720 | 0.451 | 0.000 | 0.890 | 0.139 | 0.000 | 0.590 | 0.397 | 0.207 | 0.590 | 0.476 | 0.000 | 1.040 | 0.476 | 0.286 | 0.890 | 0.539 | 0.133 | 0.780 | 0.419 | 0.000 | 0.800 | 0.415 | 0.000 | 0.460 | 0.086 | 0.850 | 0.418 | 0.164 |
| Edge Sawtooth | 1.020 | 0.405 | 0.154 | 0.850 | 0.399 | 0.186 | 0.750 | 0.445 | 0.000 | 0.740 | 0.129 | 0.000 | 0.600 | 0.397 | 0.163 | 0.600 | 0.493 | 0.000 | 0.990 | 0.480 | 0.286 | 0.990 | 0.528 | 0.133 | 0.930 | 0.425 | 0.000 | 0.620 | 0.407 | 0.000 | 0.459 | 0.086 | 0.870 | 0.417 | 0.138 |
| Elastic | 1.070 | 0.412 | 0.308 | 0.880 | 0.407 | 0.186 | 0.680 | 0.447 | 0.000 | 0.890 | 0.137 | 0.000 | 0.690 | 0.405 | 0.185 | 0.690 | 0.484 | 0.000 | 1.180 | 0.486 | 0.429 | 0.890 | 0.539 | 0.133 | 0.830 | 0.419 | 0.000 | 0.840 | 0.412 | 0.000 | 0.462 | 0.100 | 0.900 | 0.423 | 0.158 |
| Flicker | 1.080 | 0.411 | 0.308 | 0.870 | 0.405 | 0.200 | 0.740 | 0.450 | 0.000 | 1.000 | 0.132 | 0.000 | 0.550 | 0.404 | 0.196 | 0.550 | 0.490 | 0.000 | 1.220 | 0.485 | 0.286 | 0.950 | 0.534 | 0.133 | 0.720 | 0.417 | 0.000 | 0.850 | 0.419 | 0.000 | 0.464 | 0.086 | 0.900 | 0.423 | 0.158 |
| Foggy | 1.090 | 0.412 | 0.231 | 0.890 | 0.408 | 0.200 | 0.700 | 0.443 | 0.000 | 1.080 | 0.135 | 0.000 | 0.560 | 0.405 | 0.185 | 0.560 | 0.486 | 0.000 | 1.150 | 0.484 | 0.429 | 0.970 | 0.539 | 0.100 | 0.850 | 0.418 | 0.000 | 0.680 | 0.412 | 0.000 | 0.463 | 0.086 | 0.900 | 0.423 | 0.151 |
| Frame Drop | 0.610 | 0.360 | 0.154 | 0.530 | 0.368 | 0.114 | 0.530 | 0.415 | 0.000 | 0.980 | 0.135 | 0.000 | 0.530 | 0.365 | 0.109 | 0.530 | 0.476 | 0.000 | 0.870 | 0.446 | 0.286 | 0.850 | 0.524 | 0.200 | 0.640 | 0.394 | 0.000 | 0.520 | 0.407 | 0.000 | 0.444 | 0.114 | 0.600 | 0.390 | 0.118 |
| Frame Repeat | 0.560 | 0.344 | 0.231 | 0.480 | 0.352 | 0.086 | 0.480 | 0.399 | 0.067 | 1.210 | 0.156 | 0.000 | 0.380 | 0.351 | 0.109 | 0.380 | 0.456 | 0.000 | 0.800 | 0.449 | 0.000 | 0.800 | 0.512 | 0.200 | 0.590 | 0.374 | 0.000 | 0.540 | 0.397 | 0.000 | 0.434 | 0.086 | 0.570 | 0.378 | 0.105 |
| Frame Replace | 1.200 | 0.420 | 0.308 | 0.890 | 0.409 | 0.200 | 0.780 | 0.449 | 0.000 | 0.740 | 0.136 | 0.000 | 0.550 | 0.407 | 0.196 | 0.550 | 0.477 | 0.000 | 1.240 | 0.491 | 0.429 | 0.980 | 0.544 | 0.167 | 0.710 | 0.429 | 0.000 | 0.710 | 0.419 | 0.000 | 0.466 | 0.114 | 0.940 | 0.426 | 0.171 |
| Frost | 0.970 | 0.389 | 0.154 | 0.800 | 0.397 | 0.200 | 0.690 | 0.430 | 0.000 | 0.710 | 0.147 | 0.000 | 0.510 | 0.392 | 0.174 | 0.510 | 0.486 | 0.000 | 0.980 | 0.468 | 0.143 | 0.940 | 0.532 | 0.033 | 0.700 | 0.411 | 0.000 | 0.750 | 0.406 | 0.000 | 0.454 | 0.029 | 0.810 | 0.412 | 0.118 |
| Gaussian | 0.980 | 0.400 | 0.154 | 0.750 | 0.401 | 0.200 | 0.660 | 0.446 | 0.000 | 0.650 | 0.147 | 0.000 | 0.660 | 0.398 | 0.174 | 0.660 | 0.489 | 0.000 | 1.200 | 0.488 | 0.429 | 0.960 | 0.533 | 0.167 | 0.750 | 0.417 | 0.000 | 0.830 | 0.404 | 0.000 | 0.459 | 0.114 | 0.830 | 0.418 | 0.158 |
| Gaussian Blur | 0.690 | 0.364 | 0.154 | 0.600 | 0.372 | 0.143 | 0.540 | 0.416 | 0.000 | 0.850 | 0.139 | 0.000 | 0.480 | 0.370 | 0.130 | 0.480 | 0.460 | 0.000 | 0.900 | 0.455 | 0.143 | 0.820 | 0.527 | 0.133 | 0.640 | 0.397 | 0.000 | 0.650 | 0.398 | 0.000 | 0.442 | 0.071 | 0.640 | 0.393 | 0.112 |
| Glass Blur | 0.910 | 0.400 | 0.231 | 0.800 | 0.398 | 0.214 | 0.690 | 0.447 | 0.000 | 0.830 | 0.138 | 0.000 | 0.610 | 0.396 | 0.196 | 0.610 | 0.480 | 0.000 | 1.120 | 0.476 | 0.286 | 0.950 | 0.544 | 0.167 | 0.690 | 0.411 | 0.000 | 0.790 | 0.413 | 0.000 | 0.459 | 0.100 | 0.830 | 0.417 | 0.164 |
| H265 Artifacts | 0.580 | 0.354 | 0.154 | 0.530 | 0.366 | 0.143 | 0.440 | 0.410 | 0.000 | 0.950 | 0.146 | 0.000 | 0.560 | 0.363 | 0.130 | 0.560 | 0.465 | 0.000 | 0.770 | 0.435 | 0.000 | 0.770 | 0.513 | 0.033 | 0.510 | 0.378 | 0.000 | 0.620 | 0.382 | 0.000 | 0.428 | 0.014 | 0.580 | 0.384 | 0.086 |
| Impulse | 0.750 | 0.381 | 0.077 | 0.670 | 0.379 | 0.171 | 0.580 | 0.419 | 0.000 | 1.080 | 0.144 | 0.000 | 0.580 | 0.376 | 0.141 | 0.580 | 0.467 | 0.000 | 0.890 | 0.447 | 0.143 | 0.890 | 0.526 | 0.067 | 0.650 | 0.398 | 0.000 | 0.710 | 0.394 | 0.000 | 0.440 | 0.043 | 0.710 | 0.397 | 0.105 |
| Jpeg Artifact | 0.880 | 0.398 | 0.154 | 0.800 | 0.400 | 0.200 | 0.650 | 0.442 | 0.000 | 0.810 | 0.148 | 0.000 | 0.800 | 0.396 | 0.174 | 0.800 | 0.489 | 0.000 | 1.070 | 0.474 | 0.429 | 0.930 | 0.532 | 0.133 | 0.840 | 0.425 | 0.000 | 0.790 | 0.401 | 0.000 | 0.457 | 0.100 | 0.800 | 0.416 | 0.151 |
| Motion Blur | 0.570 | 0.357 | 0.154 | 0.580 | 0.365 | 0.114 | 0.520 | 0.406 | 0.000 | 0.760 | 0.134 | 0.000 | 0.420 | 0.362 | 0.109 | 0.420 | 0.444 | 0.000 | 0.890 | 0.446 | 0.143 | 0.760 | 0.515 | 0.067 | 0.520 | 0.379 | 0.000 | 0.530 | 0.381 | 0.000 | 0.428 | 0.043 | 0.610 | 0.384 | 0.086 |
| Overexposure | 1.030 | 0.408 | 0.308 | 0.860 | 0.405 | 0.214 | 0.710 | 0.445 | 0.000 | 0.870 | 0.141 | 0.000 | 0.550 | 0.402 | 0.207 | 0.550 | 0.481 | 0.000 | 1.140 | 0.482 | 0.286 | 1.020 | 0.538 | 0.133 | 0.600 | 0.406 | 0.000 | 0.600 | 0.410 | 0.000 | 0.458 | 0.086 | 0.880 | 0.421 | 0.164 |
| Poisson | 0.890 | 0.391 | 0.154 | 0.760 | 0.392 | 0.186 | 0.550 | 0.432 | 0.000 | 0.820 | 0.144 | 0.000 | 0.520 | 0.389 | 0.163 | 0.520 | 0.471 | 0.000 | 1.040 | 0.480 | 0.429 | 0.930 | 0.533 | 0.133 | 0.620 | 0.403 | 0.000 | 0.760 | 0.401 | 0.000 | 0.455 | 0.100 | 0.760 | 0.411 | 0.145 |
| Rainy | 1.000 | 0.408 | 0.231 | 0.870 | 0.406 | 0.229 | 0.690 | 0.449 | 0.000 | 0.850 | 0.142 | 0.000 | 0.650 | 0.404 | 0.207 | 0.650 | 0.484 | 0.000 | 1.170 | 0.480 | 0.429 | 0.950 | 0.539 | 0.133 | 0.680 | 0.416 | 0.000 | 0.850 | 0.411 | 0.000 | 0.461 | 0.100 | 0.880 | 0.422 | 0.171 |
| Random Block | 0.960 | 0.401 | 0.385 | 0.640 | 0.396 | 0.200 | 0.640 | 0.431 | 0.000 | 0.790 | 0.134 | 0.000 | 0.500 | 0.393 | 0.143 | 0.500 | 0.477 | 0.000 | 0.950 | 0.479 | 0.143 | 0.950 | 0.527 | 0.071 | 0.780 | 0.428 | 0.000 | 0.780 | 0.408 | 0.000 | 0.456 | 0.071 | 0.810 | 0.413 | 0.158 |
| Reflect | 1.030 | 0.407 | 0.231 | 0.890 | 0.406 | 0.214 | 0.690 | 0.442 | 0.000 | 0.770 | 0.139 | 0.000 | 0.580 | 0.402 | 0.196 | 0.580 | 0.478 | 0.000 | 1.100 | 0.479 | 0.429 | 0.920 | 0.542 | 0.100 | 0.840 | 0.415 | 0.000 | 0.820 | 0.410 | 0.000 | 0.461 | 0.086 | 0.870 | 0.421 | 0.158 |
| Resolution Degrade | 1.010 | 0.402 | 0.308 | 0.780 | 0.393 | 0.200 | 0.710 | 0.440 | 0.000 | 0.850 | 0.136 | 0.000 | 0.560 | 0.392 | 0.196 | 0.560 | 0.466 | 0.000 | 1.140 | 0.478 | 0.429 | 0.870 | 0.531 | 0.133 | 0.730 | 0.411 | 0.000 | 0.800 | 0.411 | 0.000 | 0.456 | 0.086 | 0.830 | 0.413 | 0.158 |
| Rolling Shutter | 0.960 | 0.388 | 0.308 | 0.740 | 0.387 | 0.157 | 0.670 | 0.433 | 0.000 | 0.870 | 0.142 | 0.000 | 0.640 | 0.386 | 0.163 | 0.640 | 0.471 | 0.000 | 1.040 | 0.474 | 0.286 | 0.880 | 0.520 | 0.100 | 0.830 | 0.412 | 0.000 | 0.780 | 0.407 | 0.000 | 0.452 | 0.057 | 0.790 | 0.407 | 0.125 |
| Shadow | 1.150 | 0.416 | 0.231 | 0.890 | 0.408 | 0.200 | 0.800 | 0.445 | 0.000 | 0.730 | 0.136 | 0.000 | 0.620 | 0.406 | 0.185 | 0.620 | 0.489 | 0.000 | 1.200 | 0.486 | 0.429 | 0.930 | 0.542 | 0.133 | 0.800 | 0.423 | 0.000 | 0.650 | 0.411 | 0.000 | 0.465 | 0.100 | 0.920 | 0.425 | 0.158 |
| Snow | 1.130 | 0.423 | 0.308 | 0.870 | 0.408 | 0.229 | 0.710 | 0.453 | 0.000 | 0.850 | 0.134 | 0.000 | 0.640 | 0.408 | 0.217 | 0.640 | 0.489 | 0.000 | 1.170 | 0.487 | 0.429 | 0.990 | 0.537 | 0.100 | 0.900 | 0.428 | 0.000 | 0.870 | 0.416 | 0.000 | 0.466 | 0.114 | 0.900 | 0.426 | 0.184 |
| Speckle | 0.890 | 0.393 | 0.154 | 0.760 | 0.396 | 0.186 | 0.670 | 0.435 | 0.000 | 0.850 | 0.146 | 0.000 | 0.680 | 0.392 | 0.163 | 0.680 | 0.477 | 0.000 | 1.070 | 0.474 | 0.286 | 0.840 | 0.532 | 0.100 | 0.720 | 0.401 | 0.000 | 0.770 | 0.399 | 0.000 | 0.451 | 0.071 | 0.800 | 0.412 | 0.132 |
| Stretch Squish | 0.770 | 0.379 | 0.154 | 0.670 | 0.380 | 0.186 | 0.500 | 0.422 | 0.000 | 0.760 | 0.138 | 0.000 | 0.530 | 0.378 | 0.163 | 0.530 | 0.462 | 0.000 | 0.890 | 0.464 | 0.143 | 0.890 | 0.529 | 0.133 | 0.700 | 0.396 | 0.000 | 0.500 | 0.395 | 0.000 | 0.444 | 0.071 | 0.690 | 0.399 | 0.132 |
| Target Block | 1.050 | 0.410 | 0.308 | 0.780 | 0.397 | 0.143 | 0.670 | 0.434 | 0.000 | 0.940 | 0.146 | 0.000 | 0.590 | 0.395 | 0.196 | 0.590 | 0.479 | 0.000 | 1.080 | 0.484 | 0.286 | 0.960 | 0.530 | 0.167 | 0.610 | 0.424 | 0.000 | 0.620 | 0.412 | 0.000 | 0.460 | 0.100 | 0.830 | 0.416 | 0.164 |
| Temporal Jitter | 0.860 | 0.399 | 0.231 | 0.700 | 0.392 | 0.143 | 0.500 | 0.447 | 0.000 | 0.940 | 0.145 | 0.000 | 0.760 | 0.393 | 0.141 | 0.760 | 0.465 | 0.000 | 1.120 | 0.483 | 0.429 | 0.960 | 0.534 | 0.200 | 0.680 | 0.406 | 0.000 | 0.620 | 0.412 | 0.000 | 0.456 | 0.129 | 0.780 | 0.413 | 0.145 |
| Underexposure | 1.050 | 0.416 | 0.308 | 0.880 | 0.406 | 0.214 | 0.750 | 0.445 | 0.000 | 0.770 | 0.145 | 0.000 | 0.620 | 0.405 | 0.207 | 0.620 | 0.485 | 0.000 | 1.170 | 0.487 | 0.429 | 1.030 | 0.545 | 0.133 | 0.830 | 0.422 | 0.000 | 0.870 | 0.417 | 0.000 | 0.466 | 0.100 | 0.900 | 0.424 | 0.171 |
| Zoom Blur | 0.780 | 0.371 | 0.154 | 0.630 | 0.374 | 0.143 | 0.520 | 0.420 | 0.000 | 0.840 | 0.143 | 0.000 | 0.430 | 0.372 | 0.130 | 0.430 | 0.462 | 0.000 | 0.920 | 0.455 | 0.143 | 0.700 | 0.515 | 0.033 | 0.530 | 0.389 | 0.000 | 0.650 | 0.395 | 0.000 | 0.438 | 0.029 | 0.670 | 0.394 | 0.092 |

Table 17: PLLaVA 7B performance across different question types

| Noise Type | CP | | | FP-S | | | FP-C | | | HL | | | Mean | | | LR | | | AR | | | RR | | | CSR | | | TR | | | Mean2 | | | Overall | | |
|---|---|---|---|---|---|---|---|---|---|---|---|---|---|---|---|---|---|---|---|---|---|---|---|---|---|---|---|---|---|---|---|---|---|---|---|---|
| | GPT | SBERT | Acc | GPT | SBERT | Acc | GPT | SBERT | Acc | GPT | SBERT | Acc | GPT | SBERT | Acc | GPT | SBERT | Acc | GPT | SBERT | Acc | GPT | SBERT | Acc | GPT | SBERT | Acc | GPT | SBERT | Acc | GPT | SBERT | Acc | GPT | SBERT | Acc |
| Clean | 0.890 | 0.391 | 0.000 | 0.650 | 0.383 | 0.114 | 0.570 | 0.413 | 0.000 | 0.900 | 0.144 | 0.000 | 0.710 | 0.380 | 0.087 | 0.340 | 0.460 | 0.000 | 0.950 | 0.453 | 0.000 | 0.660 | 0.502 | 0.033 | 0.460 | 0.384 | 0.000 | 0.430 | 0.384 | 0.000 | 0.580 | 0.431 | 0.014 | 0.690 | 0.397 | 0.059 |
| Bit Error | 0.700 | 0.369 | 0.000 | 0.530 | 0.372 | 0.129 | 0.420 | 0.399 | 0.000 | 0.890 | 0.137 | 0.000 | 0.570 | 0.367 | 0.098 | 0.340 | 0.462 | 0.000 | 0.840 | 0.440 | 0.000 | 0.590 | 0.486 | 0.000 | 0.420 | 0.382 | 0.000 | 0.350 | 0.379 | 0.000 | 0.520 | 0.423 | 0.000 | 0.560 | 0.385 | 0.059 |
| Bright Transform | 0.730 | 0.379 | 0.000 | 0.580 | 0.380 | 0.100 | 0.540 | 0.408 | 0.000 | 0.840 | 0.137 | 0.000 | 0.610 | 0.375 | 0.076 | 0.350 | 0.470 | 0.000 | 0.810 | 0.444 | 0.000 | 0.620 | 0.487 | 0.000 | 0.510 | 0.384 | 0.000 | 0.330 | 0.378 | 0.000 | 0.520 | 0.426 | 0.000 | 0.590 | 0.391 | 0.046 |
| Color Quantized | 0.760 | 0.381 | 0.000 | 0.620 | 0.377 | 0.114 | 0.510 | 0.398 | 0.000 | 0.900 | 0.147 | 0.000 | 0.660 | 0.373 | 0.087 | 0.470 | 0.459 | 0.000 | 1.010 | 0.449 | 0.000 | 0.760 | 0.488 | 0.000 | 0.600 | 0.389 | 0.000 | 0.410 | 0.394 | 0.000 | 0.640 | 0.431 | 0.000 | 0.660 | 0.392 | 0.053 |
| Color Shift | 0.650 | 0.377 | 0.000 | 0.590 | 0.380 | 0.100 | 0.440 | 0.405 | 0.000 | 0.820 | 0.131 | 0.000 | 0.600 | 0.374 | 0.076 | 0.410 | 0.463 | 0.000 | 0.810 | 0.447 | 0.000 | 0.600 | 0.494 | 0.033 | 0.430 | 0.379 | 0.000 | 0.390 | 0.376 | 0.000 | 0.530 | 0.425 | 0.014 | 0.580 | 0.390 | 0.053 |
| Contrast | 0.780 | 0.390 | 0.000 | 0.610 | 0.384 | 0.114 | 0.510 | 0.410 | 0.000 | 0.870 | 0.138 | 0.000 | 0.640 | 0.380 | 0.087 | 0.300 | 0.452 | 0.000 | 0.890 | 0.449 | 0.000 | 0.650 | 0.494 | 0.000 | 0.590 | 0.396 | 0.000 | 0.310 | 0.376 | 0.000 | 0.530 | 0.426 | 0.000 | 0.620 | 0.394 | 0.053 |
| Defocus Blur | 0.630 | 0.369 | 0.000 | 0.450 | 0.374 | 0.100 | 0.460 | 0.401 | 0.000 | 0.690 | 0.133 | 0.000 | 0.560 | 0.368 | 0.076 | 0.290 | 0.457 | 0.000 | 0.820 | 0.447 | 0.000 | 0.560 | 0.499 | 0.033 | 0.600 | 0.388 | 0.000 | 0.390 | 0.388 | 0.000 | 0.530 | 0.431 | 0.014 | 0.560 | 0.388 | 0.053 |
| Edge Sawtooth | 0.680 | 0.372 | 0.000 | 0.540 | 0.377 | 0.100 | 0.520 | 0.403 | 0.000 | 0.850 | 0.132 | 0.000 | 0.580 | 0.371 | 0.076 | 0.420 | 0.463 | 0.000 | 0.880 | 0.439 | 0.000 | 0.560 | 0.494 | 0.000 | 0.570 | 0.389 | 0.000 | 0.340 | 0.381 | 0.000 | 0.540 | 0.426 | 0.000 | 0.580 | 0.388 | 0.046 |
| Elastic | 0.730 | 0.379 | 0.000 | 0.590 | 0.380 | 0.100 | 0.530 | 0.400 | 0.000 | 0.850 | 0.139 | 0.000 | 0.630 | 0.375 | 0.076 | 0.410 | 0.465 | 0.000 | 0.800 | 0.443 | 0.000 | 0.710 | 0.488 | 0.000 | 0.540 | 0.390 | 0.000 | 0.340 | 0.375 | 0.000 | 0.540 | 0.424 | 0.000 | 0.610 | 0.391 | 0.046 |
| Flicker | 0.720 | 0.386 | 0.000 | 0.570 | 0.376 | 0.086 | 0.490 | 0.404 | 0.000 | 0.770 | 0.137 | 0.000 | 0.610 | 0.374 | 0.065 | 0.350 | 0.463 | 0.000 | 0.880 | 0.444 | 0.000 | 0.560 | 0.498 | 0.000 | 0.410 | 0.399 | 0.000 | 0.360 | 0.387 | 0.000 | 0.540 | 0.432 | 0.000 | 0.600 | 0.392 | 0.039 |
| Foggy | 0.710 | 0.376 | 0.000 | 0.550 | 0.377 | 0.114 | 0.450 | 0.397 | 0.000 | 0.790 | 0.139 | 0.000 | 0.580 | 0.371 | 0.087 | 0.400 | 0.452 | 0.000 | 0.800 | 0.440 | 0.000 | 0.670 | 0.494 | 0.000 | 0.480 | 0.379 | 0.000 | 0.330 | 0.380 | 0.000 | 0.530 | 0.423 | 0.000 | 0.570 | 0.387 | 0.053 |
| Frame Drop | 0.260 | 0.283 | 0.000 | 0.220 | 0.292 | 0.086 | 0.220 | 0.313 | 0.000 | 0.850 | 0.117 | 0.000 | 0.250 | 0.290 | 0.065 | 0.290 | 0.415 | 0.000 | 0.500 | 0.387 | 0.000 | 0.380 | 0.468 | 0.000 | 0.210 | 0.327 | 0.000 | 0.330 | 0.339 | 0.000 | 0.360 | 0.383 | 0.000 | 0.300 | 0.319 | 0.039 |
| Frame Repeat | 0.310 | 0.319 | 0.000 | 0.350 | 0.332 | 0.086 | 0.340 | 0.343 | 0.000 | 0.820 | 0.137 | 0.000 | 0.360 | 0.325 | 0.065 | 0.370 | 0.421 | 0.000 | 0.490 | 0.387 | 0.000 | 0.520 | 0.460 | 0.033 | 0.320 | 0.335 | 0.000 | 0.370 | 0.357 | 0.000 | 0.420 | 0.389 | 0.000 | 0.420 | 0.345 | 0.046 |
| Frame Replace | 0.820 | 0.390 | 0.077 | 0.630 | 0.381 | 0.114 | 0.560 | 0.413 | 0.000 | 0.790 | 0.127 | 0.000 | 0.670 | 0.378 | 0.098 | 0.350 | 0.454 | 0.000 | 0.900 | 0.455 | 0.000 | 0.610 | 0.498 | 0.000 | 0.510 | 0.380 | 0.000 | 0.450 | 0.403 | 0.000 | 0.580 | 0.436 | 0.000 | 0.650 | 0.397 | 0.059 |
| Frost | 0.720 | 0.371 | 0.077 | 0.530 | 0.372 | 0.114 | 0.420 | 0.388 | 0.000 | 0.730 | 0.135 | 0.000 | 0.570 | 0.366 | 0.087 | 0.370 | 0.459 | 0.000 | 0.730 | 0.441 | 0.000 | 0.610 | 0.485 | 0.033 | 0.460 | 0.376 | 0.000 | 0.360 | 0.375 | 0.000 | 0.510 | 0.421 | 0.014 | 0.560 | 0.384 | 0.059 |
| Gaussian | 0.530 | 0.352 | 0.000 | 0.490 | 0.367 | 0.114 | 0.440 | 0.401 | 0.000 | 0.890 | 0.141 | 0.000 | 0.510 | 0.361 | 0.087 | 0.390 | 0.464 | 0.000 | 0.710 | 0.435 | 0.000 | 0.580 | 0.487 | 0.000 | 0.300 | 0.382 | 0.000 | 0.300 | 0.382 | 0.000 | 0.480 | 0.425 | 0.014 | 0.510 | 0.381 | 0.053 |
| Gaussian Blur | 0.420 | 0.342 | 0.000 | 0.360 | 0.346 | 0.057 | 0.190 | 0.374 | 0.000 | 0.840 | 0.132 | 0.000 | 0.370 | 0.342 | 0.043 | 0.300 | 0.446 | 0.000 | 0.660 | 0.418 | 0.000 | 0.520 | 0.508 | 0.000 | 0.380 | 0.373 | 0.000 | 0.380 | 0.374 | 0.000 | 0.450 | 0.417 | 0.000 | 0.400 | 0.365 | 0.026 |
| Glass Blur | 0.660 | 0.367 | 0.000 | 0.450 | 0.372 | 0.114 | 0.490 | 0.410 | 0.000 | 1.060 | 0.140 | 0.000 | 0.600 | 0.368 | 0.087 | 0.370 | 0.461 | 0.000 | 0.520 | 0.445 | 0.000 | 0.520 | 0.497 | 0.033 | 0.600 | 0.397 | 0.000 | 0.400 | 0.382 | 0.000 | 0.540 | 0.430 | 0.014 | 0.580 | 0.388 | 0.059 |
| H265 Artifacts | 0.350 | 0.324 | 0.000 | 0.380 | 0.347 | 0.057 | 0.220 | 0.361 | 0.000 | 1.100 | 0.132 | 0.000 | 0.380 | 0.337 | 0.043 | 0.300 | 0.439 | 0.000 | 0.470 | 0.409 | 0.000 | 0.550 | 0.496 | 0.000 | 0.230 | 0.354 | 0.000 | 0.390 | 0.363 | 0.000 | 0.390 | 0.407 | 0.000 | 0.390 | 0.359 | 0.026 |
| Impulse | 0.320 | 0.327 | 0.000 | 0.330 | 0.336 | 0.071 | 0.270 | 0.363 | 0.000 | 0.900 | 0.135 | 0.000 | 0.340 | 0.330 | 0.054 | 0.210 | 0.442 | 0.000 | 0.560 | 0.409 | 0.000 | 0.520 | 0.478 | 0.000 | 0.260 | 0.340 | 0.000 | 0.260 | 0.361 | 0.000 | 0.370 | 0.402 | 0.000 | 0.360 | 0.351 | 0.033 |
| Jpeg Artifact | 0.600 | 0.362 | 0.000 | 0.460 | 0.369 | 0.100 | 0.380 | 0.397 | 0.000 | 0.700 | 0.137 | 0.000 | 0.500 | 0.364 | 0.076 | 0.350 | 0.448 | 0.000 | 0.660 | 0.423 | 0.000 | 0.490 | 0.493 | 0.033 | 0.420 | 0.379 | 0.000 | 0.470 | 0.375 | 0.000 | 0.470 | 0.419 | 0.000 | 0.500 | 0.381 | 0.046 |
| Motion Blur | 0.320 | 0.328 | 0.000 | 0.320 | 0.338 | 0.071 | 0.240 | 0.368 | 0.000 | 0.920 | 0.121 | 0.000 | 0.340 | 0.334 | 0.054 | 0.220 | 0.426 | 0.000 | 0.480 | 0.389 | 0.143 | 0.450 | 0.503 | 0.000 | 0.260 | 0.337 | 0.000 | 0.320 | 0.369 | 0.000 | 0.370 | 0.402 | 0.014 | 0.360 | 0.355 | 0.039 |
| Overexposure | 0.690 | 0.370 | 0.000 | 0.560 | 0.378 | 0.100 | 0.390 | 0.391 | 0.000 | 0.950 | 0.139 | 0.000 | 0.580 | 0.371 | 0.076 | 0.380 | 0.461 | 0.000 | 0.860 | 0.445 | 0.000 | 0.640 | 0.495 | 0.000 | 0.460 | 0.384 | 0.000 | 0.340 | 0.390 | 0.000 | 0.540 | 0.430 | 0.000 | 0.570 | 0.389 | 0.046 |
| Poisson | 0.520 | 0.349 | 0.077 | 0.450 | 0.363 | 0.100 | 0.350 | 0.363 | 0.000 | 0.870 | 0.136 | 0.000 | 0.480 | 0.357 | 0.076 | 0.330 | 0.465 | 0.000 | 0.560 | 0.428 | 0.000 | 0.560 | 0.495 | 0.000 | 0.490 | 0.392 | 0.000 | 0.340 | 0.368 | 0.000 | 0.500 | 0.384 | 0.000 | 0.480 | 0.374 | 0.046 |
| Rainy | 0.700 | 0.380 | 0.077 | 0.580 | 0.381 | 0.100 | 0.500 | 0.406 | 0.000 | 0.850 | 0.137 | 0.000 | 0.620 | 0.375 | 0.087 | 0.330 | 0.452 | 0.000 | 0.840 | 0.442 | 0.000 | 0.650 | 0.498 | 0.000 | 0.520 | 0.405 | 0.000 | 0.360 | 0.380 | 0.000 | 0.530 | 0.428 | 0.000 | 0.600 | 0.391 | 0.053 |
| Random Block | 0.650 | 0.365 | 0.000 | 0.450 | 0.381 | 0.100 | 0.340 | 0.390 | 0.000 | 0.940 | 0.140 | 0.000 | 0.540 | 0.371 | 0.076 | 0.330 | 0.464 | 0.000 | 0.540 | 0.441 | 0.000 | 0.410 | 0.487 | 0.000 | 0.520 | 0.386 | 0.000 | 0.340 | 0.394 | 0.040 | 0.470 | 0.430 | 0.000 | 0.520 | 0.389 | 0.053 |
| Reflect | 0.680 | 0.373 | 0.000 | 0.580 | 0.374 | 0.114 | 0.460 | 0.399 | 0.000 | 0.870 | 0.136 | 0.000 | 0.610 | 0.369 | 0.087 | 0.500 | 0.457 | 0.000 | 0.820 | 0.439 | 0.000 | 0.610 | 0.498 | 0.000 | 0.510 | 0.392 | 0.000 | 0.400 | 0.384 | 0.000 | 0.570 | 0.429 | 0.000 | 0.600 | 0.388 | 0.053 |
| Resolution Degrade | 0.620 | 0.370 | 0.000 | 0.540 | 0.379 | 0.100 | 0.530 | 0.410 | 0.000 | 1.160 | 0.140 | 0.000 | 0.590 | 0.373 | 0.076 | 0.290 | 0.437 | 0.000 | 0.700 | 0.431 | 0.000 | 0.540 | 0.502 | 0.000 | 0.520 | 0.398 | 0.000 | 0.400 | 0.387 | 0.000 | 0.500 | 0.425 | 0.000 | 0.560 | 0.390 | 0.046 |
| Rolling Shutter | 0.630 | 0.367 | 0.000 | 0.520 | 0.368 | 0.100 | 0.390 | 0.388 | 0.000 | 0.870 | 0.134 | 0.000 | 0.540 | 0.362 | 0.076 | 0.340 | 0.457 | 0.000 | 0.750 | 0.436 | 0.000 | 0.590 | 0.488 | 0.000 | 0.520 | 0.388 | 0.000 | 0.440 | 0.385 | 0.000 | 0.540 | 0.426 | 0.000 | 0.550 | 0.382 | 0.046 |
| Shadow | 0.830 | 0.388 | 0.000 | 0.620 | 0.378 | 0.114 | 0.500 | 0.402 | 0.000 | 0.870 | 0.139 | 0.000 | 0.660 | 0.375 | 0.087 | 0.420 | 0.452 | 0.000 | 0.890 | 0.448 | 0.000 | 0.670 | 0.495 | 0.000 | 0.620 | 0.388 | 0.000 | 0.380 | 0.376 | 0.000 | 0.580 | 0.425 | 0.000 | 0.650 | 0.390 | 0.053 |
| Snow | 0.810 | 0.381 | 0.077 | 0.580 | 0.379 | 0.100 | 0.470 | 0.410 | 0.000 | 0.810 | 0.137 | 0.000 | 0.630 | 0.375 | 0.087 | 0.290 | 0.463 | 0.000 | 0.830 | 0.448 | 0.000 | 0.550 | 0.488 | 0.000 | 0.590 | 0.394 | 0.000 | 0.390 | 0.384 | 0.000 | 0.530 | 0.429 | 0.000 | 0.600 | 0.392 | 0.053 |
| Speckle | 0.530 | 0.355 | 0.000 | 0.450 | 0.359 | 0.114 | 0.360 | 0.390 | 0.000 | 0.780 | 0.144 | 0.000 | 0.480 | 0.356 | 0.087 | 0.330 | 0.453 | 0.000 | 0.750 | 0.429 | 0.000 | 0.470 | 0.496 | 0.033 | 0.470 | 0.383 | 0.000 | 0.260 | 0.369 | 0.000 | 0.430 | 0.418 | 0.014 | 0.470 | 0.376 | 0.053 |
| Stretch Squish | 0.460 | 0.341 | 0.000 | 0.390 | 0.354 | 0.100 | 0.310 | 0.390 | 0.000 | 0.850 | 0.143 | 0.000 | 0.420 | 0.349 | 0.076 | 0.270 | 0.455 | 0.000 | 0.680 | 0.433 | 0.000 | 0.560 | 0.508 | 0.000 | 0.300 | 0.358 | 0.000 | 0.360 | 0.390 | 0.000 | 0.460 | 0.425 | 0.000 | 0.440 | 0.373 | 0.046 |
| Target Block | 0.640 | 0.369 | 0.000 | 0.480 | 0.356 | 0.114 | 0.390 | 0.393 | 0.000 | 0.820 | 0.137 | 0.000 | 0.520 | 0.356 | 0.087 | 0.300 | 0.452 | 0.000 | 0.810 | 0.438 | 0.000 | 0.510 | 0.482 | 0.033 | 0.470 | 0.378 | 0.000 | 0.360 | 0.384 | 0.000 | 0.490 | 0.422 | 0.014 | 0.520 | 0.376 | 0.059 |
| Temporal Jitter | 0.650 | 0.355 | 0.000 | 0.520 | 0.355 | 0.086 | 0.440 | 0.383 | 0.000 | 0.980 | 0.134 | 0.000 | 0.560 | 0.353 | 0.065 | 0.300 | 0.436 | 0.000 | 0.790 | 0.425 | 0.000 | 0.520 | 0.490 | 0.033 | 0.460 | 0.378 | 0.000 | 0.400 | 0.372 | 0.000 | 0.500 | 0.415 | 0.014 | 0.550 | 0.372 | 0.046 |
| Underexposure | 0.730 | 0.382 | 0.000 | 0.570 | 0.376 | 0.129 | 0.480 | 0.399 | 0.000 | 0.900 | 0.141 | 0.000 | 0.610 | 0.372 | 0.098 | 0.270 | 0.463 | 0.000 | 0.900 | 0.451 | 0.000 | 0.540 | 0.490 | 0.000 | 0.590 | 0.390 | 0.000 | 0.300 | 0.381 | 0.000 | 0.510 | 0.428 | 0.000 | 0.580 | 0.389 | 0.059 |
| Zoom Blur | 0.450 | 0.338 | 0.000 | 0.390 | 0.350 | 0.100 | 0.290 | 0.374 | 0.000 | 0.790 | 0.136 | 0.000 | 0.410 | 0.344 | 0.076 | 0.240 | 0.416 | 0.000 | 0.630 | 0.408 | 0.000 | 0.500 | 0.484 | 0.000 | 0.260 | 0.344 | 0.000 | 0.340 | 0.369 | 0.000 | 0.410 | 0.402 | 0.000 | 0.420 | 0.362 | 0.046 |

Table 18: Qwen2.5-VL 3B performance across different question types

| Noise Type | CP | | | FP-S | | | FP-C | | | HL | | | Mean | | | LR | | | AR | | | RR | | | CSR | | | TR | | | Mean2 | | | Overall | | |
|---|---|---|---|---|---|---|---|---|---|---|---|---|---|---|---|---|---|---|---|---|---|---|---|---|---|---|---|---|---|---|---|---|---|---|---|---|
| | GPT | SBERT | Acc | GPT | SBERT | Acc | GPT | SBERT | Acc | GPT | SBERT | Acc | GPT | SBERT | Acc | GPT | SBERT | Acc | GPT | SBERT | Acc | GPT | SBERT | Acc | GPT | SBERT | Acc | GPT | SBERT | Acc | GPT | SBERT | Acc | GPT | SBERT | Acc |
| Clean | 1.330 | 0.436 | 0.231 | 1.060 | 0.415 | 0.414 | 0.760 | 0.442 | 0.133 | 0.840 | 0.140 | 0.000 | 1.080 | 0.414 | 0.370 | 0.760 | 0.510 | 0.222 | 1.250 | 0.473 | 0.143 | 1.080 | 0.543 | 0.467 | 0.960 | 0.421 | 0.000 | 0.940 | 0.438 | 0.240 | 1.020 | 0.473 | 0.329 | 1.080 | 0.434 | 0.355 |
| Bit Error | 1.300 | 0.436 | 0.231 | 1.050 | 0.416 | 0.414 | 0.720 | 0.436 | 0.067 | 0.900 | 0.141 | 0.000 | 1.060 | 0.413 | 0.359 | 0.630 | 0.504 | 0.222 | 1.200 | 0.460 | 0.143 | 1.040 | 0.554 | 0.500 | 1.120 | 0.435 | 0.000 | 0.830 | 0.435 | 0.280 | 0.970 | 0.472 | 0.357 | 1.050 | 0.433 | 0.362 |
| Bright Transform | 1.290 | 0.428 | 0.231 | 1.010 | 0.415 | 0.371 | 0.790 | 0.435 | 0.067 | 0.790 | 0.136 | 0.000 | 1.040 | 0.410 | 0.326 | 0.730 | 0.500 | 0.333 | 1.170 | 0.464 | 0.143 | 1.040 | 0.551 | 0.333 | 1.140 | 0.441 | 0.000 | 0.840 | 0.437 | 0.200 | 0.980 | 0.473 | 0.271 | 1.030 | 0.431 | 0.303 |
| Color Quantized | 1.350 | 0.435 | 0.154 | 1.040 | 0.416 | 0.429 | 0.770 | 0.444 | 0.200 | 1.000 | 0.141 | 0.000 | 1.080 | 0.414 | 0.370 | 0.810 | 0.501 | 0.333 | 1.270 | 0.470 | 0.143 | 1.110 | 0.556 | 0.433 | 1.100 | 0.429 | 0.000 | 0.940 | 0.442 | 0.240 | 1.050 | 0.477 | 0.329 | 1.080 | 0.435 | 0.349 |
| Color Shift | 1.210 | 0.418 | 0.077 | 0.990 | 0.409 | 0.386 | 0.680 | 0.438 | 0.133 | 1.000 | 0.139 | 0.000 | 1.010 | 0.406 | 0.326 | 0.710 | 0.507 | 0.333 | 1.150 | 0.465 | 0.143 | 1.140 | 0.553 | 0.333 | 0.990 | 0.418 | 0.000 | 0.790 | 0.426 | 0.080 | 0.960 | 0.470 | 0.243 | 1.010 | 0.428 | 0.289 |
| Contrast | 1.320 | 0.425 | 0.231 | 1.030 | 0.415 | 0.343 | 0.740 | 0.435 | 0.067 | 0.970 | 0.141 | 0.000 | 1.060 | 0.411 | 0.304 | 0.650 | 0.509 | 0.111 | 1.260 | 0.470 | 0.143 | 0.990 | 0.549 | 0.400 | 1.050 | 0.427 | 0.000 | 0.840 | 0.427 | 0.200 | 0.970 | 0.472 | 0.271 | 1.050 | 0.430 | 0.296 |
| Defocus Blur | 1.200 | 0.408 | 0.000 | 0.770 | 0.397 | 0.314 | 0.610 | 0.440 | 0.200 | 0.900 | 0.141 | 0.000 | 0.850 | 0.397 | 0.261 | 0.520 | 0.499 | 0.000 | 0.980 | 0.462 | 0.143 | 1.080 | 0.554 | 0.333 | 0.950 | 0.424 | 0.000 | 0.770 | 0.434 | 0.280 | 0.880 | 0.470 | 0.257 | 0.870 | 0.421 | 0.270 |
| Edge Sawtooth | 1.170 | 0.413 | 0.231 | 0.890 | 0.397 | 0.371 | 0.630 | 0.440 | 0.067 | 0.630 | 0.133 | 0.000 | 0.920 | 0.405 | 0.326 | 0.720 | 0.489 | 0.111 | 1.120 | 0.454 | 0.143 | 1.050 | 0.558 | 0.367 | 1.040 | 0.430 | 0.000 | 0.900 | 0.432 | 0.280 | 0.940 | 0.469 | 0.286 | 0.940 | 0.425 | 0.316 |
| Elastic | 1.300 | 0.423 | 0.231 | 0.960 | 0.410 | 0.414 | 0.740 | 0.440 | 0.200 | 0.810 | 0.143 | 0.000 | 1.000 | 0.408 | 0.370 | 0.560 | 0.503 | 0.222 | 1.140 | 0.462 | 0.143 | 1.140 | 0.563 | 0.433 | 1.110 | 0.421 | 0.000 | 0.960 | 0.442 | 0.240 | 0.990 | 0.474 | 0.314 | 0.990 | 0.429 | 0.349 |
| Flicker | 1.230 | 0.420 | 0.231 | 0.960 | 0.408 | 0.357 | 0.760 | 0.438 | 0.200 | 1.050 | 0.137 | 0.000 | 1.000 | 0.406 | 0.337 | 0.680 | 0.509 | 0.333 | 1.200 | 0.462 | 0.143 | 1.080 | 0.536 | 0.400 | 1.020 | 0.423 | 0.000 | 1.000 | 0.440 | 0.240 | 1.000 | 0.470 | 0.314 | 1.000 | 0.427 | 0.329 |
| Foggy | 1.280 | 0.429 | 0.308 | 0.970 | 0.413 | 0.371 | 0.680 | 0.443 | 0.133 | 0.950 | 0.139 | 0.000 | 1.010 | 0.412 | 0.348 | 0.720 | 0.511 | 0.222 | 1.140 | 0.462 | 0.143 | 1.080 | 0.547 | 0.400 | 1.090 | 0.442 | 0.000 | 0.840 | 0.437 | 0.240 | 0.970 | 0.473 | 0.300 | 1.010 | 0.431 | 0.329 |
| Frame Drop | 0.860 | 0.352 | 0.231 | 0.700 | 0.370 | 0.229 | 0.620 | 0.384 | 0.267 | 0.890 | 0.139 | 0.000 | 0.730 | 0.361 | 0.239 | 0.400 | 0.453 | 0.222 | 0.930 | 0.419 | 0.143 | 0.890 | 0.484 | 0.300 | 0.850 | 0.373 | 0.000 | 0.740 | 0.404 | 0.160 | 0.750 | 0.426 | 0.229 | 0.750 | 0.382 | 0.237 |
| Frame Repeat | 0.750 | 0.335 | 0.154 | 0.570 | 0.342 | 0.171 | 0.550 | 0.342 | 0.067 | 0.950 | 0.139 | 0.000 | 0.620 | 0.336 | 0.163 | 0.410 | 0.440 | 0.111 | 0.840 | 0.395 | 0.143 | 0.920 | 0.491 | 0.400 | 0.750 | 0.353 | 0.000 | 0.600 | 0.388 | 0.120 | 0.700 | 0.412 | 0.243 | 0.660 | 0.361 | 0.204 |
| Frame Replace | 1.290 | 0.435 | 0.000 | 1.030 | 0.413 | 0.329 | 0.760 | 0.442 | 0.067 | 1.000 | 0.136 | 0.000 | 1.050 | 0.412 | 0.261 | 0.580 | 0.502 | 0.222 | 1.100 | 0.465 | 0.143 | 1.000 | 0.548 | 0.367 | 1.050 | 0.427 | 0.000 | 0.910 | 0.434 | 0.240 | 0.940 | 0.470 | 0.286 | 1.030 | 0.431 | 0.276 |
| Frost | 1.280 | 0.422 | 0.231 | 0.930 | 0.407 | 0.357 | 0.690 | 0.429 | 0.200 | 0.900 | 0.137 | 0.000 | 0.980 | 0.405 | 0.337 | 0.710 | 0.512 | 0.222 | 1.170 | 0.459 | 0.143 | 1.050 | 0.546 | 0.467 | 1.160 | 0.423 | 0.000 | 0.850 | 0.435 | 0.240 | 0.980 | 0.471 | 0.329 | 0.990 | 0.426 | 0.336 |
| Gaussian | 1.090 | 0.406 | 0.077 | 0.840 | 0.406 | 0.357 | 0.610 | 0.432 | 0.133 | 0.680 | 0.142 | 0.000 | 0.850 | 0.401 | 0.293 | 0.670 | 0.512 | 0.222 | 0.950 | 0.447 | 0.143 | 0.900 | 0.549 | 0.333 | 0.900 | 0.411 | 0.000 | 0.860 | 0.433 | 0.280 | 0.860 | 0.467 | 0.286 | 0.860 | 0.422 | 0.289 |
| Gaussian Blur | 1.080 | 0.390 | 0.231 | 0.720 | 0.392 | 0.314 | 0.600 | 0.423 | 0.133 | 0.940 | 0.140 | 0.000 | 0.790 | 0.387 | 0.293 | 0.460 | 0.484 | 0.222 | 0.930 | 0.449 | 0.143 | 0.950 | 0.554 | 0.267 | 0.830 | 0.431 | 0.000 | 0.830 | 0.432 | 0.280 | 0.870 | 0.465 | 0.257 | 0.820 | 0.411 | 0.283 |
| Glass Blur | 1.150 | 0.408 | 0.231 | 0.830 | 0.401 | 0.371 | 0.690 | 0.429 | 0.200 | 0.810 | 0.145 | 0.000 | 0.880 | 0.397 | 0.337 | 0.650 | 0.508 | 0.111 | 1.030 | 0.463 | 0.143 | 1.040 | 0.553 | 0.333 | 0.780 | 0.425 | 0.000 | 0.780 | 0.428 | 0.280 | 0.900 | 0.469 | 0.271 | 0.900 | 0.421 | 0.322 |
| H265 Artifacts | 0.870 | 0.385 | 0.154 | 0.760 | 0.395 | 0.414 | 0.540 | 0.415 | 0.133 | 0.810 | 0.148 | 0.000 | 0.760 | 0.387 | 0.337 | 0.570 | 0.502 | 0.222 | 0.910 | 0.455 | 0.143 | 0.990 | 0.545 | 0.367 | 0.890 | 0.414 | 0.000 | 0.770 | 0.424 | 0.280 | 0.840 | 0.462 | 0.300 | 0.790 | 0.411 | 0.329 |
| Impulse | 0.700 | 0.338 | 0.231 | 0.630 | 0.378 | 0.286 | 0.480 | 0.413 | 0.000 | 0.470 | 0.137 | 0.000 | 0.620 | 0.367 | 0.250 | 0.470 | 0.478 | 0.000 | 0.860 | 0.422 | 0.143 | 0.860 | 0.544 | 0.300 | 0.600 | 0.385 | 0.000 | 0.750 | 0.415 | 0.280 | 0.740 | 0.444 | 0.243 | 0.660 | 0.391 | 0.263 |
| Jpeg Artifact | 1.170 | 0.413 | 0.154 | 0.640 | 0.407 | 0.329 | 0.670 | 0.431 | 0.067 | 0.890 | 0.143 | 0.000 | 0.940 | 0.404 | 0.283 | 0.650 | 0.517 | 0.222 | 1.020 | 0.450 | 0.143 | 1.020 | 0.555 | 0.400 | 0.840 | 0.431 | 0.000 | 0.940 | 0.432 | 0.280 | 0.950 | 0.471 | 0.314 | 0.950 | 0.425 | 0.296 |
| Motion Blur | 0.910 | 0.378 | 0.231 | 0.650 | 0.381 | 0.271 | 0.620 | 0.430 | 0.133 | 1.050 | 0.130 | 0.000 | 0.720 | 0.379 | 0.250 | 0.420 | 0.484 | 0.222 | 0.960 | 0.441 | 0.143 | 1.030 | 0.556 | 0.300 | 0.840 | 0.413 | 0.000 | 0.860 | 0.433 | 0.280 | 0.860 | 0.462 | 0.271 | 0.770 | 0.406 | 0.263 |
| Overexposure | 1.280 | 0.423 | 0.231 | 0.990 | 0.409 | 0.357 | 0.570 | 0.430 | 0.133 | 1.000 | 0.139 | 0.000 | 1.010 | 0.406 | 0.326 | 0.630 | 0.502 | 0.333 | 1.190 | 0.457 | 0.143 | 1.050 | 0.541 | 0.400 | 1.170 | 0.435 | 0.000 | 0.980 | 0.426 | 0.280 | 0.980 | 0.467 | 0.300 | 1.010 | 0.427 | 0.316 |
| Poisson | 1.090 | 0.403 | 0.231 | 0.780 | 0.396 | 0.357 | 0.680 | 0.422 | 0.133 | 1.000 | 0.138 | 0.000 | 0.810 | 0.393 | 0.315 | 0.640 | 0.499 | 0.222 | 1.110 | 0.442 | 0.143 | 1.050 | 0.545 | 0.400 | 0.980 | 0.428 | 0.000 | 0.890 | 0.428 | 0.240 | 0.850 | 0.462 | 0.300 | 0.850 | 0.415 | 0.309 |
| Rainy | 1.270 | 0.430 | 0.154 | 0.980 | 0.411 | 0.429 | 0.750 | 0.444 | 0.133 | 0.950 | 0.140 | 0.000 | 1.020 | 0.411 | 0.359 | 0.720 | 0.503 | 0.333 | 1.120 | 0.466 | 0.143 | 0.940 | 0.552 | 0.433 | 1.060 | 0.415 | 0.000 | 0.880 | 0.436 | 0.240 | 0.950 | 0.470 | 0.329 | 1.010 | 0.430 | 0.342 |
| Random Block | 1.220 | 0.421 | 0.231 | 0.950 | 0.410 | 0.357 | 0.810 | 0.439 | 0.067 | 1.160 | 0.141 | 0.000 | 1.080 | 0.408 | 0.315 | 0.700 | 0.512 | 0.111 | 1.080 | 0.468 | 0.143 | 1.080 | 0.554 | 0.433 | 0.840 | 0.424 | 0.000 | 0.960 | 0.435 | 0.240 | 0.980 | 0.474 | 0.314 | 0.980 | 0.424 | 0.322 |
| Reflect | 1.270 | 0.425 | 0.231 | 0.980 | 0.417 | 0.386 | 0.680 | 0.438 | 0.067 | 0.950 | 0.134 | 0.000 | 1.010 | 0.412 | 0.337 | 0.760 | 0.506 | 0.333 | 1.140 | 0.458 | 0.143 | 1.010 | 0.553 | 0.367 | 1.040 | 0.426 | 0.000 | 0.890 | 0.438 | 0.160 | 0.970 | 0.473 | 0.271 | 1.010 | 0.433 | 0.309 |
| Resolution Degrade | 1.050 | 0.404 | 0.154 | 0.780 | 0.397 | 0.400 | 0.580 | 0.430 | 0.200 | 0.550 | 0.131 | 0.000 | 0.810 | 0.394 | 0.359 | 0.450 | 0.496 | 0.222 | 1.020 | 0.450 | 0.143 | 0.990 | 0.537 | 0.367 | 1.000 | 0.431 | 0.000 | 0.860 | 0.439 | 0.280 | 0.950 | 0.465 | 0.300 | 0.830 | 0.417 | 0.336 |
| Rolling Shutter | 1.230 | 0.416 | 0.231 | 0.880 | 0.400 | 0.357 | 0.640 | 0.429 | 0.067 | 0.710 | 0.140 | 0.000 | 0.920 | 0.398 | 0.315 | 0.520 | 0.483 | 0.111 | 1.040 | 0.462 | 0.143 | 1.080 | 0.549 | 0.333 | 1.120 | 0.435 | 0.000 | 0.710 | 0.414 | 0.160 | 0.880 | 0.462 | 0.229 | 0.920 | 0.419 | 0.283 |
| Shadow | 1.390 | 0.444 | 0.154 | 1.000 | 0.412 | 0.414 | 0.740 | 0.437 | 0.133 | 1.100 | 0.141 | 0.000 | 1.050 | 0.412 | 0.348 | 0.670 | 0.506 | 0.222 | 1.240 | 0.465 | 0.143 | 1.020 | 0.550 | 0.400 | 1.160 | 0.440 | 0.000 | 0.860 | 0.441 | 0.240 | 0.990 | 0.474 | 0.300 | 1.050 | 0.433 | 0.329 |
| Snow | 1.350 | 0.435 | 0.154 | 1.040 | 0.413 | 0.429 | 0.810 | 0.443 | 0.133 | 0.900 | 0.130 | 0.000 | 1.080 | 0.413 | 0.359 | 0.620 | 0.499 | 0.333 | 1.220 | 0.462 | 0.143 | 1.000 | 0.548 | 0.433 | 1.060 | 0.423 | 0.000 | 0.880 | 0.436 | 0.240 | 0.980 | 0.470 | 0.329 | 1.060 | 0.432 | 0.342 |
| Speckle | 0.970 | 0.386 | 0.077 | 0.770 | 0.396 | 0.271 | 0.530 | 0.420 | 0.067 | 0.690 | 0.132 | 0.000 | 0.770 | 0.388 | 0.217 | 0.640 | 0.504 | 0.000 | 0.910 | 0.440 | 0.143 | 0.930 | 0.545 | 0.367 | 0.810 | 0.413 | 0.000 | 0.810 | 0.422 | 0.200 | 0.870 | 0.460 | 0.243 | 0.810 | 0.411 | 0.243 |
| Stretch Squish | 0.950 | 0.386 | 0.231 | 0.910 | 0.392 | 0.357 | 0.580 | 0.434 | 0.133 | 0.840 | 0.136 | 0.000 | 0.760 | 0.388 | 0.315 | 0.420 | 0.481 | 0.222 | 1.070 | 0.454 | 0.143 | 0.960 | 0.552 | 0.267 | 0.960 | 0.429 | 0.000 | 0.720 | 0.431 | 0.280 | 0.850 | 0.464 | 0.257 | 0.800 | 0.413 | 0.289 |
| Target Block | 1.300 | 0.430 | 0.231 | 0.910 | 0.404 | 0.357 | 0.780 | 0.431 | 0.133 | 1.110 | 0.144 | 0.000 | 0.980 | 0.403 | 0.326 | 0.600 | 0.495 | 0.222 | 1.160 | 0.462 | 0.143 | 1.050 | 0.547 | 0.433 | 1.050 | 0.430 | 0.000 | 0.830 | 0.434 | 0.240 | 0.960 | 0.469 | 0.314 | 0.980 | 0.425 | 0.322 |
| Temporal Jitter | 1.160 | 0.408 | 0.308 | 0.720 | 0.400 | 0.371 | 0.530 | 0.427 | 0.133 | 0.980 | 0.144 | 0.000 | 0.760 | 0.398 | 0.326 | 0.610 | 0.488 | 0.222 | 1.120 | 0.464 | 0.143 | 0.860 | 0.538 | 0.333 | 0.930 | 0.430 | 0.000 | 0.760 | 0.415 | 0.200 | 0.920 | 0.459 | 0.286 | 0.940 | 0.418 | 0.322 |
| Underexposure | 1.290 | 0.425 | 0.308 | 1.000 | 0.415 | 0.343 | 0.770 | 0.450 | 0.133 | 1.080 | 0.134 | 0.000 | 1.040 | 0.412 | 0.304 | 0.640 | 0.497 | 0.222 | 1.170 | 0.459 | 0.143 | 1.120 | 0.554 | 0.467 | 0.870 | 0.419 | 0.000 | 0.870 | 0.434 | 0.240 | 0.980 | 0.469 | 0.329 | 1.030 | 0.431 | 0.316 |
| Zoom Blur | 1.090 | 0.404 | 0.308 | 0.780 | 0.398 | 0.371 | 0.640 | 0.431 | 0.200 | 0.980 | 0.137 | 0.000 | 0.840 | 0.395 | 0.348 | 0.500 | 0.492 | 0.333 | 1.110 | 0.465 | 0.143 | 0.980 | 0.542 | 0.300 | 0.890 | 0.407 | 0.000 | 0.800 | 0.427 | 0.280 | 0.870 | 0.463 | 0.286 | 0.870 | 0.417 | 0.316 |

Table 19: Video-ChatGPT performance across different question types

| Noise Type | CP GPT | CP SBERT | CP Acc | FP-S GPT | FP-S SBERT | FP-S Acc | FP-C GPT | FP-C SBERT | FP-C Acc | HL GPT | HL SBERT | HL Acc | Mean GPT | Mean SBERT | Mean Acc | LR GPT | LR SBERT | LR Acc | AR GPT | AR SBERT | AR Acc | RR GPT | RR SBERT | RR Acc | CSR GPT | CSR SBERT | CSR Acc | TR GPT | TR SBERT | TR Acc | Mean2 GPT | Mean2 SBERT | Mean2 Acc | Overall GPT | Overall SBERT | Overall Acc |
|---|---|---|---|---|---|---|---|---|---|---|---|---|---|---|---|---|---|---|---|---|---|---|---|---|---|---|---|---|---|---|---|---|---|---|---|---|
| Clean | 0.790 | 0.396 | 0.462 | 0.760 | 0.411 | 0.500 | 0.690 | 0.445 | 0.200 | 0.340 | 0.154 | 0.333 | 0.750 | 0.404 | 0.467 | 0.590 | 0.496 | 0.333 | 0.930 | 0.477 | 0.571 | 1.060 | 0.572 | 0.400 | 0.640 | 0.414 | 0.000 | 0.780 | 0.441 | 0.560 | 0.840 | 0.471 | 0.471 | 0.780 | 0.427 | 0.467 |
| Bit Error | 0.850 | 0.399 | 0.385 | 0.780 | 0.411 | 0.500 | 0.590 | 0.442 | 0.133 | 0.290 | 0.148 | 0.333 | 0.760 | 0.404 | 0.457 | 0.540 | 0.498 | 0.222 | 0.960 | 0.461 | 0.571 | 1.110 | 0.583 | 0.400 | 0.840 | 0.418 | 0.000 | 0.800 | 0.444 | 0.640 | 0.860 | 0.478 | 0.486 | 0.800 | 0.427 | 0.474 |
| Bright Transform | 0.810 | 0.400 | 0.385 | 0.740 | 0.409 | 0.471 | 0.650 | 0.451 | 0.133 | 0.310 | 0.154 | 0.000 | 0.740 | 0.404 | 0.435 | 0.520 | 0.494 | 0.111 | 0.890 | 0.465 | 0.571 | 1.140 | 0.584 | 0.400 | 0.850 | 0.408 | 0.000 | 0.740 | 0.441 | 0.600 | 0.840 | 0.477 | 0.457 | 0.770 | 0.427 | 0.454 |
| Color Quantized | 0.830 | 0.391 | 0.308 | 0.770 | 0.410 | 0.457 | 0.680 | 0.452 | 0.067 | 0.320 | 0.151 | 0.333 | 0.770 | 0.403 | 0.413 | 0.530 | 0.497 | 0.222 | 0.810 | 0.463 | 0.571 | 1.050 | 0.584 | 0.400 | 0.700 | 0.418 | 0.000 | 0.810 | 0.441 | 0.600 | 0.810 | 0.476 | 0.471 | 0.790 | 0.426 | 0.441 |
| Color Shift | 0.760 | 0.388 | 0.308 | 0.790 | 0.408 | 0.514 | 0.650 | 0.450 | 0.200 | 0.350 | 0.159 | 0.000 | 0.760 | 0.402 | 0.446 | 0.440 | 0.496 | 0.222 | 0.930 | 0.467 | 0.571 | 1.140 | 0.590 | 0.467 | 0.750 | 0.415 | 0.000 | 0.790 | 0.442 | 0.600 | 0.830 | 0.475 | 0.478 | 0.790 | 0.426 | 0.474 |
| Contrast | 0.790 | 0.389 | 0.231 | 0.770 | 0.409 | 0.529 | 0.630 | 0.446 | 0.200 | 0.350 | 0.158 | 0.000 | 0.750 | 0.402 | 0.457 | 0.470 | 0.498 | 0.222 | 0.900 | 0.460 | 0.571 | 1.200 | 0.586 | 0.500 | 0.700 | 0.406 | 0.000 | 0.810 | 0.441 | 0.520 | 0.840 | 0.475 | 0.486 | 0.780 | 0.425 | 0.474 |
| Defocus Blur | 0.780 | 0.384 | 0.308 | 0.760 | 0.408 | 0.457 | 0.680 | 0.445 | 0.133 | 0.350 | 0.155 | 0.000 | 0.750 | 0.401 | 0.402 | 0.500 | 0.486 | 0.222 | 0.900 | 0.461 | 0.571 | 1.080 | 0.579 | 0.400 | 0.900 | 0.411 | 0.000 | 0.740 | 0.438 | 0.560 | 0.820 | 0.474 | 0.486 | 0.770 | 0.423 | 0.441 |
| Edge Sawtooth | 0.770 | 0.389 | 0.385 | 0.740 | 0.406 | 0.486 | 0.670 | 0.442 | 0.200 | 0.400 | 0.154 | 0.333 | 0.730 | 0.400 | 0.457 | 0.500 | 0.495 | 0.222 | 0.800 | 0.454 | 0.571 | 0.970 | 0.576 | 0.400 | 0.630 | 0.388 | 0.000 | 0.770 | 0.440 | 0.600 | 0.770 | 0.470 | 0.471 | 0.750 | 0.423 | 0.474 |
| Elastic | 0.790 | 0.394 | 0.462 | 0.790 | 0.409 | 0.471 | 0.690 | 0.451 | 0.200 | 0.210 | 0.151 | 0.000 | 0.770 | 0.403 | 0.435 | 0.470 | 0.500 | 0.000 | 0.880 | 0.470 | 0.571 | 1.130 | 0.585 | 0.433 | 0.640 | 0.396 | 0.000 | 0.710 | 0.435 | 0.520 | 0.800 | 0.477 | 0.429 | 0.790 | 0.427 | 0.447 |
| Flicker | 0.770 | 0.388 | 0.231 | 0.760 | 0.405 | 0.471 | 0.640 | 0.443 | 0.133 | 0.400 | 0.154 | 0.000 | 0.740 | 0.398 | 0.391 | 0.440 | 0.490 | 0.222 | 0.830 | 0.462 | 0.571 | 1.100 | 0.586 | 0.400 | 0.700 | 0.403 | 0.000 | 0.760 | 0.438 | 0.480 | 0.800 | 0.473 | 0.429 | 0.760 | 0.422 | 0.414 |
| Foggy | 0.800 | 0.392 | 0.385 | 0.710 | 0.403 | 0.471 | 0.620 | 0.444 | 0.133 | 0.230 | 0.154 | 0.000 | 0.710 | 0.399 | 0.413 | 0.490 | 0.483 | 0.000 | 0.890 | 0.463 | 0.571 | 1.120 | 0.585 | 0.400 | 0.720 | 0.408 | 0.000 | 0.740 | 0.435 | 0.600 | 0.810 | 0.473 | 0.443 | 0.790 | 0.422 | 0.441 |
| Frame Drop | 0.540 | 0.335 | 0.231 | 0.630 | 0.382 | 0.486 | 0.550 | 0.419 | 0.267 | 0.610 | 0.152 | 0.333 | 0.610 | 0.371 | 0.435 | 0.470 | 0.480 | 0.111 | 0.600 | 0.418 | 0.571 | 0.810 | 0.558 | 0.400 | 0.470 | 0.348 | 0.000 | 0.690 | 0.431 | 0.480 | 0.650 | 0.450 | 0.414 | 0.630 | 0.397 | 0.441 |
| Frame Repeat | 0.760 | 0.376 | 0.231 | 0.830 | 0.406 | 0.500 | 0.630 | 0.433 | 0.200 | 0.160 | 0.148 | 0.000 | 0.770 | 0.395 | 0.435 | 0.480 | 0.490 | 0.222 | 0.570 | 0.470 | 0.571 | 0.980 | 0.572 | 0.433 | 0.570 | 0.385 | 0.000 | 0.740 | 0.443 | 0.520 | 0.750 | 0.469 | 0.457 | 0.770 | 0.418 | 0.454 |
| Frame Replace | 0.830 | 0.390 | 0.385 | 0.790 | 0.412 | 0.500 | 0.630 | 0.444 | 0.267 | 0.420 | 0.150 | 0.333 | 0.770 | 0.403 | 0.457 | 0.590 | 0.498 | 0.333 | 0.990 | 0.465 | 0.571 | 1.110 | 0.581 | 0.433 | 0.890 | 0.416 | 0.000 | 0.790 | 0.440 | 0.560 | 0.880 | 0.478 | 0.486 | 0.800 | 0.426 | 0.467 |
| Frost | 0.700 | 0.380 | 0.231 | 0.720 | 0.402 | 0.457 | 0.630 | 0.444 | 0.133 | 0.260 | 0.153 | 0.000 | 0.710 | 0.396 | 0.402 | 0.530 | 0.497 | 0.000 | 0.740 | 0.447 | 0.571 | 0.990 | 0.577 | 0.400 | 0.790 | 0.397 | 0.000 | 0.740 | 0.433 | 0.560 | 0.760 | 0.469 | 0.429 | 0.730 | 0.419 | 0.434 |
| Gaussian | 0.700 | 0.369 | 0.308 | 0.720 | 0.394 | 0.529 | 0.620 | 0.442 | 0.133 | 0.290 | 0.158 | 0.333 | 0.710 | 0.388 | 0.467 | 0.450 | 0.491 | 0.222 | 0.710 | 0.445 | 0.571 | 0.970 | 0.568 | 0.400 | 0.720 | 0.397 | 0.000 | 0.720 | 0.440 | 0.520 | 0.740 | 0.463 | 0.457 | 0.710 | 0.412 | 0.434 |
| Gaussian Blur | 0.680 | 0.377 | 0.385 | 0.680 | 0.394 | 0.486 | 0.560 | 0.437 | 0.200 | 0.390 | 0.154 | 0.333 | 0.670 | 0.389 | 0.435 | 0.490 | 0.487 | 0.222 | 0.750 | 0.438 | 0.571 | 1.040 | 0.577 | 0.400 | 0.600 | 0.389 | 0.000 | 0.750 | 0.435 | 0.520 | 0.760 | 0.464 | 0.443 | 0.700 | 0.412 | 0.447 |
| Glass Blur | 0.790 | 0.388 | 0.538 | 0.760 | 0.409 | 0.486 | 0.670 | 0.444 | 0.200 | 0.350 | 0.156 | 0.333 | 0.750 | 0.402 | 0.467 | 0.550 | 0.496 | 0.111 | 0.960 | 0.466 | 0.571 | 1.170 | 0.580 | 0.400 | 0.740 | 0.411 | 0.000 | 0.810 | 0.439 | 0.600 | 0.870 | 0.476 | 0.457 | 0.750 | 0.425 | 0.474 |
| H265 Artifacts | 0.720 | 0.379 | 0.385 | 0.720 | 0.401 | 0.443 | 0.540 | 0.438 | 0.067 | 0.320 | 0.150 | 0.333 | 0.690 | 0.394 | 0.402 | 0.470 | 0.488 | 0.111 | 0.790 | 0.450 | 0.571 | 1.060 | 0.576 | 0.433 | 0.750 | 0.395 | 0.000 | 0.740 | 0.437 | 0.560 | 0.780 | 0.468 | 0.457 | 0.730 | 0.417 | 0.434 |
| Impulse | 0.560 | 0.356 | 0.231 | 0.630 | 0.385 | 0.443 | 0.560 | 0.428 | 0.133 | 0.290 | 0.149 | 0.000 | 0.610 | 0.378 | 0.391 | 0.330 | 0.484 | 0.000 | 0.590 | 0.424 | 0.571 | 0.830 | 0.580 | 0.433 | 0.800 | 0.369 | 0.000 | 0.780 | 0.434 | 0.560 | 0.660 | 0.454 | 0.443 | 0.630 | 0.403 | 0.434 |
| Jpeg Artifact | 0.770 | 0.387 | 0.462 | 0.750 | 0.405 | 0.457 | 0.620 | 0.431 | 0.133 | 0.340 | 0.147 | 0.000 | 0.720 | 0.398 | 0.467 | 0.470 | 0.497 | 0.222 | 0.860 | 0.468 | 0.571 | 1.070 | 0.580 | 0.400 | 0.740 | 0.412 | 0.000 | 0.740 | 0.432 | 0.560 | 0.800 | 0.476 | 0.471 | 0.770 | 0.422 | 0.474 |
| Motion Blur | 0.700 | 0.375 | 0.308 | 0.680 | 0.397 | 0.471 | 0.620 | 0.431 | 0.133 | 0.310 | 0.151 | 0.000 | 0.670 | 0.390 | 0.413 | 0.470 | 0.491 | 0.111 | 0.790 | 0.439 | 0.571 | 0.910 | 0.577 | 0.433 | 0.910 | 0.386 | 0.000 | 0.740 | 0.447 | 0.560 | 0.730 | 0.464 | 0.443 | 0.750 | 0.414 | 0.434 |
| Overexposure | 0.760 | 0.385 | 0.385 | 0.750 | 0.405 | 0.457 | 0.550 | 0.442 | 0.133 | 0.320 | 0.151 | 0.000 | 0.720 | 0.398 | 0.413 | 0.360 | 0.492 | 0.000 | 0.860 | 0.458 | 0.571 | 1.080 | 0.584 | 0.433 | 0.580 | 0.404 | 0.000 | 0.810 | 0.433 | 0.600 | 0.760 | 0.475 | 0.486 | 0.760 | 0.423 | 0.447 |
| Poisson | 0.620 | 0.353 | 0.308 | 0.660 | 0.390 | 0.443 | 0.610 | 0.443 | 0.200 | 0.310 | 0.149 | 0.000 | 0.630 | 0.381 | 0.413 | 0.460 | 0.493 | 0.000 | 0.530 | 0.432 | 0.571 | 0.910 | 0.566 | 0.400 | 0.530 | 0.374 | 0.000 | 0.680 | 0.447 | 0.600 | 0.680 | 0.457 | 0.443 | 0.660 | 0.405 | 0.434 |
| Rainy | 0.770 | 0.388 | 0.154 | 0.730 | 0.405 | 0.443 | 0.650 | 0.444 | 0.200 | 0.420 | 0.158 | 0.000 | 0.720 | 0.399 | 0.380 | 0.480 | 0.494 | 0.222 | 0.890 | 0.460 | 0.571 | 0.950 | 0.568 | 0.433 | 0.770 | 0.416 | 0.000 | 0.790 | 0.444 | 0.520 | 0.750 | 0.474 | 0.429 | 0.750 | 0.423 | 0.414 |
| Random Block | 0.770 | 0.383 | 0.385 | 0.750 | 0.401 | 0.486 | 0.630 | 0.441 | 0.133 | 0.350 | 0.151 | 0.000 | 0.720 | 0.395 | 0.435 | 0.470 | 0.500 | 0.111 | 0.860 | 0.462 | 0.571 | 1.090 | 0.585 | 0.467 | 0.640 | 0.404 | 0.000 | 0.810 | 0.433 | 0.640 | 0.810 | 0.477 | 0.500 | 0.750 | 0.421 | 0.474 |
| Reflect | 0.750 | 0.385 | 0.385 | 0.760 | 0.408 | 0.514 | 0.580 | 0.443 | 0.133 | 0.310 | 0.153 | 0.333 | 0.730 | 0.400 | 0.467 | 0.480 | 0.494 | 0.444 | 0.860 | 0.463 | 0.571 | 1.070 | 0.582 | 0.433 | 0.900 | 0.420 | 0.000 | 0.820 | 0.436 | 0.560 | 0.830 | 0.475 | 0.486 | 0.760 | 0.423 | 0.474 |
| Resolution Degrade | 0.810 | 0.389 | 0.385 | 0.750 | 0.406 | 0.500 | 0.610 | 0.445 | 0.133 | 0.420 | 0.149 | 0.333 | 0.750 | 0.400 | 0.446 | 0.520 | 0.498 | 0.333 | 0.990 | 0.469 | 0.714 | 1.110 | 0.577 | 0.433 | 0.720 | 0.412 | 0.000 | 0.810 | 0.440 | 0.640 | 0.860 | 0.480 | 0.514 | 0.790 | 0.425 | 0.428 |
| Rolling Shutter | 0.840 | 0.388 | 0.385 | 0.790 | 0.409 | 0.429 | 0.660 | 0.446 | 0.267 | 0.230 | 0.152 | 0.000 | 0.760 | 0.401 | 0.413 | 0.460 | 0.488 | 0.000 | 0.880 | 0.464 | 0.571 | 1.000 | 0.582 | 0.467 | 0.700 | 0.403 | 0.000 | 0.740 | 0.444 | 0.520 | 0.790 | 0.471 | 0.400 | 0.780 | 0.423 | 0.461 |
| Shadow | 0.790 | 0.390 | 0.462 | 0.780 | 0.408 | 0.529 | 0.650 | 0.437 | 0.267 | 0.420 | 0.151 | 0.000 | 0.760 | 0.401 | 0.478 | 0.500 | 0.493 | 0.222 | 0.930 | 0.475 | 0.571 | 1.110 | 0.584 | 0.433 | 0.730 | 0.409 | 0.000 | 0.830 | 0.431 | 0.560 | 0.860 | 0.479 | 0.500 | 0.800 | 0.426 | 0.493 |
| Snow | 0.810 | 0.396 | 0.385 | 0.780 | 0.409 | 0.471 | 0.670 | 0.457 | 0.133 | 0.320 | 0.151 | 0.333 | 0.770 | 0.403 | 0.435 | 0.470 | 0.488 | 0.333 | 1.000 | 0.479 | 0.571 | 1.040 | 0.576 | 0.467 | 0.810 | 0.397 | 0.000 | 0.730 | 0.444 | 0.560 | 0.840 | 0.478 | 0.486 | 0.790 | 0.426 | 0.461 |
| Speckle | 0.730 | 0.377 | 0.385 | 0.720 | 0.401 | 0.500 | 0.620 | 0.441 | 0.133 | 0.260 | 0.153 | 0.333 | 0.700 | 0.393 | 0.424 | 0.370 | 0.497 | 0.222 | 0.840 | 0.459 | 0.571 | 0.840 | 0.570 | 0.467 | 0.650 | 0.393 | 0.000 | 0.710 | 0.431 | 0.560 | 0.750 | 0.470 | 0.486 | 0.720 | 0.418 | 0.461 |
| Stretch Squish | 0.770 | 0.383 | 0.308 | 0.760 | 0.401 | 0.514 | 0.640 | 0.440 | 0.200 | 0.320 | 0.151 | 0.333 | 0.740 | 0.395 | 0.457 | 0.600 | 0.457 | 0.333 | 0.810 | 0.452 | 0.571 | 1.010 | 0.576 | 0.433 | 0.700 | 0.398 | 0.000 | 0.760 | 0.436 | 0.520 | 0.800 | 0.470 | 0.471 | 0.760 | 0.419 | 0.461 |
| Target Block | 0.730 | 0.375 | 0.385 | 0.720 | 0.402 | 0.514 | 0.640 | 0.431 | 0.200 | 0.350 | 0.145 | 0.000 | 0.700 | 0.393 | 0.457 | 0.500 | 0.494 | 0.222 | 0.710 | 0.441 | 0.571 | 1.050 | 0.564 | 0.400 | 0.560 | 0.400 | 0.000 | 0.720 | 0.444 | 0.600 | 0.730 | 0.471 | 0.471 | 0.740 | 0.418 | 0.467 |
| Temporal Jitter | 0.690 | 0.365 | 0.385 | 0.670 | 0.392 | 0.457 | 0.640 | 0.434 | 0.200 | 0.440 | 0.151 | 0.000 | 0.670 | 0.384 | 0.457 | 0.450 | 0.478 | 0.000 | 0.940 | 0.465 | 0.571 | 0.960 | 0.591 | 0.367 | 0.620 | 0.379 | 0.000 | 0.800 | 0.431 | 0.600 | 0.810 | 0.458 | 0.457 | 0.700 | 0.408 | 0.467 |
| Underexposure | 0.810 | 0.391 | 0.154 | 0.760 | 0.409 | 0.457 | 0.640 | 0.444 | 0.200 | 0.340 | 0.151 | 0.333 | 0.750 | 0.402 | 0.402 | 0.380 | 0.491 | 0.222 | 0.800 | 0.465 | 0.571 | 1.050 | 0.591 | 0.367 | 0.700 | 0.416 | 0.000 | 0.800 | 0.444 | 0.560 | 0.730 | 0.479 | 0.443 | 0.770 | 0.426 | 0.428 |
| Zoom Blur | 0.670 | 0.371 | 0.385 | 0.700 | 0.398 | 0.500 | 0.580 | 0.442 | 0.200 | 0.370 | 0.155 | 0.333 | 0.680 | 0.391 | 0.457 | 0.440 | 0.496 | 0.222 | 0.710 | 0.444 | 0.571 | 0.880 | 0.569 | 0.367 | 0.530 | 0.368 | 0.000 | 0.730 | 0.434 | 0.560 | 0.700 | 0.463 | 0.443 | 0.700 | 0.415 | 0.454 |

Table 20: VideoChat2-HD performance across different question types

| Noise Type | CP GPT | CP SBERT | CP Acc | FP-S GPT | FP-S SBERT | FP-S Acc | FP-C GPT | FP-C SBERT | FP-C Acc | HL GPT | HL SBERT | HL Acc | Mean GPT | Mean SBERT | Mean Acc | LR GPT | LR SBERT | LR Acc | AR GPT | AR SBERT | AR Acc | RR GPT | RR SBERT | RR Acc | CSR GPT | CSR SBERT | CSR Acc | TR GPT | TR SBERT | TR Acc | Mean2 GPT | Mean2 SBERT | Mean2 Acc | Overall GPT | Overall SBERT | Overall Acc |
|---|---|---|---|---|---|---|---|---|---|---|---|---|---|---|---|---|---|---|---|---|---|---|---|---|---|---|---|---|---|---|---|---|---|---|---|---|
| Clean | 1.310 | 0.445 | 0.846 | 1.070 | 0.441 | 0.600 | 0.990 | 0.483 | 0.400 | 0.440 | 0.141 | 0.333 | 1.090 | 0.437 | 0.598 | 0.610 | 0.519 | 0.333 | 1.410 | 0.511 | 0.714 | 1.170 | 0.593 | 0.333 | 1.050 | 0.419 | 0.000 | 1.210 | 0.470 | 0.680 | 1.140 | 0.503 | 0.471 | 1.110 | 0.459 | 0.553 |
| Bit Error | 1.250 | 0.440 | 0.846 | 1.040 | 0.440 | 0.643 | 0.930 | 0.470 | 0.400 | 0.560 | 0.149 | 0.333 | 1.050 | 0.434 | 0.630 | 0.640 | 0.524 | 0.444 | 1.370 | 0.503 | 0.714 | 1.160 | 0.586 | 0.400 | 1.000 | 0.420 | 1.000 | 1.200 | 0.473 | 0.760 | 1.130 | 0.502 | 0.571 | 1.080 | 0.456 | 0.605 |
| Bright Transform | 1.320 | 0.441 | 0.846 | 1.020 | 0.439 | 0.629 | 0.940 | 0.478 | 0.400 | 0.420 | 0.142 | 0.333 | 1.050 | 0.434 | 0.620 | 0.610 | 0.524 | 0.333 | 1.340 | 0.501 | 0.857 | 1.140 | 0.593 | 0.367 | 1.010 | 0.429 | 0.500 | 1.170 | 0.470 | 0.680 | 1.110 | 0.503 | 0.529 | 1.070 | 0.456 | 0.586 |
| Color Quantized | 1.340 | 0.440 | 0.846 | 1.030 | 0.440 | 0.614 | 0.980 | 0.478 | 0.400 | 0.530 | 0.146 | 0.333 | 1.070 | 0.435 | 0.609 | 0.650 | 0.519 | 0.444 | 1.340 | 0.507 | 0.857 | 1.270 | 0.593 | 0.367 | 1.070 | 0.424 | 0.000 | 1.200 | 0.471 | 0.720 | 1.150 | 0.503 | 0.529 | 1.100 | 0.458 | 0.572 |
| Color Shift | 1.270 | 0.440 | 0.769 | 1.010 | 0.438 | 0.629 | 0.930 | 0.478 | 0.400 | 0.420 | 0.145 | 0.000 | 1.020 | 0.433 | 0.609 | 0.610 | 0.521 | 0.444 | 1.320 | 0.497 | 0.857 | 1.190 | 0.590 | 0.367 | 1.020 | 0.429 | 0.000 | 1.110 | 0.469 | 0.720 | 1.050 | 0.500 | 0.529 | 1.050 | 0.455 | 0.579 |
| Contrast | 1.300 | 0.444 | 0.846 | 1.040 | 0.439 | 0.657 | 0.990 | 0.476 | 0.400 | 0.440 | 0.142 | 0.000 | 1.070 | 0.435 | 0.641 | 0.690 | 0.516 | 0.444 | 1.250 | 0.498 | 0.857 | 1.200 | 0.594 | 0.433 | 1.060 | 0.425 | 0.000 | 1.090 | 0.470 | 0.720 | 1.080 | 0.500 | 0.557 | 1.080 | 0.456 | 0.605 |
| Defocus Blur | 1.250 | 0.434 | 0.846 | 1.000 | 0.435 | 0.586 | 0.980 | 0.481 | 0.400 | 0.650 | 0.153 | 0.333 | 1.040 | 0.432 | 0.587 | 0.500 | 0.515 | 0.222 | 1.340 | 0.497 | 0.714 | 1.200 | 0.589 | 0.367 | 0.990 | 0.428 | 0.500 | 1.060 | 0.468 | 0.600 | 1.060 | 0.499 | 0.471 | 1.050 | 0.453 | 0.553 |
| Edge Sawtooth | 1.230 | 0.438 | 0.769 | 1.000 | 0.435 | 0.614 | 0.980 | 0.475 | 0.400 | 0.630 | 0.150 | 0.333 | 1.030 | 0.432 | 0.609 | 0.570 | 0.520 | 0.222 | 1.290 | 0.491 | 0.857 | 1.200 | 0.591 | 0.400 | 1.010 | 0.420 | 1.000 | 1.140 | 0.469 | 0.720 | 1.080 | 0.498 | 0.543 | 1.060 | 0.453 | 0.592 |
| Elastic | 1.330 | 0.444 | 0.846 | 1.020 | 0.440 | 0.586 | 1.000 | 0.479 | 0.400 | 0.480 | 0.141 | 0.333 | 1.060 | 0.436 | 0.587 | 0.590 | 0.518 | 0.222 | 1.370 | 0.505 | 0.857 | 1.170 | 0.595 | 0.333 | 0.960 | 0.417 | 0.500 | 1.150 | 0.467 | 0.680 | 1.100 | 0.500 | 0.486 | 1.080 | 0.457 | 0.553 |
| Flicker | 1.290 | 0.441 | 0.923 | 1.010 | 0.437 | 0.629 | 0.970 | 0.482 | 0.333 | 0.560 | 0.150 | 0.333 | 1.040 | 0.434 | 0.630 | 0.660 | 0.524 | 0.444 | 1.240 | 0.497 | 0.714 | 1.170 | 0.595 | 0.333 | 1.170 | 0.412 | 0.000 | 1.170 | 0.473 | 0.720 | 1.170 | 0.501 | 0.500 | 1.060 | 0.456 | 0.572 |
| Foggy | 1.240 | 0.439 | 0.692 | 1.010 | 0.439 | 0.629 | 0.950 | 0.469 | 0.400 | 0.450 | 0.142 | 0.000 | 1.030 | 0.433 | 0.609 | 0.670 | 0.522 | 0.556 | 1.290 | 0.497 | 0.857 | 1.180 | 0.596 | 0.400 | 0.860 | 0.430 | 0.000 | 1.190 | 0.469 | 0.720 | 1.100 | 0.502 | 0.557 | 1.060 | 0.456 | 0.586 |
| Frame Drop | 1.020 | 0.404 | 0.846 | 0.830 | 0.416 | 0.571 | 0.740 | 0.451 | 0.333 | 0.690 | 0.156 | 0.333 | 0.860 | 0.411 | 0.587 | 0.540 | 0.514 | 0.444 | 1.070 | 0.464 | 0.714 | 1.080 | 0.574 | 0.400 | 0.830 | 0.406 | 0.500 | 1.060 | 0.454 | 0.640 | 0.960 | 0.481 | 0.529 | 0.900 | 0.433 | 0.566 |
| Frame Repeat | 0.970 | 0.392 | 0.692 | 0.790 | 0.413 | 0.486 | 0.860 | 0.467 | 0.200 | 0.560 | 0.156 | 0.333 | 0.830 | 0.409 | 0.467 | 0.560 | 0.516 | 0.444 | 1.090 | 0.465 | 0.714 | 1.050 | 0.566 | 0.433 | 0.830 | 0.403 | 0.000 | 0.870 | 0.448 | 0.560 | 0.900 | 0.477 | 0.486 | 0.860 | 0.431 | 0.480 |
| Frame Replace | 1.270 | 0.437 | 0.923 | 1.070 | 0.443 | 0.657 | 1.060 | 0.478 | 0.400 | 0.450 | 0.140 | 0.000 | 1.080 | 0.436 | 0.652 | 0.630 | 0.521 | 0.444 | 1.430 | 0.508 | 0.714 | 1.220 | 0.598 | 0.333 | 0.890 | 0.406 | 0.000 | 1.190 | 0.475 | 0.720 | 1.130 | 0.504 | 0.500 | 1.110 | 0.459 | 0.592 |
| Frost | 1.180 | 0.430 | 0.769 | 1.010 | 0.436 | 0.657 | 0.910 | 0.469 | 0.467 | 0.310 | 0.140 | 0.000 | 1.010 | 0.430 | 0.641 | 0.650 | 0.527 | 0.556 | 1.270 | 0.491 | 0.857 | 1.140 | 0.577 | 0.400 | 0.790 | 0.413 | 0.000 | 1.170 | 0.465 | 0.680 | 1.040 | 0.495 | 0.543 | 1.040 | 0.451 | 0.599 |
| Gaussian | 1.060 | 0.416 | 0.692 | 0.890 | 0.428 | 0.571 | 0.750 | 0.459 | 0.133 | 0.770 | 0.147 | 0.333 | 0.910 | 0.421 | 0.533 | 0.550 | 0.515 | 0.333 | 1.150 | 0.479 | 0.857 | 1.200 | 0.588 | 0.467 | 0.930 | 0.426 | 0.000 | 1.010 | 0.464 | 0.640 | 1.000 | 0.493 | 0.514 | 0.940 | 0.444 | 0.546 |
| Gaussian Blur | 1.130 | 0.413 | 0.769 | 0.900 | 0.422 | 0.571 | 0.870 | 0.454 | 0.333 | 0.610 | 0.157 | 0.333 | 0.940 | 0.418 | 0.576 | 0.520 | 0.512 | 0.556 | 1.140 | 0.475 | 0.714 | 1.090 | 0.576 | 0.367 | 0.830 | 0.404 | 0.000 | 1.040 | 0.461 | 0.640 | 0.970 | 0.485 | 0.514 | 0.960 | 0.440 | 0.546 |
| Glass Blur | 1.310 | 0.440 | 0.846 | 1.040 | 0.435 | 0.600 | 1.010 | 0.472 | 0.400 | 0.550 | 0.151 | 0.333 | 1.070 | 0.431 | 0.598 | 0.460 | 0.524 | 0.111 | 1.400 | 0.501 | 0.714 | 1.230 | 0.588 | 0.367 | 0.960 | 0.427 | 0.500 | 1.160 | 0.470 | 0.720 | 1.100 | 0.501 | 0.471 | 1.090 | 0.454 | 0.559 |
| H265 Artifacts | 1.090 | 0.418 | 0.692 | 0.940 | 0.421 | 0.614 | 0.880 | 0.464 | 0.200 | 0.680 | 0.156 | 0.000 | 0.950 | 0.417 | 0.576 | 0.680 | 0.523 | 0.444 | 1.080 | 0.473 | 0.714 | 1.140 | 0.585 | 0.433 | 0.960 | 0.411 | 0.500 | 0.990 | 0.454 | 0.560 | 0.990 | 0.487 | 0.514 | 0.970 | 0.440 | 0.546 |
| Impulse | 1.000 | 0.412 | 0.692 | 0.850 | 0.416 | 0.600 | 0.780 | 0.451 | 0.333 | 0.870 | 0.157 | 0.333 | 0.880 | 0.412 | 0.576 | 0.430 | 0.509 | 0.333 | 1.040 | 0.470 | 0.714 | 1.130 | 0.574 | 0.433 | 1.010 | 0.404 | 0.500 | 0.930 | 0.463 | 0.600 | 0.900 | 0.485 | 0.514 | 0.900 | 0.435 | 0.553 |
| Jpeg Artifact | 1.150 | 0.427 | 0.615 | 0.920 | 0.433 | 0.600 | 0.890 | 0.467 | 0.267 | 0.530 | 0.150 | 0.333 | 0.930 | 0.412 | 0.576 | 0.430 | 0.519 | 0.333 | 1.230 | 0.489 | 0.857 | 1.140 | 0.596 | 0.400 | 1.010 | 0.407 | 0.500 | 1.130 | 0.461 | 0.720 | 1.070 | 0.494 | 0.557 | 1.020 | 0.449 | 0.566 |
| Motion Blur | 1.120 | 0.415 | 0.769 | 0.920 | 0.425 | 0.543 | 0.810 | 0.454 | 0.267 | 0.760 | 0.153 | 0.333 | 0.930 | 0.419 | 0.543 | 0.490 | 0.508 | 0.556 | 1.110 | 0.477 | 0.714 | 1.110 | 0.576 | 0.433 | 1.070 | 0.418 | 0.500 | 1.070 | 0.457 | 0.680 | 1.040 | 0.484 | 0.543 | 0.990 | 0.440 | 0.539 |
| Overexposure | 1.230 | 0.436 | 0.923 | 1.010 | 0.436 | 0.600 | 0.880 | 0.469 | 0.400 | 0.420 | 0.145 | 0.333 | 1.010 | 0.430 | 0.641 | 0.600 | 0.517 | 0.444 | 1.320 | 0.494 | 0.857 | 1.140 | 0.588 | 0.477 | 1.160 | 0.421 | 0.000 | 1.090 | 0.469 | 0.760 | 1.090 | 0.497 | 0.529 | 1.040 | 0.451 | 0.592 |
| Poisson | 1.030 | 0.433 | 0.615 | 0.870 | 0.423 | 0.571 | 0.740 | 0.450 | 0.400 | 0.760 | 0.150 | 0.333 | 0.880 | 0.416 | 0.630 | 0.580 | 0.521 | 0.333 | 1.080 | 0.477 | 0.714 | 1.080 | 0.576 | 0.433 | 0.960 | 0.427 | 0.500 | 0.960 | 0.463 | 0.760 | 1.030 | 0.488 | 0.557 | 0.920 | 0.439 | 0.553 |
| Rainy | 1.210 | 0.431 | 0.692 | 1.040 | 0.441 | 0.600 | 1.000 | 0.474 | 0.400 | 0.450 | 0.143 | 0.000 | 1.060 | 0.434 | 0.587 | 0.580 | 0.521 | 0.444 | 1.390 | 0.501 | 0.714 | 1.190 | 0.595 | 0.333 | 1.100 | 0.421 | 0.000 | 1.080 | 0.468 | 0.720 | 1.080 | 0.500 | 0.514 | 1.070 | 0.456 | 0.553 |
| Random Block | 1.220 | 0.440 | 0.923 | 1.000 | 0.439 | 0.657 | 1.000 | 0.476 | 0.467 | 0.630 | 0.152 | 0.333 | 1.060 | 0.434 | 0.652 | 0.610 | 0.522 | 0.333 | 1.260 | 0.500 | 0.714 | 1.260 | 0.591 | 0.400 | 0.940 | 0.418 | 0.500 | 1.120 | 0.466 | 0.560 | 1.100 | 0.500 | 0.543 | 1.080 | 0.456 | 0.612 |
| Reflect | 1.210 | 0.433 | 0.846 | 1.000 | 0.438 | 0.643 | 0.950 | 0.471 | 0.400 | 0.440 | 0.143 | 0.333 | 1.020 | 0.432 | 0.630 | 0.640 | 0.519 | 0.556 | 1.340 | 0.495 | 0.857 | 1.210 | 0.590 | 0.367 | 0.910 | 0.414 | 0.500 | 1.130 | 0.463 | 0.760 | 1.090 | 0.496 | 0.557 | 1.050 | 0.453 | 0.599 |
| Resolution Degrade | 1.210 | 0.429 | 0.769 | 0.970 | 0.430 | 0.600 | 0.970 | 0.474 | 0.400 | 0.660 | 0.152 | 0.333 | 1.010 | 0.427 | 0.598 | 0.440 | 0.510 | 0.556 | 1.320 | 0.494 | 0.714 | 1.230 | 0.593 | 0.367 | 0.910 | 0.418 | 0.500 | 1.040 | 0.462 | 0.560 | 1.040 | 0.494 | 0.443 | 1.030 | 0.449 | 0.546 |
| Rolling Shutter | 1.210 | 0.430 | 0.769 | 0.950 | 0.429 | 0.657 | 1.000 | 0.473 | 0.400 | 0.400 | 0.149 | 0.333 | 1.040 | 0.427 | 0.652 | 0.650 | 0.520 | 0.556 | 1.300 | 0.495 | 0.714 | 1.140 | 0.596 | 0.400 | 0.950 | 0.417 | 0.000 | 1.030 | 0.473 | 0.720 | 1.030 | 0.501 | 0.495 | 1.010 | 0.449 | 0.579 |
| Shadow | 1.370 | 0.452 | 0.923 | 1.060 | 0.440 | 0.643 | 1.010 | 0.478 | 0.400 | 0.480 | 0.141 | 0.333 | 1.090 | 0.437 | 0.641 | 0.610 | 0.522 | 0.444 | 1.400 | 0.497 | 0.857 | 1.250 | 0.593 | 0.367 | 0.930 | 0.422 | 0.000 | 1.120 | 0.470 | 0.800 | 1.120 | 0.502 | 0.514 | 1.100 | 0.458 | 0.592 |
| Snow | 1.270 | 0.437 | 0.923 | 1.020 | 0.439 | 0.657 | 0.940 | 0.472 | 0.270 | 0.270 | 0.143 | 0.333 | 1.030 | 0.434 | 0.652 | 0.610 | 0.522 | 0.444 | 1.240 | 0.504 | 0.857 | 1.250 | 0.587 | 0.367 | 0.900 | 0.417 | 0.000 | 1.140 | 0.465 | 0.800 | 1.140 | 0.495 | 0.557 | 1.070 | 0.456 | 0.605 |
| Speckle | 1.070 | 0.425 | 0.769 | 0.910 | 0.429 | 0.657 | 0.850 | 0.466 | 0.267 | 0.530 | 0.143 | 0.333 | 0.920 | 0.424 | 0.576 | 0.600 | 0.522 | 0.444 | 1.250 | 0.487 | 0.714 | 1.150 | 0.587 | 0.400 | 0.950 | 0.416 | 1.000 | 1.070 | 0.461 | 0.640 | 1.040 | 0.495 | 0.586 | 0.960 | 0.447 | 0.618 |
| Stretch Squish | 1.150 | 0.416 | 0.769 | 0.990 | 0.434 | 0.600 | 0.850 | 0.468 | 0.200 | 0.730 | 0.155 | 0.333 | 0.960 | 0.431 | 0.576 | 0.570 | 0.516 | 0.333 | 1.250 | 0.480 | 0.714 | 1.080 | 0.585 | 0.333 | 0.900 | 0.403 | 0.500 | 1.090 | 0.469 | 0.640 | 1.020 | 0.489 | 0.529 | 1.050 | 0.442 | 0.539 |
| Target Block | 1.300 | 0.445 | 0.923 | 0.990 | 0.430 | 0.643 | 1.030 | 0.475 | 0.400 | 0.400 | 0.146 | 0.333 | 1.020 | 0.426 | 0.652 | 0.600 | 0.519 | 0.444 | 1.340 | 0.501 | 0.714 | 1.180 | 0.581 | 0.400 | 0.980 | 0.420 | 0.500 | 1.130 | 0.461 | 0.800 | 1.080 | 0.498 | 0.529 | 1.080 | 0.453 | 0.599 |
| Temporal Jitter | 1.170 | 0.443 | 0.769 | 0.990 | 0.430 | 0.614 | 0.990 | 0.475 | 0.500 | 0.500 | 0.147 | 0.333 | 1.070 | 0.437 | 0.598 | 0.660 | 0.519 | 0.556 | 1.180 | 0.494 | 0.714 | 1.190 | 0.587 | 0.367 | 0.950 | 0.423 | 0.000 | 1.110 | 0.461 | 0.760 | 1.110 | 0.493 | 0.514 | 1.090 | 0.449 | 0.579 |
| Underexposure | 1.300 | 0.443 | 0.846 | 1.040 | 0.441 | 0.600 | 1.030 | 0.478 | 0.333 | 0.400 | 0.147 | 0.000 | 1.070 | 0.437 | 0.598 | 0.620 | 0.520 | 0.222 | 1.300 | 0.499 | 0.857 | 1.160 | 0.587 | 0.333 | 0.960 | 0.423 | 0.000 | 1.200 | 0.471 | 0.760 | 1.110 | 0.500 | 0.514 | 1.090 | 0.458 | 0.572 |
| Zoom Blur | 1.170 | 0.423 | 0.846 | 0.940 | 0.431 | 0.586 | 0.860 | 0.466 | 0.133 | 0.650 | 0.151 | 0.333 | 0.970 | 0.425 | 0.565 | 0.450 | 0.511 | 0.556 | 1.140 | 0.475 | 0.571 | 1.140 | 0.573 | 0.400 | 0.940 | 0.404 | 0.500 | 1.090 | 0.461 | 0.640 | 0.990 | 0.485 | 0.529 | 0.990 | 0.444 | 0.553 |

Table 21: Video-LLaVA performance across different question types

| Noise Type | CP | | | FP-S | | | FP-C | | | HL | | | Mean | | | LR | | | AR | | | RR | | | CSR | | | TR | | | Mean2 | | | Overall | | |
|---|---|---|---|---|---|---|---|---|---|---|---|---|---|---|---|---|---|---|---|---|---|---|---|---|---|---|---|---|---|---|---|---|---|---|---|---|
| | GPT | SBERT | Acc | GPT | SBERT | Acc | GPT | SBERT | Acc | GPT | SBERT | Acc | GPT | SBERT | Acc | GPT | SBERT | Acc | GPT | SBERT | Acc | GPT | SBERT | Acc | GPT | SBERT | Acc | GPT | SBERT | Acc | GPT | SBERT | Acc | GPT | SBERT | Acc |
| Clean | 0.990 | 0.405 | 0.615 | 0.980 | 0.427 | 0.543 | 0.730 | 0.470 | 0.067 | 0.150 | 0.143 | 0.000 | 0.930 | 0.420 | 0.500 | 0.460 | 0.503 | 0.222 | 1.070 | 0.487 | 0.429 | 1.000 | 0.569 | 0.367 | 0.680 | 0.419 | 0.000 | 0.860 | 0.453 | 0.480 | 0.860 | 0.484 | 0.400 | 0.920 | 0.441 | 0.467 |
| Bit Error | 0.910 | 0.392 | 0.462 | 0.900 | 0.422 | 0.543 | 0.680 | 0.461 | 0.067 | 0.160 | 0.143 | 0.000 | 0.860 | 0.413 | 0.478 | 0.500 | 0.507 | 0.222 | 1.070 | 0.480 | 0.429 | 0.980 | 0.571 | 0.367 | 0.840 | 0.421 | 0.000 | 0.850 | 0.447 | 0.480 | 0.850 | 0.481 | 0.400 | 0.870 | 0.435 | 0.454 |
| Bright Transform | 0.930 | 0.400 | 0.462 | 0.910 | 0.423 | 0.557 | 0.690 | 0.461 | 0.067 | 0.180 | 0.147 | 0.000 | 0.870 | 0.415 | 0.489 | 0.480 | 0.511 | 0.222 | 1.060 | 0.480 | 0.429 | 1.060 | 0.573 | 0.367 | 0.640 | 0.404 | 0.000 | 0.860 | 0.444 | 0.520 | 0.860 | 0.480 | 0.414 | 0.880 | 0.436 | 0.467 |
| Color Quantized | 0.900 | 0.394 | 0.462 | 0.940 | 0.426 | 0.571 | 0.720 | 0.461 | 0.133 | 0.190 | 0.146 | 0.000 | 0.880 | 0.416 | 0.511 | 0.440 | 0.507 | 0.222 | 1.010 | 0.481 | 0.429 | 0.980 | 0.572 | 0.367 | 0.650 | 0.414 | 0.000 | 0.830 | 0.444 | 0.520 | 0.830 | 0.481 | 0.414 | 0.880 | 0.437 | 0.480 |
| Color Shift | 0.910 | 0.402 | 0.462 | 0.890 | 0.420 | 0.557 | 0.690 | 0.461 | 0.133 | 0.210 | 0.143 | 0.000 | 0.850 | 0.413 | 0.500 | 0.480 | 0.503 | 0.222 | 1.140 | 0.479 | 0.429 | 1.030 | 0.574 | 0.367 | 0.620 | 0.409 | 0.000 | 0.860 | 0.453 | 0.560 | 0.860 | 0.482 | 0.429 | 0.870 | 0.435 | 0.480 |
| Contrast | 1.010 | 0.404 | 0.538 | 0.950 | 0.426 | 0.557 | 0.720 | 0.467 | 0.133 | 0.160 | 0.143 | 0.000 | 0.920 | 0.418 | 0.511 | 0.430 | 0.506 | 0.222 | 1.100 | 0.483 | 0.429 | 1.070 | 0.575 | 0.367 | 0.770 | 0.421 | 0.000 | 0.850 | 0.450 | 0.520 | 0.880 | 0.483 | 0.414 | 0.920 | 0.439 | 0.480 |
| Defocus Blur | 0.980 | 0.396 | 0.615 | 0.900 | 0.422 | 0.543 | 0.690 | 0.463 | 0.133 | 0.150 | 0.143 | 0.000 | 0.870 | 0.414 | 0.511 | 0.500 | 0.507 | 0.222 | 1.010 | 0.483 | 0.286 | 1.020 | 0.567 | 0.367 | 0.620 | 0.411 | 0.000 | 0.840 | 0.447 | 0.520 | 0.840 | 0.481 | 0.400 | 0.870 | 0.435 | 0.474 |
| Edge Sawtooth | 0.970 | 0.399 | 0.385 | 0.940 | 0.421 | 0.543 | 0.720 | 0.459 | 0.067 | 0.240 | 0.144 | 0.000 | 0.900 | 0.413 | 0.467 | 0.530 | 0.507 | 0.222 | 1.060 | 0.479 | 0.429 | 0.990 | 0.569 | 0.367 | 0.690 | 0.404 | 0.000 | 0.860 | 0.448 | 0.520 | 0.860 | 0.479 | 0.414 | 0.900 | 0.435 | 0.454 |
| Elastic | 1.030 | 0.400 | 0.538 | 1.010 | 0.424 | 0.543 | 0.840 | 0.465 | 0.067 | 0.210 | 0.143 | 0.000 | 0.970 | 0.415 | 0.489 | 0.680 | 0.505 | 0.222 | 1.090 | 0.477 | 0.429 | 1.080 | 0.567 | 0.367 | 0.880 | 0.412 | 0.000 | 0.940 | 0.453 | 0.480 | 0.940 | 0.481 | 0.400 | 0.970 | 0.436 | 0.461 |
| Flicker | 0.980 | 0.402 | 0.615 | 0.930 | 0.422 | 0.557 | 0.680 | 0.460 | 0.067 | 0.260 | 0.148 | 0.000 | 0.890 | 0.415 | 0.511 | 0.480 | 0.502 | 0.222 | 1.110 | 0.484 | 0.429 | 1.020 | 0.570 | 0.367 | 0.590 | 0.415 | 0.000 | 0.860 | 0.451 | 0.520 | 0.860 | 0.482 | 0.414 | 0.890 | 0.436 | 0.480 |
| Foggy | 0.970 | 0.406 | 0.462 | 0.930 | 0.426 | 0.557 | 0.710 | 0.462 | 0.133 | 0.160 | 0.141 | 0.000 | 0.890 | 0.418 | 0.500 | 0.460 | 0.499 | 0.222 | 1.170 | 0.482 | 0.429 | 1.070 | 0.573 | 0.367 | 0.730 | 0.408 | 0.000 | 0.880 | 0.448 | 0.480 | 0.880 | 0.480 | 0.400 | 0.890 | 0.438 | 0.467 |
| Frame Drop | 0.780 | 0.381 | 0.231 | 0.750 | 0.406 | 0.500 | 0.620 | 0.449 | 0.133 | 0.160 | 0.146 | 0.000 | 0.730 | 0.398 | 0.435 | 0.520 | 0.505 | 0.222 | 0.900 | 0.460 | 0.286 | 1.200 | 0.569 | 0.367 | 0.650 | 0.402 | 0.000 | 0.850 | 0.448 | 0.440 | 0.850 | 0.475 | 0.371 | 0.770 | 0.422 | 0.414 |
| Frame Repeat | 0.790 | 0.372 | 0.462 | 0.730 | 0.401 | 0.471 | 0.590 | 0.436 | 0.067 | 0.080 | 0.136 | 0.000 | 0.710 | 0.392 | 0.424 | 0.560 | 0.515 | 0.222 | 0.880 | 0.459 | 0.286 | 1.020 | 0.563 | 0.367 | 0.590 | 0.391 | 0.000 | 0.830 | 0.444 | 0.520 | 0.830 | 0.473 | 0.400 | 0.760 | 0.418 | 0.421 |
| Frame Replace | 1.070 | 0.409 | 0.615 | 0.940 | 0.427 | 0.571 | 0.690 | 0.468 | 0.133 | 0.190 | 0.146 | 0.000 | 0.920 | 0.420 | 0.533 | 0.500 | 0.509 | 0.222 | 1.040 | 0.489 | 0.286 | 1.040 | 0.575 | 0.367 | 0.840 | 0.423 | 0.000 | 0.870 | 0.450 | 0.560 | 0.870 | 0.486 | 0.414 | 0.910 | 0.441 | 0.493 |
| Frost | 0.880 | 0.398 | 0.308 | 0.900 | 0.419 | 0.571 | 0.640 | 0.462 | 0.067 | 0.190 | 0.143 | 0.000 | 0.850 | 0.412 | 0.478 | 0.450 | 0.505 | 0.222 | 1.020 | 0.475 | 0.429 | 1.050 | 0.569 | 0.367 | 0.740 | 0.408 | 0.000 | 0.860 | 0.447 | 0.560 | 0.860 | 0.478 | 0.414 | 0.860 | 0.433 | 0.461 |
| Gaussian | 0.920 | 0.400 | 0.385 | 0.890 | 0.415 | 0.514 | 0.670 | 0.453 | 0.133 | 0.110 | 0.145 | 0.000 | 0.840 | 0.409 | 0.457 | 0.430 | 0.430 | 0.222 | 1.020 | 0.468 | 0.429 | 0.980 | 0.568 | 0.367 | 0.700 | 0.409 | 0.000 | 0.810 | 0.448 | 0.520 | 0.810 | 0.477 | 0.414 | 0.840 | 0.431 | 0.454 |
| Gaussian Blur | 0.790 | 0.371 | 0.462 | 0.780 | 0.410 | 0.514 | 0.600 | 0.446 | 0.133 | 0.210 | 0.147 | 0.000 | 0.750 | 0.400 | 0.467 | 0.480 | 0.499 | 0.222 | 0.880 | 0.454 | 0.286 | 0.900 | 0.562 | 0.367 | 0.570 | 0.406 | 0.000 | 0.790 | 0.446 | 0.560 | 0.790 | 0.471 | 0.400 | 0.770 | 0.423 | 0.454 |
| Glass Blur | 0.900 | 0.396 | 0.385 | 0.920 | 0.418 | 0.543 | 0.540 | 0.457 | 0.133 | 0.130 | 0.147 | 0.000 | 0.860 | 0.410 | 0.478 | 0.580 | 0.504 | 0.222 | 0.930 | 0.481 | 0.286 | 0.930 | 0.574 | 0.367 | 0.580 | 0.409 | 0.000 | 0.730 | 0.446 | 0.520 | 0.730 | 0.481 | 0.386 | 0.840 | 0.433 | 0.454 |
| H265 Artifacts | 0.840 | 0.379 | 0.385 | 0.770 | 0.406 | 0.543 | 0.640 | 0.441 | 0.133 | 0.150 | 0.146 | 0.000 | 0.750 | 0.397 | 0.478 | 0.580 | 0.497 | 0.222 | 0.760 | 0.447 | 0.286 | 0.930 | 0.564 | 0.367 | 0.570 | 0.405 | 0.000 | 0.750 | 0.444 | 0.480 | 0.750 | 0.470 | 0.386 | 0.760 | 0.420 | 0.447 |
| Impulse | 0.720 | 0.368 | 0.385 | 0.780 | 0.405 | 0.557 | 0.600 | 0.437 | 0.133 | 0.190 | 0.151 | 0.000 | 0.740 | 0.395 | 0.489 | 0.590 | 0.507 | 0.222 | 0.810 | 0.447 | 0.286 | 0.950 | 0.564 | 0.367 | 0.480 | 0.390 | 0.000 | 0.790 | 0.440 | 0.560 | 0.790 | 0.468 | 0.414 | 0.770 | 0.418 | 0.467 |
| Jpeg Artifact | 0.860 | 0.391 | 0.308 | 0.890 | 0.418 | 0.557 | 0.600 | 0.451 | 0.133 | 0.190 | 0.146 | 0.000 | 0.820 | 0.409 | 0.478 | 0.540 | 0.511 | 0.222 | 0.990 | 0.470 | 0.429 | 0.980 | 0.567 | 0.367 | 0.810 | 0.407 | 0.000 | 0.830 | 0.441 | 0.480 | 0.830 | 0.476 | 0.400 | 0.830 | 0.430 | 0.454 |
| Motion Blur | 0.760 | 0.376 | 0.385 | 0.800 | 0.407 | 0.529 | 0.760 | 0.451 | 0.133 | 0.300 | 0.145 | 0.000 | 0.790 | 0.400 | 0.467 | 0.560 | 0.499 | 0.222 | 0.790 | 0.452 | 0.286 | 0.970 | 0.567 | 0.367 | 0.710 | 0.404 | 0.000 | 0.800 | 0.443 | 0.440 | 0.800 | 0.470 | 0.371 | 0.800 | 0.422 | 0.434 |
| Overexposure | 0.940 | 0.403 | 0.462 | 0.880 | 0.419 | 0.557 | 0.730 | 0.458 | 0.133 | 0.190 | 0.145 | 0.000 | 0.860 | 0.412 | 0.500 | 0.450 | 0.510 | 0.222 | 1.010 | 0.465 | 0.429 | 1.010 | 0.566 | 0.367 | 0.600 | 0.413 | 0.000 | 0.810 | 0.446 | 0.520 | 0.810 | 0.476 | 0.414 | 0.860 | 0.432 | 0.474 |
| Poisson | 0.860 | 0.388 | 0.385 | 0.880 | 0.412 | 0.557 | 0.610 | 0.458 | 0.133 | 0.110 | 0.149 | 0.000 | 0.780 | 0.403 | 0.478 | 0.510 | 0.508 | 0.222 | 1.020 | 0.463 | 0.429 | 1.020 | 0.566 | 0.367 | 0.650 | 0.411 | 0.000 | 0.820 | 0.442 | 0.560 | 0.820 | 0.474 | 0.414 | 0.800 | 0.426 | 0.467 |
| Rainy | 0.930 | 0.398 | 0.462 | 0.920 | 0.423 | 0.557 | 0.690 | 0.462 | 0.133 | 0.190 | 0.145 | 0.000 | 0.880 | 0.414 | 0.500 | 0.450 | 0.506 | 0.222 | 1.030 | 0.485 | 0.429 | 0.980 | 0.572 | 0.367 | 0.740 | 0.430 | 0.000 | 0.840 | 0.453 | 0.560 | 0.840 | 0.486 | 0.429 | 0.870 | 0.437 | 0.480 |
| Random Block | 0.920 | 0.392 | 0.462 | 0.890 | 0.421 | 0.543 | 0.670 | 0.458 | 0.133 | 0.100 | 0.142 | 0.000 | 0.840 | 0.412 | 0.478 | 0.580 | 0.505 | 0.222 | 0.990 | 0.486 | 0.429 | 1.040 | 0.567 | 0.367 | 0.740 | 0.416 | 0.000 | 0.870 | 0.448 | 0.560 | 0.870 | 0.482 | 0.414 | 0.860 | 0.434 | 0.461 |
| Reflect | 0.900 | 0.400 | 0.385 | 0.910 | 0.423 | 0.543 | 0.700 | 0.459 | 0.133 | 0.180 | 0.146 | 0.000 | 0.860 | 0.414 | 0.478 | 0.480 | 0.508 | 0.222 | 1.050 | 0.472 | 0.429 | 0.930 | 0.572 | 0.367 | 0.730 | 0.420 | 0.000 | 0.830 | 0.449 | 0.560 | 0.830 | 0.481 | 0.429 | 0.860 | 0.435 | 0.467 |
| Resolution Degrade | 0.930 | 0.400 | 0.462 | 0.870 | 0.417 | 0.557 | 0.680 | 0.463 | 0.133 | 0.210 | 0.147 | 0.000 | 0.840 | 0.411 | 0.500 | 0.560 | 0.503 | 0.222 | 1.050 | 0.481 | 0.429 | 1.030 | 0.577 | 0.367 | 0.590 | 0.402 | 0.000 | 0.780 | 0.443 | 0.480 | 0.780 | 0.480 | 0.386 | 0.860 | 0.433 | 0.461 |
| Rolling Shutter | 0.940 | 0.399 | 0.462 | 0.870 | 0.416 | 0.529 | 0.590 | 0.455 | 0.133 | 0.130 | 0.142 | 0.000 | 0.830 | 0.409 | 0.478 | 0.450 | 0.504 | 0.222 | 1.010 | 0.475 | 0.429 | 1.070 | 0.579 | 0.367 | 0.840 | 0.421 | 0.000 | 0.810 | 0.447 | 0.480 | 0.810 | 0.484 | 0.400 | 0.840 | 0.433 | 0.454 |
| Shadow | 0.930 | 0.397 | 0.385 | 0.910 | 0.422 | 0.543 | 0.720 | 0.465 | 0.133 | 0.180 | 0.142 | 0.000 | 0.870 | 0.414 | 0.500 | 0.460 | 0.504 | 0.222 | 1.050 | 0.477 | 0.429 | 0.980 | 0.571 | 0.367 | 0.650 | 0.412 | 0.000 | 0.800 | 0.450 | 0.520 | 0.830 | 0.481 | 0.414 | 0.860 | 0.436 | 0.461 |
| Snow | 0.950 | 0.404 | 0.615 | 0.940 | 0.423 | 0.543 | 0.800 | 0.467 | 0.133 | 0.180 | 0.145 | 0.000 | 0.900 | 0.416 | 0.511 | 0.420 | 0.504 | 0.222 | 1.050 | 0.477 | 0.429 | 0.980 | 0.571 | 0.367 | 0.680 | 0.419 | 0.000 | 0.860 | 0.451 | 0.560 | 0.860 | 0.482 | 0.429 | 0.900 | 0.437 | 0.487 |
| Speckle | 0.880 | 0.385 | 0.462 | 0.840 | 0.416 | 0.514 | 0.700 | 0.454 | 0.133 | 0.130 | 0.149 | 0.000 | 0.810 | 0.406 | 0.467 | 0.560 | 0.513 | 0.222 | 0.980 | 0.470 | 0.286 | 0.980 | 0.569 | 0.367 | 0.830 | 0.425 | 0.000 | 0.790 | 0.440 | 0.520 | 0.780 | 0.478 | 0.386 | 0.830 | 0.433 | 0.447 |
| Stretch Squish | 0.740 | 0.372 | 0.385 | 0.820 | 0.410 | 0.543 | 0.700 | 0.453 | 0.133 | 0.210 | 0.147 | 0.000 | 0.780 | 0.400 | 0.478 | 0.490 | 0.505 | 0.222 | 0.850 | 0.457 | 0.286 | 0.950 | 0.566 | 0.367 | 0.520 | 0.392 | 0.000 | 0.770 | 0.441 | 0.520 | 0.770 | 0.471 | 0.400 | 0.790 | 0.423 | 0.454 |
| Target Block | 0.940 | 0.393 | 0.462 | 0.920 | 0.420 | 0.557 | 0.700 | 0.459 | 0.133 | 0.180 | 0.138 | 0.000 | 0.880 | 0.411 | 0.500 | 0.540 | 0.508 | 0.222 | 1.070 | 0.472 | 0.429 | 0.990 | 0.567 | 0.367 | 0.770 | 0.429 | 0.000 | 0.860 | 0.445 | 0.480 | 0.860 | 0.480 | 0.386 | 0.880 | 0.433 | 0.461 |
| Temporal Jitter | 0.880 | 0.392 | 0.231 | 0.820 | 0.415 | 0.529 | 0.610 | 0.463 | 0.150 | 0.150 | 0.138 | 0.000 | 0.790 | 0.408 | 0.446 | 0.500 | 0.508 | 0.222 | 1.070 | 0.485 | 0.429 | 0.680 | 0.576 | 0.367 | 0.680 | 0.423 | 0.000 | 0.830 | 0.448 | 0.440 | 0.830 | 0.484 | 0.386 | 0.830 | 0.432 | 0.428 |
| Underexposure | 0.990 | 0.404 | 0.462 | 0.940 | 0.426 | 0.543 | 0.740 | 0.465 | 0.133 | 0.190 | 0.145 | 0.000 | 0.910 | 0.417 | 0.489 | 0.440 | 0.501 | 0.222 | 1.230 | 0.486 | 0.429 | 1.100 | 0.573 | 0.367 | 0.790 | 0.423 | 0.000 | 0.940 | 0.448 | 0.560 | 0.940 | 0.483 | 0.429 | 0.920 | 0.439 | 0.474 |
| Zoom Blur | 0.830 | 0.384 | 0.385 | 0.790 | 0.409 | 0.529 | 0.640 | 0.455 | 0.133 | 0.130 | 0.147 | 0.000 | 0.770 | 0.402 | 0.467 | 0.570 | 0.501 | 0.111 | 0.780 | 0.456 | 0.286 | 0.930 | 0.564 | 0.367 | 0.560 | 0.394 | 0.000 | 0.770 | 0.447 | 0.480 | 0.770 | 0.471 | 0.371 | 0.790 | 0.425 | 0.441 |

