# OpenReview forum: "NoisyVideo: A Comprehensive Benchmark for Robust Video Understanding under Visual Noise"
_ICLR.cc/2026/Conference — ICLR 2026 Conference Withdrawn Submission_

### Official Review · Reviewer_Y7Tu · 2025-10-30

**Soundness:** 3
**Presentation:** 4
**Contribution:** 3
**Rating:** 6
**Confidence:** 5

**Summary:**

This paper constructs a comprehensive NOISYVideo benchmark to assess the behavior of Video-LLMs in face of various noises that may present in real situation. The video and QAs are sourced from MMBench-Video, and noises are generated in a controlled condition. The analyses are multi-faceted, covering multiple models, noise types, video genres, and question categories. It also discusses the effects of video restoration (de-noising) for improvements.

**Strengths:**

1.	NOISYVideo is a comprehensive benchmark for noise analysis in videos.
2.	The analyses are multi-faceted and soundness.
3.	The paper is well-structured and easy to read.

**Weaknesses:**

1.	Aside from the noise types, the degree/extent of a specific noise could be an important factor to affect model performance. The current analyses seem to have ignored this factor. I am thus curious about if and how to make sure different types of noises are added at the same level. Without such constraint, the comparison and analyses would be less meaningful.
2.	To help understand if the noise continues to affect more advanced models. It would be better to conduct assessments for the following models: Qwen2.5-VL 7B, Qwen3-VL, InternVL 3.5, LLaVA-Video, LLaVA-OV, Gemini 2.5 Pro, GPT-5.
3.	I notice that most of the noise types are added at a static image level. This raise a concern that the overall judgment could be dominated by the observations on such noise types, leading to biased conclusions such as Video-LLMs are majorly susceptible on noisy object attribute reasoning. I would suggest to analyze question category-wise model behaviors on corresponding subsets of noise types. For example, analyze temporal reasoning on types of temporal noises.
4.	As the noises are exclusively added to the visual content, it would be better to analyze the video encoders of different Video-LLMs to help better understand their different behaviors.
5.	To help better situate the work in the literature, related work should be moved from the appendix to the main text. If there is no space, I would suggest to condense the introduction for evaluation metrics as they are well-known and are not the contribution of this work.

**Questions:**

No

---

### Official Review · Reviewer_4dp5 · 2025-11-01

**Soundness:** 2
**Presentation:** 2
**Contribution:** 2
**Rating:** 4
**Confidence:** 3

**Summary:**

This paper introduces **NoisyVideo**, a benchmark that systematically evaluate the robustness of Video LLM (VLLM) in scenarios with visual noise. Specifically, NoisyVideo comprises encompasses 36 noise types in 8 categories, and 21924 noise-test videos. The experiment results reveal that current Video LLM suffers from performance degradation, particularly for tasks requiring fine-grained understanding and reasoning.

**Strengths:**

1. This paper tackles a problem that is very essential for current Video-LLMs, as researchers ofen focus more on the general or specifical performance of the models rather than the robustness.
2. This paper is will-motivated and contains a variety types of noise, which is much more than other benchmarks.

**Weaknesses:**

1. The contribution is limited. The source videos and questions are adopted from MMBench-Video[1],  while applying noise introduces a new dimension, the core content and evaluation tasks are unchanged. Therefore, it's more like a "noisy extension" of MMBench-Video rather than a new resource designed for robustness evaluation of Video-LLMs.

2. The noise is added to the video solely. However, in real-world scenarios, the visual noise is often mixed and comes from different sources, thus the noisy videos may not faithfully reveal the chanllenges in real-world scenarios.

3. As for the experimental setting, The input frames are fixed to 8 for all models, but sampling only 8 frames for long videos (~165s average) provides a very sparse input. This extreme sparsity may reduce the temporal context the models could use to mitigate noise effects.

4. From the provided case study, it seems like the 90% of the input frames are corrupted, just as the authors claimed, but that raises a concern about the reasonaibility of some types of visual noise, especially for temporal noise. A critical question: does evaluating model performance under such near-total loss of semantic information truly measure robustness to noise, or does it primarily probe the model's behavior under conditions of extreme data deprivation? For example, By the case study provided in Figure 12(e), if a model response the query with "one child", how to evaluate this? It is conficted with the ground truth answer, but the model make a right answer based on the input frames.

5. Lack of model evaluation. Most evaluated model purposed

6. As for the paper writing, there are some typos for evaluation:

   a. It seems like all the citations are added by "citet", it's better to follow the regulation for proper citation.

   b. In page 25 and 26 (Appendix), the captions are overlapped with the page index.

   c. In line 836, it might be $\beta = \rho - \alpha$ rather than $\beta = 1 - \alpha$

**Questions:**

1. Could you clarify the detailed implementaion for applying corruptions to the videos? Specifically, are these operations applied on the original, full video sequence before the input frames were sampled, or applied directly to the sampled frames?

---

### Official Review · Reviewer_d4fA · 2025-11-01

**Soundness:** 2
**Presentation:** 2
**Contribution:** 2
**Rating:** 4
**Confidence:** 5

**Summary:**

This paper proposes NoisyVideo, a comprehensive benchmark covering 36 noise types across 8 categories (e.g., quality, blur, temporal disruptions) and 21,924 noise-corrupted test videos. The paper evaluates several state-of-the-art Video MLLMs using this benchmark, analyzing performance degradation under different noises, video genres, and question types, and also explores the effectiveness of existing image restoration techniques.

**Strengths:**

1. The first systematic benchmark for assessing Video MLLMs’ robustness to visual noise, with 8 broad noise categories and 36 realistic noise types.

2. A multi-dimensional evaluation framework integrating subjective (GPT-based scores), objective (SBERT semantic similarity), and accuracy metrics for holistic assessment.

3. Fully open-sourcing the benchmark.

**Weaknesses:**

1. While the paper simulates video degradation, as a benchmark, it would be more compelling to incorporate real-world noisy videos (even if that means reducing the total video count) instead of relying solely on simulations. This would make the evaluation more convincing.

2. With many existing benchmarks for MLLMs, the paper currently struggles to convince readers why its NoisyVideo benchmark will be widely used. It needs more clarity on its unique value compared to other benchmarks to build trust in its practical utility.

**Questions:**

See weaknesses.

---

### Note · Authors · 2025-12-09

I have read and agree with the venue's withdrawal policy on behalf of myself and my co-authors.